# Balancing Learning Rates Across Layers: Exact Two-Step Dynamics and Optimal Scaling in Linear Neural Networks

**Tianyu Pang** [1]  **Vignesh Kothapalli** [2]  **Shenyang Deng** [1]  **Haohui Wang** [3]  **Dawei Zhou** [3]  **Yaoqing Yang** [1]

## Abstract

We study optimal learning-rate selection in two-layer and three-layer linear neural networks trained to learn linear target functions. In particular, we derive the exact closed-form expressions for the gradients and test loss after one and two steps of gradient descent, enabling a precise characterization of early training dynamics. We characterize how learning rates should scale under the gradient approximation in the first two steps, and prove that performing updates with this approximation yields a tractable surrogate loss with a tight, small approximation error. This formulation enables the theoretical analysis of layer-wise learning rates and reveals a distinct early-training regime: test loss can be minimized by unequal learning rates at the initial step, while equal learning rates become optimal in subsequent steps. Our numerical experiments validate the theory and demonstrate the importance of balancing layer-wise learning rates early during training. The code is available at: `TDCSZ327/Layer-Balancing`.

## 1. Introduction

The dynamics of gradient descent in deep neural networks are shaped not only by the architecture and initialization (Saxe et al., 2019; Goldt et al., 2020; Kunin et al., 2024) but also by the choice of learning rates for individual layers. In practice, networks often use separate learning rates across layers, or adopt layer-wise adaptive schedules (You et al., 2017; Zhou et al., 2023; Wang et al., 2025; He et al., 2026) to accelerate convergence and improve generalization. Even in simpler linear overparameterized models, however, it is unclear how the relative scale of layer-wise learning rates affects training trajectories and resulting test performance. Small deviations in early gradient updates can propagate through layers and affect learned representations in ways that are difficult to quantify.

In neural networks, the gradient updates tend to couple dominant, signal-aligned components with smaller residual terms (Ba et al., 2022; Wang et al., 2023; Kothapalli et al., 2025). Since the norms of these components vary across layers, the learning rates play a crucial role in determining the training dynamics. A theoretical analysis of these components requires a layer-by-layer training assumption that is atypical of practical settings. On the other hand, linear networks offer a rich alternative where the linear interaction between layers naturally determines the structure of the gradient (Saxe et al., 2014). In particular, each layer's update depends on the product of the weights in the other layers and the data matrix. However, the effects of layer-wise learning rates on the learning dynamics are still not well understood.

Previous works on continuous-time gradient flow analyses in linear networks suggest that the weight norms across layers may balance over time (Du et al., 2018; Ye & Du, 2021; Wang et al., 2021). Whereas kernel-based approximations predict nearly linear evolution of outputs (Jacot et al., 2018; Hu & Lu, 2022). However, these approaches do not study the effects of learning rate selection in discrete, finite-step settings on the signal-residual coupling in gradient updates. This coupling effect is further complicated by network depth. Even for three-layer linear networks, the gradient of the output with respect to an intermediate layer contains products of multiple weight matrices and the data matrix, leading to higher-order interactions that influence both the magnitude and direction of updates. Approaches that consider layers independently fail to capture these effects (Arora et al., 2018), and conventional mean-field (Mei et al., 2018) or maximal-update (Yang & Hu, 2021; Yang et al., 2022) analyses tend to rely on infinitesimal step sizes. Consequently, predicting how the choice of layer-wise learning rates affects early generalization requires a framework that can both isolate the leading components of the gradient and quantify their contribution to test performance.

Previous work on layer-wise adaptation has focused primarily on heuristic or asymptotic regimes. Methods such as per-layer decay, adaptive optimizers (Zhou et al., 2023; Liu et al., 2024), or normalization-based rescaling (You et al., 2018; Yang et al., 2022) are motivated by empirical

[1]Dartmouth College [2]Stanford University [3]Virginia Tech. Correspondence to: Yaoqing Yang <Yaoqing.Yang@dartmouth.edu>.

*Proceedings of the 43rd International Conference on Machine Learning*, Seoul, South Korea. PMLR 306, 2026. Copyright 2026 by the author(s).

improvements but do not offer explicit formulas linking learning rates to test loss. Analyses of implicit bias (Arora et al., 2019; Gidel et al., 2019) or norm balancing describe certain asymptotic trajectories, yet they do not address the finite-step dynamics where early layer-wise interactions are critical. In multi-layer settings, these interactions determine whether early updates align with the signal or are dominated by cross-layer interference, and small differences in learning rates can have a disproportionate effect on generalization.

In this paper, we develop a framework for analyzing layer-wise learning rates in two-layer and three-layer linear networks under random orthogonal initialization. Central to our approach is a gradient decomposition that separates the dominant, label-aligned component of each layer's update from smaller residual terms, allowing closed-form expressions for the test loss after one and two gradient steps. Our contributions include:

- Characterizing the dominant components of gradients and rigorously bounding residual terms in operator norm, establishing conditions under which approximate gradients accurately capture test loss dynamics.

- Showing that in two-layer networks, symmetric learning rates across layers are suboptimal after a single update due to the distinct roles of representation and readout, but become locally optimal after two updates in sufficiently wide networks, revealing a transition from asymmetric to balanced learning rates.

- Extending the analysis to three-layer networks with a scalar output, capturing richer cross-layer interactions, identifying distinct scaling regimes for admissible learning rates, and providing explicit test loss expressions that include higher-order interactions between layer updates.

- Identifying critical thresholds for learning rates ($\eta = O(h\sqrt{h})$ for two layers, $\eta = O(h)$ for three layers) beyond which gradient dynamics and test loss behavior qualitatively change, connecting with maximal-update and mean-field scaling regimes.

While our analysis is restricted to linear networks with orthogonal initialization, it provides a principled foundation for understanding how layer-wise learning rates shape early generalization, and offers insights that can guide the design of learning rate schedules in more complex architectures.

## 2. Related Work

### 2.1. Layer-wise Hyperparameter Tuning

When training and fine-tuning deep learning models, layer-wise hyperparameter tuning serves as a lightweight and memory-efficient tuning paradigm (Yao et al., 2024). It

has shown great potential to reconcile the coarse granularity of global tuning (Loshchilov & Hutter, 2016; Hu et al., 2024) and the high memory demands of parameter-wise tuning (Kingma, 2014; Liu et al., 2019; Yao et al., 2021). For example, Howard & Ruder (2018); Long et al. (2015) have shown layer-wise learning rate strategies can enhance test accuracy in both transfer learning and domain adaptation tasks. LARS and LAMB (You et al., 2017; 2018) propose "trust ratio" to assign layer-wise learning rates and mitigate gradient divergence in large-batch training. accelerating the training of large models on computer vision (CV) and natural language processing (NLP) tasks. AutoLR (Ro & Choi, 2021) automatically tunes its layer-wise learning rates according to the "role" of each layer to balance layer-wise weight variations. Adam-mini (Zhang et al., 2024) and Blockwise-LR (Wang et al., 2025) assign layer-wise learning rate based on the different Hessian block structures in Transformers (Vaswani et al., 2017). Complementing these algorithmic heuristics, in this work we provide an exact two-step characterization in two- and three-layer linear networks that links layer-wise learning-rate allocation directly to test loss, yielding a principled prescription for when asymmetric versus balanced learning rates are optimal. We include more discussion of other layer-wise parameters, such as the pruning ratio and weight decay, in Appendix B.

### 2.2. Layer-balancing Phenomenon

Prior work like Du et al. (2018); Wang et al. (2021); Ye & Du (2021) shows that the norm difference between adjacent layers of deep homogeneous models stays constant or vanishes during training, they term this as one kind of automatic layer balancing. More recently, Zhou et al. (2023); Liu et al. (2024) find that balancing weight spectra across layers helps model training, and they propose a layer-wise learning rate scheduler, called TempBalance, that allocates learning rates by assessing the heavy-tailness of each layer (a property that correlates with layer quality).

In addition, Kunin et al. (2024) study the training dynamics of linear neural networks and find that when all layers learn at similar rates, linear neural networks exhibit rapid feature learning. Likewise, Yang et al. (2022) and Yang et al. (2024) show that for square matrices, balanced learning-rates can be optimal under the maximal-update parameterization. Both results are consistent with our findings.

## 3. Preliminaries and Setup

**Notation.** For $n \in \mathbb{N}$, we denote $[n] = \{1, \cdots, n\}$. We use $O(\cdot)$ to denote the standard big-O notation and the subscript $O_d(\cdot)$ to denote the asymptotic limit of $d \to \infty$. Formally, for two sequences of real numbers $x_d$ and $y_d$, $x_d = O_d(y_d)$ represents $\lim_{d \to \infty} |x_d| \leq C_1|y_d|$ for some constant $C_1$. Similarly, $x_d = O_{d,\mathbb{P}}(y_d)$ denotes that the asymptotic inequality almost surely holds under a prob-

ability measure $\mathbb{P}$. The definitions can be extended to the standard $\Omega(\cdot), \Theta(\cdot)$ notations analogously. For two sequences of real numbers $x_d$ and $y_d$, $x_d \asymp y_d$ represents $C_2 |y_d| \leq |x_d| \leq C_1 |y_d|$, for constants $C_1, C_2 > 0$ For a real matrix $\boldsymbol{B} = (B_{ij})_{n \times m} \in \mathbb{R}^{n \times m}$, $\boldsymbol{B}^{\circ p}$ represents an element-wise $p$-power transformation such that $\boldsymbol{B}^{\circ p} = (B_{ij}^p)_{n \times m}$. $\odot$ is the matrix Hadamard product, sign(.) denotes the element-wise sign function. $\|\cdot\|$ denotes the $\ell_2$ norm for vectors and the operator norm for matrices. $\|\cdot\|_F$ denotes the Frobenius norm. $\boldsymbol{0}_{h \times d}, \boldsymbol{1}_{h \times d} \in \mathbb{R}^{h \times d}$ represent the all-zero and all-ones matrices, respectively.

**Definition 3.1** (Orthogonal initialization). *We say a random matrix $\boldsymbol{O}_1 \in \mathbb{R}^{h \times h}$ is random orthogonal if it is uniformly distributed on the orthogonal group with respect to the Haar measure, i.e. $\boldsymbol{O}_1 \boldsymbol{O}_1^\top = \boldsymbol{O}_1^\top \boldsymbol{O}_1 = \boldsymbol{I}_h$. We say a random vector $\boldsymbol{a}_1 \in \mathbb{R}^h$ is random orthogonal if it is uniformly distributed on the orthogonal group with respect to the Haar measure, i.e. $\boldsymbol{a}_1^\top \boldsymbol{a}_1 = 1$, and $\mathbb{E}[\boldsymbol{a}_1 \boldsymbol{a}_1^\top] = \frac{1}{h} \boldsymbol{I}_h$.*

**Assumption 3.2.** *For random orthogonal initialization of the NN weights, we assume data number $n$ = model width $h$ = data dimension $d$.*

**Two-layer and three-layer NNs.** For the two-layer case, we consider a linear NN as our *student* model $f(\cdot) : \mathbb{R}^h \to \mathbb{R}^h$. For an input $\boldsymbol{x}_i \in \mathbb{R}^h$, its prediction is formulated as:

$$f(\boldsymbol{x}_i) = \frac{1}{h} \boldsymbol{x}_i^\top \boldsymbol{W}_1 \boldsymbol{W}_2. \tag{1}$$

For the three-layer case, we consider a linear NN as our *student* model $f^*(\cdot) : \mathbb{R}^h \to \mathbb{R}$. For an input $\boldsymbol{x}_i \in \mathbb{R}^h$, its prediction is formulated as:

$$f^*(\boldsymbol{x}_i) = \frac{1}{\sqrt{h}} \boldsymbol{x}_i^\top \boldsymbol{W}_1 \boldsymbol{W}_2 \boldsymbol{a}. \tag{2}$$

Here $\boldsymbol{W}_1 \in \mathbb{R}^{h \times h}, \boldsymbol{W}_2 \in \mathbb{R}^{h \times h}, \boldsymbol{a} \in \mathbb{R}^h$ are the first two hidden layers and last layer weights, respectively, with random orthogonal initialization. To keep the loss well-scaled, we use a scaling coefficient of $\frac{1}{h}$ for the two-layer network and $\frac{1}{\sqrt{h}}$ for the three-layer network.

**Dataset.** We use linear *teacher* models to generate the training data of both two-layer and three-layer *student* networks under random orthogonal initialization. We sample input data $\boldsymbol{X} \in \mathbb{R}^{h \times h}$ as $h$ data points $\{\boldsymbol{x}_1, \cdots, \boldsymbol{x}_h\}$, where $\frac{1}{\sqrt{h}} \boldsymbol{X}$ is a random orthogonal matrix. To simplify the analysis, we do not consider label noise.

- **Two-layer NN Case.** For a given $\boldsymbol{x}_i \in \mathbb{R}^h$, we use a linear teacher model $F : \mathbb{R}^h \to \mathbb{R}^h$ to generate the corresponding label $\boldsymbol{y}_i \in \mathbb{R}^h$ (Du et al., 2018) as:

$$\boldsymbol{y}_i = F(\boldsymbol{x}_i) = \boldsymbol{M}^\top \boldsymbol{x}_i. \tag{3}$$

Here, $\boldsymbol{M} \in \mathbb{R}^{h \times h}$ is the *target matrix*, where $\sqrt{h} \boldsymbol{M}$ is a random orthogonal matrix. We represent $\boldsymbol{X} \in$

$\mathbb{R}^{h \times h}, \boldsymbol{Y} \in \mathbb{R}^{h \times h}$ as the input matrix and the label matrix, respectively.

- **Three-layer NN Case.** For a given $\boldsymbol{x}_i \in \mathbb{R}^h$, we use a linear teacher model $F^* : \mathbb{R}^h \to \mathbb{R}$ to generate the corresponding scalar label $y_i \in \mathbb{R}$ as:

$$y_i = F^*(\boldsymbol{x}_i) = \boldsymbol{\beta}^{*\top} \boldsymbol{x}_i. \tag{4}$$

Here random orthogonal vector $\boldsymbol{\beta}^* \in \mathbb{R}^h$ is the *target direction*. We represent $\boldsymbol{X} \in \mathbb{R}^{h \times h}, \boldsymbol{y} \in \mathbb{R}^h$ as the input matrix and the label vector, respectively.

**Training procedure.** We adopt GD as the optimizer for training. Here we consider a simple training procedure: For the two-layer NN, we apply GD updates on both the layers simultaneously. For the three-layer NN, each GD update only simultaneously trains the first two hidden layers and we fix the last layer weights $\boldsymbol{a} \in \mathbb{R}^h$. In both settings, we employ the mean-squared error as our training loss function:

$$\hat{L}_{\text{two-layer}}(f, \boldsymbol{X}, \boldsymbol{Y}) = \frac{1}{2h} \|\boldsymbol{Y} - f(\boldsymbol{X})\|_F^2 \tag{5}$$

$$\hat{L}_{\text{three-layer}}(f^*, \boldsymbol{X}, \boldsymbol{y}) = \frac{1}{2h} \|\boldsymbol{y} - f^*(\boldsymbol{X})\|^2. \tag{6}$$

Where $\boldsymbol{X} \in \mathbb{R}^{h \times h}, \boldsymbol{Y} \in \mathbb{R}^{h \times h}, \boldsymbol{y} \in \mathbb{R}^h$ are training data, and labels for two-layer and three-layer NN, respectively.

**Assumption 3.3.** *We consider a non-asymptotic setting, where $h, n \leq C$ and $C$ is a large constant.*

**Assumption 3.4.** *We aim to determine whether using the same learning rates across layers leads to minimal test loss for networks trained with a one-step or two-step GD update when $\eta_1 + \eta_2 = 2h^\alpha$, where $h^\alpha \leq$ critical threshold ($O(h\sqrt{h})$ for two-layer NN, $O(h)$ for three-layer NN).*

# 4. Overview of Main Results: Balancing Layer-wise Learning Rates

In this work, we study how layer-wise learning rates influence early training dynamics and generalization in linear neural networks. Our analysis reveals how the interplay between network width, training depth, and learning-rate scale drives the emergence of asymmetric versus balanced updates across layers.

**Gradient decomposition and leading-order approximation.** For both two-layer and three-layer networks, we decompose the exact gradients into leading-order, signal-aligned terms and smaller residual terms:

$$\boldsymbol{G}_i^t = \boldsymbol{B}_i^t - \boldsymbol{A}_i^t, \quad i = 1, 2,$$

where $\boldsymbol{A}_i^t$ captures the primary contribution from the labels, and $\boldsymbol{B}_i^t$ is higher-order in $\frac{1}{\sqrt{h}}$. For learning rates below the critical thresholds ($\eta_1, \eta_2 \leq O(h\sqrt{h})$ for two-layer,

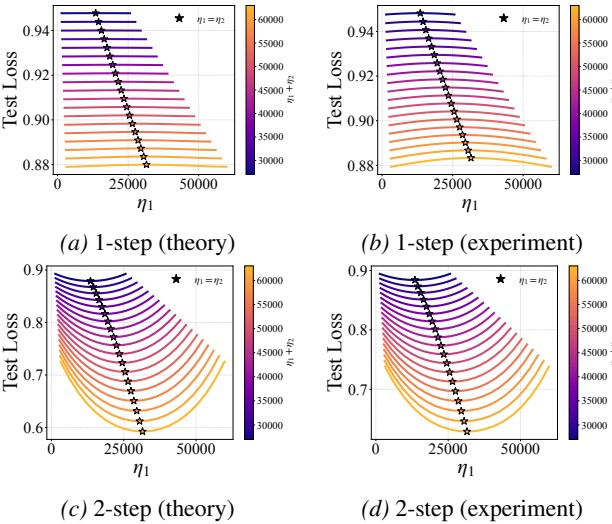

*(a)* 1-step (theory)  *(b)* 1-step (experiment)

*(c)* 2-step (theory)  *(d)* 2-step (experiment)

*Figure 1.* **2-layer NN under orthogonal initialization.** Here we set $\eta_1 + \eta_2 \leq O(h^{\frac{3}{2}})$ and $h = 1000$. We observe that the theoretical losses closely track the empirical test losses measured after either one or two updates. Moreover, a clear qualitative shift emerges: after a single update, symmetric learning rates across layers are suboptimal, whereas after two updates they become locally optimal. We discuss this in Section 6.

$\eta_1, \eta_2 \leq O(h)$ for three-layer), $\boldsymbol{B}_i^t$ is negligible in norm. This decomposition justifies replacing exact gradients with their leading-order components when computing the test loss, simplifying analysis and revealing the dominant factors that govern learning-rate balance.

**Two-layer networks: asymmetry vs. balance.** For the student network $f(\boldsymbol{x}) = \frac{1}{h}\boldsymbol{x}^\top \boldsymbol{W}_1 \boldsymbol{W}_2$ trained on labels $\boldsymbol{y} = \boldsymbol{M}^\top \boldsymbol{x}$, we find:

- *Early asymmetry:* After a single gradient step, symmetric learning rates $\eta_1 = \eta_2$ do not minimize the test loss. Early training favors asymmetric updates, as the first layer primarily absorbs the label signal while the second layer transmits it. Enforcing symmetry too early limits the network's ability to exploit this distinction.

- *Emergent balance:* After two steps, for sufficiently large width and an appropriate range of total learning-rate scale, symmetric layer-wise learning rates become locally optimal. At this stage, layers have coordinated sufficiently, making balanced updates advantageous for minimizing test loss.

**Three-layer networks: extension and scaling.** For the student network $f^*(\boldsymbol{x}) = \frac{1}{\sqrt{h}}\boldsymbol{x}^\top \boldsymbol{W}_1 \boldsymbol{W}_2 \boldsymbol{a}$ with output vector $\boldsymbol{a}$, the qualitative phenomena is similar to the two-layer NNs. Early training favors asymmetric learning rates, reflecting distinct roles of hidden layers. After multiple steps, balanced learning rates emerge as optimal, but the admis-

sible learning-rate regime is reduced to $O(h)$ due to differences in the initialization schemes of the two- and three-layer NNs, as well as the additional output layer. Two-step test loss also depends more strongly on higher-order products of $\eta_1 \eta_2$, highlighting deeper cross-layer interactions.

**Key takeaway: balancing learning rates across layers.** Our results show that optimal layer-wise learning rates are dynamic. Early-stage training benefits from asymmetry to leverage layer-specific signal propagation, while deeper or later-stage updates promote balance, enabling coordinated alignment across layers. This perspective connects explicit gradient norms, test loss formulas, and width scaling to a principled understanding of when and why learning-rate balancing is beneficial in linear NNs. In Appendix E and F, we extend the main results under orthogonal initialization to the gaussian initialization setting, obtaining a theoretical loss expression for the one-step GD update and complementing it with simulation experiments for the multi-step case, we find that similar conclusions hold.

## 5. Main Results

### 5.1. Norm Analysis of Gradient Matrices

Here we give the norm analysis of update gradient matrices under random orthogonal initialization. This analysis is an important step in simplifying the derivation of the theoretical test loss in the next Section (Section 5.3). It also provides intuition about the range of learning rates that are beneficial for model training and offers a deeper understanding of the gradient matrices. Here, we examine the norm properties of the gradient matrices during one-step and two-step updates under both the two-layer and three-layer NN settings. We take two-layer NN case as a main example. The $t$-step update equations for the two-layer NN are as follows:

$$\boldsymbol{W}_1^t = \boldsymbol{W}_1^{t-1} - \eta_1 \boldsymbol{G}_1^{t-1}; \quad \boldsymbol{W}_2^t = \boldsymbol{W}_2^{t-1} - \eta_2 \boldsymbol{G}_2^{t-1}, \quad (7)$$

where $\boldsymbol{W}_1^t, \boldsymbol{W}_2^t$ are two hidden layer weights after $t$-step update, $\eta_1$ and $\eta_2$ are the learning rate for the first layer and second layer, respectively. $\boldsymbol{W}_1^t$ and $\boldsymbol{W}_2^t$ are the updated layer weights. $\boldsymbol{G}_1^{t-1}$ and $\boldsymbol{G}_2^{t-1}$ are the corresponding $t-$step exact gradient matrices, where:

$$\boldsymbol{G}_1^{t-1} = \frac{1}{h^2}\boldsymbol{W}_1^{t-1}\boldsymbol{W}_2^{t-1}\boldsymbol{W}_2^{t-1\top} - \frac{1}{h^2}\boldsymbol{X}^\top \boldsymbol{Y}\boldsymbol{W}_2^{t-1\top},$$

$$\boldsymbol{G}_2^{t-1} = \frac{1}{h^2}\boldsymbol{W}_1^{t-1\top}\boldsymbol{W}_1^{t-1}\boldsymbol{W}_2^{t-1} - \frac{1}{h^2}\boldsymbol{W}_1^{t-1\top}\boldsymbol{X}^\top \boldsymbol{Y}.$$

Since the gradients incorporate label information, we decompose each gradient matrix into two constituent components:

$$\boldsymbol{G}_\ell^t = \boldsymbol{B}_\ell^t - \boldsymbol{A}_\ell^t, \quad \ell \in \{1, 2\}, \quad (8)$$

where the matrices $\boldsymbol{A}_\ell^t$ correspond to data-aligned gradient components, while $\boldsymbol{B}_\ell^t$ capture self-interaction effects

arising from weight Gram matrices. Specifically for $\ell = 1$:

$$A_1^t = \frac{1}{h} M W_2^{t\top}, \quad B_1^t = \frac{1}{h^2} W_1^t W_2^t W_2^{t\top}, \quad (9)$$

and for the second layer $\ell = 2$,

$$A_2^t = \frac{1}{h} W_1^{t\top} M, \quad B_2^t = \frac{1}{h^2} W_1^{t\top} W_1^t W_2^t. \quad (10)$$

The $A_\ell^t$ terms describe how the network weights align with the target matrix $M$ ( equation 3), and thus represent the primary learning signal. The $B_\ell^t$ terms arise from weight-weight interactions and act as an implicit regularization term whose magnitude grows with the norm of the weights.

**One-Step Updates and Gradient Structure.** Using this decomposition, the one-step gradient descent updates can be written as:

$$W_\ell^1 = W_\ell^0 + \eta_\ell A_\ell^0 - \eta_\ell B_\ell^0. \quad (11)$$

Under orthogonal initialization, the norms of $A_\ell^0$ concentrate around deterministic quantities, while $B_\ell^0$ is initially small due to the orthogonality of $W_1^0$ and $W_2^0$. Thus, early-stage learning is dominated by the signal-aligned term $A_\ell^0$.

**Signal-Only Reference Dynamics.** To isolate the contribution of the data-aligned terms, we consider a signal-only trajectory by removing the self-interaction components:

$$\widetilde{W_\ell^1} = W_\ell^0 + \eta_\ell A_\ell^0. \quad (12)$$

The corresponding signal-only gradient components at the next step are:

$$\widetilde{A_1^1} = \frac{1}{h} M \widetilde{W_2^1}^\top, \quad \widetilde{A_2^1} = \frac{1}{h} \widetilde{W_1^1}^\top M. \quad (13)$$

This fictitious trajectory captures pure signal propagation through the network and admits clean norm bounds that are independent of higher-order weight interactions. It serves as a reference point for comparing the true GD dynamics.

**Two-Step Updates and Higher-Order Corrections.** The true two-step updates take the form:

$$W_\ell^2 = W_\ell^1 + \eta_\ell A_\ell^1 - \eta_\ell B_\ell^1. \quad (14)$$

To facilitate comparison with the signal-only trajectory, we define the corrected approximation

$$\overline{W_\ell^2} = \widetilde{W_\ell^1} + \eta_\ell \widetilde{A_\ell^1} - \eta_\ell \widetilde{B_\ell^1}, \quad (15)$$

$$\overline{G_\ell^1} = \widetilde{B_\ell^1} - \widetilde{A_\ell^1}, \quad (16)$$

where $\widetilde{B_\ell^1}$ denotes the self-interaction term evaluated along the signal-only path. This construction allows us to quantify the deviation between $W_\ell^2$ and $\overline{W_\ell^2}$ and to show that the difference is controlled by higher-order terms in $\eta_\ell$.

## 5.2. Learning Rate Regimes and Gradient Dominance

We now formalize the effect of learning rate scaling on the relative magnitude of the signal and self-interaction components of the gradients. By analyzing the norms of the matrices $A_1^0, A_2^0, B_1^0, B_2^0, \widetilde{A_1^1}, \widetilde{A_2^1}, \widetilde{B_1^1}, \widetilde{B_2^1}$, we obtain the following characterization of the gradient structure under random orthogonal initialization.

**Proposition 5.1.** *(Two-layer NN under random orthogonal initialization.) Under Assumption 3.2, if the learning rates satisfy $\eta_1, \eta_2 \leq O(h\sqrt{h})$, then the gradients are well-approximated by their signal-aligned components:*

$$\left\| G_1^0 - A_1^0 \right\| \leq \frac{\left\| G_1^0 \right\|}{\sqrt{h} - 1}, \quad \left\| G_2^0 - A_2^0 \right\| \leq \frac{\left\| G_2^0 \right\|}{\sqrt{h} - 1},$$

$$\left\| \overline{G_1^1} - \widetilde{A_1^1} \right\| \leq \frac{\left\| \overline{G_1^1} \right\|}{\sqrt{h} - 1}, \quad \left\| \overline{G_2^1} - \widetilde{A_2^1} \right\| \leq \frac{\left\| \overline{G_2^1} \right\|}{\sqrt{h} - 1}.$$

Proposition 5.1 shows that, for sufficiently wide networks, the contribution of the self-interaction terms $B_\ell^t$ to the gradient norm is suppressed by a factor of $1/(\sqrt{h} - 1)$. Consequently, both the one-step gradients $G_\ell^0$ and the corrected two-step gradients $\overline{G_\ell^1}$ are dominated by their signal-aligned components $A_\ell^0$ and $\widetilde{A_\ell^1}$, respectively. This justifies approximating the early-stage training dynamics using the signal-only trajectory introduced previously (see complete proof in Appendix C.1.2 and C.2.1). In Figure 3 in Appendix F , we perform spectral analysis of the $A_\ell^0, \widetilde{A_\ell^1}, B_\ell^0, \widetilde{B_\ell^1}$ , $G_\ell^0$ , $\overline{G_\ell^1}$ matrices and visualize the norm gap highlighted in Proposition 5.1, we further verify that $B_\ell^0$ and $\widetilde{B_\ell^1}$ are negligible, as they are dominated by $A_\ell^0$ and $\widetilde{A_\ell^1}$.

**Large Learning Rate Regime.** The proposition also identifies a critical scaling of the learning rate at which self-interaction effects become non-negligible for two-layer neural networks. In particular, when $\eta_1 = \Theta(h\sqrt{h})$, we have:

$$\left\| W_1^1 - W_1^0 \right\|_F \asymp \left\| W_1^0 \right\|_F, \quad (17)$$

$$\left\| \overline{W_1^2} - \widetilde{W_1^1} \right\|_F \asymp \left\| \widetilde{W_1^1} \right\|_F, \quad (18)$$

and analogously for the second layer when $\eta_2 = \Theta(h\sqrt{h})$. This scaling marks a transition point where a single gradient step produces a weight update comparable in magnitude to the existing weights. Proposition 5.1 shows that when the learning rates satisfy $\eta_1, \eta_2 \leq O(h\sqrt{h})$, the signal-aligned components dominate the gradients from a norm perspective. Specifically, the sets $\{A_i^0\}_{i=1}^2$ and $\{\widetilde{A_i^1}\}_{i=1}^2$ are very close to $\{G_i^0\}_{i=1}^2$ and $\{\overline{G_i^1}\}_{i=1}^2$, respectively. This implies that $\{A_i^0\}_{i=1}^2$ and $\{\widetilde{A_i^1}\}_{i=1}^2$ serve as the leading terms in the one-step gradients $\{G_i^0\}_{i=1}^2$ and the corrected two-step gradients $\{\overline{G_i^1}\}_{i=1}^2$. This approximation substantially simplifies the

subsequent analysis of the theoretical test loss for the two-layer neural network. In particular, it allows us to replace the exact gradients $\{G_i^0\}_{i=1}^2$ and $\{\overline{G_i^1}\}_{i=1}^2$ with their leading-order counterparts $\{A_i^0\}_{i=1}^2$ and $\{\widetilde{A_i^1}\}_{i=1}^2$, respectively, as formally justified by Lemma 5.2. For the three-layer setting, we similarly replace the original gradients by their leading terms, as justified by Proposition C.4 and C.11, in order to streamline the test loss analysis.

Finally, Proposition 5.1 also identifies a critical learning-rate scaling. When the learning rates for the first and second layers are set to be on the order of $\Theta(h\sqrt{h})$, the gradient updates become comparable in magnitude to the initialized weight matrices, effectively overwhelming the initialization. This behavior mirrors the learning-rate scaling associated with the maximal update parameterization studied in Yang & Hu (2021); Yang et al. (2022). Prior work (Ba et al., 2022; Yang & Hu, 2021; Yang et al., 2022; 2023) suggests that choosing learning rates within (or below) this large-learning-rate regime can be beneficial for training. While these studies primarily focus on two-layer networks with a fixed output layer, our analysis reaches a compatible conclusion without relying on this assumption. A similar result can be obtained for the three-layer NN setting under random orthogonal initialization with $\eta_1, \eta_2$ no more than $O(h)$ (See Proposition C.4 and C.11). We provide the proof in Appendix C.1.1 and C.2.2.

### 5.3. Relationship between Test Loss and Layer-wise Learning Rates

This section characterizes how the test loss depends on the learning rates of individual layers in the linear NNs trained under random orthogonal initialization.

#### 5.3.1. TWO-LAYER NEURAL NETWORKS

Given test data $\tilde{x}_0 \sim \mathcal{N}(\mathbf{0}, I_h)$, we consider the test loss

$$L_{\text{two-layer}} = \mathbb{E}_{W_1, W_2, M, \tilde{x}_0, X} \left\| \frac{1}{h}\tilde{x}_0 W_1 W_2 - \tilde{x}_0 M \right\|^2.$$

As the exact closed-form characterization of the test loss is nontrivial, our analysis proceeds by first simplifying the training dynamics using leading-order gradient approximations, and then translating these simplified dynamics into explicit expressions for the test loss.

**Lemma 5.2.** *Under Assumption 3.2 and 3.3, for $\eta_1, \eta_2 \leq O(h\sqrt{h})$, we have*

$$\left| L_{\text{two-layer}}(W_1^1, W_2^1) - L_{\text{two-layer}}(\widetilde{W_1^1}, \widetilde{W_2^1}) \right| \leq O(h^{-1}).$$

$$\left| L_{\text{two-layer}}(W_1^2, W_2^2) - L_{\text{two-layer}}(\widetilde{W_1^2}, \widetilde{W_2^2}) \right| \leq O(h^{-\frac{1}{2}}).$$

Lemma 5.2 formalizes the idea that, for sufficiently large

width and moderate learning rates, the test loss is insensitive to higher-order gradient corrections. Specifically, when $\eta_1, \eta_2 \leq O(h\sqrt{h})$, replacing the true gradient with the signal-aligned approximations changes the test loss by at most $O(h^{-1})$ after one step and $O(h^{-1/2})$ after two steps.

Intuitively, this result builds on Proposition 5.1: since the signal components dominate the gradient norms, the parts of the update omitted in the approximation contribute only lower-order perturbations to the weights. As a consequence, the network's input–output map after one or two steps is well-approximated by the signal-only dynamics, and the resulting test loss remains essentially unchanged at leading order. This lemma is crucial because it allows us to analyze the test loss using simplified weight trajectories that admit closed-form expressions, without sacrificing asymptotic accuracy. The simplified analysis leads to the following result for two-layer networks.

**Theorem 5.3.** *Given Assumption 3.2, 3.3 and in addition assume $\eta_1$ and $\eta_2$ are no more than $O(h\sqrt{h})$, based on Proposition 5.1 and Lemma 5.2, consider the training procedure discussed in Section 3, we obtain the following test loss after one-step and two-step GD update in a two-layer neural network under random orthogonal initialization:*

$$L_{\text{two-layer}}(W_1^1, W_2^1) = \frac{\eta_1^2}{h^4} + \frac{\eta_2^2}{h^4} + \frac{2\eta_1\eta_2}{h^4} + \frac{\eta_1^2\eta_2^2}{h^7}$$
$$- \frac{2\eta_1}{h^2} - \frac{2\eta_2}{h^2} + \frac{1}{h} + \frac{2\eta_1\eta_2}{h^5} + 1$$
$$L_{\text{two-layer}}(W_1^2, W_2^2) = \frac{1}{h}(1 + \frac{\eta_1\eta_2}{h^3})^4 + \frac{16\eta_1^2\eta_2^2}{h^7}$$
$$+ \left( \frac{2(\eta_1 + \eta_2)(\eta_1\eta_2 + h^3)}{h^5} - 1 \right)^2 + (1 + \frac{\eta_1\eta_2}{h^3})^2 \frac{8\eta_1\eta_2}{h^5}$$

We provide the proof in Appendix D.1 and D.2. Theorem 5.3 provides explicit formulas for the test loss after one-step and two-step gradient descent updates in a two-layer network. While the expressions themselves are algebraically involved, their structure reveals several key phenomena.

**One-step test loss.** The one-step test loss decomposes into three types of terms: (i) Linear improvement terms (e.g., $-2\eta_1/h^2$, $-2\eta_2/h^2$), which reflect the reduction in error due to alignment with the target signal. (ii) Quadratic and interaction terms (e.g., $\eta_1^2/h^4$, $\eta_2^2/h^4$, $\eta_1\eta_2/h^4$), which capture over-updating and cross-layer coupling. (iii) Residual variance terms (e.g., $1/h + 1$), arising from the randomness of initialization and test inputs. These components make explicit how learning rates at different layers contribute asymmetrically and interactively to NN generalization.

**Two-step test loss.** For the two-step update, the test loss exhibits higher-order dependence on the product $\eta_1\eta_2$. This reflects the fact that meaningful improvement in a two-layer

linear network requires coordination between layers; updating only one layer is insufficient to substantially reduce the prediction error. The appearance of repeated factors of $(1 + \eta_1\eta_2/h^3)$ highlights the multiplicative nature of representation learning across layers. Building on this insight, we obtain the following corollary:

**Corollary 5.4.** *Suppose* $\eta_1 + \eta_2 = 2h^\alpha$ *and we consider* $0 < \alpha \leq \frac{3}{2}$. *Then, for any* $\alpha$ *in this range, the point* $\eta_1 = \eta_2 = h^\alpha$ *is not a local minimum of the loss* $L_{two\text{-}layer}(\boldsymbol{W}_1^1, \boldsymbol{W}_2^1)$. *However, for* $1 < \alpha \leq \frac{3}{2}$, *if* $h > \max\{h^*, 256\}$, *then* $\eta_1 = \eta_2 = h^\alpha$ *is a local minimum of the loss* $L_{two\text{-}layer}(\boldsymbol{W}_1^2, \boldsymbol{W}_2^2)$, *where* $h^*$ *is the root of the following equation:*

$$(1+o(1))h^{1-\alpha}+16h^{\alpha-2}+2h^{-\alpha}+8h^{\alpha-3}+6h^{3\alpha-6}-2 = 0$$

Corollary 5.4 studies the test loss landscape under a constrained learning-rate budget $\eta_1 + \eta_2 = 2h^\alpha$ (Assumption 3.4), which isolates the effect of how learning rates are allocated across layers, rather than their total magnitude.

**Asymmetric learning rates after one-step update.** The first conclusion is that the symmetric choice $\eta_1 = \eta_2$ is not a local minimum of the test loss after a single gradient descent step for any $0 < \alpha \leq \frac{3}{2}$. This result indicates that, in the initial stage of training, the test loss is optimized by an asymmetric allocation of learning rates across layers. After one update, the two layers contribute differently to the predictor: updates to the first layer primarily control the formation of internal representations, whereas updates to the second layer mainly affect the linear readout of these representations. Imposing equal learning rates at this stage restricts the network from exploiting this structural asymmetry, resulting in suboptimal test performance.

**Symmetric learning rates after two-step update.** In contrast, the second conclusion shows that for $1 < \alpha \leq \frac{3}{2}$ and sufficiently large network width, the symmetric choice $\eta_1 = \eta_2$ becomes a local minimum of the test loss after two gradient descent steps. This behavior reflects a transition in the training dynamics: after multiple updates, the learning process becomes increasingly coupled across layers, and coordinated updates yield improved generalization. In this regime, balanced learning rates facilitate effective interaction between layers, leading to optimal performance. The lower bound on $\alpha$ ensures that the learning rates are sufficiently large to induce non-negligible cross-layer effects, while remaining within a stable training regime.

Overall, this corollary identifies a phase transition in the optimal allocation of layer-wise learning rates, governed by the interaction between network width, training depth, and the overall scale of the learning rates.

### 5.3.2. THREE-LAYER NEURAL NETWORKS

We now characterize the test loss of a three-layer neural network after one-step and two-step gradient descent updates under random orthogonal initialization. Compared to the two-layer setting, the presence of a vector-valued output layer $\boldsymbol{a}$ fundamentally alters both the learning-rate scaling and the structure of the resulting test loss. Given test data $\tilde{\boldsymbol{x}}_0 \sim \mathcal{N}(\boldsymbol{0}, \boldsymbol{I}_h)$, we consider the test loss:

$$L_{\text{three-layer}} = \mathbb{E}_{\substack{\boldsymbol{W}_1^0, \boldsymbol{W}_2^0, \boldsymbol{a}, \\ \boldsymbol{\beta}^*, \tilde{\boldsymbol{x}}_0, \boldsymbol{X}}} \left( \frac{1}{\sqrt{h}} \, \tilde{\boldsymbol{x}}_0 \boldsymbol{W}_1 \boldsymbol{W}_2 \boldsymbol{a} - \tilde{\boldsymbol{x}}_0 \boldsymbol{\beta}^* \right)^2$$

**Structural distinction from the two-layer network.** In the three-layer network, the predictor takes the form $\tilde{\boldsymbol{x}}_0 \boldsymbol{W}_1 \boldsymbol{W}_2 \boldsymbol{a}$, where the output weights $\boldsymbol{a} \in \mathbb{R}^h$ are fixed throughout training. Consequently, learning in the hidden layers affects the test loss only through their joint alignment with the target vector $\boldsymbol{\beta}^*$. This additional linear mapping at the output introduces a bottleneck that attenuates the propagation of gradient updates, thereby reducing the scale at which layer-wise interactions become significant. As a result, the admissible learning-rate regime in the three-layer setting is $\eta_1, \eta_2 \leq O(h)$, which is strictly smaller than the $O(h\sqrt{h})$ regime identified for the two-layer network.

**Theorem 5.5.** *Given Assumption 3.2, 3.3 and in addition assume* $\eta_1$ *and* $\eta_2$ *are no more than* $O(h)$ *based on Proposition C.4 and C.11, consider the training procedure discussed in Section 3, we derive the test loss after one-step and two-step GD update in a three-layer neural network:*

$$L_{three\text{-}layer}(\boldsymbol{W}_1^1, \boldsymbol{W}_2^1) = \frac{\eta_1^2}{h^2} + \frac{\eta_2^2}{h^2} + \frac{2\eta_1\eta_2}{h^2} + \frac{\eta_1^2\eta_2^2}{h^4}$$
$$- \frac{2\eta_1}{h} - \frac{2\eta_2}{h} + \frac{1}{h} + \frac{2\eta_1\eta_2}{h^3} + 1$$
$$L_{three\text{-}layer}(\boldsymbol{W}_1^2, \boldsymbol{W}_2^2) = \left( \frac{2(\eta_1 + \eta_2)(h + \eta_1\eta_2)}{h^2} - 1 \right)^2$$
$$+ \frac{1}{h} + \frac{2\eta_1\eta_2}{h^2} + \frac{10\eta_1\eta_2}{h^3} + \frac{\eta_1^2\eta_2^2}{h^3}$$
$$+ \frac{37\eta_1^2\eta_2^2}{h^4} + \frac{12\eta_1^3\eta_2^3}{h^5} + \frac{\eta_1^4\eta_2^4}{h^6}$$

The first expression in Theorem 5.5 gives the test loss after a single gradient descent step. We provide the proof in Appendix D.4. Its structure mirrors that of the two-layer case, with appropriately rescaled terms. The two-step test loss exhibits a substantially richer dependence on the learning rates, involving higher-order polynomial terms in the product $\eta_1\eta_2$. We provide the proof in Appendix D.5. This behavior reflects the fact that, in a three-layer network, a meaningful reduction in test loss requires coordinated updates across both hidden layers over multiple steps. Similar to Corollary 5.4, we have the following corollary for a three-layer neural network.

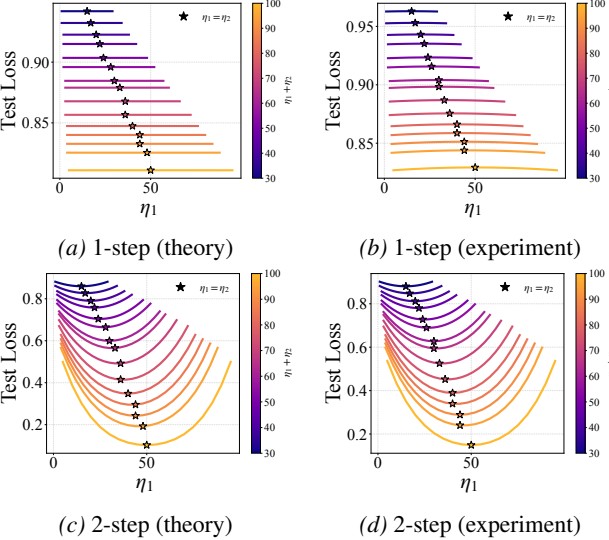

*(a)* 1-step (theory)  *(b)* 1-step (experiment)

*(c)* 2-step (theory)  *(d)* 2-step (experiment)

*Figure 2.* **3-layer NN under orthogonal initialization.** Here we set $\eta_1 + \eta_2 \leq O(h^{\frac{2}{3}})$ and $h = 1000$. We observe conclusions and results similar to those in Figure 1. We discuss more in Section 6.

**Corollary 5.6.** *Suppose $\eta_1 + \eta_2 = 2h^\alpha$ and we consider $0 < \alpha < 1$. Then, for any $\alpha$ in this range, the point $\eta_1 = \eta_2 = h^\alpha$ is not a local minimum of the loss $L_{three\text{-}layer}(\boldsymbol{W}_1^1, \boldsymbol{W}_2^1)$. However, for $0 < \alpha \leq \frac{2}{3}$, if $h > h^*$, then $\eta_1 = \eta_2 = h^\alpha$ is a local minimum of the loss $L_{three\text{-}layer}(\boldsymbol{W}_1^2, \boldsymbol{W}_2^2)$, where $h^*$ is the root of the following equation:*

$$32h^{3\alpha-2} + 33h^{\alpha-1} + 74h^{\alpha-2} + 2h^{-\alpha}$$
$$+10h^{-\alpha-1} + 36h^{3\alpha-3} + 4h^{5\alpha-4} - 8 = 0$$

**Comparison with the two-layer case.** While both two-layer and three-layer networks exhibit nontrivial dependence on layer-wise learning rates, the three-layer setting differs in two key aspects. First, the critical learning-rate scale is reduced from $O(h\sqrt{h})$ to $O(h)$ due to differences in the initialization schemes as well as the presence of the fixed output vector $\boldsymbol{a}$. Second, the two-step test loss has a stronger dependence on higher-order products of $\eta_1$ and $\eta_2$, reflecting enhanced cross-layer coupling. Overall, Theorem 5.5 demonstrates that, in three-layer networks with a vector-valued output layer, the test loss is governed by a delicate interaction between layer-wise learning rates, training depth, and network width. Although early-stage updates admit a decomposition similar to that of the two-layer case, deeper architectures amplify cross-layer interactions over successive steps, leading to a distinct learning-rate scaling regime and a richer dependence of test loss on the learning rates.

## 6. Experiments

**Orthogonal initialization.** Here we numerically validate our theoretical results under orthogonal initialization. We set $h = n = d = 1000$ and keep the model and data ini-

tialization the same as in Section 3. In Figure 1, 2 and 4, we present both theoretical simulations and empirical experiments for two- and three-layer networks, comparing the one-step and two-step test loss as a function of the learning rates across a range of $\eta_1 + \eta_2$ values below the critical threshold. Across all settings, the theoretical losses we derive closely match the observed test losses after either one or two updates. Moreover, when $\eta_1 + \eta_2 = 2h^\alpha$ lies below the critical threshold, we can see a clear qualitative shift: after a single update, symmetric learning rates across layers are suboptimal, whereas after two updates they become locally optimal in sufficiently wide networks. This reveals a transition from asymmetric to balanced layer-wise learning-rate allocation. For different $h$, see Figure 11 and 12 in Appendix F). We also find that balanced layer-wise learning-rate allocation can be locally optimal during early training over multiple steps (up to 512 update steps) under orthogonal initialization (see Figure 5, 7 and 9 in Appendix F).

**Gaussian initialization.** To demonstrate the generality of our results, we repeat the same set of experiments under gaussian initialization. We observe the same behavior as in the orthogonal case: after the first update, symmetric layer-wise learning rates are suboptimal, whereas after two updates they become locally optimal in sufficiently wide networks. This phenomenon can also further extend to multiple training steps, please see Figure 6 and 8 in Appendix F.

**Linear NN under noisy condition.** In Figure 10 in Appendix F, we consider adding label noise $\boldsymbol{\xi} \sim \mathcal{N}(0, \rho)$ to the teacher model in both two layer and three layer linear neural networks under orthogonal initialization. We still observe that after the first update, symmetric layer-wise learning rates are suboptimal, whereas after two updates they become locally optimal.

**Deep Linear NNs.** In Figure 14, we consider 4-layer and 8-layer linear neural networks, which generalize the two layer and three layer settings. We observe that, for one and two update steps, the same transition from asymmetry to balance still appears.

**Nonlinear NNs.** Here we consider a three layer nonlinear neural network, where the student model is $f(\boldsymbol{x}_i) = \frac{1}{\sqrt{h}}\sigma(\sigma(\boldsymbol{x}_i^\top \boldsymbol{W}_1)\boldsymbol{W}_2)\boldsymbol{a}$, and the teacher model is $y_i = \sigma(\boldsymbol{\beta}^{*\top}\boldsymbol{x}_i)$, with $\sigma$ being the ReLU activation. We use the same orthogonal initialization and training pipeline as in the two layer and three layer settings. In Figure 15, we visualize the test loss as a function of $\eta_1$ after 1 and 8-step updates $\eta_1 + \eta_2 < O(\sqrt{h})$. Although the curves are relatively less symmetric than in linear case, we still observe a similar asymmetry-to-balance transition, which generalizes the cases and results covered by our theoretical setup.

# 7. Discussion

**Learning rate scheduler Design.** In previous sections, we revealed the asymmetry to balance transition in layer-wise learning rate allocation and offered theoretical support for layer-wise learning rate schedulers that aim to promote layer balance at later stages of training. Here, we provide a simple example to guide their practical design. Consider a teacher model $\boldsymbol{y}_i = \boldsymbol{M}^\top \boldsymbol{x}_i$ and a student model $f(\boldsymbol{x}_i) = \boldsymbol{x}_i \boldsymbol{W}_1 \boldsymbol{W}_2$, where $\boldsymbol{x}_i$ is the input and $\boldsymbol{W}_1, \boldsymbol{W}_2$ are the two trainable matrices. Since the Frobenius norm is a classic generalization metric, we can leverage it for $\boldsymbol{W}_1$ and $\boldsymbol{W}_2$ to design a learning rate scheduler.

First, we expect the layer with the larger Frobenius norm to be assigned a smaller learning rate, due to the property of the metric. More importantly, based on the theoretical insights in our paper, we expect the learning rates of the two layers to become balanced in the later stages of training, which motivates us to promote balance between the layer norms. As a result, at each step $t$, we set the learning rates for $\boldsymbol{W}_1{}^t$ and $\boldsymbol{W}_2{}^t$ as $\eta_{\boldsymbol{W}_1}^{(t)} = \frac{2\|\boldsymbol{W}_2{}^t\|_F}{\|\boldsymbol{W}_1{}^t\|_F + \|\boldsymbol{W}_2{}^t\|_F}\mathrm{lr}, \qquad \eta_{\boldsymbol{W}_2}^{(t)} = \frac{2\|\boldsymbol{W}_1{}^t\|_F}{\|\boldsymbol{W}_1{}^t\|_F + \|\boldsymbol{W}_2{}^t\|_F}\mathrm{lr}$, where $\mathrm{lr}$ is a uniform base learning rate. As training enters the later stage, this balance-driven learning rate scheduler promotes $\big| \|\boldsymbol{W}_1\|_F - \|\boldsymbol{W}_2\|_F \big| \to 0$, and the learning rates also become balanced. It is worth noting that for this matrix-factorization type linear network, the curvature at convergence, measured by the largest Hessian eigenvalue, is related to $\big| \|\boldsymbol{W}_1\|_F - \|\boldsymbol{W}_2\|_F \big|$; in particular, smaller norm gap corresponds to a flatter solution (Wang et al., 2021). Therefore, the transition of the learning rates from asymmetry to balance also corresponds to the process by which the model gradually converges to a flatter minima.

In Figure 17, we consider a setup where $\|\boldsymbol{W}_2\|_F = 6$ and $\|\boldsymbol{W}_1\|_F = 1$ at initialization, and compare this design with a uniform learning rate used throughout training. We find that this layer-wise schedule captures the asymmetry-to-balance transition observed in our paper, and achieves lower training loss and test loss than the fully uniform baseline. More specifically, we observe that $\big| \|\boldsymbol{W}_1\|_F - \|\boldsymbol{W}_2\|_F \big|$ approaches zero in the middle and late stages of training, which corresponds to increasingly balanced learning rates.

**Step-dependent optimality of learning-rate symmetry.** Our results indicate that symmetric learning rates are suboptimal for the first step but optimal for two steps. This may not be the same as using asymmetric learning rates early and balancing them later during training. This suggests that asymmetric learning rates may be preferable at the very beginning of training, and symmetric learning rates become optimal as cross-layer interactions develop over subsequent steps, even if the initial learning-rate allocation is not optimal. We believe this also points to a practical strategy that use asymmetric learning rates early in training and

more symmetric ones later. we further clarify this question through examples involving a three-layer linear network and a CNN; please see Appendix F.

# 8. Conclusion

In this work, we provide a finite-step characterization of how layer-wise learning rates should be balanced during training in linear neural networks. By analyzing gradient descent dynamics after one and two updates, we show that symmetric learning rates across layers are generally suboptimal at initialization, with early optimization favoring asymmetric allocations that reflect the distinct roles of different layers. As training progresses, a transition occurs where sufficiently large width and appropriate scaling of the total learning rate cause balanced learning rates to be locally optimal. Thus, signaling the emergence of coordinated layer-wise updates. This transition is architecture-dependent, with deeper networks exhibiting stricter conditions under which symmetry is optimal. Our results formalize balancing learning rates across layers as a dynamical phenomenon driven by optimization and scaling, rather than a static design choice, and elucidate how depth, width, and learning-rate magnitude jointly shape this behavior.

## Acknowledgments

This work is supported by the DARPA AIQ program, the U.S. Department of Energy under Award Number DE-SC0025584, the Allocation Year 2026 DOE Mission Science Award, Dartmouth College, and Lambda AI.

## Impact Statement

This paper presents research aimed at advancing machine learning theory, particularly by providing an exact, finite-step characterization of how layer-wise learning-rate choices shape early training dynamics and generalization in multi-layer linear neural networks. Our analysis reveals a transition from initially asymmetric optimal learning rates to later balanced rates. While this work may have various potential societal implications, we do not find it necessary to highlight any specific ones here.

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

# Appendix

## A. Limitations

Our analysis is restricted to linear networks and focuses on the first few steps of gradient descent. However, unlike prior work that mainly focuses on asymptotic convergence, gradient flow, or kernel-based analyses, our work derives exact closed-form expressions for the gradients and test loss after one and two GD steps, enabling a precise characterization of early training dynamics. In particular, we directly link finite-step layer-wise learning-rate allocation to test loss. This is already nontrivial even for two-layer and three-layer linear networks: at each step, we decompose the gradient into dominant signal-aligned components and smaller residual terms, and rigorously bound the residual terms in operator norm, establishing conditions under which the approximate gradients accurately characterize the test loss dynamics. Extending these theoretical results to nonlinear activations, stochastic optimization, and adaptive learning-rate methods presents a natural and challenging direction. More broadly, understanding how and when balancing learning rates across layers emerges in realistic deep networks may provide new theoretical guidance for optimization strategies beyond the linearized or asymptotic regimes.

## B. More Related Work on Layer-wise Hyperparameter Tuning

Besides layer-wise learning rate tuning, strategies for assigning different layer-wise pruning ratios, for both unstructured and structured pruning, have been studied actively. ABCPruner (Lin et al., 2020) first proposes layer-wise pruning strategies from a heuristic way and try to reduce the search space of possible layer sparsity combinations. Lee et al. (2020) modify magnitude-based pruning by rescaling the importance scores in a layer by a factor dependent on the magnitude of surviving connections in that layer. Yin et al. (2023) allocate layer-wise pruning ratio proportional to the outlier ratio observed within each layer, thereby facilitating a more effective alignment between layer-wise weight sparsity and outlier ratios. Alphapruning (Lu et al., 2024) assigns layer-wise pruning ratios based on the heavy-tailness across layers in large language models (LLMs), undertrained layers will be pruned more to ensure the post-pruning performance does not degrade aggressively across all layers, thereby achieving layer balancing.

Furthermore, Ishii & Sato (2017); Nakamura & Hong (2019); He et al. (2025) introduce layer-wise weight decay schedulers and works such as HAWQ (Dong et al., 2019) and Q-BERT (Shen et al., 2020) propose hessian-aware layer-wise quantization approaches. Qing et al. (2024); Gao et al. (2025) allocate layer-wise experts for MoE layers in transformers based on certain representation metrics.

## C. Norm Analysis of Update Gradient Matrices

**Lemma C.1.** *(Hanson-Wright Inequality (Vershynin, 2018)). Let $\boldsymbol{x} = (x_1, \cdots, x_n) \in \mathbb{R}^n$ be a random vector with independent, mean zero, sub-gaussian coordinates. Let $\boldsymbol{A}$ be an $n \times n$ matrix. Then for every $t \geq 0$, we have*

$$\mathbb{P}\left\{\left|\boldsymbol{x}^\top \boldsymbol{A} \boldsymbol{x} - \mathbb{E}\boldsymbol{x}^\top \boldsymbol{A} \boldsymbol{x}\right| \geq t\right\} \leq 2\exp\left[-c\min(\frac{t^2}{K^4\|A\|_F^2}, \frac{t}{K^2\|A\|})\right],$$

*where $K = \max_i \|x_i\|_{\psi_2}$.*

**Lemma C.2.** *(Concentration of Lipschitz function on the sphere (Vershynin, 2018)). Consider a random vector $\boldsymbol{x} \in \sqrt{n}\mathbb{S}^{n-1}$. Given a Lipschitz function $f : \sqrt{n}\mathbb{S}^{n-1} \to \mathbb{R}$. Then*

$$\mathbb{P}\left\{|f(\boldsymbol{x}) - \mathbb{E}f(\boldsymbol{x})| \geq t\right\} \leq 2\exp(-\frac{ct^2}{\|f\|_{Lip}^2}).$$

**Lemma C.3.** *(Bernstein Inequality (Vershynin, 2018)). Let $x_1, \cdots, x_n$ be independent, mean zero, sub-exponential random variables. Then, for every $t \geq 0$, we have*

$$\mathbb{P}\left\{\left|\sum_{i=1}^n x_i\right| \geq t\right\} \leq 2\exp\left[-c\min(\frac{t^2}{\sum_{i=1}^n \|x_i\|_{\psi_1}^2}, \frac{t}{\max_i \|x_i\|_{\psi_1}})\right]$$

*where $c > 0$ is an absolute constant.*

## C.1. Norm Analysis of One-step Update Gradient Matrices

### C.1.1. THREE-LAYER NEURAL NETWORK UNDER ORTHOGONAL INITIALIZATION

**Proposition C.4.** *(Three-layer NN setting under Orthogonal initialization.) Under Assumption 3.2, we have gradient approximation,*

$$
\left\|\boldsymbol{G}_1^0 - \boldsymbol{A}_1^0\right\| \leq \frac{1}{\sqrt{h}-1}\left\|\boldsymbol{G}_1^0\right\|,
$$
$$
\left\|\boldsymbol{G}_2^0 - \boldsymbol{A}_2^0\right\| \leq \frac{1}{\sqrt{h}-1}\left\|\boldsymbol{G}_2^0\right\|. \tag{19}
$$

*And we have*

$$
Small\ lr: \eta_1 = \Theta(\sqrt{h}) \Rightarrow \left\|\boldsymbol{W}_1^1 - \boldsymbol{W}_1^0\right\| \asymp \left\|\boldsymbol{W}_1^0\right\| \tag{20}
$$
$$
\eta_2 = \Theta(\sqrt{h}) \Rightarrow \left\|\boldsymbol{W}_2^1 - \boldsymbol{W}_2^0\right\| \asymp \left\|\boldsymbol{W}_2^0\right\| \tag{21}
$$
$$
Large\ lr: \eta_1 = \Theta(h) \Rightarrow \left\|\boldsymbol{W}_1^1 - \boldsymbol{W}_1^0\right\|_F \asymp \left\|\boldsymbol{W}_1^0\right\|_F \tag{22}
$$
$$
\eta_2 = \Theta(h) \Rightarrow \left\|\boldsymbol{W}_2^1 - \boldsymbol{W}_2^0\right\|_F \asymp \left\|\boldsymbol{W}_2^0\right\|_F \tag{23}
$$

**Proof of Proposition C.4.** Note that $\boldsymbol{A}_1^0 = \frac{1}{\sqrt{h}}\boldsymbol{\beta}^* \boldsymbol{a}^\top \boldsymbol{W}_2^{0^\top}$, $\boldsymbol{A}_2^0 = \frac{1}{\sqrt{h}}\boldsymbol{W}_1^{0^\top}\boldsymbol{\beta}^* \boldsymbol{a}^\top$ are both rank-1 matrices. Here we consider the orthogonal initialization where we have $\boldsymbol{X}^\top \boldsymbol{X} = \boldsymbol{X}\boldsymbol{X}^\top = h\boldsymbol{I}, \boldsymbol{W}_1^{0^\top}\boldsymbol{W}_1^0 = \boldsymbol{W}_1^0\boldsymbol{W}_1^{0^\top} = \boldsymbol{I}, \boldsymbol{W}_2^{0^\top}\boldsymbol{W}_2^0 = \boldsymbol{W}_2^0\boldsymbol{W}_2^{0^\top} = \boldsymbol{I}, \boldsymbol{a}^\top \boldsymbol{a} = 1, \mathbb{E}[\boldsymbol{a}\boldsymbol{a}^\top] = \frac{1}{h}\boldsymbol{I}, \boldsymbol{\beta}^{*^\top}\boldsymbol{\beta}^* = 1, \mathbb{E}[\boldsymbol{\beta}^*\boldsymbol{\beta}^{*^\top}] = \frac{1}{h}\boldsymbol{I}$. Based on this,

$$
\left\|\boldsymbol{A}_1^0\right\| = \left\|\boldsymbol{A}_1^0\right\|_F = \sqrt{tr(\boldsymbol{A}_1^{0^\top}\boldsymbol{A}_1^0)} = \frac{1}{\sqrt{h}} \tag{24}
$$
$$
\left\|\boldsymbol{A}_2^0\right\| = \left\|\boldsymbol{A}_2^0\right\|_F = \sqrt{tr(\boldsymbol{A}_2^{0^\top}\boldsymbol{A}_2^0)} = \frac{1}{\sqrt{h}}. \tag{25}
$$

We also have $\boldsymbol{B}_1^0 = \frac{1}{h}\boldsymbol{W}_1^0\boldsymbol{W}_2^0\boldsymbol{a}\boldsymbol{a}^\top \boldsymbol{W}_2^{0^\top}$, $\boldsymbol{B}_2^0 = \frac{1}{h}\boldsymbol{W}_1^{0^\top}\boldsymbol{W}_1^0\boldsymbol{W}_2^0\boldsymbol{a}\boldsymbol{a}^\top$ are both rank-1 matrices. We have

$$
\left\|\boldsymbol{B}_1^0\right\| = \left\|\boldsymbol{B}_1^0\right\|_F = \sqrt{tr(\boldsymbol{B}_1^{0^\top}\boldsymbol{B}_1^0)} = \frac{1}{h} \tag{26}
$$
$$
\left\|\boldsymbol{B}_2^0\right\| = \left\|\boldsymbol{B}_2^0\right\|_F = \sqrt{tr(\boldsymbol{B}_2^{0^\top}\boldsymbol{B}_2^0)} = \frac{1}{h}. \tag{27}
$$

Since $\boldsymbol{G}_1^0 = \boldsymbol{B}_1^0 - \boldsymbol{A}_1^0$, $\boldsymbol{G}_2^0 = \boldsymbol{B}_2^0 - \boldsymbol{A}_2^0$, we obtain that

$$
\left\|\boldsymbol{G}_1^0 - \boldsymbol{A}_1^0\right\| \leq \frac{1}{\sqrt{h}}\left\|\boldsymbol{A}_1^0\right\| \leq \frac{1}{\sqrt{h}}(\left\|\boldsymbol{G}_1^0\right\| + \left\|\boldsymbol{G}_1^0 - \boldsymbol{A}_1^0\right\|)
$$
$$
\left\|\boldsymbol{G}_2^0 - \boldsymbol{A}_2^0\right\| \leq \frac{1}{\sqrt{h}}\left\|\boldsymbol{A}_2^0\right\| \leq \frac{1}{\sqrt{h}}(\left\|\boldsymbol{G}_1^0\right\| + \left\|\boldsymbol{G}_2^0 - \boldsymbol{A}_2^0\right\|).
$$

Thus, we get that

$$
\left\|\boldsymbol{G}_1^0 - \boldsymbol{A}_1^0\right\| \leq \frac{1}{\sqrt{h}-1}\left\|\boldsymbol{G}_1^0\right\|,
$$
$$
\left\|\boldsymbol{G}_2^0 - \boldsymbol{A}_2^0\right\| \leq \frac{1}{\sqrt{h}-1}\left\|\boldsymbol{G}_2^0\right\|. \tag{28}
$$

Based on this, we can get $\sqrt{h}\left\|\boldsymbol{G}_1^0\right\| = \Theta_{h,\mathbb{P}}(1), \sqrt{h}\left\|\boldsymbol{G}_1^0\right\|_F = \Theta_{h,\mathbb{P}}(1), \sqrt{h}\left\|\boldsymbol{G}_2^0\right\| = \Theta_{h,\mathbb{P}}(1), \sqrt{h}\left\|\boldsymbol{G}_2^0\right\|_F = \Theta_{h,\mathbb{P}}(1)$.

Since we have $\left\|\boldsymbol{W}_1^0\right\| = 1, \left\|\boldsymbol{W}_1^0\right\|_F = \sqrt{h}, \left\|\boldsymbol{W}_2^0\right\| = 1, \left\|\boldsymbol{W}_2^0\right\|_F = \sqrt{h}$, based on Assumption E.1, we have

$$
Small\ lr: \eta_1 = \Theta(\sqrt{h}) \Rightarrow \left\|\boldsymbol{W}_1^1 - \boldsymbol{W}_1^0\right\| \asymp \left\|\boldsymbol{W}_1^0\right\|
$$
$$
\eta_2 = \Theta(\sqrt{h}) \Rightarrow \left\|\boldsymbol{W}_2^1 - \boldsymbol{W}_2^0\right\| \asymp \left\|\boldsymbol{W}_2^0\right\|
$$
$$
Large\ lr: \eta_1 = \Theta(h) \Rightarrow \left\|\boldsymbol{W}_1^1 - \boldsymbol{W}_1^0\right\|_F \asymp \left\|\boldsymbol{W}_1^0\right\|_F
$$
$$
\eta_2 = \Theta(h) \Rightarrow \left\|\boldsymbol{W}_2^1 - \boldsymbol{W}_2^0\right\|_F \asymp \left\|\boldsymbol{W}_2^0\right\|_F
$$

$\square$

### C.1.2. TWO-LAYER NEURAL NETWORK UNDER ORTHOGONAL INITIALIZATION

**Proposition C.5.** *(Two-layer NN setting under Orthogonal initialization.) Under Assumption 3.2, we have gradient approximation,*

$$\left\|\boldsymbol{G}_1^0 - \boldsymbol{A}_1^0\right\| \leq \frac{1}{\sqrt{h}-1}\left\|\boldsymbol{G}_1^0\right\|,$$
$$\left\|\boldsymbol{G}_2^0 - \boldsymbol{A}_2^0\right\| \leq \frac{1}{\sqrt{h}-1}\left\|\boldsymbol{G}_2^0\right\|. \tag{29}$$

*And we have*

$$Large\ lr : \eta_1 = \Theta(h\sqrt{h}) \Rightarrow \left\|\boldsymbol{W}_1^1 - \boldsymbol{W}_1^0\right\|_F \asymp \left\|\boldsymbol{W}_1^0\right\|_F \tag{30}$$
$$\eta_2 = \Theta(h\sqrt{h}) \Rightarrow \left\|\boldsymbol{W}_2^1 - \boldsymbol{W}_2^0\right\|_F \asymp \left\|\boldsymbol{W}_2^0\right\|_F \tag{31}$$

**Proof of Proposition C.5.** Note that $\boldsymbol{A}_1^0 = \frac{1}{h}\boldsymbol{M}\boldsymbol{W}_2^{0\top}$, $\boldsymbol{A}_2^0 = \frac{1}{h}\boldsymbol{W}_1^{0\top}\boldsymbol{M}$. Here we consider the orthogonal initialization where we have $\boldsymbol{X}^\top\boldsymbol{X} = \boldsymbol{X}\boldsymbol{X}^\top = h\boldsymbol{I}, \boldsymbol{W}_1^{0\top}\boldsymbol{W}_1^0 = \boldsymbol{W}_1^0\boldsymbol{W}_1^{0\top} = \boldsymbol{I}, \boldsymbol{W}_2^{0\top}\boldsymbol{W}_2^0 = \boldsymbol{W}_2^0\boldsymbol{W}_2^{0\top} = \boldsymbol{I}, \boldsymbol{M}^\top\boldsymbol{M} = \boldsymbol{M}\boldsymbol{M}^\top = \frac{1}{h}\boldsymbol{I}$. Based on this,

$$\left\|\boldsymbol{A}_1^0\right\| = \frac{1}{h\sqrt{h}}, \left\|\boldsymbol{A}_1^0\right\|_F = \sqrt{tr(\boldsymbol{A}_1^{0\top}\boldsymbol{A}_1^0)} = \frac{1}{h} \tag{32}$$
$$\left\|\boldsymbol{A}_2^0\right\| = \frac{1}{h\sqrt{h}}, \left\|\boldsymbol{A}_2^0\right\|_F = \sqrt{tr(\boldsymbol{A}_2^{0\top}\boldsymbol{A}_2^0)} = \frac{1}{h}. \tag{33}$$

We also have $\boldsymbol{B}_1^0 = \frac{1}{h^2}\boldsymbol{W}_1^0\boldsymbol{W}_2^0\boldsymbol{W}_2^{0\top}$, $\boldsymbol{B}_2^0 = \frac{1}{h}\boldsymbol{W}_1^{0\top}\boldsymbol{W}_1^0\boldsymbol{W}_2^0$, so we can get that

$$\left\|\boldsymbol{B}_1^0\right\| = \frac{1}{h^2}, \left\|\boldsymbol{B}_1^0\right\|_F = \sqrt{tr(\boldsymbol{B}_1^{0\top}\boldsymbol{B}_1^0)} = \frac{1}{h\sqrt{h}} \tag{34}$$
$$\left\|\boldsymbol{B}_2^0\right\| = \frac{1}{h^2}, \left\|\boldsymbol{B}_2^0\right\|_F = \sqrt{tr(\boldsymbol{B}_2^{0\top}\boldsymbol{B}_2^0)} = \frac{1}{h\sqrt{h}}. \tag{35}$$

Since $\boldsymbol{G}_1^0 = \boldsymbol{B}_1^0 - \boldsymbol{A}_1^0$, $\boldsymbol{G}_2^0 = \boldsymbol{B}_2^0 - \boldsymbol{A}_2^0$, we obtain that

$$\left\|\boldsymbol{G}_1^0 - \boldsymbol{A}_1^0\right\| \leq \frac{1}{\sqrt{h}}\left\|\boldsymbol{A}_1^0\right\| \leq \frac{1}{\sqrt{h}}(\left\|\boldsymbol{G}_1^0\right\| + \left\|\boldsymbol{G}_1^0 - \boldsymbol{A}_1^0\right\|)$$
$$\left\|\boldsymbol{G}_2^0 - \boldsymbol{A}_2^0\right\| \leq \frac{1}{\sqrt{h}}\left\|\boldsymbol{A}_2^0\right\| \leq \frac{1}{\sqrt{h}}(\left\|\boldsymbol{G}_1^0\right\| + \left\|\boldsymbol{G}_2^0 - \boldsymbol{A}_2^0\right\|).$$

Thus, we get that

$$\left\|\boldsymbol{G}_1^0 - \boldsymbol{A}_1^0\right\| \leq \frac{1}{\sqrt{h}-1}\left\|\boldsymbol{G}_1^0\right\|,$$
$$\left\|\boldsymbol{G}_2^0 - \boldsymbol{A}_2^0\right\| \leq \frac{1}{\sqrt{h}-1}\left\|\boldsymbol{G}_2^0\right\|. \tag{36}$$

Based on this, we can get $h\sqrt{h}\left\|\boldsymbol{G}_1^0\right\| = \Theta_{h,\mathbb{P}}(1), h\left\|\boldsymbol{G}_1^0\right\|_F = \Theta_{h,\mathbb{P}}(1), h\sqrt{h}\left\|\boldsymbol{G}_2^0\right\| = \Theta_{h,\mathbb{P}}(1), h\left\|\boldsymbol{G}_2^0\right\|_F = \Theta_{h,\mathbb{P}}(1)$.

Since we have $\left\|\boldsymbol{W}_1^0\right\| = 1, \left\|\boldsymbol{W}_1^0\right\|_F = \sqrt{h}, \left\|\boldsymbol{W}_2^0\right\| = 1, \left\|\boldsymbol{W}_2^0\right\|_F = \sqrt{h}$, based on Assumption E.1, we have

$$Large\ lr : \eta_1 = \Theta(h\sqrt{h}) \Rightarrow \left\|\boldsymbol{W}_1^1 - \boldsymbol{W}_1^0\right\|_F \asymp \left\|\boldsymbol{W}_1^0\right\|_F$$
$$\eta_2 = \Theta(h\sqrt{h}) \Rightarrow \left\|\boldsymbol{W}_2^1 - \boldsymbol{W}_2^0\right\|_F \asymp \left\|\boldsymbol{W}_2^0\right\|_F$$

$\square$

C.1.3. THREE-LAYER NEURAL NETWORK UNDER GAUSSIAN INITIALIZATION

Here, we present a one-step update gradient-norm analysis for three-layer neural networks under Gaussian initialization. To ensure generality, here we incorporate noise with different $d, h, n$.

**Lemma C.6.** *Consider that $\boldsymbol{W}_1 \in \mathbb{R}^{d \times h}, \boldsymbol{W}_2 \in \mathbb{R}^{h \times h}, \boldsymbol{a} \in \mathbb{R}^h$ are the first two hidden layers and last layer weights, respectively, with entries sampled i.i.d as follows $\sqrt{d} \left[\boldsymbol{W}_1\right]_{i,j} \sim \mathcal{N}(0,1), \sqrt{h} \left[\boldsymbol{W}_2\right]_{i,j} \sim \mathcal{N}(0,1), \sqrt{h} \left[\boldsymbol{a}\right]_i \sim \mathcal{N}(0,1), \forall i \in [h], j \in [d]$. We define*

$$\boldsymbol{A}_1^0 = \frac{1}{n\sqrt{h}} \boldsymbol{X}^\top \boldsymbol{y} \boldsymbol{a}^\top \boldsymbol{W}_2^{0\top}$$

$$\boldsymbol{B}_1^0 = \frac{1}{nh} \boldsymbol{X}^\top \boldsymbol{X} \boldsymbol{W}_1^0 \boldsymbol{W}_2^0 \boldsymbol{a} \boldsymbol{a}^\top \boldsymbol{W}_2^{0\top}$$

$$\boldsymbol{A}_2^0 = \frac{1}{n\sqrt{h}} \boldsymbol{W}_1^{0\top} \boldsymbol{X}^\top \boldsymbol{y} \boldsymbol{a}^\top$$

$$\boldsymbol{B}_2^0 = \frac{1}{nh} \boldsymbol{W}_1^{0\top} \boldsymbol{X}^\top \boldsymbol{X} \boldsymbol{W}_1^0 \boldsymbol{W}_2^0 \boldsymbol{a} \boldsymbol{a}^\top,$$

*we have*

(i) $\mathbb{E} \left\| \boldsymbol{A}_1^0 \right\| \leq \mathbb{E} \left\| \boldsymbol{A}_1^0 \right\|_F \leq C \left( \frac{1}{\sqrt{h}} + \frac{d}{n\sqrt{h}} + \sqrt{\frac{1}{h} + \frac{d}{nh}} \right)$

(ii) $\mathbb{E} \left\| \boldsymbol{B}_1^0 \right\| \leq \mathbb{E} \left\| \boldsymbol{B}_1^0 \right\|_F \leq C \sqrt{1 + \frac{h}{d}} \left( \frac{1}{d} + \frac{1}{n} \right)$

(iii) $\mathbb{P} \left( \left\| \boldsymbol{A}_1^0 \right\|_F \geq C \left( \frac{6}{\sqrt{h}} + \frac{d}{n\sqrt{h}} + \frac{5\sqrt{d}}{\sqrt{nh}} \right) \right) \leq 4e^{-ch} + 6e^{-cn}$

(iv) $\mathbb{P} \left( \left\| \boldsymbol{A}_1^0 \right\| \leq \frac{\rho_e \sqrt{2d}}{8\sqrt{nh}} \right) \leq 6 \left( e^{-c \min \left\{ \frac{nd^2}{(n^2+d^2)}, \frac{nd}{n+d} \right\}} + e^{-cn} + e^{-ch} \right)$

(v) $\mathbb{P} \left( \left\| \boldsymbol{B}_1^0 \right\|_F \geq C \left( \frac{8}{h} + \frac{2d}{nh} + \frac{9\sqrt{d}}{\sqrt{nh}} + \frac{4}{\sqrt{nh}} + \frac{4}{\sqrt{hd}} \right) \right) \leq 4e^{-cn} + 8e^{-ch} + 2e^{-cd}$

(vi) $\mathbb{E} \left\| \boldsymbol{A}_2^0 \right\| \leq \mathbb{E} \left\| \boldsymbol{A}_2^0 \right\|_F \leq C \left( \frac{\sqrt{(h+d)(n+d)}(\sqrt{n+d}+\sqrt{n})}{n\sqrt{hd}} \right)$

(vii) $\mathbb{E} \left\| \boldsymbol{B}_2^0 \right\| \leq \mathbb{E} \left\| \boldsymbol{B}_2^0 \right\|_F \leq C \left( \frac{1}{d} + \frac{1}{h} + \frac{1}{n} + \frac{d}{nh} \right)$

(viii) $\mathbb{P} \left( \left\| \boldsymbol{A}_2^0 \right\|_F \geq C \left( \frac{12}{\sqrt{h}} + \frac{2d}{n\sqrt{h}} + \frac{10\sqrt{d}}{\sqrt{nh}} + \frac{6\sqrt{d}}{d} + \frac{5\sqrt{n}}{n} + \frac{\sqrt{d}}{n} \right) \right) \leq 2e^{-cd} + 2e^{-ch} + 6e^{-cn}$

(ix) $\mathbb{P} \left( \left\| \boldsymbol{A}_2^0 \right\| \leq \frac{\rho_e \sqrt{2d}}{8\sqrt{nh}} \right) \leq$
$6 \left( e^{-c \min \left\{ \frac{nd^2}{(n^2+d^2)}, \frac{nd}{n+d}, \frac{nd^4}{(n^2+d^2)(h^2+d^2)} \right\}} + e^{-cn} + e^{-ch} + e^{-cd} + nhd^2 e^{-c\sqrt{d}} \right)$

(x) $\mathbb{P} \left( \left\| \boldsymbol{B}_2^0 \right\|_F \geq C \left( \frac{4}{h} + \frac{32d}{nh} + \frac{16\sqrt{d}}{h\sqrt{n}} + \frac{4}{\sqrt{hd}} + \frac{16}{\sqrt{nh}} + \frac{4}{\sqrt{nd}} + \frac{4}{n} + \frac{16\sqrt{d}}{n\sqrt{h}} \right) \right)$
$\leq 8e^{-cd} + 6e^{-ch}$

**Remark C.7.** *Note that $\boldsymbol{A}_1^0$ is a rank-1 matrix, so the upper bound and lower bound of $\left\| \boldsymbol{A}_1^0 \right\|_F$ and $\left\| \boldsymbol{A}_1^0 \right\|$ is similar. The same is to $\boldsymbol{A}_2^0$.*

**Proof of Lemma C.6.** We analyze these matrices of interest separately.

**Part of (i).**   Notice that

$$
\begin{aligned}
\left\|A_1^0\right\|_F &\leq \frac{1}{n\sqrt{h}}\left\|X^\top X \beta^* a^\top W_2^{0\top}\right\|_F + \frac{1}{n\sqrt{h}}\left\|X^\top \xi a^\top W_2^{0\top}\right\|_F \\
&\leq \frac{1}{n\sqrt{h}}\left\|X^\top X \beta^*\right\|\left\|a^\top W_2^{0\top}\right\| + \frac{1}{n\sqrt{h}}\left\|X^\top \xi\right\|\left\|a^\top W_2^{0\top}\right\| \\
&\leq \frac{1}{n\sqrt{h}}\|X\|\left(\|X\| + \|\xi\|\right)\|a\|\left\|W_2^0\right\|.
\end{aligned}
\tag{37}
$$

Based on basic probability theory, we know that Gaussian random matrices and vectors satisfy

$$
\begin{aligned}
\mathbb{E}(\|a\|) = 1, \quad \mathbb{E}(\|\xi\|) = \rho_e\sqrt{n} \\
\mathbb{E}(\|W_2^0\|^2) \leq C_0, \quad \mathbb{E}(\|X\|^2) \leq C_1(n+d),
\end{aligned}
\tag{38}
$$

where $C_0$ and $C_1$ are consts.

Based on ( 38), we obtain

$$
\mathbb{E}(\|A_1^0\|) \leq \mathbb{E}(\|A_1^0\|)_F \leq C\left(\frac{1}{\sqrt{h}} + \frac{d}{n\sqrt{h}} + \sqrt{\frac{1}{h} + \frac{d}{nh}}\right).
$$

**Part of (ii).**   Notice that

$$
\begin{aligned}
\left\|B_1^0\right\|_F &\leq \frac{1}{nh}\left\|X^\top X W_1^0 W_2^0 a\right\|\left\|a^\top W_2^{0\top}\right\| \\
&\leq \frac{1}{nh}\|X\|^2\|W_1^0\|\|a\|^2\left\|W_2^{0\top}\right\|^2
\end{aligned}
\tag{39}
$$

Besides ( C.3), based on basic probability theory, we know that Gaussian random matrices and vectors satisfy

$$
\mathbb{E}(\|W_1^0\|^2) \leq C_0(1 + \frac{h}{d}).
\tag{40}
$$

Based on ( 38) and ( 40), we obtain

$$
\mathbb{E}(\|B_1^0\|) \leq \mathbb{E}(\|B_1^0\|)_F \leq C\sqrt{1 + \frac{h}{d}}\left(\frac{1}{d} + \frac{1}{n}\right).
$$

**Part of (iii).**   Based on random vector and matrix concentration property of $\|X\|, \|\xi\|, \|a\|$ and $\|W_2^0\|$ (e.g. derived by Lemma C.3: for any $t \geq 0$,

$$
\begin{aligned}
\mathbb{P}(|\|\xi\| - \rho_e\sqrt{n}| \geq \frac{1}{2}\rho_e\sqrt{n}) \leq 2e^{-cn}, \quad \mathbb{P}(|\|a\| - 1| \geq \frac{1}{2}) \leq 2e^{-ch}, \\
\mathbb{P}(\|X\| \geq \sqrt{n} + \sqrt{d} + t) \leq 2e^{-ct^2}, \quad \mathbb{P}(\|W_2^0\| \geq 3) \leq 2e^{-ch}
\end{aligned}
\tag{41}
$$

Hence, from ( 37) and ( 41), we arrive at

$$
\mathbb{P}\left(\|A_1^0\|_F \geq C\left(\frac{2}{\sqrt{h}} + \frac{d}{n\sqrt{h}} + \frac{3\sqrt{d}}{\sqrt{nh}} + \frac{t^2}{n\sqrt{h}} + \frac{3t}{\sqrt{nh}} + \frac{2t\sqrt{d}}{n\sqrt{h}}\right)\right) \leq 2(e^{-cn} + 2e^{-ch} + 2e^{-ct^2})
$$

Thus we can take $t = \sqrt{n}$ to obtain the result that:

$$
\mathbb{P}\left(\|A_1^0\|_F \geq C\left(\frac{6}{\sqrt{h}} + \frac{d}{n\sqrt{h}} + \frac{5\sqrt{d}}{\sqrt{nh}}\right)\right) \leq 4e^{-ch} + 6e^{-cn}.
$$

**Part of (iv).** Here we try to give the lower bound of $\|A_1\|$, since $A_1$ is rank-1 matrix, we have

$$
\begin{aligned}
\left\|A_1^0\right\|^2 &= \frac{1}{n^2 h}\left\|X^\top y a^\top W_2^{0\top}\right\|^2 = \frac{1}{n^2 h}\left\|X^\top y a^\top W_2^{0\top}\right\|_F^2 \\
&= \frac{1}{n^2 h}tr(W_2^0 a y^\top X X^\top y a^\top W_2^{0\top}) \\
&= \frac{1}{n^2 h}tr(y^\top X X^\top y a^\top W_2^{0\top} W_2^0 a) \\
&= \frac{1}{n^2 h}\left\|W_2^0 a\right\|^2 \left\|X^\top y\right\|^2 \\
&\geq \frac{1}{n^2 h}\left\|W_2^0 a\right\|^2 \left(\xi^\top X X^\top \xi + 2\beta^{*\top} X^\top X X^\top \xi\right).
\end{aligned}
$$

Following (Ba et al., 2022), we define events $E_1$, $E_2$ and $E_3$ by

$$
E_1 := \left\{|tr(X X^\top) - nd| \leq \frac{nd}{2}\right\}, \quad E_2 := \left\{\|X\| \leq \sqrt{d} + 2\sqrt{n}\right\}, \quad E_3 := \left\{\|W_2^0\| \leq 3\right\}.
$$

Based on (41) and Lemma C.3, we know that

$$
\mathbb{P}(E_1) \geq 1 - 2e^{-cn}, \quad \mathbb{P}(E_2) \geq 1 - 2e^{-cn}, \quad \mathbb{P}(E_3) \geq 1 - 2e^{-ch}.
$$

Thus condition on $E_1 \cap E_2$, it is easy to find that $\mathbb{E}\left\|X X^\top\right\|_F^2 \leq C_2 n(n^2 + d^2)$, also based on (38), by Lemma C.1, we have

$$
\mathbb{P}\left(\xi^\top X X^\top \xi \leq \frac{\rho_e^2}{2}nd - t \mid E_1 \cap E_2\right) \leq 2e^{-c\min\left\{\frac{t^2}{n(n^2+d^2)}, \frac{t}{n+d}\right\}}.
$$

Choosing $t = \frac{\rho_e^2}{4}nd$, we have

$$
\mathbb{P}\left(\xi^\top X X^\top \xi \leq \frac{\rho_e^2}{8}nd\right) \leq 2e^{-c\min\left\{\frac{nd^2}{(n^2+d^2)}, \frac{nd}{n+d}\right\}} + 4e^{-cn}. \tag{42}
$$

In addition, by Lemma C.2, condition on $E_2$, we know that

$$
\left\|\beta^{*\top} X^\top X X^\top \xi\right\|_{\psi_2} \leq C_3 \sqrt{n(n^2 + d^2)},
$$

one can easily see that

$$
\mathbb{P}\left(\left|\beta^{*\top} X^\top X X^\top \xi\right| \geq t \mid E_2\right) \leq 2e^{-\frac{ct^2}{n(n^2+d^2)}}.
$$

Thus, let $t = \frac{\rho_e^2}{32}nd$, we obtain

$$
\mathbb{P}\left(\left|\beta^{*\top} X^\top X X^\top \xi\right| \geq \frac{\rho_e^2}{32}nd\right) \leq 2e^{-\frac{cnd^2}{(n^2+d^2)}} + 2e^{-cn}. \tag{43}
$$

Similarly, by Lemma C.2, condition on $E_3$, we know that $\left\|\|W_2^0 a\| - 1\right\|_{\psi_2} \leq \frac{3C_3}{\sqrt{h}}$, we obtain that

$$
\mathbb{P}\left(\left|\|W_2^0 a\|\right| \geq t \mid E_3\right) \leq 2e^{-cht^2}.
$$

Thus, let $t = \frac{1}{2}$, we obtain

$$
\mathbb{P}\left(\|W_2^0 a\| \leq \frac{1}{2}\right) \leq 4e^{-ch}. \tag{44}
$$

Based on ( 42), ( 43) and ( 44), we arrive at

$$\mathbb{P}\left(\left\|\boldsymbol{A}_1^0\right\| \leq \frac{\rho_e\sqrt{2d}}{8\sqrt{nh}}\right) \leq 6\left(e^{-c\min\left\{\frac{nd^2}{(n^2+d^2)},\frac{nd}{n+d}\right\}} + e^{-cn} + e^{-ch}\right)$$

**Part of (v).**   Based on random matrix concentration property of $\left\|\boldsymbol{W}_1^0\right\|$, we have for any $t \geq 0$

$$\mathbb{P}(\left\|\boldsymbol{W}_1^0\right\| \geq 2 + \sqrt{\frac{h}{d}}) \leq 2e^{-cd} \tag{45}$$

Based on ( 39), ( 41) and ( 45), similar to proof of (ii), by choosing $t = \sqrt{n}$, we arrive at

$$\mathbb{P}\left(\left\|\boldsymbol{B}_1^0\right\|_F \geq C\left(\frac{8}{h} + \frac{2d}{nh} + \frac{9\sqrt{d}}{\sqrt{nh}} + \frac{4}{\sqrt{nh}} + \frac{4}{\sqrt{hd}}\right)\right) \leq 4e^{-cn} + 8e^{-ch} + 2e^{-cd}.$$

**Part of (vi).**   Notice that

$$\begin{aligned}
\left\|\boldsymbol{A}_2^0\right\|_F &\leq \frac{1}{n\sqrt{h}}\left\|\boldsymbol{W}_1^{0\top}\boldsymbol{X}^\top\boldsymbol{X}\boldsymbol{\beta}^*\boldsymbol{a}^\top\right\|_F + \frac{1}{n\sqrt{h}}\left\|\boldsymbol{W}_1^{0\top}\boldsymbol{X}^\top\boldsymbol{\xi}\boldsymbol{a}^\top\right\|_F \\
&\leq \frac{1}{n\sqrt{h}}\left\|\boldsymbol{W}_1^{0\top}\boldsymbol{X}^\top\boldsymbol{X}\boldsymbol{\beta}^*\right\|\left\|\boldsymbol{a}^\top\right\| + \frac{1}{n\sqrt{h}}\left\|\boldsymbol{W}_1^{0\top}\boldsymbol{X}^\top\boldsymbol{\xi}\right\|\left\|\boldsymbol{a}^\top\right\| \\
&\leq \frac{1}{n\sqrt{h}}\left\|\boldsymbol{W}_1^0\right\|\left\|\boldsymbol{X}\right\|\left(\left\|\boldsymbol{X}\right\| + \left\|\boldsymbol{\xi}\right\|\right)\left\|\boldsymbol{a}\right\|
\end{aligned} \tag{46}$$

Based on ( 38) and ( 40), we obtain

$$\mathbb{E}(\left\|\boldsymbol{A}_2^0\right\|) \leq \mathbb{E}(\left\|\boldsymbol{A}_2^0\right\|)_F \leq C\left(\frac{\sqrt{(h+d)(n+d)}(\sqrt{n+d}+\sqrt{n})}{n\sqrt{hd}}\right).$$

**Part of (vii).**   Notice that

$$\begin{aligned}
\left\|\boldsymbol{B}_2^0\right\|_F &\leq \frac{1}{nh}\left\|\boldsymbol{W}_1^{0\top}\boldsymbol{X}^\top\boldsymbol{X}\boldsymbol{W}_1^0\boldsymbol{W}_2^0\boldsymbol{a}\right\|\left\|\boldsymbol{a}^\top\right\| \\
&\leq \frac{1}{nh}\left\|\boldsymbol{X}\right\|^2\left\|\boldsymbol{W}_1^0\right\|^2\left\|\boldsymbol{a}\right\|^2\left\|\boldsymbol{W}_2^{0\top}\right\|
\end{aligned} \tag{47}$$

Based on ( 38) and ( 40), we obtain

$$\mathbb{E}(\left\|\boldsymbol{B}_2^0\right\|) \leq \mathbb{E}(\left\|\boldsymbol{B}_2^0\right\|)_F \leq C\left(\frac{1}{d} + \frac{1}{h} + \frac{1}{n} + \frac{d}{nh}\right).$$

**Part of (viii).**   Hence, from( 41) ( 45) and ( 46), we arrive at

$$\begin{aligned}
&\mathbb{P}\left(\left\|\boldsymbol{A}_2^0\right\|_F \geq C\left(2 + \sqrt{\frac{h}{d}}\right)\left(\frac{2}{\sqrt{h}} + \frac{d}{n\sqrt{h}} + \frac{3\sqrt{d}}{\sqrt{nh}} + \frac{t^2}{n\sqrt{h}} + \frac{3t}{\sqrt{nh}} + \frac{2t\sqrt{d}}{n\sqrt{h}}\right)\right) \\
&\leq 2(e^{-cn} + 2e^{-ch} + 2e^{-ct^2}).
\end{aligned}$$

Thus we can take $t = \sqrt{n}$ to obtain the result that:

$$\mathbb{P}\left(\left\|\boldsymbol{A}_2^0\right\|_F \geq C\left(\frac{12}{\sqrt{h}} + \frac{2d}{n\sqrt{h}} + \frac{10\sqrt{d}}{\sqrt{nh}} + \frac{6\sqrt{d}}{d} + \frac{5\sqrt{n}}{n} + \frac{\sqrt{d}}{n}\right)\right) \leq 2e^{-cd} + 2e^{-ch} + 6e^{-cn}.$$

**Part of (ix).** Here we try to give the lower bound of $\|A_2\|$, since $A_2$ is rank-1 matrix, we have

$$
\begin{aligned}
\left\|A_2^0\right\|^2 &= \frac{1}{n^2h}\left\|W_1^{0\top}X^\top ya^\top\right\|^2 = \frac{1}{n^2h}\left\|W_1^{0\top}X^\top ya^\top\right\|_F^2 \\
&= \frac{1}{n^2h}tr(ay^\top XW_1^0 W_1^{0\top}X^\top ya^\top) \\
&= \frac{1}{n^2h}tr(y^\top XW_1^0 W_1^{0\top}X^\top ya^\top a) \\
&= \frac{1}{n^2h}\|a\|^2\left\|W_2^{0\top}X^\top y\right\|^2 \\
&\geq \frac{1}{n^2h}\|a\|^2\left(\xi^\top XW_1^0 W_1^{0\top}X^\top \xi + 2\beta^{*\top}X^\top XW_1^0 W_1^{0\top}X^\top \xi\right).
\end{aligned}
$$

Following (Ba et al., 2022), we define events $E_2$, $E_4$ and $E_5$ by

$$
\begin{aligned}
E_2 &:= \left\{\|X\| \leq \sqrt{d} + 2\sqrt{n}\right\}, \\
E_4 &:= \left\{\|W_1^0\| \leq 2 + \frac{h}{d}\right\}, \\
E_5 &:= \left\{\left|tr(XW_1^0 W_1^{0\top}X^\top) - nh\right| \leq \frac{nh}{2}\right\}.
\end{aligned}
$$

Based on ( 41) and ( 45), we know that

$$
\mathbb{P}(E_2) \geq 1 - 2e^{-cn}, \quad \mathbb{P}(E_4) \geq 1 - 2e^{-cd}.
$$

We also know that by sub-gaussian and sub-exponential concentration inequality

$$
\mathbb{P}(\left|W_{1_{i,j}}^0\right| \leq 1) \geq 1 - 2e^{-cd}, \quad \mathbb{P}(\left|W_{1_{i,j}}^{0}{}^2 - 1\right| \leq 1) \geq 1 - 2e^{-c\sqrt{d}}. \tag{48}
$$

Based on Lemma C.3 and ( 48), we have

$$
\mathbb{P}(E_5) \geq 1 - 2e^{-cn} - 2nhde^{-cd} - 2nh(d^2 - d)e^{-c\sqrt{d}} \geq 1 - 2e^{-cn} - 4nhd^2 e^{-c\sqrt{d}}
$$

.

Thus condition on $E_2 \cap E_4$, it is easy to find that $\mathbb{E}\left\|XW_1^0 W_1^{0\top}X^\top\right\|_F^2 \leq C_4 n(n^2 + h^2)$, also based on ( 38), by Lemma C.1, we have

$$
\mathbb{P}\left(\xi^\top XX^\top \xi \leq \frac{\rho_e^2}{2}nh - t | E_2 \cap E_5\right) \leq 2e^{-c\min\left\{\frac{t^2}{n(n^2+h^2)}, \frac{t}{n+h}\right\}}.
$$

Choosing $t = \frac{\rho_e^2}{4}nd$, we have

$$
\mathbb{P}\left(\xi^\top XW_1^0 W_1^{0\top}X^\top \xi \leq \frac{\rho_e^2}{8}nh\right) \leq 2e^{-c\min\left\{\frac{nh^2}{(n^2+h^2)}, \frac{nh}{n+h}\right\}} + 4e^{-cn} + 4nhd^2 e^{-c\sqrt{d}}. \tag{49}
$$

In addition, by Lemma C.2, condition on $E_2$ and $E_4$, we know that

$$
\left\|\beta^{*\top}X^\top XW_1^0 W_1^{0\top}X^\top \xi\right\|_{\psi_2} \leq C_5 \frac{\sqrt{n(n^2 + d^2)(h^2 + d^2)}}{d},
$$

one can easily see that

$$\mathbb{P}\left(\left|\boldsymbol{\beta}^{*\top}\boldsymbol{X}^\top \boldsymbol{X}\boldsymbol{W}_1^0\boldsymbol{W}_1^{0\top}\boldsymbol{X}^\top\boldsymbol{\xi}\right| \geq t \middle| \boldsymbol{E}_2 \cap \boldsymbol{E}_4\right) \leq 2e^{-\frac{cd^2t^2}{n(n^2+d^2)(h^2+d^2)}}.$$

Thus, let $t = \frac{\rho_e^2}{32}nd$, we obtain

$$\mathbb{P}\left(\left|\boldsymbol{\beta}^{*\top}\boldsymbol{X}^\top \boldsymbol{X}\boldsymbol{W}_1^0\boldsymbol{W}_1^{0\top}\boldsymbol{X}^\top\boldsymbol{\xi}\right| \geq \frac{\rho_e^2}{32}nd\right) \leq 2e^{-\frac{cnd^4}{(n^2+d^2)(h^2+d^2)}} + 2e^{-cn} + 2e^{-cd}. \tag{50}$$

Based on ( 41),( 49)and( 50) , we arrive at

$$\mathbb{P}\left(\left\|\boldsymbol{A}_2^0\right\| \leq \frac{\rho_e\sqrt{2d}}{8\sqrt{nh}}\right) \leq 6\left(e^{-c\min\left\{\frac{nd^2}{(n^2+d^2)},\frac{nd}{n+d},\frac{nd^4}{(n^2+d^2)(h^2+d^2)}\right\}} + e^{-cn} + e^{-ch}\right)$$
$$+ 6\left(e^{-cd} + nhd^2e^{-c\sqrt{d}}\right)$$

**Part of (x).** Based on ( 41),( 45) and ( 47), similar to proof of (vii), by choosing $t = \sqrt{d}$, we arrive at

$$\mathbb{P}\left(\left\|\boldsymbol{B}_2^0\right\|_F \geq C\left(\frac{4}{h} + \frac{32d}{nh} + \frac{16\sqrt{d}}{h\sqrt{n}} + \frac{4}{\sqrt{hd}} + \frac{16}{\sqrt{nh}} + \frac{4}{\sqrt{nd}} + \frac{4}{n} + \frac{16\sqrt{d}}{n\sqrt{h}}\right)\right)$$
$$\leq 8e^{-cd} + 6e^{-ch}.$$

**Proposition C.8.** *(Three-layer NN setting under Gaussian initialization.) Under Assumption E.1, there exists some constant $c^* > 0$ such that for all large $n, h, d$ with probability at least $1 - 32e^{-c^*n} - 30n^4e^{-c^*\sqrt{n}}$, we have gradient approximation,*

$$\left\|\boldsymbol{G}_1^0 - \boldsymbol{A}_1^0\right\| \leq \frac{1}{\sqrt{n}-1}\left\|\boldsymbol{G}_1^0\right\|,$$
$$\left\|\boldsymbol{G}_2^0 - \boldsymbol{A}_2^0\right\| \leq \frac{1}{\sqrt{n}-1}\left\|\boldsymbol{G}_2^0\right\|. \tag{51}$$

*We obtain the norm control of gradient matrices,*

$$\sqrt{h}\left\|\boldsymbol{G}_1^0\right\| = \Theta_{d,\mathbb{P}}(1), \quad \sqrt{h}\left\|\boldsymbol{G}_1^0\right\|_F = \Theta_{d,\mathbb{P}}(1),$$
$$\sqrt{h}\left\|\boldsymbol{G}_2^0\right\| = \Theta_{d,\mathbb{P}}(1), \quad \sqrt{h}\left\|\boldsymbol{G}_2^0\right\|_F = \Theta_{d,\mathbb{P}}(1). \tag{52}$$

*Thus, we have*

$$Small\ lr : \eta_1 = \Theta(\sqrt{h}) \Rightarrow \left\|\boldsymbol{W}_1^1 - \boldsymbol{W}_1^0\right\| \asymp \left\|\boldsymbol{W}_1^0\right\| \tag{53}$$
$$\eta_2 = \Theta(\sqrt{h}) \Rightarrow \left\|\boldsymbol{W}_2^1 - \boldsymbol{W}_2^0\right\| \asymp \left\|\boldsymbol{W}_2^0\right\| \tag{54}$$
$$Large\ lr : \eta_1 = \Theta(h) \Rightarrow \left\|\boldsymbol{W}_1^1 - \boldsymbol{W}_1^0\right\|_F \asymp \left\|\boldsymbol{W}_1^0\right\|_F \tag{55}$$
$$\eta_2 = \Theta(h) \Rightarrow \left\|\boldsymbol{W}_2^1 - \boldsymbol{W}_2^0\right\|_F \asymp \left\|\boldsymbol{W}_2^0\right\|_F \tag{56}$$

**Proof of Proposition E.2.** By Lemma C.6, We know that in the proportional regime, there exist constants $C^*, c^* > 0$ such that

$$\mathbb{P}\left(\left\|\boldsymbol{G}_1^0 - \boldsymbol{A}_1^0\right\| \leq C^*\frac{1}{n}\right) \geq 1 - 14e^{-c^*n}.$$

On the other hand, part(iv) in Lemma C.6 implies that

$$\mathbb{P}\left(\left\|\boldsymbol{A}_1^0\right\| \geq C^*\frac{1}{\sqrt{n}}\right) \geq 1 - 18e^{-c^*n}.$$

Conditioning on the two events stated above, we have

$$\left\|\boldsymbol{G}_1^0 - \boldsymbol{A}_1^0\right\| \le \frac{1}{\sqrt{n}}\left\|\boldsymbol{A}_1^0\right\| \le \frac{1}{\sqrt{n}}(\left\|\boldsymbol{G}_1^0\right\| + \left\|\boldsymbol{G}_1^0 - \boldsymbol{A}_1^0\right\|).$$

We finally obtain that

$$\mathbb{P}\left(\left\|\boldsymbol{G}_1^0 - \boldsymbol{A}_1^0\right\| \le \frac{1}{\sqrt{n}-1}\left\|\boldsymbol{G}_1^0\right\|\right) \ge 1 - 34e^{-c^*n}$$

Similarly, We know that in the proportional regime, there exist constants $C^*, c^* > 0$ such that

$$\mathbb{P}\left(\left\|\boldsymbol{G}_2^0 - \boldsymbol{A}_2^0\right\| \le C^*\frac{1}{n}\right) \ge 1 - 14e^{-c^*n}.$$

On the other hand, part(iv) in Lemma C.6 implies that

$$\mathbb{P}\left(\left\|\boldsymbol{A}_2^0\right\| \ge C^*\frac{1}{\sqrt{n}}\right) \ge 1 - 30n^4 e^{-c^*\sqrt{n}}.$$

Conditioning on the two events stated above, we have

$$\left\|\boldsymbol{G}_2^0 - \boldsymbol{A}_2^0\right\| \le \frac{1}{\sqrt{n}}\left\|\boldsymbol{A}_2^0\right\| \le \frac{1}{\sqrt{n}}(\left\|\boldsymbol{G}_1^0\right\| + \left\|\boldsymbol{G}_2^0 - \boldsymbol{A}_2^0\right\|).$$

We finally obtain that

$$\mathbb{P}\left(\left\|\boldsymbol{G}_2^0 - \boldsymbol{A}_2^0\right\| \le \frac{1}{\sqrt{n}-1}\left\|\boldsymbol{G}_2^0\right\|\right) \ge 1 - 14e^{-c^*n} - 30n^4 e^{-c^*\sqrt{n}}.$$

Also we can get $\sqrt{h}\left\|\boldsymbol{G}_1^0\right\| = \Theta_{d,\mathbb{P}}(1), \sqrt{h}\left\|\boldsymbol{G}_1^0\right\|_F = \Theta_{d,\mathbb{P}}(1), \sqrt{h}\left\|\boldsymbol{G}_2^0\right\| = \Theta_{d,\mathbb{P}}(1), \sqrt{h}\left\|\boldsymbol{G}_2^0\right\|_F = \Theta_{d,\mathbb{P}}(1)$.

Since we have $\left\|\boldsymbol{W}_1^0\right\| = \Theta_{d,\mathbb{P}}(1), \left\|\boldsymbol{W}_1^0\right\|_F = \Theta_{d,\mathbb{P}}(\sqrt{h}), \left\|\boldsymbol{W}_2^0\right\| = \Theta_{d,\mathbb{P}}(1), \left\|\boldsymbol{W}_2^0\right\|_F = \Theta_{d,\mathbb{P}}(\sqrt{h})$, based on Assumption E.1, we have

$$Small\ lr : \eta_1 = \Theta(\sqrt{h}) \Rightarrow \left\|\boldsymbol{W}_1^1 - \boldsymbol{W}_1^0\right\| \asymp \left\|\boldsymbol{W}_1^0\right\|$$
$$\eta_2 = \Theta(\sqrt{h}) \Rightarrow \left\|\boldsymbol{W}_2^1 - \boldsymbol{W}_2^0\right\| \asymp \left\|\boldsymbol{W}_2^0\right\|$$
$$Large\ lr : \eta_1 = \Theta(h) \Rightarrow \left\|\boldsymbol{W}_1^1 - \boldsymbol{W}_1^0\right\|_F \asymp \left\|\boldsymbol{W}_1^0\right\|_F$$
$$\eta_2 = \Theta(h) \Rightarrow \left\|\boldsymbol{W}_2^1 - \boldsymbol{W}_2^0\right\|_F \asymp \left\|\boldsymbol{W}_2^0\right\|_F$$

$\square$

### C.1.4. TWO-LAYER NEURAL NETWORK CASE UNDER GAUSSIAN INITIALIZATION

The one-step update equations for the two-layer neural network are as follows:

$$\begin{aligned}
\boldsymbol{W}_1^1 &= \boldsymbol{W}_1^0 - \eta_1 \boldsymbol{G}_1^0 \\
\boldsymbol{W}_2^1 &= \boldsymbol{W}_2^0 - \eta_2 \boldsymbol{G}_2^0
\end{aligned} \tag{57}$$

where $W_1^0, W_2^0$ are the initial hidden layer weights, $W_1^1, W_2^1$ are the updated layer weights, $G_1^0$ and $G_2^0$ are the corresponding gradient matrix, where

$$\boldsymbol{G}_1^0 = \underbrace{\frac{1}{nh^2}\boldsymbol{X}^\top\boldsymbol{X}\boldsymbol{W}_1^0\boldsymbol{W}_2^0\boldsymbol{W}_2^{0^\top}}_{\boldsymbol{B}_1^0} - \underbrace{\frac{1}{nh}\boldsymbol{X}^\top\boldsymbol{Y}\boldsymbol{W}_2^{0^\top}}_{\boldsymbol{A}_1^0} \tag{58}$$

$$\boldsymbol{G}_2^0 = \underbrace{\frac{1}{nh^2}\boldsymbol{W}_1^{0^\top}\boldsymbol{X}^\top\boldsymbol{X}\boldsymbol{W}_1^0\boldsymbol{W}_2^0}_{\boldsymbol{B}_2^0} - \underbrace{\frac{1}{nh}\boldsymbol{W}_1^{0^\top}\boldsymbol{X}^\top\boldsymbol{Y}}_{\boldsymbol{A}_2^0} \tag{59}$$

**Proposition C.9.** *(Two-layer neural network under gaussian initialization.) There exists some constants $p^*, q^* > 0$ such that for all large $n, h, d$ with high probability at least $1 - p^* e^{-q^* n}$*

$$\left\| G_1^0 - A_1^0 \right\| \leq \frac{1}{\sqrt{n-1}} \left\| G_1^0 \right\| \tag{60}$$

$$\left\| G_2^0 - A_2^0 \right\| \leq \frac{1}{\sqrt{n-1}} \left\| G_2^0 \right\|. \tag{61}$$

*There exists some constants $p, q > 0$ such that for all large $n, h, d$ with high probability at least $1 - pe^{-qn}$*

$$Large\ lr : \eta_1 = \Theta(h\sqrt{h}) \Rightarrow \left\| W_1^1 - W_1^0 \right\|_F \asymp \left\| W_1^0 \right\|_F \tag{62}$$

$$\eta_2 = \Theta(h\sqrt{h}) \Rightarrow \left\| W_2^1 - W_2^0 \right\|_F \asymp \left\| W_2^0 \right\|_F. \tag{63}$$

**Proof of Proposition C.9.** The proof is same to proofs of Proposition E.2 and Lemma C.6.

## C.2. Norm Analysis of Two-step Update Gradient Matrices

### C.2.1. TWO-LAYER NEURAL NETWORK UNDER ORTHOGONAL INITIALIZATION

**Proposition C.10.** *(Two-layer NN setting under Orthogonal initialization.) Under Assumption 3.2, if $\eta_1, \eta_2 \leq O(h\sqrt{h})$, we have the following gradient approximation,*

$$\left\| \overline{G_1^1} - \widetilde{A_1^1} \right\| \leq \frac{1}{\sqrt{h}-1} \left\| \overline{G_1^1} \right\|,$$
$$\left\| \overline{G_2^1} - \widetilde{A_2^1} \right\| \leq \frac{1}{\sqrt{h}-1} \left\| \overline{G_2^1} \right\|. \tag{64}$$

*And we have*

$$Large\ lr : \eta_1 = \Theta(h\sqrt{h}) \Rightarrow \left\| \overline{W_1^2} - \widetilde{W_1^1} \right\|_F \asymp \left\| \widetilde{W_1^1} \right\|_F \tag{65}$$

$$\eta_2 = \Theta(h\sqrt{h}) \Rightarrow \left\| \overline{W_2^2} - \widetilde{W_2^1} \right\|_F \asymp \left\| \widetilde{W_2^1} \right\|_F \tag{66}$$

**Proof of Proposition C.10.** Note that $\widetilde{A_1^1} = \frac{1}{h} M \widetilde{W_2^1}^\top$, $\widetilde{A_2^1} = \frac{1}{h} \widetilde{W_1^1}^\top M$. Based on Lemma C.5, it is easy to get

$$\left\| \widetilde{W_1^1} - W_1^0 \right\| \asymp \left\| W_1^0 \right\|, \left\| \widetilde{W_2^1} - W_2^0 \right\| \asymp \left\| W_2^0 \right\|$$
$$\left\| \widetilde{W_1^1} - W_1^0 \right\|_F \asymp \left\| W_1^0 \right\|_F, \left\| \widetilde{W_2^1} - W_2^0 \right\|_F \asymp \left\| W_2^0 \right\|_F$$

We also have $\widetilde{B_1^1} = \frac{1}{h^2} \widetilde{W_1^1} \widetilde{W_2^1} \widetilde{W_2^1}^\top$, $\widetilde{B_2^1} = \frac{1}{h} \widetilde{W_1^1}^\top \widetilde{W_1^1} \widetilde{W_2^1}$. Thus we can obtain that

$$\left\| \widetilde{A_1^1} \right\| \asymp \left\| A_1^0 \right\|, \left\| \widetilde{A_1^1} \right\|_F \asymp \left\| A_1^0 \right\|_F, \left\| \widetilde{A_2^1} \right\| \asymp \left\| A_2^0 \right\|, \left\| \widetilde{A_2^1} \right\|_F \asymp \left\| A_2^0 \right\|_F$$
$$\left\| \widetilde{B_1^1} \right\| \asymp \left\| B_1^0 \right\|, \left\| \widetilde{B_1^1} \right\|_F \asymp \left\| B_1^0 \right\|_F, \left\| \widetilde{B_2^1} \right\| \asymp \left\| B_2^0 \right\|, \left\| \widetilde{B_2^1} \right\|_F \asymp \left\| B_2^0 \right\|_F$$

Since $\overline{G_1^1} = \widetilde{B_1^1} - \widetilde{A_1^1}, \overline{G_2^1} = \widetilde{B_2^1} - \widetilde{A_2^1}$, we obtain that

$$\left\| \overline{G_1^1} - \widetilde{A_1^1} \right\| \leq \frac{1}{\sqrt{h}} \left\| \widetilde{A_1^1} \right\| \leq \frac{1}{\sqrt{h}} (\overline{G_1^1} + \left\| \overline{G_1^1} - \widetilde{A_1^1} \right\|)$$
$$\left\| \overline{G_2^1} - \widetilde{A_2^1} \right\| \leq \frac{1}{\sqrt{h}} \left\| \widetilde{A_2^1} \right\| \leq \frac{1}{\sqrt{h}} (\overline{G_2^1} + \left\| \overline{G_2^1} - \widetilde{A_2^1} \right\|).$$

Thus, we get that

$$
\left\| \overline{\boldsymbol{G}_1^1} - \widetilde{\boldsymbol{A}_1^1} \right\| \leq \frac{1}{\sqrt{h}-1} \left\| \overline{\boldsymbol{G}_1^1} \right\|,
$$

$$
\left\| \overline{\boldsymbol{G}_2^1} - \widetilde{\boldsymbol{A}_2^1} \right\| \leq \frac{1}{\sqrt{h}-1} \left\| \overline{\boldsymbol{G}_2^1} \right\|. \tag{67}
$$

Based on this, we can get $\left\| \overline{\boldsymbol{G}_1^1} \right\| \asymp \left\| \boldsymbol{A}_1^0 \right\|$, $\left\| \overline{\boldsymbol{G}_1^1} \right\|_F = \left\| \boldsymbol{A}_1^0 \right\|_F$, $\left\| \overline{\boldsymbol{G}_2^1} \right\| = \left\| \boldsymbol{A}_2^0 \right\|$, $\left\| \overline{\boldsymbol{G}_2^1} \right\|_F = \left\| \boldsymbol{A}_2^0 \right\|_F$.

Since we have $\left\| \boldsymbol{W}_1^0 \right\| \asymp \left\| \widetilde{\boldsymbol{W}_1^1} \right\|$, $\left\| \boldsymbol{W}_1^0 \right\|_F = \left\| \widetilde{\boldsymbol{W}_1^1} \right\|_F$, $\left\| \boldsymbol{W}_2^0 \right\| = \left\| \widetilde{\boldsymbol{W}_2^1} \right\|$, $\left\| \boldsymbol{W}_2^0 \right\|_F = \left\| \widetilde{\boldsymbol{W}_2^1} \right\|_F$, based on Assumption E.1, we have

$$
Large\ lr : \eta_1 = \Theta(h\sqrt{h}) \Rightarrow \left\| \overline{\boldsymbol{W}_1^2} - \widetilde{\boldsymbol{W}_1^1} \right\|_F \asymp \left\| \widetilde{\boldsymbol{W}_1^1} \right\|_F \tag{68}
$$

$$
\eta_2 = \Theta(h\sqrt{h}) \Rightarrow \left\| \overline{\boldsymbol{W}_2^2} - \widetilde{\boldsymbol{W}_2^1} \right\|_F \asymp \left\| \widetilde{\boldsymbol{W}_2^1} \right\|_F \tag{69}
$$

$\square$

### C.2.2. THREE-LAYER NEURAL NETWORK UNDER ORTHOGONAL INITIALIZATION

**Proposition C.11.** *(Three-layer NN setting under Orthogonal initialization.) Under Assumption 3.2, if $\eta_1, \eta_2 \leq O(h)$, we have the following gradient approximation,*

$$
\left\| \overline{\boldsymbol{G}_1^1} - \widetilde{\boldsymbol{A}_1^1} \right\| \leq \frac{1}{\sqrt{h}-1} \left\| \overline{\boldsymbol{G}_1^1} \right\|,
$$

$$
\left\| \overline{\boldsymbol{G}_2^1} - \widetilde{\boldsymbol{A}_2^1} \right\| \leq \frac{1}{\sqrt{h}-1} \left\| \overline{\boldsymbol{G}_2^1} \right\|. \tag{70}
$$

*And we have*

$$
Small\ lr : \eta_1 = \Theta(\sqrt{h}) \Rightarrow \left\| \overline{\boldsymbol{W}_1^2} - \widetilde{\boldsymbol{W}_1^1} \right\| \asymp \left\| \widetilde{\boldsymbol{W}_1^1} \right\| \tag{71}
$$

$$
\eta_2 = \Theta(\sqrt{h}) \Rightarrow \left\| \overline{\boldsymbol{W}_2^2} - \widetilde{\boldsymbol{W}_2^1} \right\| \asymp \left\| \widetilde{\boldsymbol{W}_2^1} \right\| \tag{72}
$$

$$
Large\ lr : \eta_1 = \Theta(h) \Rightarrow \left\| \overline{\boldsymbol{W}_1^2} - \widetilde{\boldsymbol{W}_1^1} \right\|_F \asymp \left\| \widetilde{\boldsymbol{W}_1^1} \right\|_F \tag{73}
$$

$$
\eta_2 = \Theta(h) \Rightarrow \left\| \overline{\boldsymbol{W}_2^2} - \widetilde{\boldsymbol{W}_2^1} \right\|_F \asymp \left\| \widetilde{\boldsymbol{W}_2^1} \right\|_F \tag{74}
$$

**Proof of Proposition C.11.** Note that $\widetilde{\boldsymbol{A}_1^1} = \frac{1}{h}\boldsymbol{\beta}^* \boldsymbol{a}^\top \widetilde{\boldsymbol{W}_2^1}^\top$, $\widetilde{\boldsymbol{A}_2^1} = \frac{1}{h}\widetilde{\boldsymbol{W}_1^1}^\top \boldsymbol{\beta}^* \boldsymbol{a}^\top$. Based on Lemma C.5, it is easy to get

$$
\left\| \widetilde{\boldsymbol{W}_1^1} - \boldsymbol{W}_1^0 \right\| \asymp \left\| \boldsymbol{W}_1^0 \right\|, \left\| \widetilde{\boldsymbol{W}_2^1} - \boldsymbol{W}_2^0 \right\| \asymp \left\| \boldsymbol{W}_2^0 \right\|
$$

$$
\left\| \widetilde{\boldsymbol{W}_1^1} - \boldsymbol{W}_1^0 \right\|_F \asymp \left\| \boldsymbol{W}_1^0 \right\|_F, \left\| \widetilde{\boldsymbol{W}_2^1} - \boldsymbol{W}_2^0 \right\|_F \asymp \left\| \boldsymbol{W}_2^0 \right\|_F
$$

We also have $\widetilde{\boldsymbol{B}_1^1} = \frac{1}{h^2}\widetilde{\boldsymbol{W}_1^1}\widetilde{\boldsymbol{W}_2^1}\boldsymbol{a}\boldsymbol{a}^\top\widetilde{\boldsymbol{W}_2^1}^\top$, $\widetilde{\boldsymbol{B}_2^1} = \frac{1}{h}\widetilde{\boldsymbol{W}_1^1}^\top\widetilde{\boldsymbol{W}_1^1}\widetilde{\boldsymbol{W}_2^1}\boldsymbol{a}\boldsymbol{a}^\top$. Thus we can obtain that

$$
\left\| \widetilde{\boldsymbol{A}_1^1} \right\| \asymp \left\| \boldsymbol{A}_1^0 \right\|, \left\| \widetilde{\boldsymbol{A}_1^1} \right\|_F \asymp \left\| \boldsymbol{A}_1^0 \right\|_F, \left\| \widetilde{\boldsymbol{A}_2^1} \right\| \asymp \left\| \boldsymbol{A}_2^0 \right\|, \left\| \widetilde{\boldsymbol{A}_2^1} \right\|_F \asymp \left\| \boldsymbol{A}_2^0 \right\|_F
$$

$$
\left\| \widetilde{\boldsymbol{B}_1^1} \right\| \asymp \left\| \boldsymbol{B}_1^0 \right\|, \left\| \widetilde{\boldsymbol{B}_1^1} \right\|_F \asymp \left\| \boldsymbol{B}_1^0 \right\|_F, \left\| \widetilde{\boldsymbol{B}_2^1} \right\| \asymp \left\| \boldsymbol{B}_2^0 \right\|, \left\| \widetilde{\boldsymbol{B}_2^1} \right\|_F \asymp \left\| \boldsymbol{B}_2^0 \right\|_F
$$

Since $\overline{G_1^1} = \widetilde{B_1^1} - \widetilde{A_1^1}$, $\overline{G_2^1} = \widetilde{B_2^1} - \widetilde{A_2^1}$, we obtain that

$$\left\| \overline{G_1^1} - \widetilde{A_1^1} \right\| \le \frac{1}{\sqrt{h}} \left\| \widetilde{A_1^1} \right\| \le \frac{1}{\sqrt{h}} \left( \overline{G_1^1} + \left\| \overline{G_1^1} - \widetilde{A_1^1} \right\| \right)$$

$$\left\| \overline{G_2^1} - \widetilde{A_2^1} \right\| \le \frac{1}{\sqrt{h}} \left\| \widetilde{A_2^1} \right\| \le \frac{1}{\sqrt{h}} \left( \overline{G_2^1} + \left\| \overline{G_2^1} - \widetilde{A_2^1} \right\| \right).$$

Thus, we get that

$$\left\| \overline{G_1^1} - \widetilde{A_1^1} \right\| \le \frac{1}{\sqrt{h}-1} \left\| \overline{G_1^1} \right\|,$$
$$\left\| \overline{G_2^1} - \widetilde{A_2^1} \right\| \le \frac{1}{\sqrt{h}-1} \left\| \overline{G_2^1} \right\|. \tag{75}$$

Based on this, we can get $\left\| \overline{G_1^1} \right\| \asymp \left\| A_1^0 \right\|$, $\left\| \overline{G_1^1} \right\|_F = \left\| A_1^0 \right\|_F$, $\left\| \overline{G_2^1} \right\| = \left\| A_2^0 \right\|$, $\left\| \overline{G_2^1} \right\|_F = \left\| A_2^0 \right\|_F$.

Since we have $\left\| W_1^0 \right\| \asymp \left\| \widetilde{W_1^1} \right\|$, $\left\| W_1^0 \right\|_F = \left\| \widetilde{W_1^1} \right\|_F$, $\left\| W_2^0 \right\| = \left\| \widetilde{W_2^1} \right\|$, $\left\| W_2^0 \right\|_F = \left\| \widetilde{W_2^1} \right\|_F$, based on Assumption E.1, we have

$$Small\ lr: \eta_1 = \Theta(\sqrt{h}) \Rightarrow \left\| \overline{W_1^2} - \widetilde{W_1^1} \right\| \asymp \left\| \widetilde{W_1^1} \right\| \tag{76}$$

$$\eta_2 = \Theta(\sqrt{h}) \Rightarrow \left\| \overline{W_2^2} - \widetilde{W_2^1} \right\| \asymp \left\| \widetilde{W_2^1} \right\| \tag{77}$$

$$Large\ lr: \eta_1 = \Theta(h) \Rightarrow \left\| \overline{W_1^2} - \widetilde{W_1^1} \right\|_F \asymp \left\| \widetilde{W_1^1} \right\|_F \tag{78}$$

$$\eta_2 = \Theta(h) \Rightarrow \left\| \overline{W_2^2} - \widetilde{W_2^1} \right\|_F \asymp \left\| \widetilde{W_2^1} \right\|_F \tag{79}$$

$$\square$$

## D. Orthogonal initialization

Here we give one-step and two-step test loss under whiten initialization, also we give the gap bound between the exact test loss and the approximate test loss.

**Lemma D.1.** *Consider two stochastic random orthogonal matrices $O_1$, $O_2 \in \mathbb{R}^{h \times h}$ uniformly distributed on the orthogonal group with respect to the Haar measure (i.e. $O_1 O_1^\top = O_1^\top O_1 = I$, $O_2 O_2^\top = O_2^\top O_2 = I$), we have $\mathbb{E}[O_1] = \mathbb{E}[O_2] = 0$, $\mathbb{E}[O_1^2] = \mathbb{E}[O_2^2] = \frac{1}{h} I$, $\mathbb{E}[O_1 O_2] = 0$.*

**Proof of Lemma D.1.** For orthogonal group, one key invariance property is for any fixed $R$ in orthogonal group, the distribution of $QR$ and $RQ$ is same as $Q$. In particular, note that $-I$ is in orthogonal group, therefore, we have

$$Q \stackrel{d}{=} (-I)Q = -Q,$$

so based on this, we take a look at the expectation: $\mathbb{E}[Q] = \mathbb{E}[-Q] = -\mathbb{E}[Q]$, which means we can get $\mathbb{E}[Q_1] = \mathbb{E}[Q_2] = 0$. Also consider each row row (or column) of $Q$ is a random vector uniformly distributed on the unit sphere in $\mathbb{R}^h$. Hence by the definitions of orthogonal group, we have

$$\sum_{a=1}^{h} Q_{ia}^2 = 1 \ \Rightarrow \ \mathbb{E}[Q_{ia}^2] = \frac{1}{h},$$

furthermore, if we consider flipping the sign of one row or one column like left-multiplying by $D = diag(-1, 1, 1, \cdots, 1)$ in orthogonal group, which flips the sign of every entry in the first row, but the distribution is unchanged. Thus, the expectation of any product involving an odd number of the entries from that row must be zero. Similarly for the flipping any column, so we can get unless $i = j$ and $a = b$, $\mathbb{E}[Q_{ia}Q_{jb}] = 0$, so it is easy to get $\mathbb{E}[Q_1^2] = \mathbb{E}[Q_2^2] = \frac{1}{h}I$. Based on above arguments we can deduce that $\mathbb{E}[Q_1 Q_2] = 0$. $\square$

## D.1. Approximate one-step loss under orthogonal initialization for two-layer NN

**Theorem D.2.** *Given Assumption 3.3, 3.2, and in addition assume $\eta_1$ and $\eta_2$ are no more than $O(h\sqrt{h})$, based on Proposition 5.1 and Lemma 5.2, consider the training procedure discussed in Section 3, we obtain the following test loss after one-step and two-step GD update in a two-layer neural network under orthogonal initialization:*

$$
\begin{aligned}
L_{\text{two-layer}}(\boldsymbol{W}_1^1, \boldsymbol{W}_2^1) &= \frac{\eta_1^2}{h^4} + \frac{\eta_2^2}{h^4} + \frac{2\eta_1\eta_2}{h^4} + \frac{\eta_1^2\eta_2^2}{h^7} \\
&\quad - \frac{2\eta_1}{h^2} - \frac{2\eta_2}{h^2} + \frac{1}{h} + \frac{2\eta_1\eta_2}{h^5} + 1 \\
L_{\text{two-layer}}(\boldsymbol{W}_1^2, \boldsymbol{W}_2^2) &= \frac{1}{h}\left(1 + \frac{\eta_1\eta_2}{h^3}\right)^4 + \frac{16\eta_1^2\eta_2^2}{h^7} \\
&\quad + \left(\frac{2(\eta_1 + \eta_2)(\eta_1\eta_2 + h^3)}{h^5} - 1\right)^2 + \left(1 + \frac{\eta_1\eta_2}{h^3}\right)^2 \frac{8\eta_1\eta_2}{h^5}
\end{aligned}
\tag{80}
$$

We prove Theorem 5.3 above by the following two subsection D.1 and D.2.

For orthogonal initialization we follow assumption 3.2 that $n = h = d$.

Here we consider the whiten initialization which make the setting $\boldsymbol{X}^\top\boldsymbol{X} = \boldsymbol{X}\boldsymbol{X}^\top = h\boldsymbol{I}, \boldsymbol{W}_1^{0\top}\boldsymbol{W}_1^0 = \boldsymbol{W}_1^0\boldsymbol{W}_1^{0\top} = \boldsymbol{I}, \boldsymbol{W}_2^{0\top}\boldsymbol{W}_2^0 = \boldsymbol{W}_2^0\boldsymbol{W}_2^{0\top} = \boldsymbol{I}, \boldsymbol{M}^\top\boldsymbol{M} = \boldsymbol{M}\boldsymbol{M}^\top = \frac{1}{h}\boldsymbol{I}$.

We consider a test data $\tilde{\boldsymbol{x}}_0$ under two-layer setting, where $\frac{1}{\sqrt{h}}\tilde{\boldsymbol{x}}_0$ is an random orthogonal vector, we have

$$
\begin{aligned}
& L_{\text{two-layer}}(\boldsymbol{X}, \boldsymbol{W}_1^1, \boldsymbol{W}_2^1, \tilde{\boldsymbol{x}}_0) \\
=& \mathbb{E}_{\boldsymbol{W}_1^0, \boldsymbol{W}_2^0, \boldsymbol{\xi}, \tilde{\boldsymbol{x}}_0, \boldsymbol{X}} \left\| \frac{1}{h}\tilde{\boldsymbol{x}}_0\boldsymbol{W}_1^1\boldsymbol{W}_2^1 - \tilde{\boldsymbol{x}}_0\boldsymbol{M} \right\|_F^2 \\
=& tr\left( \mathbb{E}_{\boldsymbol{W}_1^0, \boldsymbol{W}_2^0, \boldsymbol{\xi}, \tilde{\boldsymbol{x}}_0, \boldsymbol{X}} \left[ \left(\frac{1}{h}\boldsymbol{W}_1^1\boldsymbol{W}_2^1 - \boldsymbol{M}\right)^\top \tilde{\boldsymbol{x}}_0^\top\tilde{\boldsymbol{x}}_0 \left(\frac{1}{h}\boldsymbol{W}_1^1\boldsymbol{W}_2^1 - \boldsymbol{M}\right) \right] \right) \\
=& tr\left( \mathbb{E}_{\boldsymbol{W}_1^0, \boldsymbol{W}_2^0, \boldsymbol{\xi}, \tilde{\boldsymbol{x}}_0, \boldsymbol{X}} \left[ \tilde{\boldsymbol{x}}_0^\top\tilde{\boldsymbol{x}}_0 \left(\frac{1}{h}\boldsymbol{W}_1^1\boldsymbol{W}_2^1 - \boldsymbol{M}\right) \left(\frac{1}{h}\boldsymbol{W}_1^1\boldsymbol{W}_2^1 - \boldsymbol{M}\right)^\top \right] \right) \\
=& tr\left( \mathbb{E}_{\boldsymbol{W}_1^0, \boldsymbol{W}_2^0, \boldsymbol{\xi}, \boldsymbol{X}} \left[ \left(\frac{1}{h}\boldsymbol{W}_1^1\boldsymbol{W}_2^1 - \boldsymbol{M}\right) \left(\frac{1}{h}\boldsymbol{W}_1^1\boldsymbol{W}_2^1 - \boldsymbol{M}\right)^\top \right] \right) \\
=& tr\left( \mathbb{E}_{\boldsymbol{W}_1^0, \boldsymbol{W}_2^0, \boldsymbol{\xi}, \boldsymbol{X}} \left[ \frac{1}{h^2}\boldsymbol{W}_1^1\boldsymbol{W}_2^1\boldsymbol{W}_2^{1\top}\boldsymbol{W}_1^{1\top} \right] \right) \\
& -tr\left( \mathbb{E}_{\boldsymbol{W}_1^0, \boldsymbol{W}_2^0, \boldsymbol{\xi}, \boldsymbol{X}} \left[ \frac{1}{h}\boldsymbol{M}\boldsymbol{W}_2^{1\top}\boldsymbol{W}_1^{1\top} \right] \right) \\
& -tr\left( \mathbb{E}_{\boldsymbol{W}_1^0, \boldsymbol{W}_2^0, \boldsymbol{\xi}, \boldsymbol{X}} \left[ \frac{1}{h}\boldsymbol{W}_1^1\boldsymbol{W}_2^1\boldsymbol{M}^\top \right] \right) \\
& +tr\left( \mathbb{E}\left[ \boldsymbol{M}\boldsymbol{M}^\top \right] \right).
\end{aligned}
\tag{81}
$$

Here we define $L_1, L_2, L_3, L_4$, where

$$
\begin{aligned}
L_1 &= tr\left( \mathbb{E}_{\boldsymbol{W}_1^0, \boldsymbol{W}_2^0, \boldsymbol{\xi}, \boldsymbol{X}} \left[ \frac{1}{h^2}\boldsymbol{W}_1^1\boldsymbol{W}_2^1\boldsymbol{W}_2^{1\top}\boldsymbol{W}_1^{1\top} \right] \right) \\
L_2 &= tr\left( \mathbb{E}_{\boldsymbol{W}_1^0, \boldsymbol{W}_2^0, \boldsymbol{\xi}, \boldsymbol{X}} \left[ \frac{1}{h}\boldsymbol{M}\boldsymbol{W}_2^{1\top}\boldsymbol{W}_1^{1\top} \right] \right) \\
L_3 &= tr\left( \mathbb{E}_{\boldsymbol{W}_1^0, \boldsymbol{W}_2^0, \boldsymbol{\xi}, \boldsymbol{X}} \left[ \frac{1}{h}\boldsymbol{W}_1^1\boldsymbol{W}_2^1\boldsymbol{M}^\top \right] \right) \\
L_4 &= tr\left( \mathbb{E}\left[ \boldsymbol{M}\boldsymbol{M}^\top \right] \right)
\end{aligned}
$$

Thus

$$L_{\text{two-layer}} = L_1 - L_2 - L_3 + L_4$$

We have $L_1 = \sum_{i=1}^{16} T_i$, where

$$T_1 = tr\left(\mathbb{E}_{\boldsymbol{W}_1^0, \boldsymbol{W}_2^0, \boldsymbol{X}}\left[\frac{1}{h^2}\boldsymbol{W}_2^{0\top}\boldsymbol{W}_1^{0\top}\boldsymbol{W}_1^0\boldsymbol{W}_2^0\right]\right) = \frac{1}{h},$$

$$T_2 = tr\left(\mathbb{E}_{\boldsymbol{W}_1^0, \boldsymbol{W}_2^0, \boldsymbol{X}}\left[\frac{\eta_1}{h^4}\boldsymbol{W}_2^{0\top}\boldsymbol{W}_1^{0\top}\boldsymbol{X}^\top\boldsymbol{Y}\boldsymbol{W}_2^{0\top}\boldsymbol{W}_2^0\right]\right) = 0,$$

$$T_3 = tr\left(\mathbb{E}_{\boldsymbol{W}_1^0, \boldsymbol{W}_2^0, \boldsymbol{X}}\left[\frac{\eta_2}{h^4}\boldsymbol{W}_2^{0\top}\boldsymbol{W}_1^{0\top}\boldsymbol{W}_1^0\boldsymbol{W}_1^{0\top}\boldsymbol{X}^\top\boldsymbol{Y}\right]\right) = 0,$$

$$T_4 = tr\left(\mathbb{E}_{\boldsymbol{W}_1^0, \boldsymbol{W}_2^0, \boldsymbol{X}}\left[\frac{\eta_1\eta_2}{h^6}\boldsymbol{W}_2^{0\top}\boldsymbol{W}_1^{0\top}\boldsymbol{X}^\top\boldsymbol{Y}\boldsymbol{W}_2^{0\top}\boldsymbol{W}_1^{0\top}\boldsymbol{X}^\top\boldsymbol{Y}\right]\right) = \frac{\eta_1\eta_2}{h^5},$$

$$T_5 = tr\left(\mathbb{E}_{\boldsymbol{W}_1^0, \boldsymbol{W}_2^0, \boldsymbol{X}}\left[\frac{\eta_1}{h^4}\boldsymbol{W}_2^{0\top}\boldsymbol{W}_2^0\boldsymbol{Y}^\top\boldsymbol{X}\boldsymbol{W}_1^0\boldsymbol{W}_2^0\right]\right) = 0,$$

$$T_6 = tr\left(\mathbb{E}_{\boldsymbol{W}_1^0, \boldsymbol{W}_2^0, \boldsymbol{X}}\left[\frac{\eta_1^2}{h^6}\boldsymbol{W}_2^{0\top}\boldsymbol{W}_2^0\boldsymbol{Y}^\top\boldsymbol{X}\boldsymbol{X}^\top\boldsymbol{Y}\boldsymbol{W}_2^{0\top}\boldsymbol{W}_2^0\right]\right) = \frac{\eta_1^2}{h^4},$$

$$T_7 = tr\left(\mathbb{E}_{\boldsymbol{W}_1^0, \boldsymbol{W}_2^0, \boldsymbol{X}}\left[\frac{\eta_1\eta_2}{h^6}\boldsymbol{W}_2^{0\top}\boldsymbol{W}_2^0\boldsymbol{Y}^\top\boldsymbol{X}\boldsymbol{W}_1^0\boldsymbol{W}_1^{0\top}\boldsymbol{X}^\top\boldsymbol{Y}\right]\right) = \frac{\eta_1\eta_2}{h^4},$$

$$T_8 = tr\left(\mathbb{E}_{\boldsymbol{W}_1^0, \boldsymbol{W}_2^0, \boldsymbol{X}}\left[\frac{\eta_1^2\eta_2}{h^8}\boldsymbol{W}_2^{0\top}\boldsymbol{W}_2^0\boldsymbol{Y}^\top\boldsymbol{X}\boldsymbol{X}^\top\boldsymbol{Y}\boldsymbol{W}_2^{0\top}\boldsymbol{W}_1^{0\top}\boldsymbol{X}^\top\boldsymbol{Y}\right]\right) = 0,$$

$$T_9 = tr\left(\mathbb{E}_{\boldsymbol{W}_1^0, \boldsymbol{W}_2^0, \boldsymbol{X}}\left[\frac{\eta_2}{h^4}\boldsymbol{Y}^\top\boldsymbol{X}\boldsymbol{W}_1^0\boldsymbol{W}_1^{0\top}\boldsymbol{W}_1^0\boldsymbol{W}_2^0\right]\right) = 0,$$

$$T_{10} = tr\left(\mathbb{E}_{\boldsymbol{W}_1^0, \boldsymbol{W}_2^0, \boldsymbol{X}}\left[\frac{\eta_1\eta_2}{h^6}\boldsymbol{Y}^\top\boldsymbol{X}\boldsymbol{W}_1^0\boldsymbol{W}_1^{0\top}\boldsymbol{X}^\top\boldsymbol{Y}\boldsymbol{W}_2^{0\top}\boldsymbol{W}_2^0\right]\right) = \frac{\eta_1\eta_2}{h^4},$$

$$T_{11} = tr\left(\mathbb{E}_{\boldsymbol{W}_1^0, \boldsymbol{W}_2^0, \boldsymbol{X}}\left[\frac{\eta_2^2}{h^6}\boldsymbol{Y}^\top\boldsymbol{X}\boldsymbol{W}_1^0\boldsymbol{W}_1^{0\top}\boldsymbol{W}_1^0\boldsymbol{W}_1^{0\top}\boldsymbol{X}^\top\boldsymbol{Y}\right]\right) = \frac{\eta_2^2}{h^4},$$

$$T_{12} = tr\left(\mathbb{E}_{\boldsymbol{W}_1^0, \boldsymbol{W}_2^0, \boldsymbol{X}}\left[\frac{\eta_1\eta_2^2}{h^8}\boldsymbol{Y}^\top\boldsymbol{X}\boldsymbol{W}_1^0\boldsymbol{W}_1^{0\top}\boldsymbol{X}^\top\boldsymbol{Y}\boldsymbol{W}_2^{0\top}\boldsymbol{W}_1^{0\top}\boldsymbol{X}^\top\boldsymbol{Y}\right]\right) = 0,$$

$$T_{13} = tr\left(\mathbb{E}_{\boldsymbol{W}_1^0, \boldsymbol{W}_2^0, \boldsymbol{X}}\left[\frac{\eta_1\eta_2}{h^6}\boldsymbol{Y}^\top\boldsymbol{X}\boldsymbol{W}_1^0\boldsymbol{W}_2^0\boldsymbol{Y}^\top\boldsymbol{X}\boldsymbol{W}_1^0\boldsymbol{W}_2^0\right]\right) = \frac{\eta_1\eta_2}{h^5},$$

$$T_{14} = tr\left(\mathbb{E}_{\boldsymbol{W}_1^0, \boldsymbol{W}_2^0, \boldsymbol{X}}\left[\frac{\eta_1^2\eta_2}{h^8}\boldsymbol{Y}^\top\boldsymbol{X}\boldsymbol{W}_1^0\boldsymbol{W}_2^0\boldsymbol{Y}^\top\boldsymbol{X}\boldsymbol{X}^\top\boldsymbol{Y}\boldsymbol{W}_2^{0\top}\boldsymbol{W}_2^0\right]\right) = 0,$$

$$T_{15} = tr\left(\mathbb{E}_{\boldsymbol{W}_1^0, \boldsymbol{W}_2^0, \boldsymbol{X}}\left[\frac{\eta_1\eta_2^2}{h^8}\boldsymbol{Y}^\top\boldsymbol{X}\boldsymbol{W}_1^0\boldsymbol{W}_2^0\boldsymbol{Y}^\top\boldsymbol{X}\boldsymbol{W}_1^0\boldsymbol{W}_1^{0\top}\boldsymbol{X}^\top\boldsymbol{Y}\right]\right) = 0,$$

$$T_{16} = tr\left(\mathbb{E}_{\boldsymbol{W}_1^0, \boldsymbol{W}_2^0, \boldsymbol{X}}\left[\frac{\eta_1^2\eta_2^2}{h^{10}}\boldsymbol{Y}^\top\boldsymbol{X}\boldsymbol{W}_1^0\boldsymbol{W}_2^0\boldsymbol{Y}^\top\boldsymbol{X}\boldsymbol{X}^\top\boldsymbol{Y}\boldsymbol{W}_2^{0\top}\boldsymbol{W}_1^{0\top}\boldsymbol{X}^\top\boldsymbol{Y}\right]\right) = \frac{\eta_1^2\eta_2^2}{h^7}.$$

We have $L_2 = \sum_{i=17}^{20} T_i$, where

$$T_{17} = tr\left(\mathbb{E}_{\boldsymbol{W}_1^0, \boldsymbol{W}_2^0, \boldsymbol{X}}\left[\frac{1}{h}\boldsymbol{M}\boldsymbol{W}_2^{0\top}\boldsymbol{W}_1^{0\top}\right]\right) = 0,$$

$$T_{18} = tr\left(\mathbb{E}_{\boldsymbol{W}_1^0, \boldsymbol{W}_2^0, \boldsymbol{X}}\left[\frac{\eta_1}{h^3}\boldsymbol{M}\boldsymbol{W}_2^{0\top}\boldsymbol{W}_2^0\boldsymbol{Y}^\top\boldsymbol{X}\right]\right) = \frac{\eta_1}{h^2},$$

$$T_{19} = tr\left(\mathbb{E}_{\boldsymbol{W}_1^0, \boldsymbol{W}_2^0, \boldsymbol{X}}\left[\frac{\eta_2}{h^3}\boldsymbol{M}\boldsymbol{Y}^\top\boldsymbol{X}\boldsymbol{W}_1^0\boldsymbol{W}_1^{0\top}\right]\right) = \frac{\eta_2}{h^2},$$

$$T_{20} = tr\left(\mathbb{E}_{\boldsymbol{W}_1^0, \boldsymbol{W}_2^0, \boldsymbol{X}}\left[\frac{\eta_1\eta_2}{h^5}\boldsymbol{M}\boldsymbol{Y}^\top\boldsymbol{X}\boldsymbol{W}_1^0\boldsymbol{W}_2^0\boldsymbol{Y}^\top\boldsymbol{X}\right]\right) = 0.$$

We have $L_3 = \sum_{i=21}^{24} T_i$, where

$$T_{21} = tr\left(\mathbb{E}_{\boldsymbol{W}_1^0, \boldsymbol{W}_2^0, \boldsymbol{X}}\left[\frac{1}{h}\boldsymbol{W}_1^0 \boldsymbol{W}_2^0 \boldsymbol{M}^\top\right]\right) = 0,$$

$$T_{22} = tr\left(\mathbb{E}_{\boldsymbol{W}_1^0, \boldsymbol{W}_2^0, \boldsymbol{X}}\left[\frac{\eta_1}{h^3}\boldsymbol{X}^\top \boldsymbol{Y}\boldsymbol{W}_2^{0^\top} \boldsymbol{W}_2^0 \boldsymbol{M}^\top\right]\right) = \frac{\eta_1}{h^2},$$

$$T_{23} = tr\left(\mathbb{E}_{\boldsymbol{W}_1^0, \boldsymbol{W}_2^0, \boldsymbol{X}}\left[\frac{\eta_2}{h^3}\boldsymbol{W}_1^0 \boldsymbol{W}_1^{0^\top} \boldsymbol{X}^\top \boldsymbol{Y}\boldsymbol{M}^\top\right]\right) = \frac{\eta_2}{h^2},$$

$$T_{24} = tr\left(\mathbb{E}_{\boldsymbol{W}_1^0, \boldsymbol{W}_2^0, \boldsymbol{X}}\left[\frac{\eta_1 \eta_2}{h^5}\boldsymbol{X}^\top \boldsymbol{Y}\boldsymbol{W}_2^{0^\top} \boldsymbol{W}_1^{0^\top} \boldsymbol{X}^\top \boldsymbol{Y}\boldsymbol{M}^\top\right]\right) = 0,$$

Based on the above computation, we see that for orthogonal initialization, the one-step test loss for 2-layer NN is

$$L_{\text{two-layer}}(\boldsymbol{X}, \boldsymbol{W}_1^1, \boldsymbol{W}_2^1, \tilde{\boldsymbol{x}}_0) = \frac{\eta_1^2}{h^4} + \frac{\eta_2^2}{h^4} + \frac{2\eta_1 \eta_2}{h^4} + \frac{\eta_1^2 \eta_2^2}{h^7} - \frac{2\eta_1}{h^2} - \frac{2\eta_2}{h^2} + \frac{1}{h} + \frac{2\eta_1 \eta_2}{h^5} + 1 \tag{82}$$

Here, for the one-step updated loss, we consider the following optimization problem, and we assume the following constraint $\eta_1 + \eta_2 = 2h^\alpha$, our goal is to see whether $\eta_1 = \eta_2 = h^\alpha$ is local minima or local maxima.

$$L_{\text{two-layer}}(\boldsymbol{X}, \boldsymbol{W}_1^1, \boldsymbol{W}_2^1, \tilde{\boldsymbol{x}}_0) = \frac{\eta_1^2}{h^4} + \frac{\eta_2^2}{h^4} + \frac{2\eta_1 \eta_2}{h^4} + \frac{\eta_1^2 \eta_2^2}{h^7} - 4h^{\alpha-2} + \frac{1}{h} + \frac{2\eta_1 \eta_2}{h^5} + 1 \tag{83}$$

It is easy to find that $\eta_1 = \eta_2 = h^\alpha$ is a local maxima. $\qquad\square$

### D.2. Approximate two-step loss for two-layer NN under orthogonal initialization

Here we consider the orthogonal initialization which make the setting $\boldsymbol{X}^\top \boldsymbol{X} = \boldsymbol{X}\boldsymbol{X}^\top = h\boldsymbol{I}, \boldsymbol{W}_1^{0^\top} \boldsymbol{W}_1^0 = \boldsymbol{W}_1^0 \boldsymbol{W}_1^{0^\top} = \boldsymbol{I}, \boldsymbol{W}_2^{0^\top} \boldsymbol{W}_2^0 = \boldsymbol{W}_2^0 \boldsymbol{W}_2^{0^\top} = \boldsymbol{I}, \boldsymbol{M}^\top \boldsymbol{M} = \boldsymbol{M}\boldsymbol{M}^\top = \frac{1}{h}\boldsymbol{I}$.

For the simplification, we only consider replacing $\boldsymbol{G}_1$ with $\boldsymbol{A}_1$ and $\boldsymbol{G}_2$ with $\boldsymbol{A}_2$.

$$\boldsymbol{A}_1^0 = \frac{1}{h}\boldsymbol{M}\boldsymbol{W}_2^{0^\top} \qquad\qquad \widetilde{\boldsymbol{A}_1^1} = \frac{1}{h}\boldsymbol{M}\widetilde{\boldsymbol{W}_2^1}^\top$$

$$\boldsymbol{B}_1^0 = \frac{1}{h^2}\boldsymbol{W}_1^0 \qquad\qquad \widetilde{\boldsymbol{B}_1^1} = \frac{1}{h^2}\widetilde{\boldsymbol{W}_1^1}\widetilde{\boldsymbol{W}_2^1}\widetilde{\boldsymbol{W}_2^1}^\top$$

$$\boldsymbol{A}_2^0 = \frac{1}{h}\boldsymbol{W}_1^{0^\top}\boldsymbol{M} \qquad\qquad \widetilde{\boldsymbol{A}_2^1} = \frac{1}{h}\widetilde{\boldsymbol{W}_1^1}^\top\boldsymbol{M}$$

$$\boldsymbol{B}_2^0 = \frac{1}{h^2}\boldsymbol{W}_2^0 \qquad\qquad \widetilde{\boldsymbol{B}_2^1} = \frac{1}{h^2}\widetilde{\boldsymbol{W}_1^1}^\top\widetilde{\boldsymbol{W}_1^1}\widetilde{\boldsymbol{W}_2^1}$$

Thus we have

$$\widetilde{\boldsymbol{W}_1^1} = \boldsymbol{W}_1^0 + \eta_1 \boldsymbol{A}_1^0 = \boldsymbol{W}_1^0 + \frac{\eta_1}{h}\boldsymbol{M}\boldsymbol{W}_2^{0^\top}$$

$$\widetilde{\boldsymbol{W}_2^1} = \boldsymbol{W}_2^0 + \eta_2 \boldsymbol{A}_2^0 = \boldsymbol{W}_2^0 + \frac{\eta_2}{h}\boldsymbol{W}_1^{0^\top}\boldsymbol{M}$$

$$\widetilde{\boldsymbol{W}_1^2} = \widetilde{\boldsymbol{W}_1^1} + \eta_1 \widetilde{\boldsymbol{A}_1^1} = \widetilde{\boldsymbol{W}_1^1} + \frac{\eta_1}{h}\boldsymbol{M}\widetilde{\boldsymbol{W}_2^1}^\top$$

$$= \boldsymbol{W}_1^0 + \frac{2\eta_1}{h}\boldsymbol{M}\boldsymbol{W}_2^{0^\top} + \frac{\eta_1 \eta_2}{h^3}\boldsymbol{W}_1^0$$

$$\widetilde{\boldsymbol{W}_2^2} = \widetilde{\boldsymbol{W}_2^1} + \eta_2 \widetilde{\boldsymbol{A}_2^1} = \widetilde{\boldsymbol{W}_2^1} + \frac{\eta_2}{h}\widetilde{\boldsymbol{W}_1^1}^\top\boldsymbol{M}$$

$$= \boldsymbol{W}_2^0 + \frac{2\eta_2}{h}\boldsymbol{W}_1^{0^\top}\boldsymbol{M} + \frac{\eta_1 \eta_2}{h^3}\boldsymbol{W}_2^0$$

we can derive that

$$\widetilde{\boldsymbol{W}_1^2}\widetilde{\boldsymbol{W}_2^2} = (1 + \frac{\eta_1\eta_2}{h^3})^2\boldsymbol{W}_1^0\boldsymbol{W}_2^0 + \frac{2(\eta_1 + \eta_2)(\eta_1\eta_2 + h^3)}{h^4}\boldsymbol{M} + \frac{4\eta_1\eta_2}{h^2}\boldsymbol{M}\boldsymbol{W}_2^{0\top}\boldsymbol{W}_1^{0\top}\boldsymbol{M}$$

Consider the following loss

$$L_{\text{two-layer}}(\boldsymbol{X}, \boldsymbol{W}_1^2, \boldsymbol{W}_2^2, \tilde{\boldsymbol{x}}_0)$$

$$\approx L_{\text{two-layer}}(\boldsymbol{X}, \widetilde{\boldsymbol{W}_1^2}, \widetilde{\boldsymbol{W}_2^2}, \tilde{\boldsymbol{x}}_0)$$

$$= \mathbb{E}_{\boldsymbol{W}_1^0, \boldsymbol{W}_2^0, \boldsymbol{\xi}, \tilde{\boldsymbol{x}}_0, \boldsymbol{X}}\left\| \frac{1}{h}\tilde{\boldsymbol{x}}_0\widetilde{\boldsymbol{W}_1^2}\widetilde{\boldsymbol{W}_2^2} - \tilde{\boldsymbol{x}}_0\boldsymbol{M}\right\|_F^2$$

$$= tr\left(\mathbb{E}_{\boldsymbol{W}_1^0, \boldsymbol{W}_2^0, \boldsymbol{X}}\left[\frac{1}{h^2}\widetilde{\boldsymbol{W}_1^2}\widetilde{\boldsymbol{W}_2^2}\widetilde{\boldsymbol{W}_2^2}^\top\widetilde{\boldsymbol{W}_1^2}^\top\right]\right)$$

$$- tr\left(\mathbb{E}_{\boldsymbol{W}_1^0, \boldsymbol{W}_2^0, \boldsymbol{X}}\left[\frac{1}{h}\boldsymbol{M}\widetilde{\boldsymbol{W}_2^2}^\top\widetilde{\boldsymbol{W}_1^2}^\top\right]\right)$$

$$- tr\left(\mathbb{E}_{\boldsymbol{W}_1^0, \boldsymbol{W}_2^0, \boldsymbol{X}}\left[\frac{1}{h}\widetilde{\boldsymbol{W}_1^2}\widetilde{\boldsymbol{W}_2^2}\boldsymbol{M}^\top\right]\right)$$

$$+ tr\left(\mathbb{E}\left[\boldsymbol{M}\boldsymbol{M}^\top\right]\right).$$

We first compute $tr\left(\mathbb{E}_{\boldsymbol{W}_1^0, \boldsymbol{W}_2^0, \boldsymbol{X}}\left[\frac{1}{h^2}\widetilde{\boldsymbol{W}_1^2}\widetilde{\boldsymbol{W}_2^2}\widetilde{\boldsymbol{W}_2^2}^\top\widetilde{\boldsymbol{W}_1^2}^\top\right]\right)$, we find that

$$\frac{1}{h^2}\widetilde{\boldsymbol{W}_1^2}\widetilde{\boldsymbol{W}_2^2}\widetilde{\boldsymbol{W}_2^2}^\top\widetilde{\boldsymbol{W}_1^2}^\top = \frac{1}{h^2}(1 + \frac{\eta_1\eta_2}{h^3})^4\boldsymbol{I}_h + (1 + \frac{\eta_1\eta_2}{h^3})^2\frac{2(\eta_1 + \eta_2)(\eta_1\eta_2 + h^3)}{h^6}\boldsymbol{W}_1^0\boldsymbol{W}_2^0\boldsymbol{M}^\top$$

$$+ (1 + \frac{\eta_1\eta_2}{h^3})^2\frac{4\eta_1\eta_2}{h^4}\boldsymbol{W}_1^0\boldsymbol{W}_2^0\boldsymbol{M}^\top\boldsymbol{W}_1^0\boldsymbol{W}_2^0\boldsymbol{M}^\top + (1 + \frac{\eta_1\eta_2}{h^3})^2\frac{2(\eta_1 + \eta_2)(\eta_1\eta_2 + h^3)}{h^6}\boldsymbol{M}\boldsymbol{W}_2^{0\top}\boldsymbol{W}_1^{0\top}$$

$$+ \frac{4(\eta_1 + \eta_2)^2(\eta_1\eta_2 + h^3)^2}{h^{11}}\boldsymbol{I}_h + \frac{8\eta_1\eta_2(\eta_1 + \eta_2)(\eta_1\eta_2 + h^3)}{h^7}\boldsymbol{W}_1^0\boldsymbol{W}_2^0\boldsymbol{M}^\top + \frac{16\eta_1^2\eta_2^2}{h^8}\boldsymbol{I}_h$$

$$+ (1 + \frac{\eta_1\eta_2}{h^3})^2\frac{4\eta_1\eta_2}{h^4}\boldsymbol{M}\boldsymbol{W}_2^{0\top}\boldsymbol{W}_1^{0\top}\boldsymbol{M}\boldsymbol{W}_2^{0\top}\boldsymbol{W}_1^{0\top} + \frac{8\eta_1\eta_2(\eta_1 + \eta_2)(\eta_1\eta_2 + h^3)}{h^9}\boldsymbol{M}\boldsymbol{W}_2^{0\top}\boldsymbol{W}_1^{0\top}.$$

Thus, we have

$$tr\left(\mathbb{E}_{\boldsymbol{W}_1^0, \boldsymbol{W}_2^0, \boldsymbol{\xi}, \boldsymbol{X}}\left[\frac{1}{h^2}\widetilde{\boldsymbol{W}_1^2}\widetilde{\boldsymbol{W}_2^2}\widetilde{\boldsymbol{W}_2^2}^\top\widetilde{\boldsymbol{W}_1^2}^\top\right]\right) = \frac{1}{h}(1 + \frac{\eta_1\eta_2}{h^3})^4 + \frac{4(\eta_1 + \eta_2)^2(\eta_1\eta_2 + h^3)^2}{h^{10}}$$

$$+ \frac{16\eta_1^2\eta_2^2}{h^7} + (1 + \frac{\eta_1\eta_2}{h^3})^2\frac{8\eta_1\eta_2}{h^5}$$

Following the similar way, we get that

$$tr\left(\mathbb{E}_{\boldsymbol{W}_1^0, \boldsymbol{W}_2^0, \boldsymbol{X}}\left[\frac{1}{h}\boldsymbol{M}\widetilde{\boldsymbol{W}_2^2}^\top\widetilde{\boldsymbol{W}_1^2}^\top\right]\right) = \frac{2(\eta_1 + \eta_2)(\eta_1\eta_2 + h^3)}{h^5},$$

$$tr\left(\mathbb{E}_{\boldsymbol{W}_1^0, \boldsymbol{W}_2^0, \boldsymbol{X}}\left[\frac{1}{h}\widetilde{\boldsymbol{W}_1^2}\widetilde{\boldsymbol{W}_2^2}\boldsymbol{M}^\top\right]\right) = \frac{2(\eta_1 + \eta_2)(\eta_1\eta_2 + h^3)}{h^5},$$

$$tr\left(\mathbb{E}\left[\boldsymbol{M}\boldsymbol{M}^\top\right]\right) = 1$$

We have

$$L_{\text{two-layer}}(\boldsymbol{X}, \boldsymbol{W}_1^2, \boldsymbol{W}_2^2, \tilde{\boldsymbol{x}}_0) = \frac{1}{h}(1 + \frac{\eta_1\eta_2}{h^3})^4 + \frac{4(\eta_1 + \eta_2)^2(\eta_1\eta_2 + h^3)^2}{h^{10}} + \frac{16\eta_1^2\eta_2^2}{h^7}$$

$$+ (1 + \frac{\eta_1\eta_2}{h^3})^2\frac{8\eta_1\eta_2}{h^5} - \frac{4(\eta_1 + \eta_2)(\eta_1\eta_2 + h^3)}{h^5} + 1$$

$$L_{\text{two-layer}}(\boldsymbol{X}, \boldsymbol{W}_1^2, \boldsymbol{W}_2^2, \tilde{\boldsymbol{x}}_0) = \frac{1}{h}(1 + \frac{\eta_1 \eta_2}{h^3})^4 + \frac{16\eta_1^2 \eta_2^2}{h^7} + \left(\frac{2(\eta_1 + \eta_2)(\eta_1 \eta_2 + h^3)}{h^5} - 1\right)^2 + (1 + \frac{\eta_1 \eta_2}{h^3})^2 \frac{8\eta_1 \eta_2}{h^5} \quad (84)$$

$\square$

**Corollary D.3.** *Suppose $\eta_1 + \eta_2 = 2h^\alpha$ and we consider $0 < \alpha \leq \frac{3}{2}$. Then, for any $\alpha$ in this range, the point $\eta_1 = \eta_2 = h^\alpha$ is not a local minimum of the loss $L_{two\text{-}layer}(\boldsymbol{W}_1^1, \boldsymbol{W}_2^1)$. Moreover, for $1 < \alpha \leq \frac{3}{2}$, if $h > \max\{h^*, 256\}$, then $\eta_1 = \eta_2 = h^\alpha$ is a local minimum of the loss $L_{two\text{-}layer}(\boldsymbol{W}_1^2, \boldsymbol{W}_2^2)$, where $h^*$ is the root of the following equation:*

$$(1 + o(1))h^{1-\alpha} + 16h^{\alpha-2} + 2h^{-\alpha} + 8h^{\alpha-3} + 6h^{3\alpha-6} - 2 = 0 \quad (85)$$

**Proof of Corollary 5.4.** Here, for the two-step updated loss, we consider the following optimization problem, and we assume that $\eta_1 + \eta_2 = 2h^\alpha$, we want to find whether the local minima for $L_{\text{two-layer}}$ is $\eta_1 = \eta_2 = h^\alpha$.

Since $\eta_1 + \eta_2 = 2h^\alpha$, we have

$$L_{\text{two-layer}}(\boldsymbol{X}, \boldsymbol{W}_1^2, \boldsymbol{W}_2^2, , \tilde{\boldsymbol{x}}_0) = \frac{1}{h}(1 + \frac{\eta_1(2h^\alpha - \eta_1)}{h^3})^4 + \frac{16\eta_1^2(2h^\alpha - \eta_1)^2}{h^7}$$
$$+ \left(\frac{4(\eta_1(2h^\alpha - \eta_1) + h^3)}{h^{5-\alpha}} - 1\right)^2 + (1 + \frac{\eta_1(2h^\alpha - \eta_1)}{h^3})^2 \frac{8\eta_1(2h^\alpha - \eta_1)}{h^5}$$

Taking the derivative, we have

$$L'_{\text{two-layer}}(\boldsymbol{X}, \boldsymbol{W}_1^2, \boldsymbol{W}_2^2, \tilde{\boldsymbol{x}}_0) = 2(h^\alpha - \eta_1)\left[\frac{4}{h^4}(1 + \frac{\eta_1(2h^\alpha - \eta_1)}{h^3})^3 + \frac{32\eta_1(2h^\alpha - \eta_1)}{h^7}\right]$$
$$+ 2(h^\alpha - \eta_1)\left[\frac{8}{h^{5-\alpha}}\left(\frac{4(\eta_1(2h^\alpha - \eta_1) + h^3)}{h^{5-\alpha}} - 1\right)\right]$$
$$+ 2(h^\alpha - \eta_1)\left[\frac{8}{h^5} + \frac{32\eta_1(2h^\alpha - \eta_1)}{h^8} + \frac{24\eta_1^2(2h^\alpha - \eta_1)^2}{h^{11}}\right]$$

If we let $L_{\text{two-layer}}$ is $\eta_1 = \eta_2 = h^\alpha$ be local minima, we must need

- $4 > 5 - \alpha \Rightarrow \alpha > 1$

- $3(2\alpha - 3) - 4 < \alpha - 5 \Rightarrow \alpha < \frac{8}{5}$

- $2(2\alpha - 3) - 4 < \alpha - 5 \Rightarrow \alpha < \frac{5}{3}$

- $2\alpha - 7 < \alpha - 5 \Rightarrow \alpha < 2$

- $4\alpha - 10 < \alpha - 5 \Rightarrow \alpha < \frac{5}{3}$

- $2\alpha - 8 < \alpha - 5 \Rightarrow \alpha < 3$

- $4\alpha - 11 < \alpha - 5 \Rightarrow \alpha < 2$

Take the intersection, we have $1 < \alpha < \frac{8}{5}$. Given the fixed $1 < \alpha < \frac{8}{5}$, we will give how large $h$ is to ensure that $\eta_1 = \eta_2 = h^\alpha$ will are local minima,

**Case 1.** If $\alpha = \frac{3}{2}$, we need
$$8h^{\frac{1}{2}} - 32 - 32 - 64 - o(1) > 0,$$
which means $h > 256 + o(1)$.

**Case 2.** If $1 < \alpha < \frac{3}{2}$, we find $6\alpha - 13 < 4\alpha - 10 < -4$ and $1 - \alpha > \alpha - 2$, so we need

$$(4 + o(1))h^{1-\alpha} + 32h^{\alpha-2} + (32 + o(1))h^{\alpha-2} + 8h^{-\alpha} + 32h^{\alpha-3} + 24h^{3\alpha-6} - 8 < 0,$$

which means

$$(1 + o(1))h^{1-\alpha} + 16h^{\alpha-2} + 2h^{-\alpha} + 8h^{\alpha-3} + 6h^{3\alpha-6} - 2 < 0,$$

**Case 3.** If $\frac{3}{2} < \alpha < \frac{8}{5}$, we find $6\alpha - 13 > 4\alpha - 10 > -4$, so we need

$$(4 + o(1))h^{3(2\alpha-3)-\alpha+1} + 32h^{\alpha-2} + (32 + o(1))h^{3\alpha-5} + 8h^{-\alpha} + 32h^{\alpha-3} + 24h^{3\alpha-6} - 8 < 0,$$

Which means

$$(1 + o(1))h^{5\alpha-8} + 8h^{\alpha-2} + 8h^{3\alpha-5} + 2h^{-\alpha} + 8h^{\alpha-3} + 6h^{3\alpha-6} - 2 < 0.$$

$\square$

### D.3. Bounded Loss Gap for Approximate two-step loss for two-layer NN under orthogonal initialization

Here we consider the orthogonal initialization which make the setting $X^\top X = XX^\top = hI$, $W_1^{0\top} W_1^0 = W_1^0 W_1^{0\top} = I$, $W_2^{0\top} W_2^0 = W_2^0 W_2^{0\top} = I$, $M^\top M = MM^\top = \frac{1}{h}I$.

We define that

$$A_1^0 = \frac{1}{h} M W_2^{0\top} \qquad A_1^1 = \frac{1}{h} M W_2^{1\top} \qquad \widetilde{A_1^1} = \frac{1}{h} M \widetilde{W_2^{1\top}}$$

$$B_1^0 = \frac{1}{h^2} W_1^0 \qquad B_1^1 = \frac{1}{h^2} W_1^1 W_2^1 W_2^{1\top} \qquad \widetilde{B_1^1} = \frac{1}{h^2} \widetilde{W_1^1} \widetilde{W_2^1} \widetilde{W_2^{1\top}}$$

$$A_2^0 = \frac{1}{h} W_1^{0\top} M \qquad A_2^1 = \frac{1}{h} W_1^{1\top} M \qquad \widetilde{A_2^1} = \frac{1}{h} \widetilde{W_1^{1\top}} M$$

$$B_2^0 = \frac{1}{h^2} W_2^0 \qquad B_2^1 = \frac{1}{h^2} W_1^{1\top} W_1^1 W_2^1 \qquad \widetilde{B_2^1} = \frac{1}{h^2} \widetilde{W_1^{1\top}} \widetilde{W_1^1} \widetilde{W_2^1}.$$

And we denote that

$$W_1^1 = W_1^0 + \eta_1 A_1^0 - \eta_1 B_1^0,$$
$$\widetilde{W_1^1} = W_1^0 + \eta_1 A_1^0,$$
$$W_1^2 = W_1^1 + \eta_1 A_1^1 - \eta_1 B_1^1,$$
$$\widetilde{W_1^2} = \widetilde{W_1^1} + \eta_1 \widetilde{A_1^1},$$
$$\overline{W_1^2} = \widetilde{W_1^1} + \eta_1 \widetilde{A_1^1} - \eta_1 \widetilde{B_1^1},$$
$$W_2^1 = W_2^0 + \eta_2 A_2^0 - \eta_2 B_2^0,$$
$$\widetilde{W_2^1} = W_2^0 + \eta_2 A_2^0,$$
$$W_2^2 = W_2^1 + \eta_2 A_2^1 - \eta_2 B_2^1,$$
$$\widetilde{W_2^2} = \widetilde{W_2^1} + \eta_2 \widetilde{A_2^1},$$
$$\overline{W_2^2} = \widetilde{W_2^1} + \eta_2 \widetilde{A_2^1} - \eta_2 \widetilde{B_2^1},$$

**Lemma D.4.** *Under Assumption 3.3 and 3.2, for $\eta_1$, $\eta_2 \le O(h\sqrt{h})$, we have*

$$\left| L_{\text{two-layer}}(W_1^1, W_2^1) - L_{\text{two-layer}}(\widetilde{W_1^1}, \widetilde{W_2^1}) \right| \le O(\frac{1}{h}).$$

$$\left| L_{\text{two-layer}}(W_1^2, W_2^2) - L_{\text{two-layer}}(\widetilde{W_1^2}, \widetilde{W_2^2}) \right| \le O(\frac{1}{\sqrt{h}}).$$

**Proof of Lemma 5.2.** We first give bounded loss gap for approximate one-step loss under orthogonal initialization for two-layer NN. For one step, we consider $\eta_1, \eta_2 \leq O(h^{\frac{3}{2}})$.

$$
\left| \sqrt{L_{\text{two-layer}}(\boldsymbol{X}, \boldsymbol{W}_1^1, \boldsymbol{W}_2^1, \tilde{\boldsymbol{x}}_0)} - \sqrt{L_{\text{two-layer}}(\boldsymbol{X}, \widetilde{\boldsymbol{W}_1^1}, \widetilde{\boldsymbol{W}_2^1}, \tilde{\boldsymbol{x}}_0)} \right|
$$

$$
= \left| \mathbb{E}_{\boldsymbol{W}_1^0, \boldsymbol{W}_2^0, \boldsymbol{\xi}, \tilde{\boldsymbol{x}}_0, \boldsymbol{X}} \left\| \frac{1}{h} \tilde{\boldsymbol{x}}_0 \boldsymbol{W}_1^1 \boldsymbol{W}_2^1 - \tilde{\boldsymbol{x}}_0 \boldsymbol{M} \right\|_F - \mathbb{E}_{\boldsymbol{W}_1^0, \boldsymbol{W}_2^0, \boldsymbol{\xi}, \tilde{\boldsymbol{x}}_0, \boldsymbol{X}} \left\| \frac{1}{h} \tilde{\boldsymbol{x}}_0 \widetilde{\boldsymbol{W}_1^1} \widetilde{\boldsymbol{W}_2^1} - \tilde{\boldsymbol{x}}_0 \boldsymbol{M} \right\|_F \right|
$$

$$
\leq \left| \mathbb{E}_{\boldsymbol{W}_1^0, \boldsymbol{W}_2^0, \boldsymbol{\xi}, \tilde{\boldsymbol{x}}_0, \boldsymbol{X}} \left( \left\| \frac{1}{h} \tilde{\boldsymbol{x}}_0 \boldsymbol{W}_1^1 \boldsymbol{W}_2^1 - \frac{1}{h} \tilde{\boldsymbol{x}}_0 \widetilde{\boldsymbol{W}_1^1} \widetilde{\boldsymbol{W}_2^1} \right\|_F \right) \right|
$$

$$
\leq \left| \mathbb{E}_{\boldsymbol{W}_1^0, \boldsymbol{W}_2^0, \boldsymbol{\xi}, \tilde{\boldsymbol{x}}_0, \boldsymbol{X}} \left( \left\| \frac{1}{h} \tilde{\boldsymbol{x}}_0 \right\|_F \left\| \boldsymbol{W}_1^1 \boldsymbol{W}_2^1 - \widetilde{\boldsymbol{W}_1^1} \widetilde{\boldsymbol{W}_2^1} \right\| \right) \right| \tag{86}
$$

$$
= \frac{1}{\sqrt{h}} \left| \mathbb{E}_{\boldsymbol{W}_1^0, \boldsymbol{W}_2^0, \boldsymbol{\xi}, \tilde{\boldsymbol{x}}_0, \boldsymbol{X}} \left( \left\| \boldsymbol{W}_1^1 \boldsymbol{W}_2^1 - \widetilde{\boldsymbol{W}_1^1} \widetilde{\boldsymbol{W}_2^1} \right\| \right) \right|
$$

$$
= \frac{1}{\sqrt{h}} \left| \mathbb{E}_{\boldsymbol{W}_1^0, \boldsymbol{W}_2^0, \boldsymbol{\xi} \tilde{\boldsymbol{x}}_0, \boldsymbol{X}} \left( \left\| -\eta_1 \boldsymbol{B}_1^0 \boldsymbol{W}_2^0 - \eta_1 \eta_2 \boldsymbol{B}_1^0 \boldsymbol{A}_2^0 - \eta_2 \boldsymbol{W}_1^0 \boldsymbol{B}_2^0 - \eta_1 \eta_2 \boldsymbol{A}_1^0 \boldsymbol{B}_2^0 + \eta_1 \eta_2 \boldsymbol{B}_1^0 \boldsymbol{B}_2^0 \right\| \right) \right|
$$

$$
\leq \frac{1}{\sqrt{h}} \left| \mathbb{E} \left( \left\| \eta_1 \boldsymbol{B}_1^0 \boldsymbol{W}_2^0 \right\| + \left\| \eta_1 \eta_2 \boldsymbol{B}_1^0 \boldsymbol{A}_2^0 \right\| + \left\| \eta_2 \boldsymbol{W}_1^0 \boldsymbol{B}_2^0 \right\| + \left\| \eta_1 \eta_2 \boldsymbol{A}_1^0 \boldsymbol{B}_2^0 \right\| + \left\| \eta_1 \eta_2 \boldsymbol{B}_1^0 \boldsymbol{B}_2^0 \right\| \right) \right|
$$

Consider similar techniques in Lemma C.6, we get that

$$
\mathbb{E}_{\boldsymbol{W}_1^0, \boldsymbol{W}_2^0, \boldsymbol{\xi}, \tilde{\boldsymbol{x}}_0, \boldsymbol{X}} \left\| \eta_1 \boldsymbol{B}_1^0 \boldsymbol{W}_2^0 \right\| \leq \eta_1 \left\| \boldsymbol{B}_1^0 \right\| \left\| \boldsymbol{W}_2^0 \right\| \leq \frac{\eta_1}{h^2},
$$

$$
\mathbb{E}_{\boldsymbol{W}_1^0, \boldsymbol{W}_2^0, \boldsymbol{\xi}, \tilde{\boldsymbol{x}}_0, \boldsymbol{X}} \left\| \eta_1 \eta_2 \boldsymbol{B}_1^0 \boldsymbol{A}_2^0 \right\| \leq \eta_1 \eta_2 \left\| \boldsymbol{B}_1^0 \right\| \left\| \boldsymbol{A}_2^0 \right\| \leq \frac{\eta_1 \eta_2}{h^3 \sqrt{h}},
$$

$$
\mathbb{E}_{\boldsymbol{W}_1^0, \boldsymbol{W}_2^0, \boldsymbol{\xi}, \tilde{\boldsymbol{x}}_0, \boldsymbol{X}} \left\| \eta_2 \boldsymbol{W}_1^0 \right\| \boldsymbol{B}_2^0 \leq \eta_2 \left\| \boldsymbol{B}_2^0 \right\| \left\| \boldsymbol{W}_1^0 \right\| \leq \frac{\eta_2}{h^2},
$$

$$
\mathbb{E}_{\boldsymbol{W}_1^0, \boldsymbol{W}_2^0, \boldsymbol{\xi}, \tilde{\boldsymbol{x}}_0, \boldsymbol{X}} \left\| \eta_1 \eta_2 \boldsymbol{A}_1^0 \boldsymbol{B}_2^0 \right\| \leq \eta_1 \eta_2 \left\| \boldsymbol{B}_2^0 \right\| \left\| \boldsymbol{A}_1^0 \right\| \leq \frac{\eta_1 \eta_2}{h^3 \sqrt{h}},
$$

$$
\mathbb{E}_{\boldsymbol{W}_1^0, \boldsymbol{W}_2^0, \boldsymbol{\xi}, \tilde{\boldsymbol{x}}_0, \boldsymbol{X}} \left\| \eta_1 \eta_2 \boldsymbol{B}_1^0 \boldsymbol{B}_2^0 \right\| \leq \eta_1 \eta_2 \left\| \boldsymbol{B}_2^0 \right\| \left\| \boldsymbol{B}_1^0 \right\| \leq \frac{\eta_1 \eta_2}{h^4},
$$

taking these inequalities into (117), we have

$$
\left| \sqrt{L_{\text{two-layer}}(\boldsymbol{X}, \boldsymbol{W}_1^1, \boldsymbol{W}_2^1, \tilde{\boldsymbol{x}}_0)} - \sqrt{L_{\text{two-layer}}(\boldsymbol{X}, \widetilde{\boldsymbol{W}_1^1}, \widetilde{\boldsymbol{W}_2^1}, \tilde{\boldsymbol{x}}_0)} \right|
$$

$$
\leq \frac{1}{\sqrt{h}} \left| \mathbb{E} \left( \left\| \eta_1 \boldsymbol{B}_1 \boldsymbol{W}_2^0 \right\| + \left\| \eta_1 \eta_2 \boldsymbol{B}_1 \boldsymbol{A}_2 \right\| + \left\| \eta_2 \boldsymbol{W}_1^0 \boldsymbol{B}_2 \right\| + \left\| \eta_1 \eta_2 \boldsymbol{A}_1 \boldsymbol{B}_2 \right\| + \left\| \eta_1 \eta_2 \boldsymbol{B}_1 \boldsymbol{B}_2 \right\| \right) \right| \tag{87}
$$

$$
\leq \frac{2 \eta_1 \eta_2}{h^4} + \frac{\eta_1 + \eta_2}{h^2 \sqrt{h}} + \frac{\eta_1 \eta_2}{h^4 \sqrt{h}} \leq O(\frac{\eta_1 + \eta_2}{h^2 \sqrt{h}})
$$

Also,

$$
\left| \sqrt{L_{\text{two-layer}}(\boldsymbol{X}, \boldsymbol{W}_1^1, \boldsymbol{W}_2^1, \tilde{\boldsymbol{x}}_0)} + \sqrt{L_{\text{two-layer}}(\boldsymbol{X}, \widetilde{\boldsymbol{W}_1^1}, \widetilde{\boldsymbol{W}_2^1}, \tilde{\boldsymbol{x}}_0)} \right| \leq 2 \sqrt{\max(\frac{\eta_1^2}{h^4}, \frac{\eta_1}{h^2}, \frac{\eta_1^2 \eta_2^2}{h^7}, 1)}. \tag{88}
$$

We combine (87), (88) and Assumption E.1, finally we get that

$$
\left| L_{\text{two-layer}}(\boldsymbol{X}, \boldsymbol{W}_1^1, \boldsymbol{W}_2^1, \tilde{\boldsymbol{x}}_0) - L_{\text{two-layer}}(\boldsymbol{X}, \widetilde{\boldsymbol{W}_1^1}, \widetilde{\boldsymbol{W}_2^1}, \tilde{\boldsymbol{x}}_0) \right| \leq O \left( \frac{\eta_1 + \eta_2}{h^2 \sqrt{h}} \sqrt{\max(\frac{\eta_1^2}{h^4}, \frac{\eta_1}{h^2}, \frac{\eta_1^2 \eta_2^2}{h^7}, 1)} \right)
$$

$$
\leq O(\frac{1}{h})
$$

$\square$

We are here considering the 2-step loss under orthogonal initialization for two-layer NN. We have

$$
\begin{aligned}
&\left| \sqrt{L_{\text{two-layer}}(\boldsymbol{X}, \boldsymbol{W}_1^2, \boldsymbol{W}_2^2, \tilde{\boldsymbol{x}}_0)} - \sqrt{L_{\text{two-layer}}(\boldsymbol{X}, \widetilde{\boldsymbol{W}_1^2}, \widetilde{\boldsymbol{W}_2^2}, \tilde{\boldsymbol{x}}_0)} \right| \\
\leq &\left| \sqrt{L_{\text{two-layer}}(\boldsymbol{X}, \boldsymbol{W}_1^2, \boldsymbol{W}_2^2, \tilde{\boldsymbol{x}}_0)} - \sqrt{L_{\text{two-layer}}(\boldsymbol{X}, \overline{\boldsymbol{W}_1^2}, \overline{\boldsymbol{W}_2^2}, \tilde{\boldsymbol{x}}_0)} \right| \\
+ &\left| \sqrt{L_{\text{two-layer}}(\boldsymbol{X}, \overline{\boldsymbol{W}_1^2}, \overline{\boldsymbol{W}_2^2}, \tilde{\boldsymbol{x}}_0)} - \sqrt{L_{\text{two-layer}}(\boldsymbol{X}, \widetilde{\boldsymbol{W}_1^2}, \widetilde{\boldsymbol{W}_2^2}, \tilde{\boldsymbol{x}}_0)} \right|
\end{aligned}
\tag{89}
$$

We first give bounded loss gap for $\left| \sqrt{L_{\text{two-layer}}(\boldsymbol{X}, \overline{\boldsymbol{W}_1^2}, \overline{\boldsymbol{W}_2^2}, \tilde{\boldsymbol{x}}_0)} - \sqrt{L_{\text{two-layer}}(\boldsymbol{X}, \widetilde{\boldsymbol{W}_1^2}, \widetilde{\boldsymbol{W}_2^2}, \tilde{\boldsymbol{x}}_0)} \right|$. Similar to (86), we have

$$
\begin{aligned}
&\left| \sqrt{L_{\text{two-layer}}(\boldsymbol{X}, \overline{\boldsymbol{W}_1^2}, \overline{\boldsymbol{W}_2^2}, \tilde{\boldsymbol{x}}_0)} - \sqrt{L_{\text{two-layer}}(\boldsymbol{X}, \widetilde{\boldsymbol{W}_1^2}, \widetilde{\boldsymbol{W}_2^2}, \tilde{\boldsymbol{x}}_0)} \right| \\
= &\left| \mathbb{E}_{\boldsymbol{W}_1^0, \boldsymbol{W}_2^0, \boldsymbol{\xi}, \tilde{\boldsymbol{x}}_0, \boldsymbol{X}} \left\| \frac{1}{h} \tilde{\boldsymbol{x}}_0 \overline{\boldsymbol{W}_1^2} \overline{\boldsymbol{W}_2^2} - \tilde{\boldsymbol{x}}_0 \boldsymbol{M} \right\|_F - \mathbb{E}_{\boldsymbol{W}_1^0, \boldsymbol{W}_2^0, \boldsymbol{\xi}, \tilde{\boldsymbol{x}}_0, \boldsymbol{X}} \left\| \frac{1}{h} \tilde{\boldsymbol{x}}_0 \widetilde{\boldsymbol{W}_1^2} \widetilde{\boldsymbol{W}_2^2} - \tilde{\boldsymbol{x}}_0 \boldsymbol{M} \right\|_F \right| \\
\leq &\left| \mathbb{E}_{\boldsymbol{W}_1^0, \boldsymbol{W}_2^0, \boldsymbol{\xi}, \tilde{\boldsymbol{x}}_0, \boldsymbol{X}} \left( \left\| \frac{1}{h} \tilde{\boldsymbol{x}}_0 \overline{\boldsymbol{W}_1^2} \overline{\boldsymbol{W}_2^2} - \frac{1}{h} \tilde{\boldsymbol{x}}_0 \widetilde{\boldsymbol{W}_1^2} \widetilde{\boldsymbol{W}_2^2} \right\|_F \right) \right| \\
\leq &\left| \mathbb{E}_{\boldsymbol{W}_1^0, \boldsymbol{W}_2^0, \boldsymbol{\xi}, \tilde{\boldsymbol{x}}_0, \boldsymbol{X}} \left( \left\| \frac{1}{h} \tilde{\boldsymbol{x}}_0 \right\|_F \left\| \overline{\boldsymbol{W}_1^2} \overline{\boldsymbol{W}_2^2} - \widetilde{\boldsymbol{W}_1^2} \widetilde{\boldsymbol{W}_2^2} \right\| \right) \right| \\
= &\frac{1}{\sqrt{h}} \left| \mathbb{E}_{\boldsymbol{W}_1^0, \boldsymbol{W}_2^0, \boldsymbol{\xi}, \tilde{\boldsymbol{x}}_0, \boldsymbol{X}} \left( \left\| \overline{\boldsymbol{W}_1^2} \overline{\boldsymbol{W}_2^2} - \widetilde{\boldsymbol{W}_1^2} \widetilde{\boldsymbol{W}_2^2} \right\| \right) \right| \\
= &\frac{1}{\sqrt{h}} \left| \mathbb{E}_{\boldsymbol{W}_1^0, \boldsymbol{W}_2^0, \boldsymbol{\xi} \tilde{\boldsymbol{x}}_0, \boldsymbol{X}} \left( \left\| -\eta_1 \widetilde{\boldsymbol{B}_1^1} \widetilde{\boldsymbol{W}_2^1} - \eta_1 \eta_2 \widetilde{\boldsymbol{B}_1^1} \widetilde{\boldsymbol{A}_2^1} - \eta_2 \widetilde{\boldsymbol{W}_1^1} \widetilde{\boldsymbol{B}_2^1} - \eta_1 \eta_2 \widetilde{\boldsymbol{A}_1^1} \widetilde{\boldsymbol{B}_2^1} + \eta_1 \eta_2 \widetilde{\boldsymbol{B}_1^1} \widetilde{\boldsymbol{B}_2^1} \right\| \right) \right| \\
\leq &\frac{1}{\sqrt{h}} \left| \mathbb{E} \left( \left\| \eta_1 \widetilde{\boldsymbol{B}_1^1} \widetilde{\boldsymbol{W}_2^1} \right\| + \left\| \eta_1 \eta_2 \widetilde{\boldsymbol{B}_1^1} \widetilde{\boldsymbol{A}_2^1} \right\| + \left\| \eta_2 \widetilde{\boldsymbol{W}_1^1} \widetilde{\boldsymbol{B}_2^1} \right\| + \left\| \eta_1 \eta_2 \widetilde{\boldsymbol{A}_1^1} \widetilde{\boldsymbol{B}_2^1} \right\| + \left\| \eta_1 \eta_2 \widetilde{\boldsymbol{B}_1^1} \widetilde{\boldsymbol{B}_2^1} \right\| \right) \right|
\end{aligned}
\tag{90}
$$

We know that

$$
\begin{aligned}
\widetilde{\boldsymbol{W}_1^1} &= \boldsymbol{W}_1^0 + \eta_1 \boldsymbol{A}_1^0 & \widetilde{\boldsymbol{W}_2^1} &= \boldsymbol{W}_2^0 + \eta_2 \boldsymbol{A}_2^0 \\
\boldsymbol{A}_1^0 &= \frac{1}{h} \boldsymbol{M} \boldsymbol{W}_2^{0\top} & \widetilde{\boldsymbol{A}_1^1} &= \frac{1}{h} \boldsymbol{M} \widetilde{\boldsymbol{W}_2^1}^\top \\
\boldsymbol{B}_1^0 &= \frac{1}{h^2} \boldsymbol{W}_1^0 & \widetilde{\boldsymbol{B}_1^1} &= \frac{1}{h^2} \widetilde{\boldsymbol{W}_1^1} \widetilde{\boldsymbol{W}_2^1} \widetilde{\boldsymbol{W}_2^1}^\top \\
\boldsymbol{A}_2^0 &= \frac{1}{h} \boldsymbol{W}_1^{0\top} \boldsymbol{M} & \widetilde{\boldsymbol{A}_2^1} &= \frac{1}{h} \widetilde{\boldsymbol{W}_1^1}^\top \boldsymbol{M} \\
\boldsymbol{B}_2^0 &= \frac{1}{h^2} \boldsymbol{W}_2^0 & \widetilde{\boldsymbol{B}_2^1} &= \frac{1}{h^2} \widetilde{\boldsymbol{W}_1^1}^\top \widetilde{\boldsymbol{W}_1^1} \widetilde{\boldsymbol{W}_2^1}.
\end{aligned}
$$

Thus, we have

$$\widetilde{A}_1^1 = \frac{1}{h} M W_2^{0\top} + \frac{\eta_2}{h^3} W_1^0$$

$$\widetilde{A}_2^1 = \frac{1}{h} W_1^{0\top} M + \frac{\eta_2}{h^3} W_2^0$$

$$\widetilde{B}_1^1 = \left( \frac{1}{h^2} + \frac{\eta_1 \eta_2 + \eta_2^2}{h^5} \right) W_1^0 + \left( \frac{\eta_1 + \eta_2}{h^3} + \frac{\eta_1 \eta_2^2}{h^6} \right) M W_2^{0\top}$$

$$\qquad + \frac{\eta_2}{h^3} W_1^0 W_2^0 M^\top W_1^0 + \frac{\eta_1 \eta_2}{h^4} M W_2^{0\top} W_1^{0\top} M W_2^{0\top}$$

$$\widetilde{B}_2^1 = \left( \frac{1}{h^2} + \frac{\eta_1 \eta_2 + \eta_1^2}{h^5} \right) W_2^0 + \left( \frac{\eta_1 + \eta_2}{h^3} + \frac{\eta_1^2 \eta_2}{h^6} \right) W_1^{0\top} M$$

$$\qquad + \frac{\eta_1}{h^3} W_2^0 M^\top W_1^0 W_2^0 + \frac{\eta_1 \eta_2}{h^4} W_1^{0\top} M W_2^{0\top} W_1^{0\top} M$$

We consider $\eta_1, \eta_2 \leq O(h^{\frac{3}{2}})$, it is easy to find that

$$\left\| \widetilde{W_1^1} \right\| \leq \| W_1^0 \| + \frac{\eta_1}{h} \| M \| \| W_2^0 \|$$

$$\leq O(1) + \frac{\eta_1}{h\sqrt{h}} = O(1)$$

$$\left\| \widetilde{W_2^1} \right\| \leq \| W_2^0 \| + \frac{\eta_1}{h} \| M \| \| W_2^0 \|$$

$$\leq O(1) + \frac{\eta_2}{h\sqrt{h}} = O(1)$$

$$\left\| \widetilde{A_1^1} \right\| \leq \frac{1}{h} \| M \| \left\| \widetilde{W_2^1} \right\| \leq O(\frac{1}{h\sqrt{h}})$$

$$\left\| \widetilde{A_2^1} \right\| \leq \frac{1}{h} \| M \| \left\| \widetilde{W_1^1} \right\| \leq O(\frac{1}{h\sqrt{h}})$$

$$\left\| \widetilde{B_1^1} \right\| \leq \frac{1}{h^2} \left\| \widetilde{W_1^1} \right\| \left\| \widetilde{W_2^1} \right\|^2 \leq O(\frac{1}{h^2})$$

$$\left\| \widetilde{B_2^1} \right\| \leq \frac{1}{h^2} \left\| \widetilde{W_2^1} \right\| \left\| \widetilde{W_1^1} \right\|^2 \leq O(\frac{1}{h^2}).$$

Combining (90), we have

$$\frac{1}{\sqrt{h}} \left| \mathbb{E} \left( \left\| \eta_1 \widetilde{B}_1^1 \widetilde{W}_2^1 \right\| + \left\| \eta_1 \eta_2 \widetilde{B}_1^1 \widetilde{A}_2^1 \right\| + \left\| \eta_2 \widetilde{W}_1^1 \widetilde{B}_2^1 \right\| + \left\| \eta_1 \eta_2 \widetilde{A}_1^1 \widetilde{B}_2^1 \right\| + \left\| \eta_1 \eta_2 \widetilde{B}_1^1 \widetilde{B}_2^1 \right\| \right) \right|$$

$$\leq \frac{2\eta_1 \eta_2}{h^4} + \frac{\eta_1 + \eta_2}{h^2 \sqrt{h}} + \frac{\eta_1 \eta_2}{h^4 \sqrt{h}} \leq O(\frac{\eta_1 + \eta_2}{h^2 \sqrt{h}})$$

Follow the same way to (88), we can obtain that

$$\left| \sqrt{L_{\text{two-layer}}(X, \overline{W_1^2}, \overline{W_2^2}, \tilde{x}_0)} + \sqrt{L_{\text{two-layer}}(X, \widetilde{W_1^2}, \widetilde{W_2^2}, \tilde{x}_0)} \right| \leq O(1).$$

Finally we get that

$$\left| L_{\text{two-layer}}(X, \overline{W_1^2}, \overline{W_2^2}, \tilde{x}_0) - L_{\text{two-layer}}(X, \widetilde{W_1^2}, \widetilde{W_2^2}, \tilde{x}_0) \right| \leq O\left( \frac{\eta_1 + \eta_2}{h^2 \sqrt{h}} \right) \leq O(\frac{1}{h})$$

We now give bounded loss gap for $\left| \sqrt{L_{\text{two-layer}}(X, W_1^2, W_2^2, \tilde{x}_0)} - \sqrt{L_{\text{two-layer}}(X, \overline{W_1^2}, \overline{W_2^2}, \tilde{x}_0)} \right|$.

We have

$$
\begin{aligned}
\boldsymbol{W}_1^2 &= \boldsymbol{W}_1^1 + \eta_1 \boldsymbol{A}_1^1 - \eta_1 \boldsymbol{B}_1^1 = \boldsymbol{W}_1^0 + \eta_1 \boldsymbol{A}_1^0 - \eta_1 \boldsymbol{B}_1^0 + \eta_1 \boldsymbol{A}_1^1 - \eta_1 \boldsymbol{B}_1^1, \\
\overline{\boldsymbol{W}_1^2} &= \widetilde{\boldsymbol{W}_1^1} + \eta_1 \widetilde{\boldsymbol{A}_1^1} - \eta_1 \widetilde{\boldsymbol{B}_1^1} = \boldsymbol{W}_1^0 + \eta_1 \boldsymbol{A}_1^0 + \eta_1 \widetilde{\boldsymbol{A}_1^1} - \eta_1 \widetilde{\boldsymbol{B}_1^1}, \\
\boldsymbol{W}_2^2 &= \boldsymbol{W}_2^1 + \eta_2 \boldsymbol{A}_2^1 - \eta_2 \boldsymbol{B}_2^1 = \boldsymbol{W}_2^0 + \eta_2 \boldsymbol{A}_2^0 - \eta_2 \boldsymbol{B}_2^0 + \eta_2 \boldsymbol{A}_2^1 - \eta_2 \boldsymbol{B}_2^1, \\
\overline{\boldsymbol{W}_2^2} &= \widetilde{\boldsymbol{W}_2^1} + \eta_2 \widetilde{\boldsymbol{A}_2^1} - \eta_2 \widetilde{\boldsymbol{B}_2^1} = \boldsymbol{W}_2^0 + \eta_2 \boldsymbol{A}_2^0 + \eta_2 \widetilde{\boldsymbol{A}_2^1} - \eta_2 \widetilde{\boldsymbol{B}_2^1},
\end{aligned}
$$

For $\boldsymbol{W}_1^2 \boldsymbol{W}_2^2$, we have

$$
\begin{aligned}
\boldsymbol{W}_1^2 \boldsymbol{W}_2^2 = {}& \boldsymbol{W}_1^0 \boldsymbol{W}_2^0 + \eta_1 \boldsymbol{A}_1^0 \boldsymbol{W}_2^0 - \eta_1 \boldsymbol{B}_1^0 \boldsymbol{W}_2^0 + \eta_1 \boldsymbol{A}_1^1 \boldsymbol{W}_2^0 - \eta_1 \boldsymbol{B}_1^1 \boldsymbol{W}_2^0 \\
& + \eta_2 \boldsymbol{W}_1^0 \boldsymbol{A}_2^0 + \eta_1 \eta_2 \boldsymbol{A}_1^0 \boldsymbol{A}_2^0 - \eta_1 \eta_2 \boldsymbol{B}_1^0 \boldsymbol{A}_2^0 + \eta_1 \eta_2 \boldsymbol{A}_1^1 \boldsymbol{A}_2^0 - \eta_1 \eta_2 \boldsymbol{B}_1^1 \boldsymbol{A}_2^0 \\
& - \eta_2 \boldsymbol{W}_1^0 \boldsymbol{B}_2^0 - \eta_1 \eta_2 \boldsymbol{A}_1^0 \boldsymbol{B}_2^0 + \eta_1 \eta_2 \boldsymbol{B}_1^0 \boldsymbol{B}_2^0 - \eta_1 \eta_2 \boldsymbol{A}_1^1 \boldsymbol{B}_2^0 + \eta_1 \eta_2 \boldsymbol{B}_1^1 \boldsymbol{B}_2^0 \\
& + \eta_2 \boldsymbol{W}_1^0 \boldsymbol{A}_2^1 + \eta_1 \eta_2 \boldsymbol{A}_1^0 \boldsymbol{A}_2^1 - \eta_1 \eta_2 \boldsymbol{B}_1^0 \boldsymbol{A}_2^1 + \eta_1 \eta_2 \boldsymbol{A}_1^1 \boldsymbol{A}_2^1 - \eta_1 \eta_2 \boldsymbol{B}_1^1 \boldsymbol{A}_2^1 \\
& - \eta_2 \boldsymbol{W}_1^0 \boldsymbol{B}_2^1 - \eta_1 \eta_2 \boldsymbol{A}_1^0 \boldsymbol{B}_2^1 + \eta_1 \eta_2 \boldsymbol{B}_1^0 \boldsymbol{B}_2^1 - \eta_1 \eta_2 \boldsymbol{A}_1^1 \boldsymbol{B}_2^1 + \eta_1 \eta_2 \boldsymbol{B}_1^1 \boldsymbol{B}_2^1,
\end{aligned}
\tag{91}
$$

For $\overline{\boldsymbol{W}_1^2 \boldsymbol{W}_2^2}$, we have

$$
\begin{aligned}
\overline{\boldsymbol{W}_1^2 \boldsymbol{W}_2^2} = {}& \boldsymbol{W}_1^0 \boldsymbol{W}_2^0 + \eta_1 \boldsymbol{A}_1^0 \boldsymbol{W}_2^0 + \eta_1 \widetilde{\boldsymbol{A}_1^1} \boldsymbol{W}_2^0 - \eta_1 \widetilde{\boldsymbol{B}_1^1} \boldsymbol{W}_2^0 \\
& + \eta_2 \boldsymbol{W}_1^0 \boldsymbol{A}_2^0 + \eta_1 \eta_2 \boldsymbol{A}_1^0 \boldsymbol{A}_2^0 + \eta_1 \eta_2 \widetilde{\boldsymbol{A}_1^1} \boldsymbol{A}_2^0 - \eta_1 \eta_2 \widetilde{\boldsymbol{B}_1^1} \boldsymbol{A}_2^0 \\
& + \eta_2 \boldsymbol{W}_1^0 \widetilde{\boldsymbol{A}_2^1} + \eta_1 \eta_2 \boldsymbol{A}_1^0 \widetilde{\boldsymbol{A}_2^1} + \eta_1 \eta_2 \widetilde{\boldsymbol{A}_1^1} \widetilde{\boldsymbol{A}_2^1} - \eta_1 \eta_2 \widetilde{\boldsymbol{B}_1^1} \widetilde{\boldsymbol{A}_2^1} \\
& - \eta_2 \boldsymbol{W}_1^0 \widetilde{\boldsymbol{B}_2^1} - \eta_1 \eta_2 \boldsymbol{A}_1^0 \widetilde{\boldsymbol{B}_2^1} - \eta_1 \eta_2 \widetilde{\boldsymbol{A}_1^1} \widetilde{\boldsymbol{B}_2^1} + \eta_1 \eta_2 \widetilde{\boldsymbol{B}_1^1} \widetilde{\boldsymbol{B}_2^1}
\end{aligned}
\tag{92}
$$

Based on (91) and (92), we have

$$
\begin{aligned}
\boldsymbol{W}_1^2 \boldsymbol{W}_2^2 - \overline{\boldsymbol{W}_1^2 \boldsymbol{W}_2^2} = {}& -\eta_1 \boldsymbol{B}_1^0 \boldsymbol{W}_2^0 + \eta_1 (\boldsymbol{A}_1^1 - \widetilde{\boldsymbol{A}_1^1}) \boldsymbol{W}_2^0 - \eta_1 (\boldsymbol{B}_1^1 - \widetilde{\boldsymbol{B}_1^1}) \boldsymbol{W}_2^0 \\
& - \eta_1 \eta_2 \boldsymbol{B}_1^0 \boldsymbol{A}_2^0 + \eta_1 \eta_2 (\boldsymbol{A}_1^1 - \widetilde{\boldsymbol{A}_1^1}) \boldsymbol{A}_2^0 - \eta_1 \eta_2 (\boldsymbol{B}_1^1 - \widetilde{\boldsymbol{B}_1^1}) \boldsymbol{A}_2^0 \\
& + \eta_2 \boldsymbol{W}_1^0 (\boldsymbol{A}_2^1 - \widetilde{\boldsymbol{A}_2^1}) + \eta_1 \eta_2 \boldsymbol{A}_1^0 (\boldsymbol{A}_2^1 - \widetilde{\boldsymbol{A}_2^1}) - \eta_1 \eta_2 \boldsymbol{B}_1^0 \boldsymbol{A}_2^1 \\
& + \eta_1 \eta_2 \boldsymbol{A}_1^1 \boldsymbol{A}_2^1 - \eta_1 \eta_2 \boldsymbol{B}_1^1 \boldsymbol{A}_2^1 - \eta_1 \eta_2 \widetilde{\boldsymbol{A}_1^1} \widetilde{\boldsymbol{A}_2^1} + \eta_1 \eta_2 \widetilde{\boldsymbol{B}_1^1} \widetilde{\boldsymbol{A}_2^1} \\
& + \eta_2 \boldsymbol{W}_1^0 (\boldsymbol{B}_2^1 - \widetilde{\boldsymbol{B}_2^1}) + \eta_1 \eta_2 \boldsymbol{A}_1^0 (\boldsymbol{B}_2^1 - \widetilde{\boldsymbol{B}_2^1}) + \eta_1 \eta_2 \boldsymbol{B}_1^0 \boldsymbol{B}_2^1 \\
& - \eta_1 \eta_2 \boldsymbol{A}_1^1 \boldsymbol{B}_2^1 + \eta_1 \eta_2 \boldsymbol{B}_1^1 \boldsymbol{B}_2^1 + \eta_1 \eta_2 \widetilde{\boldsymbol{A}_1^1} \widetilde{\boldsymbol{B}_2^1} - \eta_1 \eta_2 \widetilde{\boldsymbol{B}_1^1} \widetilde{\boldsymbol{B}_2^1} \\
& - \eta_2 \boldsymbol{W}_1^0 \boldsymbol{B}_2^0 - \eta_1 \eta_2 \boldsymbol{A}_1^0 \boldsymbol{B}_2^0 + \eta_1 \eta_2 \boldsymbol{B}_1^0 \boldsymbol{B}_2^0 - \eta_1 \eta_2 \boldsymbol{A}_1^1 \boldsymbol{B}_2^0 + \eta_1 \eta_2 \boldsymbol{B}_1^1 \boldsymbol{B}_2^0
\end{aligned}
$$

We know that

$$\|\boldsymbol{A}_1^0\| \le O(\frac{1}{h\sqrt{h}}), \|\boldsymbol{A}_2^0\| \le O(\frac{1}{h\sqrt{h}}), \|\boldsymbol{B}_1^0\| \le O(\frac{1}{h^2}), \|\boldsymbol{B}_2^0\| \le O(\frac{1}{h^2})$$

$$\left\|\widetilde{\boldsymbol{W}_1^1}\right\| \le \|\boldsymbol{W}_1^0\| + \frac{\eta_1}{h}\|\boldsymbol{M}\|\|\boldsymbol{W}_2^0\| \le O(1) + \frac{\eta_1}{h\sqrt{h}} = O(1)$$

$$\left\|\widetilde{\boldsymbol{W}_2^1}\right\| \le \|\boldsymbol{W}_2^0\| + \frac{\eta_1}{h}\|\boldsymbol{M}\|\|\boldsymbol{W}_2^0\| \le O(1) + \frac{\eta_2}{h\sqrt{h}} = O(1)$$

$$\|\boldsymbol{W}_1^1\| \le \|\boldsymbol{W}_1^0\| + \eta_1\|\boldsymbol{A}_1^0\| + \eta_1\|\boldsymbol{B}_1^0\| \le O(1)$$

$$\|\boldsymbol{W}_2^1\| \le \|\boldsymbol{W}_2^0\| + \eta_2\|\boldsymbol{A}_2^0\| + \eta_2\|\boldsymbol{B}_2^0\| \le O(1)$$

$$\left\|\widetilde{\boldsymbol{A}_1^1}\right\| \le \frac{1}{h}\|\boldsymbol{M}\|\left\|\widetilde{\boldsymbol{W}_2^1}\right\| \le O(\frac{1}{h\sqrt{h}}), \|\boldsymbol{A}_1^1\| \le \frac{1}{h}\|\boldsymbol{M}\|\|\boldsymbol{W}_2^1\| \le O(\frac{1}{h\sqrt{h}})$$

$$\left\|\widetilde{\boldsymbol{A}_2^1}\right\| \le \frac{1}{h}\|\boldsymbol{M}\|\left\|\widetilde{\boldsymbol{W}_1^1}\right\| \le O(\frac{1}{h\sqrt{h}}), \|\boldsymbol{A}_2^1\| \le \frac{1}{h}\|\boldsymbol{M}\|\|\boldsymbol{W}_1^1\| \le O(\frac{1}{h\sqrt{h}})$$

$$\left\|\widetilde{\boldsymbol{B}_1^1}\right\| \le \frac{1}{h^2}\left\|\widetilde{\boldsymbol{W}_1^1}\right\|\left\|\widetilde{\boldsymbol{W}_2^1}\right\|^2 \le O(\frac{1}{h^2}), \|\boldsymbol{B}_1^1\| \le \frac{1}{h^2}\|\boldsymbol{W}_1^1\|\|\boldsymbol{W}_2^1\|^2 \le O(\frac{1}{h^2})$$

$$\left\|\widetilde{\boldsymbol{B}_2^1}\right\| \le \frac{1}{h^2}\left\|\widetilde{\boldsymbol{W}_2^1}\right\|\left\|\widetilde{\boldsymbol{W}_1^1}\right\|^2 \le O(\frac{1}{h^2}), \|\boldsymbol{B}_2^1\| \le \frac{1}{h^2}\|\boldsymbol{W}_2^1\|\|\boldsymbol{W}_1^1\|^2 \le O(\frac{1}{h^2}).$$

We find that

$$\left\|\boldsymbol{W}_1^2\boldsymbol{W}_2^2 - \overline{\boldsymbol{W}_1^2\boldsymbol{W}_2^2}\right\| \le 3\frac{\eta_1}{h^2} + 2\frac{\eta_1}{h\sqrt{h}} + 12\frac{\eta_1\eta_2}{h^3\sqrt{h}} + 6\frac{\eta_1\eta_2}{h^3} + 5\frac{\eta_1\eta_2}{h^4} + 2\frac{\eta_2}{h\sqrt{h}} + 3\frac{\eta_2}{h^2} \le O(\frac{\eta_1+\eta_2}{h\sqrt{h}}) \qquad (93)$$

Similar to (90)

$$
\begin{aligned}
&\left|\sqrt{L_{\text{two-layer}}(\boldsymbol{X}, \boldsymbol{W}_1^2, \boldsymbol{W}_2^2, \tilde{\boldsymbol{x}}_0)} - \sqrt{L_{\text{two-layer}}(\boldsymbol{X}, \overline{\boldsymbol{W}_1^2}, \overline{\boldsymbol{W}_2^2}, \tilde{\boldsymbol{x}}_0)}\right| \\
&= \left|\mathbb{E}_{\boldsymbol{W}_1^0,\boldsymbol{W}_2^0,\boldsymbol{\xi},\tilde{\boldsymbol{x}}_0,\boldsymbol{X}}\left\|\frac{1}{h}\tilde{\boldsymbol{x}}_0\overline{\boldsymbol{W}_1^2\boldsymbol{W}_2^2} - \tilde{\boldsymbol{x}}_0\boldsymbol{M}\right\|_F - \mathbb{E}_{\boldsymbol{W}_1^0,\boldsymbol{W}_2^0,\boldsymbol{\xi},\tilde{\boldsymbol{x}}_0,\boldsymbol{X}}\left\|\frac{1}{h}\tilde{\boldsymbol{x}}_0\widetilde{\boldsymbol{W}_1^2}\widetilde{\boldsymbol{W}_2^2} - \tilde{\boldsymbol{x}}_0\boldsymbol{M}\right\|_F\right| \\
&\le \left|\mathbb{E}_{\boldsymbol{W}_1^0,\boldsymbol{W}_2^0,\boldsymbol{\xi},\tilde{\boldsymbol{x}}_0,\boldsymbol{X}}\left(\left\|\frac{1}{h}\tilde{\boldsymbol{x}}_0\overline{\boldsymbol{W}_1^2\boldsymbol{W}_2^2} - \frac{1}{h}\tilde{\boldsymbol{x}}_0\widetilde{\boldsymbol{W}_1^2}\widetilde{\boldsymbol{W}_2^2}\right\|_F\right)\right| \\
&\le \left|\mathbb{E}_{\boldsymbol{W}_1^0,\boldsymbol{W}_2^0,\boldsymbol{\xi},\tilde{\boldsymbol{x}}_0,\boldsymbol{X}}\left(\left\|\frac{1}{h}\tilde{\boldsymbol{x}}_0\right\|_F\left\|\overline{\boldsymbol{W}_1^2\boldsymbol{W}_2^2} - \widetilde{\boldsymbol{W}_1^2}\widetilde{\boldsymbol{W}_2^2}\right\|\right)\right| \\
&= \frac{1}{\sqrt{h}}\left|\mathbb{E}_{\boldsymbol{W}_1^0,\boldsymbol{W}_2^0,\boldsymbol{\xi},\tilde{\boldsymbol{x}}_0,\boldsymbol{X}}\left(\left\|\overline{\boldsymbol{W}_1^2\boldsymbol{W}_2^2} - \widetilde{\boldsymbol{W}_1^2}\widetilde{\boldsymbol{W}_2^2}\right\|\right)\right| \\
&\le O(\frac{\eta_1+\eta_2}{h^2})
\end{aligned}
\qquad (94)
$$

Follow the same way to (88), we can obtain that

$$\left|\sqrt{L_{\text{two-layer}}(\boldsymbol{X}, \boldsymbol{W}_1^2, \boldsymbol{W}_2^2, \tilde{\boldsymbol{x}}_0)} + \sqrt{L_{\text{two-layer}}(\boldsymbol{X}, \overline{\boldsymbol{W}_1^2}, \overline{\boldsymbol{W}_2^2}, \tilde{\boldsymbol{x}}_0)}\right| \le O(1).$$

Finally we get that

$$\left|L_{\text{two-layer}}(\boldsymbol{X}, \boldsymbol{W}_1^2, \boldsymbol{W}_2^2, \tilde{\boldsymbol{x}}_0) - L_{\text{two-layer}}(\boldsymbol{X}, \overline{\boldsymbol{W}_1^2}, \overline{\boldsymbol{W}_2^2}, \tilde{\boldsymbol{x}}_0)\right| \le O\left(\frac{\eta_1+\eta_2}{h^2}\right) \le O(\frac{1}{\sqrt{h}})$$

Thus, due to (89), we have

$$\left| \sqrt{L_{\text{two-layer}}(\boldsymbol{X}, \boldsymbol{W}_1^2, \boldsymbol{W}_2^2, \tilde{\boldsymbol{x}}_0)} - \sqrt{L_{\text{two-layer}}(\boldsymbol{X}, \widetilde{\boldsymbol{W}_1^2}, \widetilde{\boldsymbol{W}_2^2}, \tilde{\boldsymbol{x}}_0)} \right|$$

$$\leq \left| \sqrt{L_{\text{two-layer}}(\boldsymbol{X}, \boldsymbol{W}_1^2, \boldsymbol{W}_2^2, \tilde{\boldsymbol{x}}_0)} - \sqrt{L_{\text{two-layer}}(\boldsymbol{X}, \overline{\boldsymbol{W}_1^2}, \overline{\boldsymbol{W}_2^2}, \tilde{\boldsymbol{x}}_0)} \right| \tag{95}$$

$$+ \left| \sqrt{L_{\text{two-layer}}(\boldsymbol{X}, \overline{\boldsymbol{W}_1^2}, \overline{\boldsymbol{W}_2^2}, \tilde{\boldsymbol{x}}_0)} - \sqrt{L_{\text{two-layer}}(\boldsymbol{X}, \widetilde{\boldsymbol{W}_1^2}, \widetilde{\boldsymbol{W}_2^2}, \tilde{\boldsymbol{x}}_0)} \right|$$

$$\leq O(\frac{1}{\sqrt{h}}) + O(\frac{1}{h}) = O(\frac{1}{\sqrt{h}}).$$

$\square$

### D.4. Approximate one-step loss under orthogonal initialization for three-layer NN

**Theorem D.5.** *Given Assumption 3.3, 3.2, and in addition assume $\eta_1$ and $\eta_2$ are no more than $O(h)$ based on Proposition C.4 and C.11, consider the training procedure discussed in Section 3, we derive the test loss after one-step and two-step GD update in a three-layer neural network:*

$$L_{\text{two-layer}}(\boldsymbol{W}_1^1, \boldsymbol{W}_2^1) = \frac{\eta_1^2}{h^2} + \frac{\eta_2^2}{h^2} + \frac{2\eta_1\eta_2}{h^2} + \frac{\eta_1^2\eta_2^2}{h^4}$$

$$- \frac{2\eta_1}{h} - \frac{2\eta_2}{h} + \frac{1}{h} + \frac{2\eta_1\eta_2}{h^3} + 1$$

$$L_{\text{three-layer}}(\boldsymbol{W}_1^2, \boldsymbol{W}_2^2) = \left( \frac{2(\eta_1 + \eta_2)(h + \eta_1\eta_2)}{h^2} - 1 \right)^2 \tag{96}$$

$$+ \frac{1}{h} + \frac{2\eta_1\eta_2}{h^2} + \frac{10\eta_1\eta_2}{h^3} + \frac{\eta_1^2\eta_2^2}{h^3}$$

$$+ \frac{37\eta_1^2\eta_2^2}{h^4} + \frac{12\eta_1^3\eta_2^3}{h^5} + \frac{\eta_1^4\eta_2^4}{h^6}$$

We prove Theorem 5.5 above by the following two subsection D.4 and D.5.

For orthogonal initialization we assume $n = h = d$.

Here we consider the orthogonal initialization where we make the setting $\boldsymbol{X}^\top\boldsymbol{X} = \boldsymbol{X}\boldsymbol{X}^\top = h\boldsymbol{I}, \boldsymbol{W}_1^{0^\top}\boldsymbol{W}_1^0 = \boldsymbol{W}_1^0\boldsymbol{W}_1^{0^\top} = \boldsymbol{I}, \boldsymbol{W}_2^{0^\top}\boldsymbol{W}_2^0 = \boldsymbol{W}_2^0\boldsymbol{W}_2^{0^\top} = \boldsymbol{I}, \boldsymbol{a}^\top\boldsymbol{a} = 1, \mathbb{E}[\boldsymbol{a}\boldsymbol{a}^\top] = \frac{1}{h}\boldsymbol{I}, \boldsymbol{\beta}^{*^\top}\boldsymbol{\beta}^* = 1, \mathbb{E}[\boldsymbol{\beta}^*\boldsymbol{\beta}^{*^\top}] = \frac{1}{h}\boldsymbol{I}.$

We consider a test data $\tilde{x}_0$ under three-layer setting, where $\frac{1}{\sqrt{h}}\tilde{x}_0$ is an random orthogonal vector, we have

$$
\begin{aligned}
&L(\boldsymbol{X}, \boldsymbol{W}_1^1, \boldsymbol{W}_2^1, \boldsymbol{a}, \tilde{\boldsymbol{x}}_0) \\
=&\mathbb{E}_{\boldsymbol{W}_1^0, \boldsymbol{W}_2^0, \boldsymbol{a}, \boldsymbol{\xi}, \tilde{\boldsymbol{x}}_0, \boldsymbol{X}} \left( \frac{1}{\sqrt{h}}\tilde{\boldsymbol{x}}_0 \boldsymbol{W}_1^1 \boldsymbol{W}_2^1 \boldsymbol{a} - \tilde{\boldsymbol{x}}_0 \boldsymbol{\beta}^* \right)^2 \\
=&\mathbb{E}_{\boldsymbol{W}_1^0, \boldsymbol{W}_2^0, \boldsymbol{a}, \boldsymbol{\xi}, \tilde{\boldsymbol{x}}_0, \boldsymbol{X}} \left[ \left( \frac{1}{\sqrt{h}}\boldsymbol{W}_1^1 \boldsymbol{W}_2^1 \boldsymbol{a} - \boldsymbol{\beta}^* \right)^\top \tilde{\boldsymbol{x}}_0^\top \tilde{\boldsymbol{x}}_0 \left( \frac{1}{\sqrt{h}}\boldsymbol{W}_1^1 \boldsymbol{W}_2^1 \boldsymbol{a} - \boldsymbol{\beta}^* \right) \right] \\
=&tr \left( \mathbb{E}_{\boldsymbol{W}_1^0, \boldsymbol{W}_2^0, \boldsymbol{a}, \boldsymbol{\xi}, \tilde{\boldsymbol{x}}_0, \boldsymbol{X}} \left[ \tilde{\boldsymbol{x}}_0^\top \tilde{\boldsymbol{x}}_0 \left( \frac{1}{\sqrt{h}}\boldsymbol{W}_1^1 \boldsymbol{W}_2^1 \boldsymbol{a} - \boldsymbol{\beta}^* \right) \left( \frac{1}{\sqrt{h}}\boldsymbol{W}_1^1 \boldsymbol{W}_2^1 \boldsymbol{a} - \boldsymbol{\beta}^* \right)^\top \right] \right) \\
=&tr \left( \mathbb{E}_{\boldsymbol{W}_1^0, \boldsymbol{W}_2^0, \boldsymbol{a}, \boldsymbol{\xi}, \boldsymbol{X}} \left[ \left( \frac{1}{\sqrt{h}}\boldsymbol{W}_1^1 \boldsymbol{W}_2^1 \boldsymbol{a} - \boldsymbol{\beta}^* \right) \left( \frac{1}{\sqrt{h}}\boldsymbol{W}_1^1 \boldsymbol{W}_2^1 \boldsymbol{a} - \boldsymbol{\beta}^* \right)^\top \right] \right) \\
=&tr \left( \mathbb{E}_{\boldsymbol{W}_1^0, \boldsymbol{W}_2^0, \boldsymbol{a}, \boldsymbol{\xi}, \boldsymbol{X}} \left[ \frac{1}{h}\boldsymbol{W}_1^1 \boldsymbol{W}_2^1 \boldsymbol{a}\boldsymbol{a}^\top \boldsymbol{W}_2^{1\top} \boldsymbol{W}_1^{1\top} \right] \right) \\
&-tr \left( \mathbb{E}_{\boldsymbol{W}_1^0, \boldsymbol{W}_2^0, \boldsymbol{a}, \boldsymbol{\xi}, \boldsymbol{X}} \left[ \frac{1}{\sqrt{h}}\boldsymbol{\beta}^* \boldsymbol{a}^\top \boldsymbol{W}_2^{1\top} \boldsymbol{W}_1^{1\top} \right] \right) \\
&-tr \left( \mathbb{E}_{\boldsymbol{W}_1^0, \boldsymbol{W}_2^0, \boldsymbol{a}, \boldsymbol{\xi}, \boldsymbol{X}} \left[ \frac{1}{\sqrt{h}}\boldsymbol{W}_1^1 \boldsymbol{W}_2^1 \boldsymbol{a}\boldsymbol{\beta}^{*\top} \right] \right) + tr \left( \mathbb{E} \left[ \boldsymbol{\beta}^* \boldsymbol{\beta}^{*\top} \right] \right).
\end{aligned}
\tag{97}
$$

Here we define $L_1, L_2, L_3, L_4$, where

$$
\begin{aligned}
L_1 &= tr \left( \mathbb{E}_{\boldsymbol{W}_1^0, \boldsymbol{W}_2^0, \boldsymbol{a}, \boldsymbol{\xi}, \boldsymbol{X}} \left[ \frac{1}{h}\boldsymbol{W}_1^1 \boldsymbol{W}_2^1 \boldsymbol{a}\boldsymbol{a}^\top \boldsymbol{W}_2^{1\top} \boldsymbol{W}_1^{1\top} \right] \right) \\
L_2 &= tr \left( \mathbb{E}_{\boldsymbol{W}_1^0, \boldsymbol{W}_2^0, \boldsymbol{a}, \boldsymbol{\xi}, \boldsymbol{X}} \left[ \frac{1}{\sqrt{h}}\boldsymbol{\beta}^* \boldsymbol{a}^\top \boldsymbol{W}_2^{1\top} \boldsymbol{W}_1^{1\top} \right] \right) \\
L_3 &= tr \left( \mathbb{E}_{\boldsymbol{W}_1^0, \boldsymbol{W}_2^0, \boldsymbol{a}, \boldsymbol{\xi}, \boldsymbol{X}} \left[ \frac{1}{\sqrt{h}}\boldsymbol{W}_1^1 \boldsymbol{W}_2^1 \boldsymbol{a}\boldsymbol{\beta}^{*\top} \right] \right) \\
L_4 &= tr \left( \mathbb{E} \left[ \boldsymbol{\beta}^* \boldsymbol{\beta}^{*\top} \right] \right)
\end{aligned}
$$

Thus

$$
L_{\text{three-layer}} = L_1 - L_2 - L_3 + L_4
$$

We have $L_1 = \sum_{i=1}^{16} T_i$, where

$$T_1 = tr\left(\mathbb{E}_{\boldsymbol{W}_1^0, \boldsymbol{W}_2^0, \boldsymbol{a}, \boldsymbol{X}}\left[\frac{1}{h}\boldsymbol{a}\boldsymbol{a}^\top \boldsymbol{W}_2^{0\top}\boldsymbol{W}_1^{0\top}\boldsymbol{W}_1^0\boldsymbol{W}_2^0\right]\right) = \frac{1}{h},$$

$$T_2 = tr\left(\mathbb{E}_{\boldsymbol{W}_1^0, \boldsymbol{W}_2^0, \boldsymbol{a}, \boldsymbol{X}}\left[\frac{\eta_1}{h^2\sqrt{h}}\boldsymbol{a}\boldsymbol{a}^\top \boldsymbol{W}_2^{0\top}\boldsymbol{W}_1^{0\top}\boldsymbol{X}^\top\boldsymbol{y}\boldsymbol{a}^\top \boldsymbol{W}_2^{0\top}\boldsymbol{W}_2^0\right]\right) = 0,$$

$$T_3 = tr\left(\mathbb{E}_{\boldsymbol{W}_1^0, \boldsymbol{W}_2^0, \boldsymbol{a}, \boldsymbol{X}}\left[\frac{\eta_2}{h^2\sqrt{h}}\boldsymbol{a}\boldsymbol{a}^\top \boldsymbol{W}_2^{0\top}\boldsymbol{W}_1^{0\top}\boldsymbol{W}_1^0\boldsymbol{W}_1^{0\top}\boldsymbol{X}^\top\boldsymbol{y}\boldsymbol{a}^\top\right]\right) = 0,$$

$$T_4 = tr\left(\mathbb{E}_{\boldsymbol{W}_1^0, \boldsymbol{W}_2^0, \boldsymbol{a}, \boldsymbol{X}}\left[\frac{\eta_1\eta_2}{h^4}\boldsymbol{a}\boldsymbol{a}^\top \boldsymbol{W}_2^{0\top}\boldsymbol{W}_1^{0\top}\boldsymbol{X}^\top\boldsymbol{y}\boldsymbol{a}^\top \boldsymbol{W}_2^{0\top}\boldsymbol{W}_1^{0\top}\boldsymbol{X}^\top\boldsymbol{y}\boldsymbol{a}^\top\right]\right) = \frac{\eta_1\eta_2}{h^3},$$

$$T_5 = tr\left(\mathbb{E}_{\boldsymbol{W}_1^0, \boldsymbol{W}_2^0, \boldsymbol{a}, \boldsymbol{X}}\left[\frac{\eta_1}{h^2\sqrt{h}}\boldsymbol{a}\boldsymbol{a}^\top \boldsymbol{W}_2^{0\top}\boldsymbol{W}_2^0\boldsymbol{a}\boldsymbol{y}^\top\boldsymbol{X}\boldsymbol{W}_1^0\boldsymbol{W}_2^0\right]\right) = 0,$$

$$T_6 = tr\left(\mathbb{E}_{\boldsymbol{W}_1^0, \boldsymbol{W}_2^0, \boldsymbol{a}, \boldsymbol{X}}\left[\frac{\eta_1^2}{h^4}\boldsymbol{a}\boldsymbol{a}^\top \boldsymbol{W}_2^{0\top}\boldsymbol{W}_2^0\boldsymbol{a}\boldsymbol{y}^\top\boldsymbol{X}\boldsymbol{X}^\top\boldsymbol{y}\boldsymbol{a}^\top \boldsymbol{W}_2^{0\top}\boldsymbol{W}_2^0\right]\right) = \frac{\eta_1^2}{h^2},$$

$$T_7 = tr\left(\mathbb{E}_{\boldsymbol{W}_1^0, \boldsymbol{W}_2^0, \boldsymbol{a}, \boldsymbol{X}}\left[\frac{\eta_1\eta_2}{h^4}\boldsymbol{a}\boldsymbol{a}^\top \boldsymbol{W}_2^{0\top}\boldsymbol{W}_2^0\boldsymbol{a}\boldsymbol{y}^\top\boldsymbol{X}\boldsymbol{W}_1^0\boldsymbol{W}_1^{0\top}\boldsymbol{X}^\top\boldsymbol{y}\boldsymbol{a}^\top\right]\right) = \frac{\eta_1\eta_2}{h^2},$$

$$T_8 = tr\left(\mathbb{E}_{\boldsymbol{W}_1^0, \boldsymbol{W}_2^0, \boldsymbol{a}, \boldsymbol{X}}\left[\frac{\eta_1^2\eta_2}{h^5\sqrt{h}}\boldsymbol{a}\boldsymbol{a}^\top \boldsymbol{W}_2^{0\top}\boldsymbol{W}_2^0\boldsymbol{a}\boldsymbol{y}^\top\boldsymbol{X}\boldsymbol{X}^\top\boldsymbol{y}\boldsymbol{a}^\top \boldsymbol{W}_2^{0\top}\boldsymbol{W}_1^{0\top}\boldsymbol{X}^\top\boldsymbol{y}\boldsymbol{a}^\top\right]\right) = 0,$$

$$T_9 = tr\left(\mathbb{E}_{\boldsymbol{W}_1^0, \boldsymbol{W}_2^0, \boldsymbol{a}, \boldsymbol{X}}\left[\frac{\eta_2}{h^2\sqrt{h}}\boldsymbol{a}\boldsymbol{a}^\top \boldsymbol{a}\boldsymbol{y}^\top\boldsymbol{X}\boldsymbol{W}_1^0\boldsymbol{W}_1^{0\top}\boldsymbol{W}_1^0\boldsymbol{W}_2^0\right]\right) = 0,$$

$$T_{10} = tr\left(\mathbb{E}_{\boldsymbol{W}_1^0, \boldsymbol{W}_2^0, \boldsymbol{a}, \boldsymbol{X}}\left[\frac{\eta_1\eta_2}{h^4}\boldsymbol{a}\boldsymbol{a}^\top \boldsymbol{a}\boldsymbol{y}^\top\boldsymbol{X}\boldsymbol{W}_1^0\boldsymbol{W}_1^{0\top}\boldsymbol{X}^\top\boldsymbol{y}\boldsymbol{a}^\top \boldsymbol{W}_2^{0\top}\boldsymbol{W}_2^0\right]\right) = \frac{\eta_1\eta_2}{h^2},$$

$$T_{11} = tr\left(\mathbb{E}_{\boldsymbol{W}_1^0, \boldsymbol{W}_2^0, \boldsymbol{a}, \boldsymbol{X}}\left[\frac{\eta_2^2}{h^4}\boldsymbol{a}\boldsymbol{a}^\top \boldsymbol{a}\boldsymbol{y}^\top\boldsymbol{X}\boldsymbol{W}_1^0\boldsymbol{W}_1^{0\top}\boldsymbol{W}_1^0\boldsymbol{W}_1^{0\top}\boldsymbol{X}^\top\boldsymbol{y}\boldsymbol{a}^\top\right]\right) = \frac{\eta_2^2}{h^2},$$

$$T_{12} = tr\left(\mathbb{E}_{\boldsymbol{W}_1^0, \boldsymbol{W}_2^0, \boldsymbol{a}, \boldsymbol{X}}\left[\frac{\eta_1\eta_2^2}{h^5\sqrt{h}}\boldsymbol{a}\boldsymbol{a}^\top \boldsymbol{a}\boldsymbol{y}^\top\boldsymbol{X}\boldsymbol{W}_1^0\boldsymbol{W}_1^{0\top}\boldsymbol{X}^\top\boldsymbol{y}\boldsymbol{a}^\top \boldsymbol{W}_2^{0\top}\boldsymbol{W}_1^{0\top}\boldsymbol{X}^\top\boldsymbol{y}\boldsymbol{a}^\top\right]\right) = 0,$$

$$T_{13} = tr\left(\mathbb{E}_{\boldsymbol{W}_1^0, \boldsymbol{W}_2^0, \boldsymbol{a}, \boldsymbol{X}}\left[\frac{\eta_1\eta_2}{h^4}\boldsymbol{a}\boldsymbol{a}^\top \boldsymbol{a}\boldsymbol{y}^\top\boldsymbol{X}\boldsymbol{W}_1^0\boldsymbol{W}_2^0\boldsymbol{a}\boldsymbol{y}^\top\boldsymbol{X}\boldsymbol{W}_1^0\boldsymbol{W}_2^0\right]\right) = \frac{\eta_1\eta_2}{h^3},$$

$$T_{14} = tr\left(\mathbb{E}_{\boldsymbol{W}_1^0, \boldsymbol{W}_2^0, \boldsymbol{a}, \boldsymbol{X}}\left[\frac{\eta_1^2\eta_2}{h^5\sqrt{h}}\boldsymbol{a}\boldsymbol{a}^\top \boldsymbol{a}\boldsymbol{y}^\top\boldsymbol{X}\boldsymbol{W}_1^0\boldsymbol{W}_2^0\boldsymbol{a}\boldsymbol{y}^\top\boldsymbol{X}\boldsymbol{X}^\top\boldsymbol{y}\boldsymbol{a}^\top \boldsymbol{W}_2^{0\top}\boldsymbol{W}_2^0\right]\right) = 0,$$

$$T_{15} = tr\left(\mathbb{E}_{\boldsymbol{W}_1^0, \boldsymbol{W}_2^0, \boldsymbol{a}, \boldsymbol{X}}\left[\frac{\eta_1\eta_2^2}{h^5\sqrt{h}}\boldsymbol{a}\boldsymbol{a}^\top \boldsymbol{a}\boldsymbol{y}^\top\boldsymbol{X}\boldsymbol{W}_1^0\boldsymbol{W}_2^0\boldsymbol{a}\boldsymbol{y}^\top\boldsymbol{X}\boldsymbol{W}_1^0\boldsymbol{W}_1^{0\top}\boldsymbol{X}^\top\boldsymbol{y}\boldsymbol{a}^\top\right]\right) = 0,$$

$$T_{16} = tr\left(\mathbb{E}_{\boldsymbol{W}_1^0, \boldsymbol{W}_2^0, \boldsymbol{a}, \boldsymbol{X}}\left[\frac{\eta_1^2\eta_2^2}{h^7}\boldsymbol{a}\boldsymbol{a}^\top \boldsymbol{a}\boldsymbol{y}^\top\boldsymbol{X}\boldsymbol{W}_1^0\boldsymbol{W}_2^0\boldsymbol{a}\boldsymbol{y}^\top\boldsymbol{X}\boldsymbol{X}^\top\boldsymbol{y}\boldsymbol{a}^\top \boldsymbol{W}_2^{0\top}\boldsymbol{W}_1^{0\top}\boldsymbol{X}^\top\boldsymbol{y}\boldsymbol{a}^\top\right]\right) = \frac{\eta_1^2\eta_2^2}{h^4}.$$

We have $L_2 = \sum_{i=17}^{20} T_i$, where

$$T_{17} = tr\left(\mathbb{E}_{\boldsymbol{W}_1^0, \boldsymbol{W}_2^0, \boldsymbol{a}, \boldsymbol{X}}\left[\frac{1}{\sqrt{h}}\boldsymbol{\beta}^*\boldsymbol{a}^\top \boldsymbol{W}_2^{0\top}\boldsymbol{W}_1^{0\top}\right]\right) = 0,$$

$$T_{18} = tr\left(\mathbb{E}_{\boldsymbol{W}_1^0, \boldsymbol{W}_2^0, \boldsymbol{a}, \boldsymbol{X}}\left[\frac{\eta_1}{h^2}\boldsymbol{\beta}^*\boldsymbol{a}^\top \boldsymbol{W}_2^{0\top}\boldsymbol{W}_2^0\boldsymbol{a}\boldsymbol{y}^\top\boldsymbol{X}\right]\right) = \frac{\eta_1}{h},$$

$$T_{19} = tr\left(\mathbb{E}_{\boldsymbol{W}_1^0, \boldsymbol{W}_2^0, \boldsymbol{a}, \boldsymbol{X}}\left[\frac{\eta_2}{h^2}\boldsymbol{\beta}^*\boldsymbol{a}^\top \boldsymbol{a}\boldsymbol{y}^\top\boldsymbol{X}\boldsymbol{W}_1^0\boldsymbol{W}_1^{0\top}\right]\right) = \frac{\eta_2}{h},$$

$$T_{20} = tr\left(\mathbb{E}_{\boldsymbol{W}_1^0, \boldsymbol{W}_2^0, \boldsymbol{a}, \boldsymbol{X}}\left[\frac{\eta_1\eta_2}{h^3\sqrt{h}}\boldsymbol{\beta}^*\boldsymbol{a}^\top \boldsymbol{a}\boldsymbol{y}^\top\boldsymbol{X}\boldsymbol{W}_1^0\boldsymbol{W}_2^0\boldsymbol{a}\boldsymbol{y}^\top\boldsymbol{X}\right]\right) = 0.$$

We have $L_3 = \sum_{i=21}^{24} T_i$, where

$$T_{21} = tr\left(\mathbb{E}_{\boldsymbol{W}_1^0, \boldsymbol{W}_2^0, \boldsymbol{a}, \boldsymbol{X}}\left[\frac{1}{\sqrt{h}}\boldsymbol{W}_1^0\boldsymbol{W}_2^0\boldsymbol{a}\boldsymbol{\beta}^{*\top}\right]\right) = 0,$$

$$T_{22} = tr\left(\mathbb{E}_{\boldsymbol{W}_1^0, \boldsymbol{W}_2^0, \boldsymbol{a}, \boldsymbol{X}}\left[\frac{\eta_1}{h^2}\boldsymbol{X}^\top\boldsymbol{y}\boldsymbol{a}^\top\boldsymbol{W}_2^{0\top}\boldsymbol{W}_2^0\boldsymbol{a}\boldsymbol{\beta}^{*\top}\right]\right) = \frac{\eta_1}{h},$$

$$T_{23} = tr\left(\mathbb{E}_{\boldsymbol{W}_1^0, \boldsymbol{W}_2^0, \boldsymbol{a}, \boldsymbol{X}}\left[\frac{\eta_2}{h^2}\boldsymbol{W}_1^0\boldsymbol{W}_1^{0\top}\boldsymbol{X}^\top\boldsymbol{y}\boldsymbol{a}^\top\boldsymbol{a}\boldsymbol{\beta}^{*\top}\right]\right) = \frac{\eta_2}{h},$$

$$T_{24} = tr\left(\mathbb{E}_{\boldsymbol{W}_1^0, \boldsymbol{W}_2^0, \boldsymbol{a}, \boldsymbol{X}}\left[\frac{\eta_1\eta_2}{h^3\sqrt{h}}\boldsymbol{X}^\top\boldsymbol{y}\boldsymbol{a}^\top\boldsymbol{W}_2^{0\top}\boldsymbol{W}_1^{0\top}\boldsymbol{X}^\top\boldsymbol{y}\boldsymbol{a}^\top\boldsymbol{a}\boldsymbol{\beta}^{*\top}\right]\right) = 0,$$

Based on the above computation, we see that for orthogonal initialization, the one-step test loss for 3-layer NN is

$$L_{\text{three-layer}}(\boldsymbol{X}, \boldsymbol{W}_1^1, \boldsymbol{W}_2^1, \boldsymbol{a}^1, \tilde{\boldsymbol{x}}_0) = \frac{\eta_1^2}{h^2} + \frac{\eta_2^2}{h^2} + \frac{2\eta_1\eta_2}{h^2} + \frac{\eta_1^2\eta_2^2}{h^4} - \frac{2\eta_1}{h} - \frac{2\eta_2}{h} + \frac{1}{h} + \frac{2\eta_1\eta_2}{h^3} + 1 \qquad (98)$$

Here, for the one-step updated loss, we consider the following optimization problem, and we assume the following constraint $\eta_1 + \eta_2 = 2h^\alpha$, our goal is to see whether $\eta_1 = \eta_2 = h^\alpha$ is local minima or local maxima.

$$L_{\text{three-layer}}(\boldsymbol{X}, \boldsymbol{W}_1^1, \boldsymbol{W}_2^1, \boldsymbol{a}^1, \tilde{\boldsymbol{x}}_0) = \frac{\eta_1^2}{h^2} + \frac{\eta_2^2}{h^2} + \frac{2\eta_1\eta_2}{h^2} + \frac{\eta_1^2\eta_2^2}{h^4} - 4h^{\alpha-1} + \frac{1}{h} + 1 + O(\frac{\eta_1\eta_2}{h^3}) \qquad (99)$$

It is easy to find that $\eta_1 = \eta_2 = h^\alpha$ is a local maxima. $\qquad\square$

### D.5. Approximate two-step loss for three-layer NN under orthogonal initialization

For orthogonal initialization we assume $n = h = d$.

Here we consider the orthogonal initialization where we make the setting $\boldsymbol{X}^\top\boldsymbol{X} = \boldsymbol{X}\boldsymbol{X}^\top = h\boldsymbol{I}, \boldsymbol{W}_1^{0\top}\boldsymbol{W}_1^0 = \boldsymbol{W}_1^0\boldsymbol{W}_1^{0\top} = \boldsymbol{I}, \boldsymbol{W}_2^{0\top}\boldsymbol{W}_2^0 = \boldsymbol{W}_2^0\boldsymbol{W}_2^{0\top} = \boldsymbol{I}, \boldsymbol{a}^\top\boldsymbol{a} = 1, \mathbb{E}[\boldsymbol{a}\boldsymbol{a}^\top] = \frac{1}{h}\boldsymbol{I}, \boldsymbol{\beta}^{*\top}\boldsymbol{\beta}^* = 1, \mathbb{E}[\boldsymbol{\beta}^*\boldsymbol{\beta}^{*\top}] = \frac{1}{h}\boldsymbol{I}$.

For the simplification, we only consider replacing $\boldsymbol{G}_1$ with $\boldsymbol{A}_1$ and $\boldsymbol{G}_2$ with $\boldsymbol{A}_2$. We consider $s = \frac{1}{h}$

$$\boldsymbol{A}_1^0 = \frac{1}{\sqrt{h}}\boldsymbol{\beta}^*\boldsymbol{a}^\top\boldsymbol{W}_2^{0\top} \qquad\qquad \widetilde{\boldsymbol{A}_1^1} = \frac{1}{\sqrt{h}}\boldsymbol{\beta}^*\boldsymbol{a}^\top\widetilde{\boldsymbol{W}_2^1}^\top$$

$$\boldsymbol{B}_1^0 = \frac{1}{h}\boldsymbol{W}_1^0\boldsymbol{W}_2^0\boldsymbol{a}\boldsymbol{a}^\top\boldsymbol{W}_2^{0\top} \qquad\qquad \widetilde{\boldsymbol{B}_1^1} = \frac{1}{h}\widetilde{\boldsymbol{W}_1^1}\widetilde{\boldsymbol{W}_2^1}\boldsymbol{a}\boldsymbol{a}^\top\widetilde{\boldsymbol{W}_2^1}^\top$$

$$\boldsymbol{A}_2^0 = \frac{1}{\sqrt{h}}\boldsymbol{W}_1^{0\top}\boldsymbol{\beta}^*\boldsymbol{a}^\top \qquad\qquad \widetilde{\boldsymbol{A}_2^1} = \frac{1}{\sqrt{h}}\widetilde{\boldsymbol{W}_1^1}^\top\boldsymbol{\beta}^*\boldsymbol{a}^\top$$

$$\boldsymbol{B}_2^0 = \frac{1}{h}\boldsymbol{W}_2^0\boldsymbol{a}\boldsymbol{a}^\top \qquad\qquad \widetilde{\boldsymbol{B}_2^1} = \frac{1}{h}\widetilde{\boldsymbol{W}_1^1}^\top\widetilde{\boldsymbol{W}_1^1}\widetilde{\boldsymbol{W}_2^1}\boldsymbol{a}\boldsymbol{a}^\top$$

Thus we have

$$\widetilde{W_1^1} = W_1^0 + \eta_1 A_1^0 = W_1^0 + \frac{\eta_1}{\sqrt{h}} \beta^* a^\top W_2^{0\top}$$

$$\widetilde{W_2^1} = W_2^0 + \eta_2 A_2^0 = W_2^0 + \frac{\eta_2}{\sqrt{h}} W_1^{0\top} \beta^* a^\top$$

$$\widetilde{W_1^2} = \widetilde{W_1^1} + \eta_1 \widetilde{A_1^1} = \widetilde{W_1^1} + \frac{\eta_1}{\sqrt{h}} \beta^* a^\top \widetilde{W_2^1}^\top$$

$$= W_1^0 + \frac{2\eta_1}{\sqrt{h}} \beta^* a^\top W_2^{0\top} + \frac{\eta_1 \eta_2}{h} \beta^* \beta^{*\top} W_1^0$$

$$\widetilde{W_2^2} = \widetilde{W_2^1} + \eta_2 \widetilde{A_2^1} = \widetilde{W_2^1} + \frac{\eta_2}{\sqrt{h}} \widetilde{W_1^1}^\top \beta^* a^\top$$

$$= W_2^0 + \frac{2\eta_2}{\sqrt{h}} W_1^{0\top} \beta^* a^\top + \frac{\eta_1 \eta_2}{h} W_2^0 a a^\top$$

we can derive that

$$
\begin{aligned}
\widetilde{W_1^2}\widetilde{W_2^2} &= W_1^0 W_2^0 + \frac{2\eta_1}{\sqrt{h}} \beta^* a^\top + \frac{\eta_1 \eta_2}{h} \beta^* \beta^{*\top} W_1^0 W_2^0 \\
&\quad + \frac{2\eta_2}{\sqrt{h}} \beta^* a^\top + \frac{4\eta_1 \eta_2}{h} \beta^* a^\top W_2^{0\top} W_1^{0\top} \beta^* a^\top + \frac{2\eta_1 \eta_2^2}{h\sqrt{h}} \beta^* a^\top \\
&\quad + \frac{\eta_1 \eta_2}{h} W_1^0 W_2^0 a a^\top + \frac{2\eta_1^2 \eta_2}{h\sqrt{h}} \beta^* a^\top + \frac{\eta_1^2 \eta_2^2}{h^2} \beta^* \beta^{*\top} W_1^0 W_2^0 a a^\top \\
&= W_1^0 W_2^0 + (\frac{2\eta_1}{\sqrt{h}} + \frac{2\eta_2}{\sqrt{h}} + \frac{2\eta_1 \eta_2^2}{h\sqrt{h}} + \frac{2\eta_1^2 \eta_2}{h\sqrt{h}}) \beta^* a^\top + \frac{\eta_1 \eta_2}{h} \beta^* \beta^{*\top} W_1^0 W_2^0 \\
&\quad + \frac{4\eta_1 \eta_2}{h} \beta^* a^\top W_2^{0\top} W_1^{0\top} \beta^* a^\top + \frac{\eta_1 \eta_2}{h} W_1^0 W_2^0 a a^\top + \frac{\eta_1^2 \eta_2^2}{h^2} \beta^* \beta^{*\top} W_1^0 W_2^0 a a^\top \\
\widetilde{W_2^2}^\top \widetilde{W_1^2}^\top &= W_2^{0\top} W_1^{0\top} + (\frac{2\eta_1}{\sqrt{h}} + \frac{2\eta_2}{\sqrt{h}} + \frac{2\eta_1 \eta_2^2}{h\sqrt{h}} + \frac{2\eta_1^2 \eta_2}{h\sqrt{h}}) a \beta^{*\top} + \frac{\eta_1 \eta_2}{h} W_2^{0\top} W_1^{0\top} \beta^* \beta^{*\top} \\
&\quad + \frac{4\eta_1 \eta_2}{h} a \beta^{*\top} W_1^0 W_2^0 a \beta^{*\top} + \frac{\eta_1 \eta_2}{h} a a^\top W_2^{0\top} W_1^{0\top} + \frac{\eta_1^2 \eta_2^2}{h^2} a a^\top W_2^{0\top} W_1^{0\top} \beta^* \beta^{*\top}
\end{aligned}
$$

Thus, we have

$$tr\left(\mathbb{E}_{\boldsymbol{W}_1^0,\boldsymbol{W}_2^0,\boldsymbol{a},\boldsymbol{\beta}^*,\boldsymbol{X}}\left[\frac{1}{h}\widetilde{\boldsymbol{W}_1^2}\widetilde{\boldsymbol{W}_2^2}\boldsymbol{a}\boldsymbol{a}^\top\widetilde{\boldsymbol{W}_2^2}^\top\widetilde{\boldsymbol{W}_1^2}^\top\right]\right) = tr\left(\mathbb{E}_{\boldsymbol{W}_1^0,\boldsymbol{W}_2^0,\boldsymbol{a},\boldsymbol{\beta}^*,\boldsymbol{X}}\left[\frac{1}{h}\boldsymbol{a}\boldsymbol{a}^\top\widetilde{\boldsymbol{W}_2^2}^\top\widetilde{\boldsymbol{W}_1^2}^\top\widetilde{\boldsymbol{W}_1^2}\widetilde{\boldsymbol{W}_2^2}\right]\right)$$

$$= \frac{1}{h} + \frac{2\eta_1\eta_2}{h^2} + \frac{2\eta_1\eta_2}{h^3} + \frac{21\eta_1^2\eta_2^2}{h^4}$$

$$+ \frac{4\eta_1\eta_2}{h^2}tr\left(\mathbb{E}\left[\boldsymbol{a}\boldsymbol{a}^\top\boldsymbol{a}\boldsymbol{\beta}^{*\top}\boldsymbol{W}_1^0\boldsymbol{W}_2^0\boldsymbol{a}\boldsymbol{\beta}^{*\top}\boldsymbol{W}_1^0\boldsymbol{W}_2^0\right]\right)$$

$$+ \frac{4((\eta_1+\eta_2)^2(h+\eta_1\eta_2))}{h^4} + \frac{4\eta_1^3\eta_2^3}{h^5} + \frac{\eta_1^4\eta_2^4}{h^6}$$

$$+ \frac{4\eta_1^2\eta_2^2}{h^3}tr\left(\mathbb{E}\left[\boldsymbol{a}\boldsymbol{a}^\top\boldsymbol{a}\boldsymbol{\beta}^{*\top}\boldsymbol{W}_1^0\boldsymbol{W}_2^0\boldsymbol{a}\boldsymbol{\beta}^{*\top}\boldsymbol{W}_1^0\boldsymbol{W}_2^0\right]\right)$$

$$+ \frac{4\eta_1\eta_2}{h^2}tr\left(\mathbb{E}\left[\boldsymbol{a}\boldsymbol{a}^\top\boldsymbol{W}_2^{0\top}\boldsymbol{W}_1^{0\top}\boldsymbol{\beta}^*\boldsymbol{a}^\top\boldsymbol{W}_2^{0\top}\boldsymbol{W}_1^{0\top}\boldsymbol{\beta}^*\boldsymbol{a}^\top\right]\right)$$

$$+ \frac{4\eta_1^2\eta_2^2}{h^3}tr\left(\mathbb{E}\left[\boldsymbol{a}\boldsymbol{a}^\top\boldsymbol{W}_2^{0\top}\boldsymbol{W}_1^{0\top}\boldsymbol{\beta}^*\boldsymbol{a}^\top\boldsymbol{W}_2^{0\top}\boldsymbol{W}_1^{0\top}\boldsymbol{\beta}^*\boldsymbol{a}^\top\right]\right)$$

$$+ \frac{4\eta_1^2\eta_2^2}{h^3}tr\left(\mathbb{E}\left[\boldsymbol{a}\boldsymbol{a}^\top\boldsymbol{W}_2^{0\top}\boldsymbol{W}_1^{0\top}\boldsymbol{\beta}^*\boldsymbol{a}^\top\boldsymbol{W}_2^{0\top}\boldsymbol{W}_1^{0\top}\boldsymbol{\beta}^*\boldsymbol{a}^\top\right]\right)$$

$$+ \frac{4\eta_1^3\eta_2^3}{h^4}tr\left(\mathbb{E}\left[\boldsymbol{a}\boldsymbol{a}^\top\boldsymbol{W}_2^{0\top}\boldsymbol{W}_1^{0\top}\boldsymbol{\beta}^*\boldsymbol{a}^\top\boldsymbol{W}_2^{0\top}\boldsymbol{W}_1^{0\top}\boldsymbol{\beta}^*\boldsymbol{a}^\top\right]\right)$$

$$+ \frac{4\eta_1^2\eta_2^2}{h^3}tr\left(\mathbb{E}\left[\boldsymbol{a}\boldsymbol{a}^\top\boldsymbol{a}\boldsymbol{\beta}^{*\top}\boldsymbol{W}_1^0\boldsymbol{W}_2^0\boldsymbol{a}\boldsymbol{\beta}^{*\top}\boldsymbol{W}_1^0\boldsymbol{W}_2^0\right]\right)$$

$$+ \frac{4\eta_1^3\eta_2^3}{h^4}tr\left(\mathbb{E}\left[\boldsymbol{a}\boldsymbol{a}^\top\boldsymbol{W}_2^{0\top}\boldsymbol{W}_1^{0\top}\boldsymbol{\beta}^*\boldsymbol{a}^\top\boldsymbol{W}_2^{0\top}\boldsymbol{W}_1^{0\top}\boldsymbol{\beta}^*\boldsymbol{a}^\top\right]\right)$$

$$= \frac{1}{h} + \frac{2\eta_1\eta_2}{h^2} + \frac{10\eta_1\eta_2}{h^3} + \frac{\eta_1^2\eta_2^2}{h^3} + \frac{37\eta_1^2\eta_2^2}{h^4}$$

$$+ \frac{4((\eta_1+\eta_2)^2(h+\eta_1\eta_2)^2)}{h^4} + \frac{12\eta_1^3\eta_2^3}{h^5} + \frac{\eta_1^4\eta_2^4}{h^6}$$

$$tr\left(\mathbb{E}_{\boldsymbol{W}_1^0,\boldsymbol{W}_2^0,\boldsymbol{a},\boldsymbol{\beta}^*,\boldsymbol{X}}\left[\frac{1}{\sqrt{h}}\boldsymbol{\beta}^*\boldsymbol{a}^\top\widetilde{\boldsymbol{W}_2^2}^\top\widetilde{\boldsymbol{W}_1^2}^\top\right]\right) = \frac{2(\eta_1+\eta_2)(h+\eta_1\eta_2)}{h^2}$$

$$tr\left(\mathbb{E}_{\boldsymbol{W}_1^0,\boldsymbol{W}_2^0,\boldsymbol{a},\boldsymbol{\beta}^*,\boldsymbol{X}}\left[\frac{1}{\sqrt{h}}\widetilde{\boldsymbol{W}_1^2}\widetilde{\boldsymbol{W}_2^2}\boldsymbol{a}\boldsymbol{\beta}^{*\top}\right]\right) = \frac{2(\eta_1+\eta_2)(h+\eta_1\eta_2)}{h^2}$$

$$tr\left(\mathbb{E}_{\boldsymbol{W}_1^0,\boldsymbol{W}_2^0,\boldsymbol{a},\boldsymbol{\beta}^*,\boldsymbol{X}}\left[\boldsymbol{\beta}^*\boldsymbol{\beta}^{*\top}\right]\right) = 1$$

Thus, we have

$$L_{\text{three-layer}}(\boldsymbol{X},\boldsymbol{W}_1^2,\boldsymbol{W}_2^2,\boldsymbol{a},\tilde{\boldsymbol{x}}_0) = \left(\frac{2(\eta_1+\eta_2)(h+\eta_1\eta_2)}{h^2} - 1\right)^2 + \frac{1}{h} + \frac{2\eta_1\eta_2}{h^2} + \frac{10\eta_1\eta_2}{h^3} + \frac{\eta_1^2\eta_2^2}{h^3} + \frac{37\eta_1^2\eta_2^2}{h^4} + \frac{12\eta_1^3\eta_2^3}{h^5} + \frac{\eta_1^4\eta_2^4}{h^6}$$

$$(100)$$

$$\square$$

**Corollary D.6.** *Suppose $\eta_1 + \eta_2 = 2h^\alpha$ and we consider $0 < \alpha < 1$. Then, for any $\alpha$ in this range, the point $\eta_1 = \eta_2 = h^\alpha$ is not a local minimum of the loss $L_{three\text{-}layer}(\boldsymbol{W}_1^1,\boldsymbol{W}_2^1)$. Moreover, for $0 < \alpha \le \frac{2}{3}$, if $h > h^*$, then $\eta_1 = \eta_2 = h^\alpha$ is a local minimum of the loss $L_{three\text{-}layer}(\boldsymbol{W}_1^2,\boldsymbol{W}_2^2)$, where $h^*$ is the root of the following equation:*

$$32h^{3\alpha-2} + 33h^{\alpha-1} + 74h^{\alpha-2} + 2h^{-\alpha}$$

$$+10h^{-\alpha-1} + 36h^{3\alpha-3} + 4h^{5\alpha-4} - 8 = 0$$

$$(101)$$

**Proof of Corollary D.6.** Here, for the two-step updated loss, we consider the following optimization problem, and we assume that $\eta_1 + \eta_2 = 2h^\alpha$, we want to find whether the local minima for $L_{\text{three-layer}}$ is $\eta_1 = \eta_2 = h^\alpha$.

Since $\eta_1 + \eta_2 = 2h^\alpha$, we have

$$L_{\text{three-layer}}(\boldsymbol{X}, \boldsymbol{W}_1^2, \boldsymbol{W}_2^2, \boldsymbol{a}, \tilde{\boldsymbol{x}}_0) = \left(\frac{4(h + \eta_1(2h^\alpha - \eta_1))}{h^{2-\alpha}} - 1\right)^2 + \frac{1}{h} + \frac{2\eta_1(2h^\alpha - \eta_1)}{h^2} + \frac{10\eta_1(2h^\alpha - \eta_1)}{h^3}$$
$$+ \frac{\eta_1^2(2h^\alpha - \eta_1)^2}{h^3} + \frac{37\eta_1^2(2h^\alpha - \eta_1)^2}{h^4} + \frac{12\eta_1^3(2h^\alpha - \eta_1)^3}{h^5} + \frac{\eta_1^4(2h^\alpha - \eta_1)^4}{h^6}$$

Taking the derivative, we have

$$L'_{\text{three-layer}}(\boldsymbol{X}, \boldsymbol{W}_1^2, \boldsymbol{W}_2^2, \boldsymbol{a}, \tilde{\boldsymbol{x}}_0) = 2(h^\alpha - \eta_1)\left[\frac{8}{h^{2-\alpha}}\left(\frac{4(\eta_1(2h^\alpha - \eta_1) + h)}{h^{2-\alpha}} - 1\right)\right]$$
$$+ 2(h^\alpha - \eta_1)\left[\frac{2}{h^2} + \frac{10}{h^3} + \frac{(h + 74)\eta_1(2h^\alpha - \eta_1)}{h^4} + \frac{36\eta_1^2(2h^\alpha - \eta_1)^2}{h^5} + \frac{4\eta_1^3(2h^\alpha - \eta_1)^3}{h^6}\right]$$

If we let $L_{\text{two-layer}}$ is $\eta_1 = \eta_2 = h^\alpha$ be local minima, we must need

- $2 > 2 - \alpha \Rightarrow \alpha > 0$
- $2\alpha - 3 < \alpha - 2 \Rightarrow \alpha < 1$
- $2\alpha - 4 < \alpha - 2 \Rightarrow \alpha < 2$
- $4\alpha - 5 < \alpha - 2 \Rightarrow \alpha < 1$
- $6\alpha - 6 < \alpha - 2 \Rightarrow \alpha < \frac{4}{5}$
- $2\alpha - 3 < \alpha - 2 \Rightarrow \alpha < 1$
- $4\alpha - 4 < \alpha - 2 \Rightarrow \alpha < \frac{2}{3}$

Take the intersection, we have $0 < \alpha < \frac{2}{3}$. Given the fixed $0 < \alpha < \frac{2}{3}$, we will give how large $h$ is to ensure that $\eta_1 = \eta_2 = h^\alpha$ will be the local minima,

We need
$$32h^{3\alpha-2} + 33h^{\alpha-1} + 74h^{\alpha-2} + 2h^{-\alpha} + 10h^{-\alpha-1} + 36h^{3\alpha-3} + 4h^{5\alpha-4} - 8 < 0.$$

$\square$

# E. Gaussian Initialization

In this section, to obtain more general and practical results, we extend the one-step loss analysis to gaussian initialization while also accounting for label noise.

**Assumption E.1.** *For gaussian initialization, we consider more general case with $d = h$, we also assume $c_1 n \le h \le C_1 n$, where $C_1, c_1$ are finite constants.*

**Dataset.** Here we use linear *teacher* models to generate the training data of both two-layer and three-layer *student* networks under gaussian initialization. We sample $n$ data points $\{\boldsymbol{x}_1, \cdots, \boldsymbol{x}_n\}$ from the isotropic Gaussian $\boldsymbol{x}_i \sim \mathcal{N}(\boldsymbol{0}_h, \boldsymbol{I}_h), \forall i \in [n]$ as our input data.

- **Two-layer NN Case.** For a given $\boldsymbol{x}_i \in \mathbb{R}^h$, we use a linear *teacher* model $F : \mathbb{R}^h \to \mathbb{R}^h$ to generate the corresponding label $\boldsymbol{y}_i \in \mathbb{R}^h$ (Du et al., 2018) as follows

$$\boldsymbol{y}_i = F(\boldsymbol{x}_i) + \boldsymbol{\xi}_i' = \boldsymbol{M}^\top \boldsymbol{x}_i + \boldsymbol{\xi}_i'. \tag{102}$$

Here, $\boldsymbol{M} \in \mathbb{R}^{h \times h}$ with entries sampled i.i.d as follows $h[\boldsymbol{M}]_{i,j} \sim \mathcal{N}(0, 1)$ is the *target matrix*, and $\boldsymbol{\xi}_i' \in \mathbb{R}^h$ with entries sampled i.i.d as follows $\sqrt{h}[\boldsymbol{\xi}_i']_j \sim \mathcal{N}(0, \rho_e^2)$ is the independent additive label noise. We represent $\boldsymbol{X} \in \mathbb{R}^{n \times h}, \boldsymbol{Y} \in \mathbb{R}^{n \times h}$ as the input matrix and the label matrix, respectively.

- **Three-layer NN Case.** For a given $\boldsymbol{x}_i \in \mathbb{R}^h$, we use a linear teacher model $F^* : \mathbb{R}^h \to \mathbb{R}$ to generate the corresponding scalar label $y_i \in \mathbb{R}$ as follows:

$$y_i = F^*(\boldsymbol{x}_i) + \xi_i = \boldsymbol{\beta}^{*\top}\boldsymbol{x}_i + \boldsymbol{\xi}_i. \tag{103}$$

Here $\boldsymbol{\beta}^* \in \mathbb{R}^h$ with $\sqrt{h}\boldsymbol{\beta}^* \sim \mathcal{N}(0,1)$ is the *target direction*, and $\boldsymbol{\xi}_i \sim \mathcal{N}(0, \rho_e^2)$ is the independent additive label noise. We represent $\boldsymbol{X} \in \mathbb{R}^{n \times h}, \boldsymbol{y} \in \mathbb{R}^n$ as the input matrix and the label vector, respectively.

**Model.** For two-layer and three-layer NNs, we consider the entries sampled i.i.d follows $\sqrt{h}\left[\boldsymbol{W}_1^0\right]_{i,j} \sim \mathcal{N}(0,1)$, $\sqrt{h}\left[\boldsymbol{W}_2^0\right]_{i,j} \sim \mathcal{N}(0,1)$, $\sqrt{h}\left[\boldsymbol{a}\right]_i \sim \mathcal{N}(0,1), \forall i \in [h], j \in [h]$.

### E.1. Norm Analysis of One-step Update Gradient Matrices Under Gaussian Initialization

We first give the norm analysis of one-step update gradient matrices under gaussian initialization. This analysis is an important step in simplifying the derivation of the theoretical test loss in the next Section (Section E.2). It also provides intuition about the range of learning rates that are beneficial for model training and offers a deeper understanding of the gradient matrices. Here, we follow the work of Ba et al. (2022) and examine the norm properties of the hidden layers' gradient matrices during a one-step update under both the three-layer and two-layer NN settings. We give the norm analysis of three-layer NN setting as an example, for the norm analysis of two-layer NN setting, please see Appendix C.1.4.

The one-step update equations for the three-layer NN are as follows:

$$\boldsymbol{W}_1^1 = \boldsymbol{W}_1^0 - \eta_1 \boldsymbol{G}_1^0, \quad \boldsymbol{W}_2^1 = \boldsymbol{W}_2^0 - \eta_2 \boldsymbol{G}_2^0, \tag{104}$$

where $\boldsymbol{W}_1^0, \boldsymbol{W}_2^0$ are the initial hidden layer weights, $\eta_1$ and $\eta_2$ are the learning rate for the first layer and second layer, respectively. $\boldsymbol{W}_1^1$ and $\boldsymbol{W}_2^1$ are the updated layer weights. $\boldsymbol{G}_1$ and $\boldsymbol{G}_2$ are the corresponding gradient matrices, where

$$\boldsymbol{G}_1^0 = \underbrace{\frac{1}{nh}\boldsymbol{X}^\top \boldsymbol{X} \boldsymbol{W}_1^0 \boldsymbol{W}_2^0 \boldsymbol{a}\boldsymbol{a}^\top \boldsymbol{W}_2^{0\top}}_{\boldsymbol{B}_1^0} - \underbrace{\frac{1}{n\sqrt{h}}\boldsymbol{X}^\top \boldsymbol{y} \boldsymbol{a}^\top \boldsymbol{W}_2^{0\top}}_{\boldsymbol{A}_1^0}, \tag{105}$$

$$\boldsymbol{G}_2^0 = \underbrace{\frac{1}{nh}\boldsymbol{W}_1^{0\top} \boldsymbol{X}^\top \boldsymbol{X} \boldsymbol{W}_1^0 \boldsymbol{W}_2^0 \boldsymbol{a}\boldsymbol{a}^\top}_{\boldsymbol{B}_2^0} - \underbrace{\frac{1}{n\sqrt{h}}\boldsymbol{W}_1^{0\top} \boldsymbol{X}^\top \boldsymbol{y} \boldsymbol{a}^\top}_{\boldsymbol{A}_2^0}. \tag{106}$$

By analyzing the norm of $\boldsymbol{A}_1^0, \boldsymbol{A}_2^0, \boldsymbol{B}_1^0, \boldsymbol{B}_2^0$, we have the following proposition:

**Proposition E.2.** *(Three-layer NN setting under gaussian initialization.) Under Assumption E.1, there exists some constant $c^* > 0$ such that for all large $n, h$ with probability at least $1 - 32e^{-c^* n} - 30n^4 e^{-c^* \sqrt{n}}$, we have gradient approximation,*

$$\left\|\boldsymbol{G}_1^0 - \boldsymbol{A}_1^0\right\| \leq \frac{1}{\sqrt{n-1}} \left\|\boldsymbol{G}_1^0\right\|,$$

$$\left\|\boldsymbol{G}_2^0 - \boldsymbol{A}_2^0\right\| \leq \frac{1}{\sqrt{n-1}} \left\|\boldsymbol{G}_2^0\right\|. \tag{107}$$

*We obtain the norm control of gradient matrices,*

$$\sqrt{h}\left\|\boldsymbol{G}_1^0\right\| = \Theta_{h,\mathbb{P}}(1), \quad \sqrt{h}\left\|\boldsymbol{G}_1^0\right\|_F = \Theta_{h,\mathbb{P}}(1),$$

$$\sqrt{h}\left\|\boldsymbol{G}_2^0\right\| = \Theta_{h,\mathbb{P}}(1), \quad \sqrt{h}\left\|\boldsymbol{G}_2^0\right\|_F = \Theta_{h,\mathbb{P}}(1). \tag{108}$$

*Thus, we have*

$$Small\ lr : \eta_1 = \Theta(\sqrt{h}) \Rightarrow \left\|\boldsymbol{W}_1^1 - \boldsymbol{W}_1^0\right\| \asymp \left\|\boldsymbol{W}_1^0\right\| \tag{109}$$

$$\eta_2 = \Theta(\sqrt{h}) \Rightarrow \left\|\boldsymbol{W}_2^1 - \boldsymbol{W}_2^0\right\| \asymp \left\|\boldsymbol{W}_2^0\right\| \tag{110}$$

$$Large\ lr : \eta_1 = \Theta(h) \Rightarrow \left\|\boldsymbol{W}_1^1 - \boldsymbol{W}_1^0\right\|_F \asymp \left\|\boldsymbol{W}_1^0\right\|_F \tag{111}$$

$$\eta_2 = \Theta(h) \Rightarrow \left\|\boldsymbol{W}_2^1 - \boldsymbol{W}_2^0\right\|_F \asymp \left\|\boldsymbol{W}_2^0\right\|_F \tag{112}$$

We provide the complete proof in the Appendix C.1.3. A similar result can be obtained for the two-layer NN setting. See Proposition C.9, we provide the proof in Appendix C.1.4.

Proposition E.2 shows that in terms of norm, $\{A_i^0\}_{i=1}^2$ is very close to $\{G_i^0\}_{i=1}^2$, which means $\{A_i^0\}_{i=1}^2$ serves as the leading term in $\{G_i^0\}_{i=1}^2$. This approximation can significantly simplify the subsequent analysis of the theoretical test loss for the three-layer NN when we replace the gradients $\{G_i^0\}_{i=1}^2$ with their approximated version $\{A_i^0\}_{i=1}^2$. For the two-layer NN setting, we similarly replace the original gradient with its leading term, as justified by Proposition C.9 and Lemma E.3, to simplify the test loss analysis.

### E.2. Relationship between Test Loss and Layer-wise Learning Rates

In this section, we first derive the theoretical test loss after a one-step update for both two-layer and three-layer neural networks under our setup. Based on this theoretical test loss, we vary the learning rates for each layer in these two networks. We aim to determine whether using the same learning rates across layers leads to minimal test loss for networks trained with a one-step GD update when $\eta_1 + \eta_2 = 2h^\alpha$, where $h^\alpha \leq$ *Large lr*.

#### E.2.1. TWO-LAYER NEURAL NETWORKS

Given test data $\tilde{x}_0 \sim \mathcal{N}(\mathbf{0}, I_h)$, we consider the test loss

$$L_{\text{two-layer}} = \mathbb{E}_{W_1^0, W_2^0, \xi', M, \tilde{x}_0, X} \left\| \frac{1}{\sqrt{h}} \tilde{x}_0 W_1 W_2 - \tilde{x}_0 M \right\|^2.$$

The key lemma in this subsection uses the approximate gradient to replace the true gradient updates, thereby simplifying the analysis of the test loss.

**Lemma E.3.** *We define the following matrices:*

$$A_1^{0'} = \frac{1}{nh} X^\top Y W_2^{0\top}, B_1^{0'} = \frac{1}{nh^2} X^\top X W_1^0 W_2^0 W_2^{0\top},$$
$$A_2^{0'} = \frac{1}{nh} W_1^{0\top} X^\top Y, B_2^{0'} = \frac{1}{nh^2} W_1^{0\top} X^\top X W_1^0 W_2^0.$$

*We also define* $\widetilde{W_1^1}' = W_1^0 + \eta_1 A_1^{0'}$ *and* $\widetilde{W_2^1}' = W_2^0 + \eta_2 A_2^{0'}$. *Then, under Assumption 3.3 and E.1, for* $\eta_1, \eta_2$ *no more than* $O(h\sqrt{h})$, *we have*

$$\left| L_{\text{two-layer}}(W_1^{1'}, W_2^{1'}) - L_{\text{two-layer}}(\widetilde{W_1^1}', \widetilde{W_2^1}') \right| \leq O(\frac{1}{h}).$$

The simplified analysis leads to the following result for two-layer networks.

**Theorem E.4.** *Given Assumption 3.3, E.1, and in addition assume* $\eta_1$ *and* $\eta_2$ *are no more than* $O(h\sqrt{h})$, *based on Proposition C.9 and Lemma E.3, consider the training procedure discussed in Section 3, we obtain the following test loss after one-step GD update in a two-layer neural network under gaussian initialization:*

$$
\begin{aligned}
L_{\text{two-layer}} = {} & \frac{2\eta_1^2}{h^4} + \frac{2\eta_1^2(1+\rho_e^2)}{nh^3} + \frac{2\eta_2^2}{h^4} + \frac{2\eta_2^2(1+\rho_e^2)}{nh^3} \\
& - 2\frac{\eta_1}{h^2} - 2\frac{\eta_2}{h^2} + \frac{\eta_1^2\eta_2^2}{n^2h^5} + \frac{2\eta_1\eta_2}{h^4} + \frac{2\eta_1\eta_2}{nh^3} + \frac{2\eta_1\eta_2\rho_e^2}{nh^3} \\
& + \frac{\eta_1^2\eta_2^2(\rho_e^2+1)^2}{n^2h^5} + \frac{2\eta_1^2\eta_2^2}{h^7} + \frac{2\eta_1^2\eta_2^2\rho_e^2}{n^3h^4} + 1 + \frac{1}{h} \\
& + O(\frac{\eta_1^2}{h^5}) + O(\frac{\eta_2^5}{h^2}) + O(\frac{\eta_1\eta_2}{h^5}) + O(\frac{\eta_1^2\eta_2^2}{h^8}).
\end{aligned}
\tag{113}
$$

The complete proof is provided in Appendix E.3.

**Analysis of Special Cases.** Here, we consider a special case. Specifically, we take $h = n = d$ and $\rho_e = 0$, under which the loss simplifies to:

$$L_{\text{two-layer}} = \frac{4\eta_1^2}{h^4} + \frac{4\eta_2^2}{h^4} - 2\frac{\eta_1}{h^2} - 2\frac{\eta_2}{h^2} + \frac{4\eta_1\eta_2}{h^4} + \frac{4\eta_1^2\eta_2^2}{h^7}$$
$$+ 1 + \frac{1}{h} + O(\frac{\eta_1^2}{h^5}) + O(\frac{\eta_2^5}{h^2}) + O(\frac{\eta_1\eta_2}{h^5}) + O(\frac{\eta_1^2\eta_2^2}{h^8}). \tag{114}$$

Taking special case ( 114) as an example 114, we obtain the following corollary for two-layer neural network under gaussian initialization.

**Corollary E.5.** *Suppose $\eta_1 + \eta_2 = 2h^\alpha$ and we consider $0 < \alpha \le \frac{3}{2}$. Then, for any $\alpha$ in this range, the point $\eta_1 = \eta_2 = h^\alpha$ is not a local minimum of the loss $L_{two\text{-}layer}(\boldsymbol{W}_1^{1'}, \boldsymbol{W}_2^{1'})$.*

We do simulations in Figure 6 in Appendix F to support Corollary E.5.

E.2.2. THREE-LAYER NEURAL NETWORKS

Given test data $\tilde{\boldsymbol{x}}_0 \sim \mathcal{N}(\boldsymbol{0}, \boldsymbol{I}_d)$, we consider the test loss

$$L_{\text{three-layer}} = \mathbb{E}_{\boldsymbol{W}_1^0, \boldsymbol{W}_2^0, \boldsymbol{a}, \boldsymbol{\xi}, \tilde{\boldsymbol{x}}_0, \boldsymbol{X}} \left( \frac{1}{\sqrt{h}} \tilde{\boldsymbol{x}}_0 \boldsymbol{W}_1 \boldsymbol{W}_2 \boldsymbol{a} - \tilde{\boldsymbol{x}}_0 \boldsymbol{\beta}^* \right)^2$$

**Theorem E.6.** *Given Assumption 3.3, E.1, and in addition assume $\eta_1$ and $\eta_2$ are no more than $O(h)$ based on Proposition E.2, consider the training procedure discussed in Section 3, we derive the test loss after one-step GD update in a three-layer neural network:*

$$L_{three\text{-}layer} = \frac{\eta_1^2}{h^2} + \frac{\eta_1^2(1 + \rho_e^2)}{hn} - 2\frac{\eta_1}{h} + \frac{2\eta_2^2}{h^2}$$
$$+ \frac{2\eta_2^2(1 + \rho_e^2)}{nh} - 2\frac{\eta_2}{h} + \frac{2\eta_1\eta_2}{h^2} + \frac{2\eta_1\eta_2(1 + \rho_e^2)}{nh}$$
$$+ \frac{\eta_1^2\eta_2^2\rho_e^2}{nh^3} + \frac{\eta_1^2\eta_2^2\rho_e^2}{n^2h^2} + \frac{4\eta_1^2\eta_2^2}{n^2h^2} + 1$$
$$+ O\left(\frac{\eta_1^2}{h^3}\right) + O\left(\frac{\eta_2^2}{h^3}\right) + O\left(\frac{\eta_1\eta_2}{h^3}\right) + O\left(\frac{\eta_1^2\eta_2^2}{h^5}\right). \tag{115}$$

The complete proof is provided in Appendix E.4.

**Analysis of Special Cases.** Here we consider a special case. Specifically, we take $h = n = d$ and $\rho_e = 0$, under which the loss becomes:

$$L_{\text{three-layer}} = \frac{2\eta_1^2}{h^2} - \frac{2\eta_1}{h} + \frac{4\eta_2^2}{h^2} - \frac{2\eta_2}{h} + \frac{4\eta_1\eta_2}{h^2} + \frac{4\eta_1^2\eta_2^2 h}{n^2h^3}$$
$$+ 1 + O\left(\frac{\eta_1^2}{h^3}\right) + O\left(\frac{\eta_2^2}{h^3}\right) + O\left(\frac{\eta_1\eta_2}{h^3}\right) + O\left(\frac{\eta_1^2\eta_2^2}{h^5}\right). \tag{116}$$

Taking special case ( 116) as an example, we obtain the following corollary for three-layer neural network under gaussian initialization.

**Corollary E.7.** *Suppose $\eta_1 + \eta_2 = 2h^\alpha$ and we consider $0 < \alpha < 1$. Then, for any $\alpha$ in this range, the point $\eta_1 = \eta_2 = h^\alpha$ is not a local minimum of the loss $L_{three\text{-}layer}(\boldsymbol{W}_1^1, \boldsymbol{W}_2^1)$.*

We do simulations in Figure 8 in Appendix F to support Corollary E.7.

### E.3. Two-layer NN Test Loss under Gaussian initialization

**Lemma E.8.** *Consider that*

$$A_1^0 = \frac{1}{nh} X^\top Y W_2^{0\top}$$

$$B_1^0 = \frac{1}{nh^2} X^\top X W_1^0 W_2^0 W_2^{0\top}$$

$$A_2^0 = \frac{1}{nh} W_1^{0\top} X^\top Y$$

$$B_2^0 = \frac{1}{nh^2} W_1^{0\top} X^\top X W_1^0 W_2^0,$$

*under Assumption 3.3, E.1, consider $\widetilde{W_1^1} = W_1^0 + \eta_1 A_1^0$ and $\widetilde{W_2^1} = W_2^0 + \eta_2 A_2^0$, then for $\eta_1, \eta_2 \sim O(h\sqrt{h})$, we have*

$$\left| L_{\text{two-layer}}(X, W_1^1, W_2^1, \tilde{x}_0) - L_{\text{two-layer}}(X, \widetilde{W_1^1}, \widetilde{W_2^1}, \tilde{x}_0) \right| \le O(\frac{1}{h})$$

**Proof of Lemma E.8.**

$$\left| \sqrt{L_{\text{two-layer}}(X, W_1^1, W_2^1, \tilde{x}_0)} - \sqrt{L_{\text{two-layer}}(X, \widetilde{W_1^1}, \widetilde{W_2^1}, \tilde{x}_0)} \right|$$

$$= \left| \mathbb{E}_{W_1^0, W_2^0, \xi, \tilde{x}_0, X} \left\| \frac{1}{h} \tilde{x}_0 W_1^1 W_2^1 - \tilde{x}_0 M \right\|_F - \mathbb{E}_{W_1^0, W_2^0, \xi, \tilde{x}_0, X} \left\| \frac{1}{h} \tilde{x}_0 \widetilde{W_1^1} \widetilde{W_2^1} - \tilde{x}_0 M \right\|_F \right|$$

$$\le \left| \mathbb{E}_{W_1^0, W_2^0, \xi, \tilde{x}_0, X} \left( \left\| \frac{1}{h} \tilde{x}_0 W_1^1 W_2^1 - \frac{1}{h} \tilde{x}_0 \widetilde{W_1^1} \widetilde{W_2^1} \right\|_F \right) \right|$$

$$\le \left| \mathbb{E}_{W_1^0, W_2^0, \xi, \tilde{x}_0, X} \left( \left\| \frac{1}{h} \tilde{x}_0 \right\|_F \left\| W_1^1 W_2^1 - \widetilde{W_1^1} \widetilde{W_2^1} \right\| \right) \right| \tag{117}$$

$$= \sqrt{\frac{1}{h}} \left| \mathbb{E}_{W_1^0, W_2^0, \xi, \tilde{x}_0, X} \left( \left\| W_1^1 W_2^1 - \widetilde{W_1^1} \widetilde{W_2^1} \right\| \right) \right|$$

$$= \sqrt{\frac{1}{h}} \left| \mathbb{E}_{W_1^0, W_2^0, \xi \tilde{x}_0, X} \left( \left\| -\eta_1 B_1^0 W_2^0 - \eta_1 \eta_2 B_1^0 A_2^0 - \eta_2 W_1^0 B_2^0 - \eta_1 \eta_2 A_1^0 B_2^0 + \eta_1 \eta_2 B_1^0 B_2^0 \right\| \right) \right|$$

$$\le \sqrt{\frac{1}{h}} \left| \mathbb{E} \left( \left\| \eta_1 B_1^0 W_2^0 \right\| + \left\| \eta_1 \eta_2 B_1^0 A_2^0 \right\| + \left\| \eta_2 W_1^0 B_2^0 \right\| + \left\| \eta_1 \eta_2 A_1^0 B_2^0 \right\| + \left\| \eta_1 \eta_2 B_1^0 B_2^0 \right\| \right) \right|$$

Consider similar techniques in Lemma C.6, we get that

$$\mathbb{E}_{W_1^0, W_2^0, \xi, \tilde{x}_0, X} \left\| \eta_1 B_1^0 W_2^0 \right\| \le \eta_1 \left\| B_1^0 \right\| \left\| W_2^0 \right\| \le O(\frac{\eta_1}{h^2}),$$

$$\mathbb{E}_{W_1^0, W_2^0, \xi, \tilde{x}_0, X} \left\| \eta_1 \eta_2 B_1^0 A_2^0 \right\| \le \eta_1 \eta_2 \left\| B_1^0 \right\| \left\| A_2^0 \right\| \le O(\frac{\eta_1 \eta_2}{h^3 \sqrt{h}}),$$

$$\mathbb{E}_{W_1^0, W_2^0, \xi, \tilde{x}_0, X} \left\| \eta_2 W_1^0 B_2^0 \right\| \le \eta_2 \left\| B_2^0 \right\| \left\| W_1^0 \right\| \le O(\frac{\eta_2}{h^2}),$$

$$\mathbb{E}_{W_1^0, W_2^0, \xi, \tilde{x}_0, X} \left\| \eta_1 \eta_2 A_1^0 B_2^0 \right\| \le \eta_1 \eta_2 \left\| B_2^0 \right\| \left\| A_1^0 \right\| \le O(\frac{\eta_1 \eta_2}{h^3 \sqrt{h}}),$$

$$\mathbb{E}_{W_1^0, W_2^0, \xi, \tilde{x}_0, X} \left\| \eta_1 \eta_2 B_1^0 B_2^0 \right\| \le \eta_1 \eta_2 \left\| B_2^0 \right\| \left\| B_1^0 \right\| \le O(\frac{\eta_1 \eta_2}{h^4}),$$

taking these inequalities into (117), we have

$$\left| \sqrt{L_{\text{two-layer}}(X, W_1^1, W_2^1, \tilde{x}_0)} - \sqrt{L_{\text{two-layer}}(X, \widetilde{W_1^1}, \widetilde{W_2^1}, \tilde{x}_0)} \right|$$

$$\le \sqrt{\frac{1}{h}} \left| \mathbb{E} \left( \left\| \eta_1 B_1^0 W_2^0 \right\| + \left\| \eta_1 \eta_2 B_1^0 A_2^0 \right\| + \left\| \eta_2 W_1^0 B_2^0 \right\| + \left\| \eta_1 \eta_2 A_1^0 B_2^0 \right\| + \left\| \eta_1 \eta_2 B_1^0 B_2^0 \right\| \right) \right| \tag{118}$$

$$\le O(\frac{\eta_1 + \eta_2}{h^2 \sqrt{h}}) + O(\frac{\eta_1 \eta_2}{h^4})$$

Also, based on the Assumption 3.3 and theorem E.9, we have

$$\left| \sqrt{L_{\text{two-layer}}(\boldsymbol{X}, \boldsymbol{W}_1^1, \boldsymbol{W}_2^1, \tilde{\boldsymbol{x}}_0)} + \sqrt{L_{\text{two-layer}}(\boldsymbol{X}, \widetilde{\boldsymbol{W}_1^1}, \widetilde{\boldsymbol{W}_2^1}, \tilde{\boldsymbol{x}}_0)} \right| \le O(1). \tag{119}$$

We combine (118), (119) and Assumption E.1, and assume $\eta_1, \eta_2 \sim O(h\sqrt{h})$, finally we get that

$$\left| L_{\text{two-layer}}(\boldsymbol{X}, \boldsymbol{W}_1^1, \boldsymbol{W}_2^1, \tilde{\boldsymbol{x}}_0) - L_{\text{two-layer}}(\boldsymbol{X}, \widetilde{\boldsymbol{W}_1^1}, \widetilde{\boldsymbol{W}_2^1}, \tilde{\boldsymbol{x}}_0) \right| \le O(\frac{\eta_1 + \eta_2}{h^2\sqrt{h}}) \le O(\frac{1}{h})$$

$\square$

**Theorem E.9.** *Given Assumption 3.3, E.1, and in addition assume $\eta_1$ and $\eta_2$ are no more than $O(h\sqrt{h})$ based on Proposition C.9, consider training procedure discussed in section 3, we derive the test loss after one-step GD update in a two-layer neural network under guassian initialization:*

$$\begin{aligned}
L_{\text{two-layer}} =\ & \frac{2\eta_1^2}{h^4} + \frac{2\eta_1^2(1+\rho_e^2)}{nh^3} - 2\frac{\eta_1}{h^2} \\
& + \frac{2\eta_2^2}{h^4} + \frac{2\eta_2^2(1+\rho_e^2)}{nh^3} - 2\frac{\eta_2}{h^2} + \frac{\eta_1^2\eta_2^2}{n^2h^5} \\
& + \frac{2\eta_1\eta_2}{h^4} + \frac{2\eta_1\eta_2}{nh^3} + \frac{2\eta_1\eta_2\rho_e^2}{nh^3} + \frac{\eta_1^2\eta_2^2(\rho_e^2+1)^2}{n^2h^5} + \frac{2\eta_1^2\eta_2^2}{h^7} + \frac{2\eta_1^2\eta_2^2\rho_e^2}{n^3h^4} \\
& + 1 + \frac{1}{h} + O(\frac{\eta_1^2}{h^5}) + O(\frac{\eta_2^5}{h^2}) + O(\frac{\eta_1\eta_2}{h^5}) + O(\frac{\eta_1^2\eta_2^2}{h^8})
\end{aligned} \tag{120}$$

**Proof of Theorem E.4.** We consider a test data $\tilde{\boldsymbol{x}}_0 \sim \mathcal{N}(\boldsymbol{0}, \boldsymbol{I}_d) \in \mathbb{R}^{1\times d}$ under two-layer setting , we have

$$\begin{aligned}
& L_{\text{two-layer}}(\boldsymbol{X}, \boldsymbol{W}_1^1, \boldsymbol{W}_2^1, \tilde{\boldsymbol{x}}_0) \\
=& \mathbb{E}_{\boldsymbol{W}_1^0, \boldsymbol{W}_2^0, \boldsymbol{\xi}, \tilde{\boldsymbol{x}}_0, \boldsymbol{X}} \left\| \frac{1}{h}\tilde{\boldsymbol{x}}_0\boldsymbol{W}_1^1\boldsymbol{W}_2^1 - \tilde{\boldsymbol{x}}_0\boldsymbol{M} \right\|_F^2 \\
=& tr\left( \mathbb{E}_{\boldsymbol{W}_1^0, \boldsymbol{W}_2^0, \boldsymbol{\xi}, \tilde{\boldsymbol{x}}_0, \boldsymbol{X}} \left[ \left( \frac{1}{h}\boldsymbol{W}_1^1\boldsymbol{W}_2^1 - \boldsymbol{M} \right)^\top \tilde{\boldsymbol{x}}_0^\top\tilde{\boldsymbol{x}}_0 \left( \frac{1}{h}\boldsymbol{W}_1^1\boldsymbol{W}_2^1 - \boldsymbol{M} \right) \right] \right) \\
=& tr\left( \mathbb{E}_{\boldsymbol{W}_1^0, \boldsymbol{W}_2^0, \boldsymbol{\xi}, \tilde{\boldsymbol{x}}_0, \boldsymbol{X}} \left[ \tilde{\boldsymbol{x}}_0^\top\tilde{\boldsymbol{x}}_0 \left( \frac{1}{h}\boldsymbol{W}_1^1\boldsymbol{W}_2^1 - \boldsymbol{M} \right) \left( \frac{1}{h}\boldsymbol{W}_1^1\boldsymbol{W}_2^1 - \boldsymbol{M} \right)^\top \right] \right) \\
=& tr\left( \mathbb{E}_{\boldsymbol{W}_1^0, \boldsymbol{W}_2^0, \boldsymbol{\xi}, \boldsymbol{X}} \left[ \left( \frac{1}{h}\boldsymbol{W}_1^1\boldsymbol{W}_2^1 - \boldsymbol{M} \right) \left( \frac{1}{h}\boldsymbol{W}_1^1\boldsymbol{W}_2^1 - \boldsymbol{M} \right)^\top \right] \right) \\
=& tr\left( \mathbb{E}_{\boldsymbol{W}_1^0, \boldsymbol{W}_2^0, \boldsymbol{\xi}, \boldsymbol{X}} \left[ \frac{1}{h^2}\boldsymbol{W}_1^1\boldsymbol{W}_2^1\boldsymbol{W}_2^{1\top}\boldsymbol{W}_1^{1\top} \right] \right) \\
& -tr\left( \mathbb{E}_{\boldsymbol{W}_1^0, \boldsymbol{W}_2^0, \boldsymbol{\xi}, \boldsymbol{X}} \left[ \frac{1}{h}\boldsymbol{M}\boldsymbol{W}_2^{1\top}\boldsymbol{W}_1^{1\top} \right] \right) \\
& -tr\left( \mathbb{E}_{\boldsymbol{W}_1^0, \boldsymbol{W}_2^0, \boldsymbol{\xi}, \boldsymbol{X}} \left[ \frac{1}{h}\boldsymbol{W}_1^1\boldsymbol{W}_2^1\boldsymbol{M}^\top \right] \right) \\
& +tr\left( \mathbb{E}\left[ \boldsymbol{M}\boldsymbol{M}^\top \right] \right).
\end{aligned} \tag{121}$$

Here we define $L_1, L_2, L_3, L_4$, where

$$\begin{aligned}
L_1 &= tr\left( \mathbb{E}_{\boldsymbol{W}_1^0, \boldsymbol{W}_2^0, \boldsymbol{\xi}, \boldsymbol{X}} \left[ \frac{1}{h^2}\boldsymbol{W}_1^1\boldsymbol{W}_2^1\boldsymbol{W}_2^{1\top}\boldsymbol{W}_1^{1\top} \right] \right) \\
L_2 &= tr\left( \mathbb{E}_{\boldsymbol{W}_1^0, \boldsymbol{W}_2^0, \boldsymbol{\xi}, \boldsymbol{X}} \left[ \frac{1}{h}\boldsymbol{M}\boldsymbol{W}_2^{1\top}\boldsymbol{W}_1^{1\top} \right] \right) \\
L_3 &= tr\left( \mathbb{E}_{\boldsymbol{W}_1^0, \boldsymbol{W}_2^0, \boldsymbol{\xi}, \boldsymbol{X}} \left[ \frac{1}{h}\boldsymbol{W}_1^1\boldsymbol{W}_2^1\boldsymbol{M}^\top \right] \right) \\
L_4 &= tr\left( \mathbb{E}\left[ \boldsymbol{M}\boldsymbol{M}^\top \right] \right)
\end{aligned}$$

Thus

$$L_{\text{two-layer}} = L_1 - L_2 - L_3 + L_4$$

Consider the exact gradient update,

$$\boldsymbol{W}_1^1 = \boldsymbol{W}_1^0 + \eta_1 \boldsymbol{A}_1^0 - \eta_1 \boldsymbol{B}_1^0$$
$$\boldsymbol{W}_2^1 = \boldsymbol{W}_2^0 + \eta_2 \boldsymbol{A}_2^0 - \eta_2 \boldsymbol{B}_2^0.$$

We have

$$\boldsymbol{W}_1^1 \boldsymbol{W}_2^1 = \boldsymbol{W}_1^0 \boldsymbol{W}_2^0 + \eta_1 \boldsymbol{A}_1^0 \boldsymbol{W}_2^0 - \eta_1 \boldsymbol{B}_1^0 \boldsymbol{W}_2^0 + \eta_2 \boldsymbol{W}_1^0 \boldsymbol{A}_2^0 + \eta_1 \eta_2 \boldsymbol{A}_1^0 \boldsymbol{A}_2^0$$
$$- \eta_1 \eta_2 \boldsymbol{B}_1^0 \boldsymbol{A}_2^0 - \eta_2 \boldsymbol{W}_1^0 \boldsymbol{B}_2^0 - \eta_1 \eta_2 \boldsymbol{A}_1^0 \boldsymbol{B}_2^0 + \eta_1 \eta_2 \boldsymbol{B}_1^0 \boldsymbol{B}_2^0,$$

where

$$\boldsymbol{A}_1^0 \boldsymbol{B}_2^0 = \frac{1}{n^2 h^3} \boldsymbol{X}^\top \boldsymbol{Y} \boldsymbol{W}_2^{0\top} \boldsymbol{W}_1^{0\top} \boldsymbol{X}^\top \boldsymbol{X} \boldsymbol{W}_1^0 \boldsymbol{W}_2^0$$

$$\boldsymbol{B}_1^0 \boldsymbol{A}_2^0 = \frac{1}{n^2 h^3} \boldsymbol{X}^\top \boldsymbol{X} \boldsymbol{W}_1^0 \boldsymbol{W}_2^0 \boldsymbol{W}_2^{0\top} \boldsymbol{W}_1^{0\top} \boldsymbol{X}^\top \boldsymbol{Y}$$

$$\boldsymbol{B}_1^0 \boldsymbol{B}_2^0 = \frac{1}{n^2 h^4} \boldsymbol{X}^\top \boldsymbol{X} \boldsymbol{W}_1^0 \boldsymbol{W}_2^0 \boldsymbol{W}_2^{0\top} \boldsymbol{W}_1^{0\top} \boldsymbol{X}^\top \boldsymbol{X} \boldsymbol{W}_1^0 \boldsymbol{W}_2^0.$$

Based on Lemma E.8, we consider replacing $\boldsymbol{W}_2^1, \boldsymbol{W}_2^1$ with $\widetilde{\boldsymbol{W}_1^1}, \widetilde{\boldsymbol{W}_2^1}$.

Thus we have

$$\boldsymbol{W}_1^1 \boldsymbol{W}_2^1 \approx \widetilde{\boldsymbol{W}_1^1} \widetilde{\boldsymbol{W}_2^1} = \boldsymbol{W}_1^0 \boldsymbol{W}_2^0 + \frac{\eta_1}{nh} \boldsymbol{X}^\top \boldsymbol{Y} \boldsymbol{W}_2^{0\top} \boldsymbol{W}_2^0$$
$$+ \frac{\eta_2}{nh} \boldsymbol{W}_1^0 \boldsymbol{W}_1^{0\top} \boldsymbol{X}^\top \boldsymbol{Y}$$
$$+ \frac{\eta_1 \eta_2}{n^2 h^2} \boldsymbol{X}^\top \boldsymbol{Y} \boldsymbol{W}_2^{0\top} \boldsymbol{W}_1^{0\top} \boldsymbol{X}^\top \boldsymbol{Y}$$

$$\boldsymbol{W}_2^{1\top} \boldsymbol{W}_1^{1\top} \approx \left( \widetilde{\boldsymbol{W}_1^1} \widetilde{\boldsymbol{W}_2^1} \right)^\top = \boldsymbol{W}_2^{0\top} \boldsymbol{W}_1^{0\top} + \frac{\eta_1}{nh} \boldsymbol{W}_2^{0\top} \boldsymbol{W}_2^0 \boldsymbol{Y}^\top \boldsymbol{X}$$
$$+ \frac{\eta_2}{nh} \boldsymbol{Y}^\top \boldsymbol{X} \boldsymbol{W}_1^0 \boldsymbol{W}_1^{0\top}$$
$$+ \frac{\eta_1 \eta_2}{n^2 h^2} \boldsymbol{Y}^\top \boldsymbol{X} \boldsymbol{W}_1^0 \boldsymbol{W}_2^0 \boldsymbol{Y}^\top \boldsymbol{X}$$

We have $L_1 = \sum_{i=1}^{16} T_i$, where

$$T_1 = tr\left(\mathbb{E}_{\boldsymbol{W}_1^0, \boldsymbol{W}_2^0, \boldsymbol{\xi}, \boldsymbol{X}}\left[\frac{1}{h^2}\boldsymbol{W}_2^{0\top}\boldsymbol{W}_1^{0\top}\boldsymbol{W}_1^0\boldsymbol{W}_2^0\right]\right),$$

$$T_2 = tr\left(\mathbb{E}_{\boldsymbol{W}_1^0, \boldsymbol{W}_2^0, \boldsymbol{\xi}, \boldsymbol{X}}\left[\frac{\eta_1}{nh^3}\boldsymbol{W}_2^{0\top}\boldsymbol{W}_1^{0\top}\boldsymbol{X}^\top\boldsymbol{Y}\boldsymbol{W}_2^{0\top}\boldsymbol{W}_2^0\right]\right),$$

$$T_3 = tr\left(\mathbb{E}_{\boldsymbol{W}_1^0, \boldsymbol{W}_2^0, \boldsymbol{\xi}, \boldsymbol{X}}\left[\frac{\eta_2}{nh^2}\boldsymbol{W}_2^{0\top}\boldsymbol{W}_1^{0\top}\boldsymbol{W}_1^0\boldsymbol{W}_1^{0\top}\boldsymbol{X}^\top\boldsymbol{Y}\right]\right),$$

$$T_4 = tr\left(\mathbb{E}_{\boldsymbol{W}_1^0, \boldsymbol{W}_2^0, \boldsymbol{\xi}, \boldsymbol{X}}\left[\frac{\eta_1\eta_2}{n^2h^4}\boldsymbol{W}_2^{0\top}\boldsymbol{W}_1^{0\top}\boldsymbol{X}^\top\boldsymbol{Y}\boldsymbol{W}_2^{0\top}\boldsymbol{W}_1^{0\top}\boldsymbol{X}^\top\boldsymbol{Y}\right]\right),$$

$$T_5 = tr\left(\mathbb{E}_{\boldsymbol{W}_1^0, \boldsymbol{W}_2^0, \boldsymbol{\xi}, \boldsymbol{X}}\left[\frac{\eta_1}{nh^3}\boldsymbol{W}_2^{0\top}\boldsymbol{W}_2^0\boldsymbol{Y}^\top\boldsymbol{X}\boldsymbol{W}_1^0\boldsymbol{W}_2^0\right]\right),$$

$$T_6 = tr\left(\mathbb{E}_{\boldsymbol{W}_1^0, \boldsymbol{W}_2^0, \boldsymbol{\xi}, \boldsymbol{X}}\left[\frac{\eta_1^2}{n^2h^4}\boldsymbol{W}_2^{0\top}\boldsymbol{W}_2^0\boldsymbol{Y}^\top\boldsymbol{X}\boldsymbol{X}^\top\boldsymbol{Y}\boldsymbol{W}_2^{0\top}\boldsymbol{W}_2^0\right]\right),$$

$$T_7 = tr\left(\mathbb{E}_{\boldsymbol{W}_1^0, \boldsymbol{W}_2^0, \boldsymbol{\xi}, \boldsymbol{X}}\left[\frac{\eta_1\eta_2}{n^2h^4}\boldsymbol{W}_2^{0\top}\boldsymbol{W}_2^0\boldsymbol{Y}^\top\boldsymbol{X}\boldsymbol{W}_1^0\boldsymbol{W}_1^{0\top}\boldsymbol{X}^\top\boldsymbol{Y}\right]\right),$$

$$T_8 = tr\left(\mathbb{E}_{\boldsymbol{W}_1^0, \boldsymbol{W}_2^0, \boldsymbol{\xi}, \boldsymbol{X}}\left[\frac{\eta_1^2\eta_2}{n^3h^5}\boldsymbol{W}_2^{0\top}\boldsymbol{W}_2^0\boldsymbol{Y}^\top\boldsymbol{X}\boldsymbol{X}^\top\boldsymbol{Y}\boldsymbol{W}_2^{0\top}\boldsymbol{W}_1^{0\top}\boldsymbol{X}^\top\boldsymbol{Y}\right]\right),$$

$$T_9 = tr\left(\mathbb{E}_{\boldsymbol{W}_1^0, \boldsymbol{W}_2^0, \boldsymbol{\xi}, \boldsymbol{X}}\left[\frac{\eta_2}{nh^3}\boldsymbol{Y}^\top\boldsymbol{X}\boldsymbol{W}_1^0\boldsymbol{W}_1^{0\top}\boldsymbol{W}_1^0\boldsymbol{W}_2^0\right]\right),$$

$$T_{10} = tr\left(\mathbb{E}_{\boldsymbol{W}_1^0, \boldsymbol{W}_2^0, \boldsymbol{\xi}, \boldsymbol{X}}\left[\frac{\eta_1\eta_2}{n^2h^4}\boldsymbol{Y}^\top\boldsymbol{X}\boldsymbol{W}_1^0\boldsymbol{W}_1^{0\top}\boldsymbol{X}^\top\boldsymbol{Y}\boldsymbol{W}_2^{0\top}\boldsymbol{W}_2^0\right]\right),$$

$$T_{11} = tr\left(\mathbb{E}_{\boldsymbol{W}_1^0, \boldsymbol{W}_2^0, \boldsymbol{\xi}, \boldsymbol{X}}\left[\frac{\eta_2^2}{n^2h^4}\boldsymbol{Y}^\top\boldsymbol{X}\boldsymbol{W}_1^0\boldsymbol{W}_1^{0\top}\boldsymbol{W}_1^0\boldsymbol{W}_1^{0\top}\boldsymbol{X}^\top\boldsymbol{Y}\right]\right),$$

$$T_{12} = tr\left(\mathbb{E}_{\boldsymbol{W}_1^0, \boldsymbol{W}_2^0, \boldsymbol{\xi}, \boldsymbol{X}}\left[\frac{\eta_1\eta_2^2}{n^3h^5\sqrt{h}}\boldsymbol{Y}^\top\boldsymbol{X}\boldsymbol{W}_1^0\boldsymbol{W}_1^{0\top}\boldsymbol{X}^\top\boldsymbol{Y}\boldsymbol{W}_2^{0\top}\boldsymbol{W}_1^{0\top}\boldsymbol{X}^\top\boldsymbol{Y}\right]\right),$$

$$T_{13} = tr\left(\mathbb{E}_{\boldsymbol{W}_1^0, \boldsymbol{W}_2^0, \boldsymbol{\xi}, \boldsymbol{X}}\left[\frac{\eta_1\eta_2}{n^2h^4}\boldsymbol{Y}^\top\boldsymbol{X}\boldsymbol{W}_1^0\boldsymbol{W}_2^0\boldsymbol{Y}^\top\boldsymbol{X}\boldsymbol{W}_1^0\boldsymbol{W}_2^0\right]\right),$$

$$T_{14} = tr\left(\mathbb{E}_{\boldsymbol{W}_1^0, \boldsymbol{W}_2^0, \boldsymbol{\xi}, \boldsymbol{X}}\left[\frac{\eta_1^2\eta_2}{n^3h^5\sqrt{h}}\boldsymbol{Y}^\top\boldsymbol{X}\boldsymbol{W}_1^0\boldsymbol{W}_2^0\boldsymbol{Y}^\top\boldsymbol{X}\boldsymbol{X}^\top\boldsymbol{Y}\boldsymbol{W}_2^{0\top}\boldsymbol{W}_2^0\right]\right),$$

$$T_{15} = tr\left(\mathbb{E}_{\boldsymbol{W}_1^0, \boldsymbol{W}_2^0, \boldsymbol{\xi}, \boldsymbol{X}}\left[\frac{\eta_1\eta_2^2}{n^3h^5\sqrt{h}}\boldsymbol{Y}^\top\boldsymbol{X}\boldsymbol{W}_1^0\boldsymbol{W}_2^0\boldsymbol{Y}^\top\boldsymbol{X}\boldsymbol{W}_1^0\boldsymbol{W}_1^{0\top}\boldsymbol{X}^\top\boldsymbol{Y}\right]\right),$$

$$T_{16} = tr\left(\mathbb{E}_{\boldsymbol{W}_1^0, \boldsymbol{W}_2^0, \boldsymbol{\xi}, \boldsymbol{X}}\left[\frac{\eta_1^2\eta_2^2}{n^4h^6}\boldsymbol{Y}^\top\boldsymbol{X}\boldsymbol{W}_1^0\boldsymbol{W}_2^0\boldsymbol{Y}^\top\boldsymbol{X}\boldsymbol{X}^\top\boldsymbol{Y}\boldsymbol{W}_2^{0\top}\boldsymbol{W}_1^{0\top}\boldsymbol{X}^\top\boldsymbol{Y}\right]\right).$$

We have $L_2 = \sum_{i=17}^{20} T_i$, where

$$T_{17} = tr\left(\mathbb{E}_{\boldsymbol{W}_1^0, \boldsymbol{W}_2^0, \boldsymbol{\xi}, \boldsymbol{X}}\left[\frac{1}{h}\boldsymbol{M}\boldsymbol{W}_2^{0\top}\boldsymbol{W}_1^{0\top}\right]\right),$$

$$T_{18} = tr\left(\mathbb{E}_{\boldsymbol{W}_1^0, \boldsymbol{W}_2^0, \boldsymbol{\xi}, \boldsymbol{X}}\left[\frac{\eta_1}{nh^2}\boldsymbol{M}\boldsymbol{W}_2^{0\top}\boldsymbol{W}_2^0\boldsymbol{Y}^\top\boldsymbol{X}\right]\right),$$

$$T_{19} = tr\left(\mathbb{E}_{\boldsymbol{W}_1^0, \boldsymbol{W}_2^0, \boldsymbol{\xi}, \boldsymbol{X}}\left[\frac{\eta_2}{nh^2}\boldsymbol{M}\boldsymbol{Y}^\top\boldsymbol{X}\boldsymbol{W}_1^0\boldsymbol{W}_1^{0\top}\right]\right),$$

$$T_{20} = tr\left(\mathbb{E}_{\boldsymbol{W}_1^0, \boldsymbol{W}_2^0, \boldsymbol{\xi}, \boldsymbol{X}}\left[\frac{\eta_1\eta_2}{n^2h^3}\boldsymbol{M}\boldsymbol{Y}^\top\boldsymbol{X}\boldsymbol{W}_1^0\boldsymbol{W}_2^0\boldsymbol{Y}^\top\boldsymbol{X}\right]\right).$$

We have $L_3 = \sum_{i=21}^{24} T_i$, where

$$T_{21} = tr\left(\mathbb{E}_{\boldsymbol{W}_1^0, \boldsymbol{W}_2^0, \boldsymbol{\xi}, \boldsymbol{X}}\left[\frac{1}{h}\boldsymbol{W}_1^0\boldsymbol{W}_2^0\boldsymbol{M}^\top\right]\right),$$

$$T_{22} = tr\left(\mathbb{E}_{\boldsymbol{W}_1^0, \boldsymbol{W}_2^0, \boldsymbol{\xi}, \boldsymbol{X}}\left[\frac{\eta_1}{nh^2}\boldsymbol{X}^\top\boldsymbol{Y}\boldsymbol{W}_2^{0\top}\boldsymbol{W}_2^0\boldsymbol{M}^\top\right]\right),$$

$$T_{23} = tr\left(\mathbb{E}_{\boldsymbol{W}_1^0, \boldsymbol{W}_2^0, \boldsymbol{\xi}, \boldsymbol{X}}\left[\frac{\eta_2}{nh^2}\boldsymbol{W}_1^0\boldsymbol{W}_1^{0\top}\boldsymbol{X}^\top\boldsymbol{Y}\boldsymbol{M}^\top\right]\right),$$

$$T_{24} = tr\left(\mathbb{E}_{\boldsymbol{W}_1^0, \boldsymbol{W}_2^0, \boldsymbol{\xi}, \boldsymbol{X}}\left[\frac{\eta_1\eta_2}{n^2h^3}\boldsymbol{X}^\top\boldsymbol{Y}\boldsymbol{W}_2^{0\top}\boldsymbol{W}_1^{0\top}\boldsymbol{X}^\top\boldsymbol{Y}\boldsymbol{M}^\top\right]\right),$$

Thus, we obtain that

$$L_{\text{two-layer}} = \sum_{i=1}^{16} T_i - \sum_{i=17}^{20} T_i - \sum_{i=21}^{24} T_i + L_4$$

**Analysis of $T_1$.**

$$T_1 = tr\left(\mathbb{E}_{\boldsymbol{W}_1^0, \boldsymbol{W}_2^0, \boldsymbol{\xi}, \boldsymbol{X}}\left[\frac{1}{h^2}\boldsymbol{W}_2^{0\top}\boldsymbol{W}_1^{0\top}\boldsymbol{W}_1^0\boldsymbol{W}_2^0\right]\right) = \frac{1}{h} \tag{122}$$

$\square$

**Analysis of $T_4$ and $T_{13}$.**

$$T_4 = tr\left(\mathbb{E}_{\boldsymbol{W}_1^0, \boldsymbol{W}_2^0, \boldsymbol{\xi}, \boldsymbol{X}}\left[\frac{\eta_1\eta_2}{n^2h^4}\boldsymbol{W}_2^{0\top}\boldsymbol{W}_1^{0\top}\boldsymbol{X}^\top\boldsymbol{Y}\boldsymbol{W}_2^{0\top}\boldsymbol{W}_1^{0\top}\boldsymbol{X}^\top\boldsymbol{Y}\right]\right)$$

$$= \frac{\eta_1\eta_2}{n^2h^4}\mathbb{E}\sum_{i=1}^{h}\sum_{k=1}^{h}\sum_{m=1}^{d}\sum_{p=1}^{n}\sum_{q=1}^{d}\sum_{t=1}^{h}\sum_{k=1}^{h}\sum_{m=1}^{d}\sum_{p=1}^{n}\sum_{q=1}^{d}$$

$$\boldsymbol{W}_{2\,ki}^0\boldsymbol{W}_{1\,mk}^0\boldsymbol{X}_{pm}\boldsymbol{X}_{pq}\boldsymbol{M}_{qt}\boldsymbol{W}_{2\,kt}^0\boldsymbol{W}_{1\,mk}^0\boldsymbol{X}_{pm}\boldsymbol{X}_{pq}\boldsymbol{M}_{qi} \tag{123}$$

$$+ \frac{\eta_1\eta_2}{n^2h^4}\mathbb{E}\sum_{i=1}^{h}\sum_{k=1}^{h}\sum_{m=1}^{d}\sum_{p=1}^{n}\sum_{t=1}^{h}\sum_{k=1}^{h}\sum_{m=1}^{d}\sum_{p=1}^{n}$$

$$\boldsymbol{W}_{2\,ki}^0\boldsymbol{W}_{1\,mk}^0\boldsymbol{X}_{pm}\boldsymbol{\xi}_{pt}\boldsymbol{W}_{2\,kt}^0\boldsymbol{W}_{1\,mk}^0\boldsymbol{X}_{pm}\boldsymbol{\xi}_{pi}$$

For $Term_1 = \frac{\eta_1\eta_2}{n^2h^4}\mathbb{E}\sum_{i=1}^{h}\sum_{k=1}^{h}\sum_{m=1}^{d}\sum_{p=1}^{n}\sum_{q=1}^{d}\sum_{t=1}^{h}\sum_{k=1}^{h}\sum_{m=1}^{d}\sum_{p=1}^{n}\sum_{q=1}^{d}$
$\boldsymbol{W}_{2\,ki}^0\boldsymbol{W}_{1\,mk}^0\boldsymbol{X}_{pm}\boldsymbol{X}_{pq}\boldsymbol{M}_{qt}\boldsymbol{W}_{2\,kt}^0\boldsymbol{W}_{1\,mk}^0\boldsymbol{X}_{pm}\boldsymbol{X}_{pq}\boldsymbol{M}_{qi}$, we consider the following cases:

**Case 1.** $k = k, i = t, m = m, q = q, p = p, m \neq q$.

$$Term_1^1 = \frac{\eta_1\eta_2}{n^2h^5} \times h \times h \times (d^2 - d) \times n \times \frac{1}{d} \times \frac{1}{d} \times \frac{1}{h} = \frac{\eta_1\eta_2}{nh^4} + O(\frac{\eta_1\eta_2}{nh^5}).$$

**Case 2.** $k = k, i = t, m = m, q = q, p \neq p, m = q$.

$$Term_1^2 = \frac{\eta_1\eta_2}{n^2h^5} \times h \times h \times d \times (n^2 - n) \times \frac{1}{d} \times \frac{1}{d} \times \frac{1}{h} = \frac{\eta_1\eta_2}{h^5} + O(\frac{\eta_1\eta_2}{nh^5}).$$

**Case 3.** $k = k, i = t, m = m, q = q, p = p, m = q$.

$$Term_1^3 = \frac{\eta_1\eta_2}{n^2h^5} \times h \times h \times d \times n \times \frac{1}{d} \times \frac{1}{d} \times \frac{1}{h} \times 3 = O(\frac{\eta_1\eta_2}{nh^5}).$$

Thus we have

$$Term_1 = \frac{\eta_1\eta_2}{nh^4} + \frac{\eta_1\eta_2}{h^5} + O(\frac{\eta_1\eta_2}{nh^5})$$

For $Term_2 = \frac{\eta_1\eta_2}{n^2h^4}\mathbb{E}\sum_{i=1}^{h}\sum_{k=1}^{h}\sum_{m=1}^{d}\sum_{p=1}^{n}\sum_{t=1}^{h}\sum_{k=1}^{h}\sum_{m=1}^{d}\sum_{p=1}^{n}$
$\boldsymbol{W}^0_{2\,ki}\boldsymbol{W}^0_{1\,mk}\boldsymbol{X}_{pm}\boldsymbol{\xi}_{pt}\boldsymbol{W}^0_{2\,kt}\boldsymbol{W}^0_{1\,mk}\boldsymbol{X}_{pm}\boldsymbol{\xi}_{pi}$, we consider the following case:

**Case 1.** $k = k, i = t, m = m, p = p.$

$$Term_2 = \frac{\eta_1\eta_2}{n^2h^5} \times h \times h \times d \times n \times \frac{1}{d} \times \frac{1}{h} \times \rho_e^2 = \frac{\eta_1\eta_2\rho_e^2}{nh^4}.$$

Combine $Term_1$ and $Term_2$

We finally get that

$$T_4 = O(\frac{\eta_1\eta_2}{h^5}) \tag{124}$$

$\square$

Since it is easy to see that $T_4 = T_{13}$, we have

$$T_{13} = O(\frac{\eta_1\eta_2}{h^5}) \tag{125}$$

$\square$

**Analysis of $T_6$ and $T_{11}$.**

$$
\begin{aligned}
T_6 &= tr\left(\mathbb{E}_{\boldsymbol{W}^0_1,\boldsymbol{W}^0_2,\boldsymbol{\xi},\boldsymbol{X}}\left[\frac{\eta_1^2}{n^2h^4}\boldsymbol{W}_2^{0\top}\boldsymbol{W}_2^0\boldsymbol{Y}^\top\boldsymbol{X}\boldsymbol{X}^\top\boldsymbol{Y}\boldsymbol{W}_2^{0\top}\boldsymbol{W}_2^0\right]\right) \\
&= tr\left(\mathbb{E}_{\boldsymbol{W}^0_1,\boldsymbol{W}^0_2,\boldsymbol{\xi},\boldsymbol{X}}\left[\frac{\eta_1^2}{n^2h^4}\boldsymbol{W}_2^{0\top}\boldsymbol{W}_2^0\boldsymbol{W}_2^{0\top}\boldsymbol{W}_2^0\boldsymbol{Y}^\top\boldsymbol{X}\boldsymbol{X}^\top\boldsymbol{Y}\right]\right)
\end{aligned}
$$

Similar to  159, we have

$$\mathbb{E}(\boldsymbol{W}_2^{0\top}\boldsymbol{W}_2^0\boldsymbol{W}_2^{0\top}\boldsymbol{W}_2^0) = \left(2 + \frac{1}{h}\right)\boldsymbol{I}_h.$$

Taking $\mathbb{E}(\boldsymbol{W}_2^{0\top}\boldsymbol{W}_2^0\boldsymbol{W}_2^{0\top}\boldsymbol{W}_2^0)$ into $T_6$, we have

$$
\begin{aligned}
T_6 &= \frac{\eta_1^2}{n^2h^4}\left(2 + \frac{1}{h}\right)tr\left(\mathbb{E}_{\boldsymbol{W}^0_1,\boldsymbol{W}^0_2,\boldsymbol{\xi},\boldsymbol{X}}\left[\boldsymbol{Y}^\top\boldsymbol{X}\boldsymbol{X}^\top\boldsymbol{Y}\right]\right) \\
&= \frac{\eta_1^2}{n^2h^4}\left(2 + \frac{1}{h}\right)tr\left(\mathbb{E}_{\boldsymbol{W}^0_1,\boldsymbol{W}^0_2,\boldsymbol{\xi},\boldsymbol{X}}\left[\boldsymbol{M}^\top\boldsymbol{X}^\top\boldsymbol{X}\boldsymbol{X}^\top\boldsymbol{X}\boldsymbol{M} + \boldsymbol{\xi}^\top\boldsymbol{X}\boldsymbol{X}^\top\boldsymbol{\xi}\right]\right) \\
&= \frac{\eta_1^2}{n^2h^4}\left(2 + \frac{1}{h}\right)tr\left(\mathbb{E}_{\boldsymbol{W}^0_1,\boldsymbol{W}^0_2,\boldsymbol{\xi},\boldsymbol{X}}\left[\boldsymbol{M}^\top\boldsymbol{X}^\top\boldsymbol{X}\boldsymbol{X}^\top\boldsymbol{X}\boldsymbol{M}\right]\right) \\
&\quad + \frac{\eta_1^2}{n^2h^4}\left(2 + \frac{1}{h}\right)tr\left(\mathbb{E}_{\boldsymbol{W}^0_1,\boldsymbol{W}^0_2,\boldsymbol{\xi},\boldsymbol{X}}\left[\boldsymbol{\xi}^\top\boldsymbol{X}\boldsymbol{X}^\top\boldsymbol{\xi}\right]\right) \\
&= \frac{\eta_1^2}{n^2h^4d}\left(2 + \frac{1}{h}\right)\left(n^2d + nd^2 + nd\right) + \frac{\eta_1^2}{n^2h^5}\left(2 + \frac{1}{h}\right)\left(nhd\rho_e^2\right) \\
&= \frac{2\eta_1^2}{h^4} + \frac{2\eta_1^2d}{nh^4} + \frac{2\eta_1^2d\rho_e^2}{nh^4} + O(\frac{\eta_1^2}{h^5})
\end{aligned}
\tag{126}
$$

$\square$

Similar to $T_6$, For $T_{11}$ we have

$$
\begin{aligned}
T_{11} &= tr\left(\mathbb{E}_{\boldsymbol{W}_1^0, \boldsymbol{W}_2^0, \boldsymbol{\xi}, \boldsymbol{X}}\left[\frac{\eta_2^2}{n^2 h^4} \boldsymbol{Y}^\top \boldsymbol{X} \boldsymbol{W}_1^0 \boldsymbol{W}_1^{0\top} \boldsymbol{W}_1^0 \boldsymbol{W}_1^{0\top} \boldsymbol{X}^\top \boldsymbol{Y}\right]\right) \\
&= tr\left(\mathbb{E}_{\boldsymbol{W}_1^0, \boldsymbol{W}_2^0, \boldsymbol{\xi}, \boldsymbol{X}}\left[\frac{\eta_2^2}{n^2 h^4} \boldsymbol{X}^\top \boldsymbol{Y} \boldsymbol{Y}^\top \boldsymbol{X} \boldsymbol{W}_1^0 \boldsymbol{W}_1^{0\top} \boldsymbol{W}_1^0 \boldsymbol{W}_1^{0\top}\right]\right) \\
&= \frac{\eta_2^2}{n^2 h^4}\left(\frac{h^2 + hd + h}{d^2}\right) tr\left(\mathbb{E}_{\boldsymbol{W}_1^0, \boldsymbol{W}_2^0, \boldsymbol{\xi}, \boldsymbol{X}}\left[\boldsymbol{X}^\top \boldsymbol{Y} \boldsymbol{Y}^\top \boldsymbol{X}\right]\right) \\
&= \frac{\eta_2^2}{n^2 h^4}\left(\frac{h^2 + hd + h}{d^2}\right) tr\left(\mathbb{E}_{\boldsymbol{W}_1^0, \boldsymbol{W}_2^0, \boldsymbol{\xi}, \boldsymbol{X}}\left[\boldsymbol{X}^\top \boldsymbol{X} \boldsymbol{M} \boldsymbol{M}^\top \boldsymbol{X}^\top \boldsymbol{X} + \boldsymbol{X}^\top \boldsymbol{\xi} \boldsymbol{\xi}^\top \boldsymbol{X}\right]\right) \\
&= \frac{\eta_2^2}{n^2 h^4 d}\left(\frac{h^2 + hd + h}{d^2}\right)\left(n^2 d + nd^2 + nd\right) + \frac{\eta_2^2}{n^2 h^5}\left(\frac{h^2 + hd + h}{d^2}\right)\left(ndh\rho_e^2\right) \\
&= \frac{\eta_2^2}{d^2 h^2} + \frac{\eta_2^2}{dh^3} + \frac{\eta_2^2}{ndh^2} + \frac{\eta_2^2}{nh^3} + \frac{\eta_2^2 \rho_e^2}{ndh^2} + \frac{\eta_2^2 \rho_e^2}{nh^3} + O\left(\frac{\eta_2^2}{h^5}\right)
\end{aligned}
\tag{127}
$$

$\square$

**Analysis of $T_7$ and $T_{10}$.**

$$
\begin{aligned}
T_7 &= tr\left(\mathbb{E}_{\boldsymbol{W}_1^0, \boldsymbol{W}_2^0, \boldsymbol{\xi}, \boldsymbol{X}}\left[\frac{\eta_1 \eta_2}{n^2 h^4} \boldsymbol{W}_2^{0\top} \boldsymbol{W}_2^0 \boldsymbol{Y}^\top \boldsymbol{X} \boldsymbol{W}_1^0 \boldsymbol{W}_1^{0\top} \boldsymbol{X}^\top \boldsymbol{Y}\right]\right) \\
&= \frac{\eta_1 \eta_2}{n^2 h^4} tr\left(\mathbb{E}_{\boldsymbol{W}_1^0, \boldsymbol{W}_2^0, \boldsymbol{\xi}, \boldsymbol{X}}\left[\boldsymbol{X}^\top \boldsymbol{Y} \boldsymbol{W}_2^{0\top} \boldsymbol{W}_2^0 \boldsymbol{Y}^\top \boldsymbol{X} \boldsymbol{W}_1^0 \boldsymbol{W}_1^{0\top}\right]\right) \\
&= \frac{\eta_1 \eta_2}{n^2 h^3 d} tr\left(\mathbb{E}_{\boldsymbol{W}_1^0, \boldsymbol{W}_2^0, \boldsymbol{\xi}, \boldsymbol{X}}\left[\boldsymbol{X}^\top \boldsymbol{Y} \boldsymbol{W}_2^{0\top} \boldsymbol{W}_2^0 \boldsymbol{Y}^\top \boldsymbol{X}\right]\right) \\
&= \frac{\eta_1 \eta_2}{n^2 h^3 d} tr\left(\mathbb{E}_{\boldsymbol{W}_1^0, \boldsymbol{W}_2^0, \boldsymbol{\xi}, \boldsymbol{X}}\left[\boldsymbol{Y}^\top \boldsymbol{X} \boldsymbol{X}^\top \boldsymbol{Y} \boldsymbol{W}_2^{0\top} \boldsymbol{W}_2^0\right]\right) \\
&= \frac{\eta_1 \eta_2}{n^2 h^3 d} tr\left(\mathbb{E}_{\boldsymbol{W}_1^0, \boldsymbol{W}_2^0, \boldsymbol{\xi}, \boldsymbol{X}}\left[\boldsymbol{Y}^\top \boldsymbol{X} \boldsymbol{X}^\top \boldsymbol{Y}\right]\right) \\
&= \frac{\eta_1 \eta_2}{n^2 d^2 h^3}\left(n^2 d + nd^2 + nd\right) + \frac{\eta_1 \eta_2}{n^2 h^4 d}\left(nhd\rho_e^2\right) \\
&= \frac{\eta_1 \eta_2}{dh^3} + \frac{\eta_1 \eta_2}{nh^3} + \frac{\eta_1 \eta_2 \rho_e^2}{nh^3} + O\left(\frac{\eta_1 \eta_2}{h^5}\right)
\end{aligned}
\tag{128}
$$

$\square$

It is easy to find

$$
T_{10} = T_7 = \frac{\eta_1 \eta_2}{dh^3} + \frac{\eta_1 \eta_2}{nh^3} + \frac{\eta_1 \eta_2 \rho_e^2}{nh^3} + O\left(\frac{\eta_1 \eta_2}{h^5}\right)
\tag{129}
$$

$\square$

**Analysis of $T_{16}$**

$$
\begin{aligned}
T_{16} &= tr\left(\mathbb{E}_{\boldsymbol{W}_1^0, \boldsymbol{W}_2^0, \boldsymbol{\xi}, \boldsymbol{X}}\left[\frac{\eta_1^2 \eta_2^2}{n^4 h^6} \boldsymbol{Y}^\top \boldsymbol{X} \boldsymbol{W}_1^0 \boldsymbol{W}_2^0 \boldsymbol{Y}^\top \boldsymbol{X} \boldsymbol{X}^\top \boldsymbol{Y} \boldsymbol{W}_2^{0\top} \boldsymbol{W}_1^{0\top} \boldsymbol{X}^\top \boldsymbol{Y}\right]\right) \\
&= \frac{\eta_1^2 \eta_2^2}{n^4 h^6} tr\left(\mathbb{E}\left[\boldsymbol{M}^\top \boldsymbol{X}^\top \boldsymbol{X} \boldsymbol{W}_1^0 \boldsymbol{W}_2^0 \boldsymbol{M}^\top \boldsymbol{X}^\top \boldsymbol{X} \boldsymbol{X}^\top \boldsymbol{X} \boldsymbol{M} \boldsymbol{W}_2^{0\top} \boldsymbol{W}_1^{0\top} \boldsymbol{X}^\top \boldsymbol{X} \boldsymbol{M}\right]\right) \\
&\quad + \frac{\eta_1^2 \eta_2^2}{n^4 h^6} tr\left(\mathbb{E}\left[\boldsymbol{\xi}^\top \boldsymbol{X} \boldsymbol{W}_1^0 \boldsymbol{W}_2^0 \boldsymbol{\xi}^\top \boldsymbol{X} \boldsymbol{X}^\top \boldsymbol{\xi} \boldsymbol{W}_2^{0\top} \boldsymbol{W}_1^{0\top} \boldsymbol{X}^\top \boldsymbol{\xi}\right]\right) \\
&\quad + \frac{2\eta_1^2 \eta_2^2}{n^4 h^6} tr\left(\mathbb{E}\left[\boldsymbol{\xi}^\top \boldsymbol{X} \boldsymbol{W}_1^0 \boldsymbol{W}_2^0 \boldsymbol{\xi}^\top \boldsymbol{X} \boldsymbol{X}^\top \boldsymbol{X} \boldsymbol{M} \boldsymbol{W}_2^{0\top} \boldsymbol{W}_1^{0\top} \boldsymbol{X}^\top \boldsymbol{X} \boldsymbol{M}\right]\right) \\
&\quad + \frac{\eta_1^2 \eta_2^2}{n^4 h^6} tr\left(\mathbb{E}\left[\boldsymbol{M}^\top \boldsymbol{X}^\top \boldsymbol{X} \boldsymbol{W}_1^0 \boldsymbol{W}_2^0 \boldsymbol{\xi}^\top \boldsymbol{X} \boldsymbol{X}^\top \boldsymbol{\xi} \boldsymbol{W}_2^{0\top} \boldsymbol{W}_1^{0\top} \boldsymbol{X}^\top \boldsymbol{X} \boldsymbol{M}\right]\right) \\
&\quad + \frac{\eta_1^2 \eta_2^2}{n^4 h^6} tr\left(\mathbb{E}\left[\boldsymbol{\xi}^\top \boldsymbol{X} \boldsymbol{W}_1^0 \boldsymbol{W}_2^0 \boldsymbol{M}^\top \boldsymbol{X}^\top \boldsymbol{X} \boldsymbol{X}^\top \boldsymbol{X} \boldsymbol{M} \boldsymbol{W}_2^{0\top} \boldsymbol{W}_1^{0\top} \boldsymbol{X}^\top \boldsymbol{\xi}\right]\right)
\end{aligned}
\tag{130}
$$

For $\frac{\eta_1^2 \eta_2^2}{n^4 h^6} tr\left(\mathbb{E}\left[\boldsymbol{M}^\top \boldsymbol{X}^\top \boldsymbol{X} \boldsymbol{W}_1^0 \boldsymbol{W}_2^0 \boldsymbol{M}^\top \boldsymbol{X}^\top \boldsymbol{X} \boldsymbol{X}^\top \boldsymbol{X} \boldsymbol{M} \boldsymbol{W}_2^{0\top} \boldsymbol{W}_1^{0\top} \boldsymbol{X}^\top \boldsymbol{X} \boldsymbol{M}\right]\right)$, we have

$$\frac{\eta_1^2 \eta_2^2}{n^4 h^6} tr\left(\mathbb{E}\left[\boldsymbol{M}^\top \boldsymbol{X}^\top \boldsymbol{X} \boldsymbol{W}_1^0 \boldsymbol{W}_2^0 \boldsymbol{M}^\top \boldsymbol{X}^\top \boldsymbol{X} \boldsymbol{X}^\top \boldsymbol{X} \boldsymbol{M} \boldsymbol{W}_2^{0\top} \boldsymbol{W}_1^{0\top} \boldsymbol{X}^\top \boldsymbol{X} \boldsymbol{M}\right]\right)$$

$$=\frac{\eta_1^2 \eta_2^2}{n^4 h^6} \mathbb{E} \sum_{i=1}^{h}\sum_{k=1}^{d}\sum_{m=1}^{n}\sum_{q=1}^{d}\sum_{s=1}^{h}\sum_{t=1}^{h}\sum_{b=1}^{d}\sum_{p=1}^{n}\sum_{c=1}^{d}\sum_{p=1}^{n}\sum_{b=1}^{d}\sum_{t=1}^{h}\sum_{s=1}^{h}\sum_{q=1}^{d}\sum_{m=1}^{n}\sum_{k=1}^{d}$$

$$\boldsymbol{M}_{ki}\boldsymbol{X}_{mk}\boldsymbol{X}_{mq}\boldsymbol{W}_{1\,qs}^0\boldsymbol{W}_{2\,st}^0\boldsymbol{M}_{bt}\boldsymbol{X}_{pb}\boldsymbol{X}_{pc}\boldsymbol{X}_{pc}\boldsymbol{X}_{pb}\boldsymbol{M}_{bt}\boldsymbol{W}_{2\,st}^0\boldsymbol{W}_{1\,qs}^0\boldsymbol{X}_{mq}\boldsymbol{X}_{mk}\boldsymbol{M}_{ki}$$

$$=\frac{\eta_1^2 \eta_2^2}{n^4 h^6} \mathbb{E} \sum_{i=1}^{h}\sum_{k=1}^{d}\sum_{m=1}^{n}\sum_{q=1}^{d}\sum_{s=1}^{h}\sum_{t=1}^{h}\sum_{b=1}^{d}\sum_{p=1}^{n}\sum_{c=1}^{d}\sum_{p=1}^{n}\sum_{b=1}^{d}\sum_{m=1}^{n}\sum_{k=1}^{d}$$

$$\boldsymbol{M}_{ki}\boldsymbol{X}_{mk}\boldsymbol{X}_{mq}\boldsymbol{W}_{1\,qs}^{0\,2}\boldsymbol{W}_{2\,st}^{0\,2}\boldsymbol{M}_{bt}\boldsymbol{X}_{pb}\boldsymbol{X}_{pc}\boldsymbol{X}_{pc}\boldsymbol{X}_{pb}\boldsymbol{M}_{bt}\boldsymbol{X}_{mq}\boldsymbol{X}_{mk}\boldsymbol{M}_{ki}$$

**We focus only on the case dominated by the leading term.** Since other cases will be $O(\frac{\eta_1^2 \eta_2^2}{h^8})$.

**Case 1.** $b = b, k = k, i \neq t, m = m, p = p, m \neq p, k \neq q, b \neq c.$

$$=\frac{\eta_1^2 \eta_2^2}{n^4 h^6} \mathbb{E} \sum_{i=1}^{h}\sum_{k=1}^{d}\sum_{m=1}^{n}\sum_{q=1}^{d}\sum_{s=1}^{h}\sum_{t=1}^{h}\sum_{b=1}^{d}\sum_{p=1}^{n}\sum_{c=1}^{d}\sum_{p=1}^{n}\sum_{m=1}^{n}$$

$$\boldsymbol{M}_{ki}^2\boldsymbol{W}_{1\,qs}^{0\,2}\boldsymbol{W}_{2\,st}^{0\,2}\boldsymbol{M}_{bt}^2\boldsymbol{X}_{mk}\boldsymbol{X}_{mq}\boldsymbol{X}_{pb}\boldsymbol{X}_{pc}\boldsymbol{X}_{pc}\boldsymbol{X}_{pb}\boldsymbol{X}_{mq}\boldsymbol{X}_{mk}$$

$$=\frac{\eta_1^2 \eta_2^2}{n^4 h^6} \mathbb{E} \sum_{i=1}^{h}\sum_{k=1}^{d}\sum_{m=1}^{n}\sum_{q=1}^{d}\sum_{s=1}^{h}\sum_{t=1}^{h}\sum_{b=1}^{d}\sum_{p=1}^{n}\sum_{c=1}^{d}$$

$$\boldsymbol{M}_{ki}^2\boldsymbol{W}_{1\,qs}^{0\,2}\boldsymbol{W}_{2\,st}^{0\,2}\boldsymbol{M}_{bt}^2\boldsymbol{X}_{mk}^2\boldsymbol{X}_{mq}^2\boldsymbol{X}_{pb}^2\boldsymbol{X}_{pc}^2$$

$$=\frac{\eta_1^2 \eta_2^2}{n^4 h^8} \times \frac{1}{d} \times \frac{1}{d} \times \frac{1}{d} \times \frac{1}{h} \times h \times (d^2 - d) \times (n^2 - n) \times (d^2 - d) \times (h^2 - h)$$

$$=\frac{\eta_1^2 \eta_2^2 d}{n^2 h^5} + O(\frac{\eta_1^2 \eta_2^2}{h^8}).$$

**Case 2.** $b = b, k = k, i \neq t, k = q, b = c, m \neq m, p \neq p, k \neq b,.$

Similar to Case 1 , we have

$$=\frac{\eta_1^2 \eta_2^2}{n^4 h^6} \mathbb{E} \sum_{i=1}^{h}\sum_{k=1}^{d}\sum_{m=1}^{n}\sum_{q=1}^{d}\sum_{s=1}^{h}\sum_{t=1}^{h}\sum_{b=1}^{d}\sum_{p=1}^{n}\sum_{c=1}^{d}\sum_{p=1}^{n}\sum_{m=1}^{n}$$

$$\boldsymbol{M}_{ki}^2\boldsymbol{W}_{1\,qs}^{0\,2}\boldsymbol{W}_{2\,st}^{0\,2}\boldsymbol{M}_{bt}^2\boldsymbol{X}_{mk}\boldsymbol{X}_{mq}\boldsymbol{X}_{pb}\boldsymbol{X}_{pc}\boldsymbol{X}_{pc}\boldsymbol{X}_{pb}\boldsymbol{X}_{mq}\boldsymbol{X}_{mk}$$

$$=\frac{\eta_1^2 \eta_2^2}{n^4 h^6} \mathbb{E} \sum_{i=1}^{h}\sum_{k=1}^{d}\sum_{m=1}^{n}\sum_{s=1}^{h}\sum_{t=1}^{h}\sum_{b=1}^{d}\sum_{p=1}^{n}\sum_{m=1}^{n}\sum_{p=1}^{n}$$

$$\boldsymbol{M}_{ki}^2\boldsymbol{W}_{1\,qs}^{0\,2}\boldsymbol{W}_{2\,st}^{0\,2}\boldsymbol{M}_{bt}^2\boldsymbol{X}_{mk}^2\boldsymbol{X}_{mk}^2\boldsymbol{X}_{pb}^2\boldsymbol{X}_{pb}^2$$

$$=\frac{\eta_1^2 \eta_2^2}{n^4 h^8} \times \frac{1}{d} \times \frac{1}{d} \times \frac{1}{d} \times \frac{1}{h} \times h \times (n^2 - n) \times (n^2 - n) \times (d^2 - d) \times (h^2 - h)$$

$$=\frac{\eta_1^2 \eta_2^2}{d h^6} + O(\frac{\eta_1^2 \eta_2^2}{h^8}).$$

$\square$

**Case 3.** $b = b, k = k, i \neq t, k = q, p = p, m \neq p, m \neq m, b \neq c.$

$$
\begin{aligned}
=&\frac{\eta_1^2 \eta_2^2}{n^4 h^6} \mathbb{E} \sum_{i=1}^{h} \sum_{k=1}^{d} \sum_{m=1}^{n} \sum_{q=1}^{d} \sum_{s=1}^{h} \sum_{t=1}^{h} \sum_{b=1}^{d} \sum_{p=1}^{n} \sum_{c=1}^{d} \sum_{p=1}^{n} \sum_{m=1}^{n} \\
&\boldsymbol{M}_{ki}{}^2 \boldsymbol{W}_{1\,qs}^0{}^2 \boldsymbol{W}_{2\,st}^0{}^2 \boldsymbol{M}_{bt}{}^2 \boldsymbol{X}_{mk} \boldsymbol{X}_{mq} \boldsymbol{X}_{pb} \boldsymbol{X}_{pc} \boldsymbol{X}_{pc} \boldsymbol{X}_{pb} \boldsymbol{X}_{mq} \boldsymbol{X}_{mk} \\
=&\frac{\eta_1^2 \eta_2^2}{n^4 h^6} \mathbb{E} \sum_{i=1}^{h} \sum_{k=1}^{d} \sum_{m=1}^{n} \sum_{m=1}^{n} \sum_{s=1}^{h} \sum_{t=1}^{h} \sum_{b=1}^{d} \sum_{p=1}^{n} \sum_{c=1}^{d} \\
&\boldsymbol{M}_{ki}{}^2 \boldsymbol{W}_{1\,qs}^0{}^2 \boldsymbol{W}_{2\,st}^0{}^2 \boldsymbol{M}_{bt}{}^2 \boldsymbol{X}_{mk}^2 \boldsymbol{X}_{mq}^2 \boldsymbol{X}_{pb}^2 \boldsymbol{X}_{pc}^2 \\
=&\frac{\eta_1^2 \eta_2^2}{n^4 h^8} \times \frac{1}{d} \times \frac{1}{d} \times \frac{1}{d} \times \frac{1}{h} \times h \times (n^2 - n) \times (n^2 - n) \times (d^2 - d) \times (h^2 - h) \\
=&\frac{\eta_1^2 \eta_2^2}{d h^6} + O(\frac{\eta_1^2 \eta_2^2}{h^8}).
\end{aligned}
$$

**Case 4.** $b = b, k = k, m = m, i \neq t, , b = c, k \neq q, p \neq p, k \neq b,.$

Similar to Case 1 , we have

$$
\begin{aligned}
=&\frac{\eta_1^2 \eta_2^2}{n^4 h^6} \mathbb{E} \sum_{i=1}^{h} \sum_{k=1}^{d} \sum_{m=1}^{n} \sum_{q=1}^{d} \sum_{s=1}^{h} \sum_{t=1}^{h} \sum_{b=1}^{d} \sum_{p=1}^{n} \sum_{c=1}^{d} \sum_{p=1}^{n} \sum_{m=1}^{n} \\
&\boldsymbol{M}_{ki}{}^2 \boldsymbol{W}_{1\,qs}^0{}^2 \boldsymbol{W}_{2\,st}^0{}^2 \boldsymbol{M}_{bt}{}^2 \boldsymbol{X}_{mk} \boldsymbol{X}_{mq} \boldsymbol{X}_{pb} \boldsymbol{X}_{pc} \boldsymbol{X}_{pc} \boldsymbol{X}_{pb} \boldsymbol{X}_{mq} \boldsymbol{X}_{mk} \\
=&\frac{\eta_1^2 \eta_2^2}{n^4 h^6} \mathbb{E} \sum_{i=1}^{h} \sum_{k=1}^{d} \sum_{m=1}^{n} \sum_{s=1}^{h} \sum_{t=1}^{h} \sum_{b=1}^{d} \sum_{p=1}^{n} \sum_{q=1}^{d} \sum_{p=1}^{n} \\
&\boldsymbol{M}_{ki}{}^2 \boldsymbol{W}_{1\,qs}^0{}^2 \boldsymbol{W}_{2\,st}^0{}^2 \boldsymbol{M}_{bt}{}^2 \boldsymbol{X}_{mk}^2 \boldsymbol{X}_{mq}^2 \boldsymbol{X}_{pb}^2 \boldsymbol{X}_{pb}^2 \\
=&\frac{\eta_1^2 \eta_2^2}{n^4 h^8} \times \frac{1}{d} \times \frac{1}{d} \times \frac{1}{d} \times \frac{1}{h} \times h \times (d^2 - d) \times (n^2 - n) \times (d^2 - d) \times (h^2 - h) \\
=&\frac{\eta_1^2 \eta_2^2 d}{n^2 h^5} + O(\frac{\eta_1^2 \eta_2^2}{h^8}).
\end{aligned}
$$

$\square$

For $\frac{\eta_1^2 \eta_2^2}{n^4 h^6} tr \left( \mathbb{E} \left[ \boldsymbol{\xi}^\top \boldsymbol{X} \boldsymbol{W}_1^0 \boldsymbol{W}_2^0 \boldsymbol{\xi}^\top \boldsymbol{X} \boldsymbol{X}^\top \boldsymbol{\xi} \boldsymbol{W}_2^{0\top} \boldsymbol{W}_1^{0\top} \boldsymbol{X}^\top \boldsymbol{\xi} \right] \right)$, we have

$$
\begin{aligned}
&\frac{\eta_1^2 \eta_2^2}{n^4 h^6} tr \left( \mathbb{E} \left[ \boldsymbol{\xi}^\top \boldsymbol{X} \boldsymbol{W}_1^0 \boldsymbol{W}_2^0 \boldsymbol{\xi}^\top \boldsymbol{X} \boldsymbol{X}^\top \boldsymbol{\xi} \boldsymbol{W}_2^{0\top} \boldsymbol{W}_1^{0\top} \boldsymbol{X}^\top \boldsymbol{\xi} \right] \right) \\
=&\frac{\eta_1^2 \eta_2^2}{n^4 h^3} \mathbb{E} \sum_{i=1}^{h} \sum_{k=1}^{n} \sum_{q=1}^{d} \sum_{s=1}^{h} \sum_{t=1}^{h} \sum_{b=1}^{n} \sum_{c=1}^{d} \sum_{b=1}^{n} \sum_{t=1}^{h} \sum_{s=1}^{h} \sum_{q=1}^{d} \sum_{k=1}^{n} \\
&\boldsymbol{\xi}_{ki} \boldsymbol{X}_{kq} \boldsymbol{W}_{1\,qs}^0 \boldsymbol{W}_{2\,st}^0 \boldsymbol{\xi}_{bt} \boldsymbol{X}_{bc} \boldsymbol{X}_{bc} \boldsymbol{\xi}_{bt} \boldsymbol{W}_{2\,st}^0 \boldsymbol{W}_{1\,qs}^0 \boldsymbol{X}_{kq} \boldsymbol{\xi}_{ki} \\
=&\frac{\eta_1^2 \eta_2^2}{n^4 h^6} \mathbb{E} \sum_{i=1}^{h} \sum_{k=1}^{n} \sum_{q=1}^{d} \sum_{s=1}^{h} \sum_{t=1}^{h} \sum_{b=1}^{n} \sum_{c=1}^{d} \sum_{b=1}^{n} \sum_{k=1}^{n} \boldsymbol{\xi}_{ki} \boldsymbol{X}_{kq} \boldsymbol{W}_{1\,qs}^{0}{}^2 \boldsymbol{W}_{2\,st}^{0}{}^2 \boldsymbol{\xi}_{bt} \boldsymbol{X}_{bc} \boldsymbol{X}_{bc} \boldsymbol{\xi}_{bt} \boldsymbol{X}_{kq} \boldsymbol{\xi}_{ki}
\end{aligned}
$$

**We focus only on the case dominated by the leading term.** Since other cases will be $O(\frac{\eta_1^2 \eta_2^2}{h^8})$.

**Case 1.** $b = b, k = k, i \neq t, k \neq b$**.**

$$= \frac{\eta_1^2 \eta_2^2}{n^4 h^6} \mathbb{E} \sum_{i=1}^{h} \sum_{k=1}^{n} \sum_{q=1}^{d} \sum_{s=1}^{h} \sum_{t=1}^{h} \sum_{b=1}^{n} \sum_{c=1}^{d} {\boldsymbol{\xi}_{ki}}^2 \boldsymbol{X}_{kq}^2 \boldsymbol{W}_{1\,qs}^{0\,2} \boldsymbol{W}_{2\,st}^{0\,2} {\boldsymbol{\xi}_{bt}}^2 \boldsymbol{X}_{bc}^2$$

$$= \frac{\eta_1^2 \eta_2^2}{n^4 h^8} \times \rho_e^2 \times \rho_e^2 \times \frac{1}{d} \times \frac{1}{h} \times (h^2 - h) \times (n^2 - n) \times d \times d \times h$$

$$= \frac{\eta_1^2 \eta_2^2 d \rho_e^4}{n^2 h^6} + O(\frac{\eta_1^2 \eta_2^2}{h^8}).$$

$\square$

For $\frac{2\eta_1^2 \eta_2^2}{n^4 h^6} tr \left( \mathbb{E} \left[ \boldsymbol{\xi}^\top \boldsymbol{X} \boldsymbol{W}_1^0 \boldsymbol{W}_2^0 \boldsymbol{\xi}^\top \boldsymbol{X} \boldsymbol{X}^\top \boldsymbol{X} \boldsymbol{M} \boldsymbol{W}_2^{0\top} \boldsymbol{W}_1^{0\top} \boldsymbol{X}^\top \boldsymbol{X} \boldsymbol{M} \right] \right)$, we have

$$\frac{2\eta_1^2 \eta_2^2}{n^4 h^6} tr \left( \mathbb{E} \left[ \boldsymbol{\xi}^\top \boldsymbol{X} \boldsymbol{W}_1^0 \boldsymbol{W}_2^0 \boldsymbol{\xi}^\top \boldsymbol{X} \boldsymbol{X}^\top \boldsymbol{X} \boldsymbol{M} \boldsymbol{W}_2^{0\top} \boldsymbol{W}_1^{0\top} \boldsymbol{X}^\top \boldsymbol{X} \boldsymbol{M} \right] \right)$$

$$= \frac{\eta_1^2 \eta_2^2}{n^4 h^6} \mathbb{E} \sum_{i=1}^{h} \sum_{k=1}^{n} \sum_{q=1}^{d} \sum_{s=1}^{h} \sum_{t=1}^{h} \sum_{b=1}^{n} \sum_{c=1}^{d} \sum_{b=1}^{d} \sum_{t=1}^{h} \sum_{s=1}^{h} \sum_{q=1}^{d} \sum_{k=1}^{d} \sum_{p=1}^{n} \sum_{m=1}^{n}$$

$$\boldsymbol{\xi}_{ki} \boldsymbol{X}_{kq} \boldsymbol{W}_{1\,qs}^0 \boldsymbol{W}_{2\,st}^0 \boldsymbol{\xi}_{bt} \boldsymbol{X}_{bc} \boldsymbol{X}_{pc} \boldsymbol{X}_{pb} \boldsymbol{M}_{bt} \boldsymbol{W}_{2\,st}^0 \boldsymbol{W}_{1\,qs}^0 \boldsymbol{X}_{mq} \boldsymbol{X}_{mk} \boldsymbol{M}_{ki}$$

$$= \frac{\eta_1^2 \eta_2^2}{n^4 h^6} \mathbb{E} \sum_{i=1}^{h} \sum_{k=1}^{n} \sum_{q=1}^{d} \sum_{s=1}^{h} \sum_{b=1}^{n} \sum_{c=1}^{d} \sum_{p=1}^{n} \sum_{m=1}^{n} \sum_{k=1}^{d}$$

$$ {\boldsymbol{\xi}_{ki}}^2 \boldsymbol{X}_{kq} \boldsymbol{W}_{1\,qs}^{0\,2} \boldsymbol{W}_{2\,si}^{0\,2} \boldsymbol{X}_{kc} \boldsymbol{X}_{pc} \boldsymbol{X}_{pk} \boldsymbol{M}_{ki}^2 \boldsymbol{X}_{mq} \boldsymbol{X}_{mk}$$

**We focus only on the case dominated by the leading term.**

**Case 1.** $q = c = k, k \neq p \neq m$**.**

$$\frac{\eta_1^2 \eta_2^2}{n^4 h^6} \mathbb{E} \sum_{i=1}^{h} \sum_{k=1}^{n} \sum_{q=1}^{d} \sum_{s=1}^{h} \sum_{b=1}^{n} \sum_{p=1}^{n} \sum_{m=1}^{n} {\boldsymbol{\xi}_{ki}}^2 \boldsymbol{X}_{kq}^2 \boldsymbol{W}_{1\,qs}^{0\,2} \boldsymbol{W}_{2\,si}^{0\,2} \boldsymbol{X}_{pq}^2 \boldsymbol{M}_{ki}^2 \boldsymbol{X}_{mq}^2$$

$$= \frac{\eta_1^2 \eta_2^2}{n^4 h^8} \times \rho_e^2 \times \frac{1}{d} \times \frac{1}{h} \times \frac{1}{d} \times h \times h \times d \times (n^3 - n)$$

$$= O(\frac{\eta_1^2 \eta_2^2}{h^8}).$$

$\square$

For $\frac{\eta_1^2 \eta_2^2}{n^4 h^6} tr \left( \mathbb{E} \left[ \boldsymbol{M}^\top \boldsymbol{X}^\top \boldsymbol{X} \boldsymbol{W}_1^0 \boldsymbol{W}_2^0 \boldsymbol{\xi}^\top \boldsymbol{X} \boldsymbol{X}^\top \boldsymbol{\xi} \boldsymbol{W}_2^{0\top} \boldsymbol{W}_1^{0\top} \boldsymbol{X}^\top \boldsymbol{X} \boldsymbol{M} \right] \right)$, we have

$$\frac{\eta_1^2 \eta_2^2}{n^4 h^6} tr \left( \mathbb{E} \left[ \boldsymbol{M}^\top \boldsymbol{X}^\top \boldsymbol{X} \boldsymbol{W}_1^0 \boldsymbol{W}_2^0 \boldsymbol{\xi}^\top \boldsymbol{X} \boldsymbol{X}^\top \boldsymbol{\xi} \boldsymbol{W}_2^{0\top} \boldsymbol{W}_1^{0\top} \boldsymbol{X}^\top \boldsymbol{X} \boldsymbol{M} \right] \right)$$

$$= \frac{\eta_1^2 \eta_2^2}{n^4 h^5 d} tr \left( \mathbb{E} \left[ \boldsymbol{X}^\top \boldsymbol{X} \boldsymbol{W}_1^0 \boldsymbol{W}_2^0 \boldsymbol{\xi}^\top \boldsymbol{X} \boldsymbol{X}^\top \boldsymbol{\xi} \boldsymbol{W}_2^{0\top} \boldsymbol{W}_1^{0\top} \boldsymbol{X}^\top \boldsymbol{X} \right] \right)$$

$$= \frac{\eta_1^2 \eta_2^2}{n^4 h^5 d} \mathbb{E} \sum_{i=1}^{d} \sum_{m=1}^{n} \sum_{q=1}^{d} \sum_{s=1}^{h} \sum_{t=1}^{h} \sum_{b=1}^{d} \sum_{c=1}^{d} \sum_{b=1}^{d} \sum_{t=1}^{h} \sum_{q=1}^{d} \sum_{s=1}^{h} \sum_{m=1}^{n}$$

$$\boldsymbol{X}_{mi} \boldsymbol{X}_{mq} \boldsymbol{W}_{1\,qs}^0 \boldsymbol{W}_{2\,st}^0 \boldsymbol{\xi}_{bt} \boldsymbol{X}_{bc} \boldsymbol{X}_{bc} \boldsymbol{\xi}_{bt} \boldsymbol{W}_{2\,st}^0 \boldsymbol{W}_{1\,qs}^0 \boldsymbol{X}_{mq} \boldsymbol{X}_{mi}$$

$$= \frac{\eta_1^2 \eta_2^2}{n^4 h^5 d} \mathbb{E} \sum_{i=1}^{d} \sum_{m=1}^{n} \sum_{q=1}^{d} \sum_{s=1}^{h} \sum_{t=1}^{h} \sum_{b=1}^{d} \sum_{c=1}^{d} \sum_{m=1}^{n} \boldsymbol{X}_{mi} \boldsymbol{X}_{mq} \boldsymbol{W}_{1\,qs}^{0\,2} \boldsymbol{W}_{2\,st}^{0\,2} \boldsymbol{\xi}_{bt}^2 \boldsymbol{X}_{bc}^2 \boldsymbol{X}_{mq} \boldsymbol{X}_{mi}$$

**We focus only on the case dominated by the leading term.**

**Case 1.** $q = i, m \neq m$**.**

$$\frac{\eta_1^2\eta_2^2}{n^4h^5d}\mathbb{E}\sum_{i=1}^{d}\sum_{m=1}^{n}\sum_{q=1}^{d}\sum_{s=1}^{h}\sum_{t=1}^{h}\sum_{b=1}^{d}\sum_{c=1}^{d}\sum_{m=1}^{n}\boldsymbol{X}_{mi}\boldsymbol{X}_{mq}\boldsymbol{W_1^0}_{qs}^{2}\boldsymbol{W_2^0}_{st}^{2}\boldsymbol{\xi}_{bt}^2\boldsymbol{X}_{bc}^2\boldsymbol{X}_{mq}^2\boldsymbol{X}_{mi}$$

$$=\frac{\eta_1^2\eta_2^2}{n^4h^5d}\mathbb{E}\sum_{i=1}^{d}\sum_{m=1}^{n}\sum_{s=1}^{h}\sum_{t=1}^{h}\sum_{b=1}^{d}\sum_{c=1}^{d}\sum_{m=1}^{n}\boldsymbol{X}_{mi}^2\boldsymbol{W_1^0}_{qs}^{2}\boldsymbol{W_2^0}_{st}^{2}\boldsymbol{\xi}_{bt}^2\boldsymbol{X}_{bc}^2\boldsymbol{X}_{mi}^2$$

$$=\frac{\eta_1^2\eta_2^2}{n^4h^7d}\times\rho_e^2\times\frac{1}{d}\times\frac{1}{h}\times(n^2-n)\times d\times h\times h\times d\times d$$

$$=\frac{\eta_1^2\eta_2^2 d\rho_e^2}{n^2h^6}+O(\frac{\eta_1^2\eta_2^2}{h^8}).$$

**Case 2.** $m=m,q\neq i$.

$$\frac{\eta_1^2\eta_2^2}{n^4h^5d}\mathbb{E}\sum_{i=1}^{d}\sum_{m=1}^{n}\sum_{q=1}^{d}\sum_{s=1}^{h}\sum_{t=1}^{h}\sum_{b=1}^{d}\sum_{c=1}^{d}\sum_{m=1}^{n}\boldsymbol{X}_{mi}\boldsymbol{X}_{mq}\boldsymbol{W_1^0}_{qs}^{2}\boldsymbol{W_2^0}_{st}^{2}\boldsymbol{\xi}_{bt}^2\boldsymbol{X}_{bc}^2\boldsymbol{X}_{mq}\boldsymbol{X}_{mi}$$

$$=\frac{\eta_1^2\eta_2^2}{n^4h^7d}\mathbb{E}\sum_{i=1}^{d}\sum_{m=1}^{n}\sum_{s=1}^{h}\sum_{t=1}^{h}\sum_{b=1}^{d}\sum_{c=1}^{d}\sum_{q=1}^{d}\boldsymbol{X}_{mi}^2\boldsymbol{W_1^0}_{qs}^{2}\boldsymbol{W_2^0}_{st}^{2}\boldsymbol{\xi}_{bt}^2\boldsymbol{X}_{bc}^2\boldsymbol{X}_{mq}^2$$

$$=\frac{\eta_1^2\eta_2^2}{n^4h^2d}\times\rho_e^2\times\frac{1}{d}\times\frac{1}{h}\times(d^2-d)\times n\times h\times h\times d\times d$$

$$=\frac{\eta_1^2\eta_2^2 d^2\rho_e^2}{n^3h^6}+O(\frac{\eta_1^2\eta_2^2}{h^8})$$

It is easy to see

$$\frac{\eta_1^2\eta_2^2}{n^4h^6}tr\left(\mathbb{E}\left[\boldsymbol{M}^\top\boldsymbol{X}^\top\boldsymbol{X}\boldsymbol{W_1^0}\boldsymbol{W_2^0}\boldsymbol{\xi}^\top\boldsymbol{X}\boldsymbol{X}^\top\boldsymbol{\xi}\boldsymbol{W_2^0}^\top\boldsymbol{W_1^0}^\top\boldsymbol{X}^\top\boldsymbol{X}\boldsymbol{M}\right]\right)$$

$$=\frac{\eta_1^2\eta_2^2}{n^4h^6}tr\left(\mathbb{E}\left[\boldsymbol{\xi}^\top\boldsymbol{X}\boldsymbol{W_1^0}\boldsymbol{W_2^0}\boldsymbol{M}^\top\boldsymbol{X}^\top\boldsymbol{X}\boldsymbol{X}^\top\boldsymbol{X}\boldsymbol{M}\boldsymbol{W_2^0}^\top\boldsymbol{W_1^0}^\top\boldsymbol{X}^\top\boldsymbol{\xi}\right]\right)$$

$$=\frac{\eta_1^2\eta_2^2 d\rho_e^2}{n^2h^6}+\frac{\eta_1^2\eta_2^2 d^2\rho_e^2}{n^3h^6}+O(\frac{\eta_1^2\eta_2^2}{h^8})$$

Finally we get

$$T_{16}=\frac{\eta_1^2\eta_2^2 d}{n^2h^6}+\frac{\eta_1^2\eta_2^2}{dh^6}+\frac{\eta_1^2\eta_2^2 d\rho_e^4}{n^2h^6}+\frac{2\eta_1^2\eta_2^2 d\rho_e^2}{n^2h^6}+\frac{2\eta_1^2\eta_2^2 d^2\rho_e^2}{n^3h^6}+O(\frac{\eta_1^2\eta_2^2}{h^8}) \tag{131}$$

$\square$

**Analysis of $T_2$, $T_3$, $T_5$, $T_8$, $T_9$, $T_{12}$, $T_{14}$, $T_{15}$, $T_{17}$, $T_{20}$, $T_{21}$ and $T_{24}$.** All terms involve the product of an odd number of identical random matrices with zero mean, and due to their independence from other random matrices, these terms are all 0. $\square$

**Analysis of $T_{18}$.**

$$T_{18}=tr\left(\mathbb{E}_{\boldsymbol{W_1^0},\boldsymbol{W_2^0},\boldsymbol{\xi},\boldsymbol{X}}\left[\frac{\eta_1}{nh^2}\boldsymbol{M}\boldsymbol{W_2^0}^\top\boldsymbol{W_2^0}\boldsymbol{Y}^\top\boldsymbol{X}\right]\right)$$

$$=\frac{\eta_1}{nh^2}tr\left(\mathbb{E}_{\boldsymbol{W_1^0},\boldsymbol{W_2^0},\boldsymbol{\xi},\boldsymbol{X}}\left[\boldsymbol{W_2^0}^\top\boldsymbol{W_2^0}\boldsymbol{Y}^\top\boldsymbol{X}\boldsymbol{M}\right]\right)$$

$$=\frac{\eta_1}{nh^2}tr\left(\mathbb{E}_{\boldsymbol{W_1^0},\boldsymbol{W_2^0},\boldsymbol{\xi},\boldsymbol{X}}\left[\boldsymbol{M}^\top\boldsymbol{X}^\top\boldsymbol{X}\boldsymbol{M}\right]\right)$$

$$=\frac{\eta_1}{h^2}. \tag{132}$$

$\square$

Similar to $T_{18}$, we can get

$$T_{19} = \frac{\eta_2}{dh} \tag{133}$$

$$T_{22} = \frac{\eta_1}{h^2} \tag{134}$$

$$T_{23} = \frac{\eta_2}{dh}. \tag{135}$$

$\square$

Finally, we obtain the exact loss

$$
\begin{aligned}
L_{\text{two-layer}} =\ & \frac{2\eta_1^2}{h^4} + \frac{2\eta_1^2 d(1+\rho_e^2)}{nh^4} - 2\frac{\eta_1}{h^2} \\
& + \frac{\eta_2^2}{d^2 h^2} + \frac{\eta_2^2}{dh^3} + \frac{\eta_2^2(1+\rho_e^2)}{ndh^2} + \frac{\eta_2^2(1+\rho_e^2)}{nh^3} - 2\frac{\eta_2}{dh} + \frac{\eta_1^2\eta_2^2 d}{n^2 h^6} \\
& + \frac{2\eta_1\eta_2}{dh^3} + \frac{2\eta_1\eta_2}{nh^3} + \frac{2\eta_1\eta_2\rho_e^2}{nh^3} + \frac{\eta_1^2\eta_2^2 d(\rho_e^2+1)^2}{n^2 h^6} + \frac{2\eta_1^2\eta_2^2}{dh^6} + \frac{2\eta_1^2\eta_2^2 d^2\rho_e^2}{n^3 h^6} \\
& + 1 + \frac{1}{h} + O(\frac{\eta_1^2}{h^5}) + O(\frac{\eta_2^5}{h^5}) + O(\frac{\eta_1\eta_2}{h^2}) + O(\frac{\eta_1^2\eta_2^2}{h^8})
\end{aligned}
\tag{136}
$$

Under Assumption E.1, we have

$$
\begin{aligned}
L_{\text{two-layer}} =\ & \frac{2\eta_1^2}{h^4} + \frac{2\eta_1^2(1+\rho_e^2)}{nh^3} - 2\frac{\eta_1}{h^2} \\
& + \frac{2\eta_2^2}{h^4} + \frac{2\eta_2^2(1+\rho_e^2)}{nh^3} - 2\frac{\eta_2}{h^2} + \frac{\eta_1^2\eta_2^2}{n^2 h^5} \\
& + \frac{2\eta_1\eta_2}{h^4} + \frac{2\eta_1\eta_2}{nh^3} + \frac{2\eta_1\eta_2\rho_e^2}{nh^3} + \frac{\eta_1^2\eta_2^2(\rho_e^2+1)^2}{n^2 h^5} + \frac{2\eta_1^2\eta_2^2}{h^7} + \frac{2\eta_1^2\eta_2^2\rho_e^2}{n^3 h^4} \\
& + 1 + \frac{1}{h} + O(\frac{\eta_1^2}{h^5}) + O(\frac{\eta_2^5}{h^2}) + O(\frac{\eta_1\eta_2}{h^5}) + O(\frac{\eta_1^2\eta_2^2}{h^8})
\end{aligned}
\tag{137}
$$

$\square$

## E.4. Three-layer NN Test Loss under Gaussian initialization

**Theorem E.10.** *Given Assumption 3.3, E.1, and in addition assume $\eta_1$ and $\eta_2$ are no more than $O(h)$ based on Proposition E.2, consider training procedure discussed in section 3, we derive the test loss after one-step GD update in a three-layer neural network:*

$$
\begin{aligned}
L_{three-layer} =\ & \frac{\eta_1^2}{h^2} + \frac{\eta_1^2(1+\rho_e^2)}{hn} - 2\frac{\eta_1}{h} + \frac{2\eta_2^2}{h^2} + \frac{2\eta_2^2(1+\rho_e^2)}{nh} - 2\frac{\eta_2}{h} \\
& + \frac{2\eta_1\eta_2}{h^2} + \frac{2\eta_1\eta_2(1+\rho_e^2)}{nh} + \frac{\eta_1^2\eta_2^2\rho_e^2}{nh^3} + \frac{\eta_1^2\eta_2^2 d\rho_e^2}{n^2 h^3} + \frac{4\eta_1^2\eta_2^2}{n^2 h^2} \\
& + 1 + O(\frac{\eta_1^2}{h^3}) + O(\frac{\eta_2^2}{h^3}) + O(\frac{\eta_1\eta_2}{h^3}) + O\left(\frac{\eta_1^2\eta_2^2}{h^5}\right)
\end{aligned}
\tag{138}
$$

**Proof of Theorem E.6.** Similar to appendix E.3, we consider a test data $\tilde{x}_0 \sim \mathcal{N}(\mathbf{0}, \boldsymbol{I}_d) \in \mathbb{R}^{1 \times d}$.

$$
\begin{aligned}
& L(\boldsymbol{X}, \boldsymbol{W}_1^1, \boldsymbol{W}_2^1, \boldsymbol{a}, \tilde{\boldsymbol{x}}_0) \\
=& \mathbb{E}_{\boldsymbol{W}_1^0, \boldsymbol{W}_2^0, \boldsymbol{a}, \boldsymbol{\xi}, \tilde{\boldsymbol{x}}_0, \boldsymbol{X}} \left( \frac{1}{\sqrt{h}} \tilde{\boldsymbol{x}}_0 \boldsymbol{W}_1^1 \boldsymbol{W}_2^1 \boldsymbol{a} - \tilde{\boldsymbol{x}}_0 \boldsymbol{\beta}^* \right)^2 \\
=& \mathbb{E}_{\boldsymbol{W}_1^0, \boldsymbol{W}_2^0, \boldsymbol{a}, \boldsymbol{\xi}, \tilde{\boldsymbol{x}}_0, \boldsymbol{X}} \left[ \left( \frac{1}{\sqrt{h}} \boldsymbol{W}_1^1 \boldsymbol{W}_2^1 \boldsymbol{a} - \boldsymbol{\beta}^* \right)^\top \tilde{\boldsymbol{x}}_0^\top \tilde{\boldsymbol{x}}_0 \left( \frac{1}{\sqrt{h}} \boldsymbol{W}_1^1 \boldsymbol{W}_2^1 \boldsymbol{a} - \boldsymbol{\beta}^* \right) \right] \\
=& tr \left( \mathbb{E}_{\boldsymbol{W}_1^0, \boldsymbol{W}_2^0, \boldsymbol{a}, \boldsymbol{\xi}, \tilde{\boldsymbol{x}}_0, \boldsymbol{X}} \left[ \tilde{\boldsymbol{x}}_0^\top \tilde{\boldsymbol{x}}_0 \left( \frac{1}{\sqrt{h}} \boldsymbol{W}_1^1 \boldsymbol{W}_2^1 \boldsymbol{a} - \boldsymbol{\beta}^* \right) \left( \frac{1}{\sqrt{h}} \boldsymbol{W}_1^1 \boldsymbol{W}_2^1 \boldsymbol{a} - \boldsymbol{\beta}^* \right)^\top \right] \right) \\
=& tr \left( \mathbb{E}_{\boldsymbol{W}_1^0, \boldsymbol{W}_2^0, \boldsymbol{a}, \boldsymbol{\xi}, \boldsymbol{X}} \left[ \left( \frac{1}{\sqrt{h}} \boldsymbol{W}_1^1 \boldsymbol{W}_2^1 \boldsymbol{a} - \boldsymbol{\beta}^* \right) \left( \frac{1}{\sqrt{h}} \boldsymbol{W}_1^1 \boldsymbol{W}_2^1 \boldsymbol{a} - \boldsymbol{\beta}^* \right)^\top \right] \right) \\
=& tr \left( \mathbb{E}_{\boldsymbol{W}_1^0, \boldsymbol{W}_2^0, \boldsymbol{a}, \boldsymbol{\xi}, \boldsymbol{X}} \left[ \frac{1}{h} \boldsymbol{W}_1^1 \boldsymbol{W}_2^1 \boldsymbol{a} \boldsymbol{a}^\top \boldsymbol{W}_2^{1^\top} \boldsymbol{W}_1^{1^\top} \right] \right) \\
& - tr \left( \mathbb{E}_{\boldsymbol{W}_1^0, \boldsymbol{W}_2^0, \boldsymbol{a}, \boldsymbol{\xi}, \boldsymbol{X}} \left[ \frac{1}{\sqrt{h}} \boldsymbol{\beta}^* \boldsymbol{a}^\top \boldsymbol{W}_2^{1^\top} \boldsymbol{W}_1^{1^\top} \right] \right) \\
& - tr \left( \mathbb{E}_{\boldsymbol{W}_1^0, \boldsymbol{W}_2^0, \boldsymbol{a}, \boldsymbol{\xi}, \boldsymbol{X}} \left[ \frac{1}{\sqrt{h}} \boldsymbol{W}_1^1 \boldsymbol{W}_2^1 \boldsymbol{a} \boldsymbol{\beta}^{*\top} \right] \right) + tr \left( \mathbb{E} \left[ \boldsymbol{\beta}^* \boldsymbol{\beta}^{*\top} \right] \right).
\end{aligned}
\tag{139}
$$

Here we define $L_1, L_2, L_3, L_4$, where

$$
\begin{aligned}
L_1 &= tr \left( \mathbb{E}_{\boldsymbol{W}_1^0, \boldsymbol{W}_2^0, \boldsymbol{a}, \boldsymbol{\xi}, \boldsymbol{X}} \left[ \frac{1}{h} \boldsymbol{W}_1^1 \boldsymbol{W}_2^1 \boldsymbol{a} \boldsymbol{a}^\top \boldsymbol{W}_2^{1^\top} \boldsymbol{W}_1^{1^\top} \right] \right) \\
L_2 &= tr \left( \mathbb{E}_{\boldsymbol{W}_1^0, \boldsymbol{W}_2^0, \boldsymbol{a}, \boldsymbol{\xi}, \boldsymbol{X}} \left[ \frac{1}{\sqrt{h}} \boldsymbol{\beta}^* \boldsymbol{a}^\top \boldsymbol{W}_2^{1^\top} \boldsymbol{W}_1^{1^\top} \right] \right) \\
L_3 &= tr \left( \mathbb{E}_{\boldsymbol{W}_1^0, \boldsymbol{W}_2^0, \boldsymbol{a}, \boldsymbol{\xi}, \boldsymbol{X}} \left[ \frac{1}{\sqrt{h}} \boldsymbol{W}_1^1 \boldsymbol{W}_2^1 \boldsymbol{a} \boldsymbol{\beta}^{*\top} \right] \right) \\
L_4 &= tr \left( \mathbb{E} \left[ \boldsymbol{\beta}^* \boldsymbol{\beta}^{*\top} \right] \right)
\end{aligned}
$$

Thus

$$
L_{\text{three-layer}} = L_1 - L_2 - L_3 + L_4
$$

Due to Proposition E.2 and Appendix C.1.3, we know that the norm of $G_1$ is dominated by $A_1$, the norm of $G_2$ is dominated by $A_2$, to simplify the gradient, we consider the following approximation.

$$
\boldsymbol{W}_1^1 \approx \boldsymbol{W}_1^0 + \frac{\eta_1}{n\sqrt{h}} \boldsymbol{X}^\top \boldsymbol{y} \boldsymbol{a}^\top \boldsymbol{W}_2^{0^\top}.
\tag{140}
$$

$$
\boldsymbol{W}_2^1 \approx \boldsymbol{W}_2^0 + \frac{\eta_2}{n\sqrt{h}} \boldsymbol{W}_1^{0^\top} \boldsymbol{X}^\top \boldsymbol{y} \boldsymbol{a}^\top.
\tag{141}
$$

Thus we have

$$
\begin{aligned}
\boldsymbol{W}_1^1 \boldsymbol{W}_2^1 \approx & \boldsymbol{W}_1^0 \boldsymbol{W}_2^0 + \frac{\eta_1}{n\sqrt{h}} \boldsymbol{X}^\top \boldsymbol{y} \boldsymbol{a}^\top \boldsymbol{W}_2^{0^\top} \boldsymbol{W}_2^0 \\
& + \frac{\eta_2}{n\sqrt{h}} \boldsymbol{W}_1^0 \boldsymbol{W}_1^{0^\top} \boldsymbol{X}^\top \boldsymbol{y} \boldsymbol{a}^\top \\
& + \frac{\eta_1 \eta_2}{n^2 h} \boldsymbol{X}^\top \boldsymbol{y} \boldsymbol{a}^\top \boldsymbol{W}_2^{0^\top} \boldsymbol{W}_1^{0^\top} \boldsymbol{X}^\top \boldsymbol{y} \boldsymbol{a}^\top,
\end{aligned}
\tag{142}
$$

$$\boldsymbol{W}_2^{1\top}\boldsymbol{W}_1^{1\top} \approx \boldsymbol{W}_2^{0\top}\boldsymbol{W}_1^{0\top} + \frac{\eta_1}{n\sqrt{h}}\boldsymbol{W}_2^{0\top}\boldsymbol{W}_2^0\boldsymbol{a}\boldsymbol{y}^\top\boldsymbol{X}$$

$$+ \frac{\eta_2}{n\sqrt{h}}\boldsymbol{a}\boldsymbol{y}^\top\boldsymbol{X}\boldsymbol{W}_1^0\boldsymbol{W}_1^{0\top} \tag{143}$$

$$+ \frac{\eta_1\eta_2}{n^2h}\boldsymbol{a}\boldsymbol{y}^\top\boldsymbol{X}\boldsymbol{W}_1^0\boldsymbol{W}_2^0\boldsymbol{a}\boldsymbol{y}^\top\boldsymbol{X},$$

we take ( 140) and ( 141) into $L_1, L_2, L_3$.

We have $L_1 = \sum_{i=1}^{16} T_i$, where

$$T_1 = tr\left(\mathbb{E}_{\boldsymbol{W}_1^0,\boldsymbol{W}_2^0,\boldsymbol{a},\boldsymbol{\xi},\boldsymbol{X}}\left[\frac{1}{h}\boldsymbol{a}\boldsymbol{a}^\top\boldsymbol{W}_2^{0\top}\boldsymbol{W}_1^{0\top}\boldsymbol{W}_1^0\boldsymbol{W}_2^0\right]\right),$$

$$T_2 = tr\left(\mathbb{E}_{\boldsymbol{W}_1^0,\boldsymbol{W}_2^0,\boldsymbol{a},\boldsymbol{\xi},\boldsymbol{X}}\left[\frac{\eta_1}{nh\sqrt{h}}\boldsymbol{a}\boldsymbol{a}^\top\boldsymbol{W}_2^{0\top}\boldsymbol{W}_1^{0\top}\boldsymbol{X}^\top\boldsymbol{y}\boldsymbol{a}^\top\boldsymbol{W}_2^{0\top}\boldsymbol{W}_2^0\right]\right),$$

$$T_3 = tr\left(\mathbb{E}_{\boldsymbol{W}_1^0,\boldsymbol{W}_2^0,\boldsymbol{a},\boldsymbol{\xi},\boldsymbol{X}}\left[\frac{\eta_2}{nh\sqrt{h}}\boldsymbol{a}\boldsymbol{a}^\top\boldsymbol{W}_2^{0\top}\boldsymbol{W}_1^{0\top}\boldsymbol{W}_1^0\boldsymbol{W}_1^{0\top}\boldsymbol{X}^\top\boldsymbol{y}\boldsymbol{a}^\top\right]\right),$$

$$T_4 = tr\left(\mathbb{E}_{\boldsymbol{W}_1^0,\boldsymbol{W}_2^0,\boldsymbol{a},\boldsymbol{\xi},\boldsymbol{X}}\left[\frac{\eta_1\eta_2}{n^2h^2}\boldsymbol{a}\boldsymbol{a}^\top\boldsymbol{W}_2^{0\top}\boldsymbol{W}_1^{0\top}\boldsymbol{X}^\top\boldsymbol{y}\boldsymbol{a}^\top\boldsymbol{W}_2^{0\top}\boldsymbol{W}_1^{0\top}\boldsymbol{X}^\top\boldsymbol{y}\boldsymbol{a}^\top\right]\right),$$

$$T_5 = tr\left(\mathbb{E}_{\boldsymbol{W}_1^0,\boldsymbol{W}_2^0,\boldsymbol{a},\boldsymbol{\xi},\boldsymbol{X}}\left[\frac{\eta_1}{nh\sqrt{h}}\boldsymbol{a}\boldsymbol{a}^\top\boldsymbol{W}_2^{0\top}\boldsymbol{W}_2^0\boldsymbol{a}\boldsymbol{y}^\top\boldsymbol{X}\boldsymbol{W}_1^0\boldsymbol{W}_2^0\right]\right),$$

$$T_6 = tr\left(\mathbb{E}_{\boldsymbol{W}_1^0,\boldsymbol{W}_2^0,\boldsymbol{a},\boldsymbol{\xi},\boldsymbol{X}}\left[\frac{\eta_1^2}{n^2h^2}\boldsymbol{a}\boldsymbol{a}^\top\boldsymbol{W}_2^{0\top}\boldsymbol{W}_2^0\boldsymbol{a}\boldsymbol{y}^\top\boldsymbol{X}\boldsymbol{X}^\top\boldsymbol{y}\boldsymbol{a}^\top\boldsymbol{W}_2^{0\top}\boldsymbol{W}_2^0\right]\right),$$

$$T_7 = tr\left(\mathbb{E}_{\boldsymbol{W}_1^0,\boldsymbol{W}_2^0,\boldsymbol{a},\boldsymbol{\xi},\boldsymbol{X}}\left[\frac{\eta_1\eta_2}{n^2h^2}\boldsymbol{a}\boldsymbol{a}^\top\boldsymbol{W}_2^{0\top}\boldsymbol{W}_2^0\boldsymbol{a}\boldsymbol{y}^\top\boldsymbol{X}\boldsymbol{W}_1^0\boldsymbol{W}_1^{0\top}\boldsymbol{X}^\top\boldsymbol{y}\boldsymbol{a}^\top\right]\right),$$

$$T_8 = tr\left(\mathbb{E}_{\boldsymbol{W}_1^0,\boldsymbol{W}_2^0,\boldsymbol{a},\boldsymbol{\xi},\boldsymbol{X}}\left[\frac{\eta_1^2\eta_2}{n^3h^2\sqrt{h}}\boldsymbol{a}\boldsymbol{a}^\top\boldsymbol{W}_2^{0\top}\boldsymbol{W}_2^0\boldsymbol{a}\boldsymbol{y}^\top\boldsymbol{X}\boldsymbol{X}^\top\boldsymbol{y}\boldsymbol{a}^\top\boldsymbol{W}_2^{0\top}\boldsymbol{W}_1^{0\top}\boldsymbol{X}^\top\boldsymbol{y}\boldsymbol{a}^\top\right]\right),$$

$$T_9 = tr\left(\mathbb{E}_{\boldsymbol{W}_1^0,\boldsymbol{W}_2^0,\boldsymbol{a},\boldsymbol{\xi},\boldsymbol{X}}\left[\frac{\eta_2}{nh\sqrt{h}}\boldsymbol{a}\boldsymbol{a}^\top\boldsymbol{a}\boldsymbol{y}^\top\boldsymbol{X}\boldsymbol{W}_1^0\boldsymbol{W}_1^{0\top}\boldsymbol{W}_1^0\boldsymbol{W}_2^0\right]\right),$$

$$T_{10} = tr\left(\mathbb{E}_{\boldsymbol{W}_1^0,\boldsymbol{W}_2^0,\boldsymbol{a},\boldsymbol{\xi},\boldsymbol{X}}\left[\frac{\eta_1\eta_2}{n^2h^2}\boldsymbol{a}\boldsymbol{a}^\top\boldsymbol{a}\boldsymbol{y}^\top\boldsymbol{X}\boldsymbol{W}_1^0\boldsymbol{W}_1^{0\top}\boldsymbol{X}^\top\boldsymbol{y}\boldsymbol{a}^\top\boldsymbol{W}_2^{0\top}\boldsymbol{W}_2^0\right]\right),$$

$$T_{11} = tr\left(\mathbb{E}_{\boldsymbol{W}_1^0,\boldsymbol{W}_2^0,\boldsymbol{a},\boldsymbol{\xi},\boldsymbol{X}}\left[\frac{\eta_2^2}{n^2h^2}\boldsymbol{a}\boldsymbol{a}^\top\boldsymbol{a}\boldsymbol{y}^\top\boldsymbol{X}\boldsymbol{W}_1^0\boldsymbol{W}_1^{0\top}\boldsymbol{W}_1^0\boldsymbol{W}_1^{0\top}\boldsymbol{X}^\top\boldsymbol{y}\boldsymbol{a}^\top\right]\right),$$

$$T_{12} = tr\left(\mathbb{E}_{\boldsymbol{W}_1^0,\boldsymbol{W}_2^0,\boldsymbol{a},\boldsymbol{\xi},\boldsymbol{X}}\left[\frac{\eta_1\eta_2^2}{n^3h^2\sqrt{h}}\boldsymbol{a}\boldsymbol{a}^\top\boldsymbol{a}\boldsymbol{y}^\top\boldsymbol{X}\boldsymbol{W}_1^0\boldsymbol{W}_1^{0\top}\boldsymbol{X}^\top\boldsymbol{y}\boldsymbol{a}^\top\boldsymbol{W}_2^{0\top}\boldsymbol{W}_1^{0\top}\boldsymbol{X}^\top\boldsymbol{y}\boldsymbol{a}^\top\right]\right),$$

$$T_{13} = tr\left(\mathbb{E}_{\boldsymbol{W}_1^0,\boldsymbol{W}_2^0,\boldsymbol{a},\boldsymbol{\xi},\boldsymbol{X}}\left[\frac{\eta_1\eta_2}{n^2h^2}\boldsymbol{a}\boldsymbol{a}^\top\boldsymbol{a}\boldsymbol{y}^\top\boldsymbol{X}\boldsymbol{W}_1^0\boldsymbol{W}_2^0\boldsymbol{a}\boldsymbol{y}^\top\boldsymbol{X}\boldsymbol{W}_1^0\boldsymbol{W}_2^0\right]\right),$$

$$T_{14} = tr\left(\mathbb{E}_{\boldsymbol{W}_1^0,\boldsymbol{W}_2^0,\boldsymbol{a},\boldsymbol{\xi},\boldsymbol{X}}\left[\frac{\eta_1^2\eta_2}{n^3h^2\sqrt{h}}\boldsymbol{a}\boldsymbol{a}^\top\boldsymbol{a}\boldsymbol{y}^\top\boldsymbol{X}\boldsymbol{W}_1^0\boldsymbol{W}_2^0\boldsymbol{a}\boldsymbol{y}^\top\boldsymbol{X}\boldsymbol{X}^\top\boldsymbol{y}\boldsymbol{a}^\top\boldsymbol{W}_2^{0\top}\boldsymbol{W}_2^0\right]\right),$$

$$T_{15} = tr\left(\mathbb{E}_{\boldsymbol{W}_1^0,\boldsymbol{W}_2^0,\boldsymbol{a},\boldsymbol{\xi},\boldsymbol{X}}\left[\frac{\eta_1\eta_2^2}{n^3h^2\sqrt{h}}\boldsymbol{a}\boldsymbol{a}^\top\boldsymbol{a}\boldsymbol{y}^\top\boldsymbol{X}\boldsymbol{W}_1^0\boldsymbol{W}_2^0\boldsymbol{a}\boldsymbol{y}^\top\boldsymbol{X}\boldsymbol{W}_1^0\boldsymbol{W}_1^{0\top}\boldsymbol{X}^\top\boldsymbol{y}\boldsymbol{a}^\top\right]\right),$$

$$T_{16} = tr\left(\mathbb{E}_{\boldsymbol{W}_1^0,\boldsymbol{W}_2^0,\boldsymbol{a},\boldsymbol{\xi},\boldsymbol{X}}\left[\frac{\eta_1^2\eta_2^2}{n^4h^3}\boldsymbol{a}\boldsymbol{a}^\top\boldsymbol{a}\boldsymbol{y}^\top\boldsymbol{X}\boldsymbol{W}_1^0\boldsymbol{W}_2^0\boldsymbol{a}\boldsymbol{y}^\top\boldsymbol{X}\boldsymbol{X}^\top\boldsymbol{y}\boldsymbol{a}^\top\boldsymbol{W}_2^{0\top}\boldsymbol{W}_1^{0\top}\boldsymbol{X}^\top\boldsymbol{y}\boldsymbol{a}^\top\right]\right).$$

We have $L_2 = \sum_{i=17}^{20} T_i$, where

$$T_{17} = tr\left(\mathbb{E}_{\boldsymbol{W}_1^0,\boldsymbol{W}_2^0,\boldsymbol{a},\boldsymbol{\xi},\boldsymbol{X}}\left[\frac{1}{\sqrt{h}}\boldsymbol{\beta}^*\boldsymbol{a}^\top\boldsymbol{W}_2^{0\top}\boldsymbol{W}_1^{0\top}\right]\right),$$

$$T_{18} = tr\left(\mathbb{E}_{\boldsymbol{W}_1^0,\boldsymbol{W}_2^0,\boldsymbol{a},\boldsymbol{\xi},\boldsymbol{X}}\left[\frac{\eta_1}{nh}\boldsymbol{\beta}^*\boldsymbol{a}^\top\boldsymbol{W}_2^{0\top}\boldsymbol{W}_2^0\boldsymbol{a}\boldsymbol{y}^\top\boldsymbol{X}\right]\right),$$

$$T_{19} = tr\left(\mathbb{E}_{\boldsymbol{W}_1^0,\boldsymbol{W}_2^0,\boldsymbol{a},\boldsymbol{\xi},\boldsymbol{X}}\left[\frac{\eta_2}{nh}\boldsymbol{\beta}^*\boldsymbol{a}^\top\boldsymbol{a}\boldsymbol{y}^\top\boldsymbol{X}\boldsymbol{W}_1^0\boldsymbol{W}_1^{0\top}\right]\right),$$

$$T_{20} = tr\left(\mathbb{E}_{\boldsymbol{W}_1^0,\boldsymbol{W}_2^0,\boldsymbol{a},\boldsymbol{\xi},\boldsymbol{X}}\left[\frac{\eta_1\eta_2}{n^2h\sqrt{h}}\boldsymbol{\beta}^*\boldsymbol{a}^\top\boldsymbol{a}\boldsymbol{y}^\top\boldsymbol{X}\boldsymbol{W}_1^0\boldsymbol{W}_2^0\boldsymbol{a}\boldsymbol{y}^\top\boldsymbol{X}\right]\right).$$

We have $L_3 = \sum_{i=21}^{24} T_i$, where

$$T_{21} = tr\left(\mathbb{E}_{\boldsymbol{W}_1^0,\boldsymbol{W}_2^0,\boldsymbol{a},\boldsymbol{\xi},\boldsymbol{X}}\left[\frac{1}{\sqrt{h}}\boldsymbol{W}_1^0\boldsymbol{W}_2^0\boldsymbol{a}\boldsymbol{\beta}^{*\top}\right]\right),$$

$$T_{22} = tr\left(\mathbb{E}_{\boldsymbol{W}_1^0,\boldsymbol{W}_2^0,\boldsymbol{a},\boldsymbol{\xi},\boldsymbol{X}}\left[\frac{\eta_1}{nh}\boldsymbol{X}^\top\boldsymbol{y}\boldsymbol{a}^\top\boldsymbol{W}_2^{0\top}\boldsymbol{W}_2^0\boldsymbol{a}\boldsymbol{\beta}^{*\top}\right]\right),$$

$$T_{23} = tr\left(\mathbb{E}_{\boldsymbol{W}_1^0,\boldsymbol{W}_2^0,\boldsymbol{a},\boldsymbol{\xi},\boldsymbol{X}}\left[\frac{\eta_2}{nh}\boldsymbol{W}_1^0\boldsymbol{W}_1^{0\top}\boldsymbol{X}^\top\boldsymbol{y}\boldsymbol{a}^\top\boldsymbol{a}\boldsymbol{\beta}^{*\top}\right]\right),$$

$$T_{24} = tr\left(\mathbb{E}_{\boldsymbol{W}_1^0,\boldsymbol{W}_2^0,\boldsymbol{a},\boldsymbol{\xi},\boldsymbol{X}}\left[\frac{\eta_1\eta_2}{n^2h\sqrt{h}}\boldsymbol{X}^\top\boldsymbol{y}\boldsymbol{a}^\top\boldsymbol{W}_2^{0\top}\boldsymbol{W}_1^{0\top}\boldsymbol{X}^\top\boldsymbol{y}\boldsymbol{a}^\top\boldsymbol{a}\boldsymbol{\beta}^{*\top}\right]\right),$$

Thus, we obtain that

$$L = \sum_{i=1}^{16} T_i - \sum_{i=17}^{20} T_i - \sum_{i=21}^{24} T_i + L_4$$

**Analysis of $T_1$.**

$$T_1 = tr\left(\mathbb{E}_{\boldsymbol{W}_1^0,\boldsymbol{W}_2^0,\boldsymbol{a},\boldsymbol{\xi},\boldsymbol{X}}\left[\frac{1}{h}\boldsymbol{a}\boldsymbol{a}^\top\boldsymbol{W}_2^{0\top}\boldsymbol{W}_1^{0\top}\boldsymbol{W}_1^0\boldsymbol{W}_2^0\right]\right) = \frac{1}{h} \tag{144}$$

$\square$

**Analysis of $T_4$ and $T_{13}$.**  For $T_4$, we have

$$\begin{aligned}
T_4 &= tr\left(\mathbb{E}_{\boldsymbol{W}_1^0,\boldsymbol{W}_2^0,\boldsymbol{a},\boldsymbol{\xi},\boldsymbol{X}}\left[\frac{\eta_1\eta_2}{n^2h^2}\boldsymbol{a}\boldsymbol{a}^\top\boldsymbol{W}_2^{0\top}\boldsymbol{W}_1^{0\top}\boldsymbol{X}^\top\boldsymbol{y}\boldsymbol{a}^\top\boldsymbol{W}_2^{0\top}\boldsymbol{W}_1^{0\top}\boldsymbol{X}^\top\boldsymbol{y}\boldsymbol{a}^\top\right]\right) \\
&= \frac{\eta_1\eta_2}{n^2h^2}\mathbb{E}tr\left(\left[\boldsymbol{a}\boldsymbol{a}^\top\boldsymbol{W}_2^{0\top}\boldsymbol{W}_1^{0\top}\boldsymbol{X}^\top\boldsymbol{y}\boldsymbol{y}^\top\boldsymbol{X}\boldsymbol{W}_1^0\boldsymbol{W}_2^0\boldsymbol{a}\boldsymbol{a}^\top\right]\right) \\
&= \frac{\eta_1\eta_2}{n^2h^2}tr\left(\mathbb{E}\left[\boldsymbol{a}\boldsymbol{a}^\top\boldsymbol{a}\boldsymbol{a}^\top\right]\mathbb{E}\left[\boldsymbol{W}_2^{0\top}\boldsymbol{W}_1^{0\top}\boldsymbol{X}^\top\boldsymbol{y}\boldsymbol{y}^\top\boldsymbol{X}\boldsymbol{W}_1^0\boldsymbol{W}_2^0\right]\right)
\end{aligned} \tag{145}$$

For $\mathbb{E}\left[\boldsymbol{a}\boldsymbol{a}^\top\boldsymbol{a}\boldsymbol{a}^\top\right]$, similar to (159), we have

$$\mathbb{E}\left(\boldsymbol{a}\boldsymbol{a}^\top\boldsymbol{a}\boldsymbol{a}^\top\right) = \left(\frac{1}{h} + \frac{2}{h^2}\right)\boldsymbol{I}_h \tag{146}$$

By taking ( 146) in to $T_4$, we have

$$
\begin{aligned}
T_4 &= \frac{\eta_1 \eta_2}{n^2 h^2} \left( \frac{1}{h} + \frac{2}{h^2} \right) tr \left( \mathbb{E} \left[ \boldsymbol{W}_2^{0\top} \boldsymbol{W}_1^{0\top} \boldsymbol{X}^\top \boldsymbol{y} \boldsymbol{y}^\top \boldsymbol{X} \boldsymbol{W}_1^0 \boldsymbol{W}_2^0 \right] \right) \\
&= \frac{\eta_1 \eta_2}{n^2 h^2} \left( \frac{1}{h} + \frac{2}{h^2} \right) tr \left( \mathbb{E} \left[ \boldsymbol{W}_2^0 \boldsymbol{W}_2^{0\top} \boldsymbol{W}_1^{0\top} \boldsymbol{X}^\top \boldsymbol{y} \boldsymbol{y}^\top \boldsymbol{X} \boldsymbol{W}_1^0 \right] \right) \\
&= \frac{\eta_1 \eta_2}{n^2 h^2} \left( \frac{1}{h} + \frac{2}{h^2} \right) tr \left( \mathbb{E} \left[ \boldsymbol{W}_1^{0\top} \boldsymbol{X}^\top \boldsymbol{y} \boldsymbol{y}^\top \boldsymbol{X} \boldsymbol{W}_1^0 \right] \right) \\
&= \frac{\eta_1 \eta_2}{n^2 h^2} \left( \frac{1}{h} + \frac{2}{h^2} \right) tr \left( \mathbb{E} \left[ \boldsymbol{W}_1^0 \boldsymbol{W}_1^{0\top} \boldsymbol{X}^\top \boldsymbol{y} \boldsymbol{y}^\top \boldsymbol{X} \right] \right) \\
&= \frac{\eta_1 \eta_2}{n^2 h d} \left( \frac{1}{h} + \frac{2}{h^2} \right) tr \left( \mathbb{E} \left[ \boldsymbol{X}^\top \boldsymbol{y} \boldsymbol{y}^\top \boldsymbol{X} \right] \right) \\
&= \frac{\eta_1 \eta_2}{n^2 h d} \left( \frac{1}{h} + \frac{2}{h^2} \right) \left( n^2 + nd(1 + \rho_e^2) + n \right) \\
&= O \left( \frac{\eta_1 \eta_2}{h^3} \right).
\end{aligned}
\tag{147}
$$

$\square$

It is easy to find that $T_4 = T_{13}$, so we have

$$
T_{13} = O \left( \frac{\eta_1 \eta_2}{h^3} \right).
\tag{148}
$$

$\square$

**Analysis of $T_6$ and $T_{11}$.**   For $T_6$, we have

$$
\begin{aligned}
T_6 &= \frac{\eta_1^2}{n^2 h^2} tr \left( \mathbb{E}_{\boldsymbol{W}_1^0, \boldsymbol{W}_2^0, \boldsymbol{a}, \boldsymbol{\xi}, \boldsymbol{X}} \left[ \boldsymbol{a} \boldsymbol{a}^\top \boldsymbol{W}_2^{0\top} \boldsymbol{W}_2^0 \boldsymbol{a} \boldsymbol{y}^\top \boldsymbol{X} \boldsymbol{X}^\top \boldsymbol{y} \boldsymbol{a}^\top \boldsymbol{W}_2^{0\top} \boldsymbol{W}_2^0 \right] \right) \\
&= \frac{\eta_1^2}{n^2 h^2} tr \left( \mathbb{E}_{\boldsymbol{W}_1^0, \boldsymbol{W}_2^0, \boldsymbol{a}, \boldsymbol{\xi}, \boldsymbol{X}} \left[ \boldsymbol{a}^\top \boldsymbol{W}_2^{0\top} \boldsymbol{W}_2^0 \boldsymbol{a} \boldsymbol{a}^\top \boldsymbol{W}_2^{0\top} \boldsymbol{W}_2^0 \boldsymbol{a} \boldsymbol{y}^\top \boldsymbol{X} \boldsymbol{X}^\top \boldsymbol{y} \right] \right) \\
&= \frac{\eta_1^2}{n^2 h^2} \mathbb{E}_{\boldsymbol{W}_1^0, \boldsymbol{W}_2^0, \boldsymbol{a}, \boldsymbol{\xi}, \boldsymbol{X}} \left[ \boldsymbol{a}^\top \boldsymbol{W}_2^{0\top} \boldsymbol{W}_2^0 \boldsymbol{a} \boldsymbol{a}^\top \boldsymbol{W}_2^{0\top} \boldsymbol{W}_2^0 \boldsymbol{a} \boldsymbol{y}^\top \boldsymbol{X} \boldsymbol{X}^\top \boldsymbol{y} \right]
\end{aligned}
$$

let $\boldsymbol{S}_2 = \boldsymbol{W}_2^{0\top} \boldsymbol{W}_2^0$. By replacing them into $T_6$,

$$
\begin{aligned}
T_6 &= \frac{\eta_1^2}{n^2 h^2} \mathbb{E}_{\boldsymbol{W}_1^0, \boldsymbol{W}_2^0, \boldsymbol{a}, \boldsymbol{\xi}, \boldsymbol{X}} \left[ \boldsymbol{a}^\top \boldsymbol{W}_2^{0\top} \boldsymbol{W}_2^0 \boldsymbol{a} \boldsymbol{a}^\top \boldsymbol{W}_2^{0\top} \boldsymbol{W}_2^0 \boldsymbol{a} \boldsymbol{y}^\top \boldsymbol{X} \boldsymbol{X}^\top \boldsymbol{y} \right] \\
&= \frac{\eta_1^2}{n^2 h^2} \mathbb{E} \left[ \boldsymbol{a}^\top \boldsymbol{S}_2 \boldsymbol{a} \boldsymbol{a}^\top \boldsymbol{S}_2 \boldsymbol{a} \right] \mathbb{E} \left[ \boldsymbol{y}^\top \boldsymbol{X} \boldsymbol{X}^\top \boldsymbol{y} \right].
\end{aligned}
$$

Here we first analyze $\mathbb{E} \left[ \boldsymbol{a}^\top \boldsymbol{S}_2 \boldsymbol{a} \boldsymbol{a}^\top \boldsymbol{S}_2 \boldsymbol{a} \right] = \mathbb{E} \left[ \left( \sum_{i,j} a_i S_{2_{ij}} a_j \right) \left( \sum_{p,q} a_p S_{2_{pq}} a_q \right) \right]$, This can be reduced to the following three cases, since the expectations in all other cases are zero.

**Case 1** $(i, j) = (p, q), i \neq j$**.**

$$
\begin{aligned}
\mathbb{E} \left[ \boldsymbol{a}^\top \boldsymbol{S}_2 \boldsymbol{a} \boldsymbol{a}^\top \boldsymbol{S}_2 \boldsymbol{a} \right]_1 &= \mathbb{E} \left[ \left( \sum_{\substack{i,j \\ i \neq j}} a_i S_{2_{ij}} a_j \right)^2 \right] \\
&= \sum_{\substack{i,j \\ i \neq j}} \mathbb{E} a_i^2 \mathbb{E} S_{2_{ij}}^2 \mathbb{E} a_j^2 \\
&= (h^2 - h) \times \frac{1}{h} \times \frac{1}{h} \times \frac{1}{h} \times h \times \frac{1}{h} \\
&= \frac{1}{h} - \frac{1}{h^2}
\end{aligned}
\tag{149}
$$

**Case 2** $(i, j) = (q, p), i \neq j$**.** Same to Case 1,

$$\mathbb{E} \left[ \boldsymbol{a}^\top \boldsymbol{S}_2 \boldsymbol{a} \boldsymbol{a}^\top \boldsymbol{S}_2 \boldsymbol{a} \right]_2 = \frac{1}{h} - \frac{1}{h^2} \tag{150}$$

**Case 3** $i = j, p = q$**.**

$$\begin{aligned}
\mathbb{E} \left[ \boldsymbol{a}^\top \boldsymbol{S}_2 \boldsymbol{a} \boldsymbol{a}^\top \boldsymbol{S}_2 \boldsymbol{a} \right]_3 &= \mathbb{E} \left[ \left( \sum_i S_{2_{ii}} a_i^2 \right) \left( \sum_p S_{2_{pp}} a_p^2 \right) \right] \\
&= \sum_{i,p} \left[ \mathbb{E} \left( S_{2_{ii}} S_{2_{pp}} \right) \mathbb{E} \left( a_i^2 a_p^2 \right) \right] \\
&= \sum_{i=p} \left[ \mathbb{E} \left( S_{2_{ii}}^2 \right) \mathbb{E} \left( a_i^4 \right) \right] + \sum_{i \neq p} \left[ \mathbb{E} \left( S_{2_{ii}} S_{2_{pp}} \right) \mathbb{E} \left( a_i^2 a_p^2 \right) \right].
\end{aligned} \tag{151}$$

For $\mathbb{E} S_{2_{ii}}^2$, we have

$$\begin{aligned}
\mathbb{E} S_{2_{ii}}^2 &= \mathbb{E} S_{2_{11}}^2 \\
&= \mathbb{E} \left( W_{2_{11}}^2 + W_{2_{21}}^2 + \cdots + W_{2_{h1}}^2 \right)^2 \\
&= h \times \frac{3}{h^2} + (h^2 - h) \times \frac{1}{h} \times \frac{1}{h} \\
&= 1 + \frac{2}{h}.
\end{aligned} \tag{152}$$

It is east to see

$$\mathbb{E} a_i^4 = \frac{3}{h^2} \tag{153}$$

$$\mathbb{E}_{i \neq p} \left( S_{2_{ii}} S_{2_{pp}} \right) = \mathbb{E} S_{2_{ii}} \mathbb{E} S_{2_{pp}} = 1 \tag{154}$$

$$\mathbb{E}_{i \neq p} \left( a_{ii}^2 a_{pp}^2 \right) = \mathbb{E} a_{ii}^2 \mathbb{E} a_{pp}^2 = \frac{1}{h^2} \tag{155}$$

by combining ( 152) to ( 155), we have

$$\begin{aligned}
\mathbb{E} \left[ \boldsymbol{a}^\top \boldsymbol{S}_2 \boldsymbol{a} \boldsymbol{a}^\top \boldsymbol{S}_2 \boldsymbol{a} \right]_3 &= h \times (1 + \frac{1}{h^2}) \times \frac{3}{h^2} + (h^2 - h) \times \frac{1}{h^2} \\
&= 1 + \frac{2}{h} + \frac{6}{h^2}.
\end{aligned} \tag{156}$$

By combining ( 149),( 150)and( 156), we have

$$\mathbb{E} \left[ \boldsymbol{a}^\top \boldsymbol{S}_2 \boldsymbol{a} \boldsymbol{a}^\top \boldsymbol{S}_2 \boldsymbol{a} \right] = 1 + \frac{4}{h} + \frac{4}{h^2}. \tag{157}$$

$\square$

We then analyze $\mathbb{E} \left[ \boldsymbol{y}^\top \boldsymbol{X} \boldsymbol{X}^\top \boldsymbol{y} \right]$. note that $\boldsymbol{y} = \boldsymbol{X} \boldsymbol{\beta}^* + \boldsymbol{\xi}$, we have

$$\begin{aligned}
\mathbb{E} \left[ \boldsymbol{y}^\top \boldsymbol{X} \boldsymbol{X}^\top \boldsymbol{y} \right] &= \mathbb{E} \left[ \left( \boldsymbol{\beta}^{*\top} \boldsymbol{X}^\top \boldsymbol{X} + \boldsymbol{\xi}^\top \boldsymbol{X} \right) \left( \boldsymbol{X}^\top \boldsymbol{X} \boldsymbol{\beta}^* + \boldsymbol{X}^\top \boldsymbol{\xi} \right) \right] \\
&= \mathbb{E} \left( \boldsymbol{\beta}^{*\top} \boldsymbol{X}^\top \boldsymbol{X} \boldsymbol{X}^\top \boldsymbol{X} \boldsymbol{\beta}^* \right) + \mathbb{E} \left( \boldsymbol{\xi}^\top \boldsymbol{X} \boldsymbol{X}^\top \boldsymbol{\xi} \right) \\
&= tr \left[ \mathbb{E} \left( \boldsymbol{X}^\top \boldsymbol{X} \boldsymbol{X}^\top \boldsymbol{X} \right) \mathbb{E} \left( \boldsymbol{\beta}^* \boldsymbol{\beta}^{*\top} \right) \right] + tr \left[ \mathbb{E} \left( \boldsymbol{X} \boldsymbol{X}^\top \right) \mathbb{E} \left( \boldsymbol{\xi} \boldsymbol{\xi}^\top \right) \right].
\end{aligned} \tag{158}$$

For $\mathbb{E}\left(\boldsymbol{X}^{\top}\boldsymbol{X}\boldsymbol{X}^{\top}\boldsymbol{X}\right)$, we show that

$$
\begin{aligned}
\mathbb{E}\left(\boldsymbol{X}^{\top}\boldsymbol{X}\right)_{ii}^{2} &= \mathbb{E}\left(\boldsymbol{X}^{\top}\boldsymbol{X}\right)_{11}^{2} \\
&= \mathbb{E}\left(\sum_{m=1}^{n} X_{m1}^{2}\right)^{2} + \mathbb{E}\left[\sum_{j=2}^{d}\left(\sum_{k=1}^{n} X_{k1}X_{kj}\right)^{2}\right] \\
&= 3n + (n^{2}-n) + (d-1)n \\
&= n^{2} + nd + n \\
\mathbb{E}\left(\boldsymbol{X}^{\top}\boldsymbol{X}\right)_{ij}^{2} &= \mathbb{E}\left[\left(\sum_{m=1}^{n} X_{m1}^{2}\right)\left(\sum_{k=1}^{n} X_{k1}X_{k2}\right)\right] \\
&\quad + \mathbb{E}\left[\left(\sum_{m=1}^{n} X_{m2}^{2}\right)\left(\sum_{k=1}^{n} X_{k2}X_{k1}\right)\right] \\
&\quad + \sum_{j=3}^{n}\mathbb{E}\left[\left(\sum_{k=1}^{n} X_{k1}X_{kj}\right)\left(\sum_{p=1}^{n} X_{pj}X_{p2}\right)\right] \\
&= 0.
\end{aligned}
\tag{159}
$$

Which means $\mathbb{E}\left(\boldsymbol{X}^{\top}\boldsymbol{X}\boldsymbol{X}^{\top}\boldsymbol{X}\right) = (n^{2}+nd+n)\boldsymbol{I}_{d}$.

It is easy to see $\mathbb{E}\left(\boldsymbol{\beta}^{*}\boldsymbol{\beta}^{*\top}\right) = \frac{1}{d}\boldsymbol{I}_{d}$, $\mathbb{E}\left(\boldsymbol{X}\boldsymbol{X}^{\top}\right) = n\boldsymbol{I}_{n}$, $\mathbb{E}\left(\boldsymbol{\xi}\boldsymbol{\xi}^{\top}\right) = \rho_{e}^{2}\boldsymbol{I}_{n}$. Thus, consider ( 158) we have

$$
\mathbb{E}\left[\boldsymbol{y}^{\top}\boldsymbol{X}\boldsymbol{X}^{\top}\boldsymbol{y}\right] = n^{2} + nd(1+\rho_{e}^{2}) + n.
\tag{160}
$$

$\square$

By ( 157) and ( 160), we finally get

$$
\begin{aligned}
T_{6} &= \frac{\eta_{1}^{2}\left(n^{2}+nd(1+\rho_{e}^{2})+n\right)\left(1+\frac{4}{h}+\frac{4}{h^{4}}\right)}{n^{2}h^{2}} \\
&= \frac{\eta_{1}^{2}}{h^{2}} + \frac{\eta_{1}^{2}d(1+\rho_{e}^{2})}{h^{2}n} + O\left(\frac{\eta_{1}^{2}}{h^{3}}\right)
\end{aligned}
\tag{161}
$$

$\square$

For $T_{11}$, similar to $T_{6}$, let $\boldsymbol{W}_{1}^{0}\boldsymbol{W}_{1}^{0\top} = \boldsymbol{S}_{1}$, we have

$$
\begin{aligned}
T_{11} &= tr\left(\mathbb{E}_{\boldsymbol{W}_{1}^{0},\boldsymbol{W}_{2}^{0},\boldsymbol{a},\boldsymbol{\xi},\boldsymbol{X}}\left[\frac{\eta_{2}^{2}}{n^{2}h^{2}}\boldsymbol{a}\boldsymbol{a}^{\top}\boldsymbol{a}\boldsymbol{y}^{\top}\boldsymbol{X}\boldsymbol{W}_{1}^{0}\boldsymbol{W}_{1}^{0\top}\boldsymbol{W}_{1}^{0}\boldsymbol{W}_{1}^{0\top}\boldsymbol{X}^{\top}\boldsymbol{y}\boldsymbol{a}^{\top}\right]\right) \\
&= tr\left(\mathbb{E}_{\boldsymbol{W}_{1}^{0},\boldsymbol{W}_{2}^{0},\boldsymbol{a},\boldsymbol{\xi},\boldsymbol{X}}\left[\frac{\eta_{2}^{2}}{n^{2}h^{2}}\boldsymbol{a}^{\top}\boldsymbol{a}\boldsymbol{a}^{\top}\boldsymbol{a}\boldsymbol{y}^{\top}\boldsymbol{X}\boldsymbol{W}_{1}^{0}\boldsymbol{W}_{1}^{0\top}\boldsymbol{W}_{1}^{0}\boldsymbol{W}_{1}^{0\top}\boldsymbol{X}^{\top}\boldsymbol{y}\right]\right) \\
&= \frac{\eta_{2}^{2}}{n^{2}h^{2}}tr\left(\mathbb{E}\left(\boldsymbol{a}^{\top}\boldsymbol{a}\boldsymbol{a}^{\top}\boldsymbol{a}\right)\mathbb{E}\left(\boldsymbol{y}^{\top}\boldsymbol{X}\boldsymbol{W}_{1}^{0}\boldsymbol{W}_{1}^{0\top}\boldsymbol{W}_{1}^{0}\boldsymbol{W}_{1}^{0\top}\boldsymbol{X}^{\top}\boldsymbol{y}\right)\right) \\
&= \frac{\eta_{2}^{2}}{n^{2}h^{2}}tr\left(\mathbb{E}\left(\boldsymbol{a}^{\top}\boldsymbol{a}\boldsymbol{a}^{\top}\boldsymbol{a}\right)\mathbb{E}\left(\boldsymbol{y}^{\top}\boldsymbol{X}\boldsymbol{S}_{1}^{2}\boldsymbol{X}^{\top}\boldsymbol{y}\right)\right)
\end{aligned}
$$

For $\mathbb{E}\left(\boldsymbol{a}^{\top}\boldsymbol{a}\boldsymbol{a}^{\top}\boldsymbol{a}\right)$, we have

$$\mathbb{E}\left(\boldsymbol{a}^\top \boldsymbol{a}\boldsymbol{a}^\top \boldsymbol{a}\right) = \mathbb{E}\left(\sum_{i=1}^h a_i^2\right)^2$$
$$= h\mathbb{E}\left(a_1^4\right) + (h^2 - h)\mathbb{E}\left(a_1^2 a_2^2\right)$$
$$= h \times \frac{3}{h^2} + (h^2 - h) \times \frac{1}{h} \times \frac{1}{h} \tag{162}$$
$$= 1 + \frac{2}{h}$$

Take (162) into $T_6$, we have

$$T_{11} = \frac{\eta_2^2}{n^2 h^2}\left(1 + \frac{2}{h}\right) tr\mathbb{E}\left[(\boldsymbol{y}^\top \boldsymbol{X}\boldsymbol{S}_1^2 \boldsymbol{X}^\top \boldsymbol{y})\right]$$
$$= \frac{\eta_2^2}{n^2 h^2}\left(1 + \frac{2}{h}\right) tr\mathbb{E}\left[(\boldsymbol{X}^\top \boldsymbol{y}\boldsymbol{y}^\top \boldsymbol{X}\boldsymbol{S}_1^2)\right] \tag{163}$$
$$= \frac{\eta_2^2}{n^2 h^2}\left(1 + \frac{2}{h}\right) tr\mathbb{E}\left[(\boldsymbol{X}^\top \boldsymbol{y}\boldsymbol{y}^\top \boldsymbol{X})\mathbb{E}\left(\boldsymbol{S}_1^2\right)\right]$$

.

Similar to compute $\mathbb{E}\left(\boldsymbol{X}^\top \boldsymbol{X}\boldsymbol{X}^\top \boldsymbol{X}\right)$, we have

$$\mathbb{E}\left(\boldsymbol{S}_1^2\right) = \frac{h^2 + h + hd}{d^2}\boldsymbol{I}_d. \tag{164}$$

By taking (164) into $T_{11}$

$$T_{11} = \frac{\eta_2^2}{n^2 h^2}\left(1 + \frac{2}{h}\right)\frac{h^2 + h + hd}{d^2} tr\mathbb{E}\left[(\boldsymbol{X}^\top \boldsymbol{y}\boldsymbol{y}^\top \boldsymbol{X})\right]. \tag{165}$$

By (160), we have

$$T_{11} = \frac{\eta_2^2}{n^2 h^2}\left(1 + \frac{2}{h}\right)\left(\frac{h^2 + h + hd}{d^2}\right)\left(n^2 + nd(1 + \rho_e^2) + n\right)$$
$$= \frac{\eta_2^2}{d^2} + \frac{\eta_2^2}{hd} + \frac{\eta_2^2(1 + \rho_e^2)}{nh} + \frac{\eta_2^2(1 + \rho_e^2)}{nd} + O(\frac{\eta_2^2}{h^3}). \tag{166}$$

$\square$

**Analysis of $T_7$ and $T_{10}$.** We have

$$T_7 = tr\left(\mathbb{E}_{\boldsymbol{W}_1^0, \boldsymbol{W}_2^0, \boldsymbol{a}, \boldsymbol{\xi}, \boldsymbol{X}}\left[\frac{\eta_1 \eta_2}{n^2 h^2}\boldsymbol{a}\boldsymbol{a}^\top \boldsymbol{W}_2^{0\top}\boldsymbol{W}_2^0 \boldsymbol{a}\boldsymbol{y}^\top \boldsymbol{X}\boldsymbol{W}_1^0 \boldsymbol{W}_1^{0\top}\boldsymbol{X}^\top \boldsymbol{y}\boldsymbol{a}^\top\right]\right)$$
$$= \frac{\eta_1 \eta_2}{n^2 h^2}\mathbb{E}tr\left(\left[\boldsymbol{W}_2^{0\top}\boldsymbol{W}_2^0 \boldsymbol{a}\boldsymbol{y}^\top \boldsymbol{X}\boldsymbol{W}_1^0 \boldsymbol{W}_1^{0\top}\boldsymbol{X}^\top \boldsymbol{y}\boldsymbol{a}^\top \boldsymbol{a}\boldsymbol{a}^\top\right]\right)$$
$$= \frac{\eta_1 \eta_2}{n^2 h^2}\mathbb{E}tr\left(\left[\boldsymbol{a}\boldsymbol{y}^\top \boldsymbol{X}\boldsymbol{W}_1^0 \boldsymbol{W}_1^{0\top}\boldsymbol{X}^\top \boldsymbol{y}\boldsymbol{a}^\top \boldsymbol{a}\boldsymbol{a}^\top\right]\right)$$
$$= \frac{\eta_1 \eta_2}{n^2 h^2}\mathbb{E}tr\left(\left[\boldsymbol{W}_1^0 \boldsymbol{W}_1^{0\top}\boldsymbol{X}^\top \boldsymbol{y}\boldsymbol{a}^\top \boldsymbol{a}\boldsymbol{a}^\top \boldsymbol{a}\boldsymbol{y}^\top \boldsymbol{X}\right]\right) \tag{167}$$
$$= \frac{\eta_1 \eta_2}{n^2 hd}\mathbb{E}tr\left(\left[\boldsymbol{X}^\top \boldsymbol{y}\boldsymbol{a}^\top \boldsymbol{a}\boldsymbol{a}^\top \boldsymbol{a}\boldsymbol{y}^\top \boldsymbol{X}\right]\right)$$
$$= \frac{\eta_1 \eta_2}{n^2 hd}\mathbb{E}tr\left(\left[\boldsymbol{y}^\top \boldsymbol{X}\boldsymbol{X}^\top \boldsymbol{y}\boldsymbol{a}^\top \boldsymbol{a}\boldsymbol{a}^\top \boldsymbol{a}\right]\right)$$
$$= \frac{\eta_1 \eta_2}{n^2 hd}\mathbb{E}\left[\boldsymbol{y}^\top \boldsymbol{X}\boldsymbol{X}^\top \boldsymbol{y}\right]\mathbb{E}\left[\boldsymbol{a}^\top \boldsymbol{a}\boldsymbol{a}^\top \boldsymbol{a}\right].$$

By (160) and (162), we show

$$T_7 = \frac{\eta_1 \eta_2}{n^2 hd}\left(n^2 + nd(1 + \rho_e^2) + n\right)\left(1 + \frac{2}{h}\right)$$
$$= \frac{\eta_1 \eta_2}{hd} + \frac{\eta_1 \eta_2(1 + \rho_e^2)}{nh} + O(\frac{\eta_1 \eta_2}{h^3}). \tag{168}$$

$\square$

It is easy to find that $T_7 = T_{10}$, so we have

$$T_{10} = \frac{\eta_1\eta_2}{nh} + \frac{\eta_1\eta_2(1+\rho_e^2)}{nh} + O(\frac{\eta_1\eta_2}{h^3}). \tag{169}$$

$\square$

**Analysis of $T_{16}$.**

$$\begin{aligned}
T_{16} &= \frac{\eta_1^2\eta_2^2}{n^4h^3} tr\left(\mathbb{E}_{\boldsymbol{W}_1^0,\boldsymbol{W}_2^0,\boldsymbol{a},\boldsymbol{\xi},\boldsymbol{X}}\left[\boldsymbol{a}\boldsymbol{a}^\top\boldsymbol{a}\boldsymbol{y}^\top\boldsymbol{X}\boldsymbol{W}_1^0\boldsymbol{W}_2^0\boldsymbol{a}\boldsymbol{y}^\top\boldsymbol{X}\boldsymbol{X}^\top\boldsymbol{y}\boldsymbol{a}^\top\boldsymbol{W}_2^{0\top}\boldsymbol{W}_1^{0\top}\boldsymbol{X}^\top\boldsymbol{y}\boldsymbol{a}^\top\right]\right) \\
&= \frac{\eta_1^2\eta_2^2}{n^4h^3} tr\left(\mathbb{E}\left[\boldsymbol{a}\boldsymbol{a}^\top\boldsymbol{a}\left[\boldsymbol{y}^\top\boldsymbol{X}\boldsymbol{W}_1^0\boldsymbol{W}_2^0\boldsymbol{a}\right]^\top\boldsymbol{y}^\top\boldsymbol{X}\boldsymbol{X}^\top\boldsymbol{y}\left[\boldsymbol{a}^\top\boldsymbol{W}_2^{0\top}\boldsymbol{W}_1^{0\top}\boldsymbol{X}^\top\boldsymbol{y}\right]^\top\boldsymbol{a}^\top\right]\right) \\
&= \frac{\eta_1^2\eta_2^2}{n^4h^3} tr\left(\mathbb{E}\left[\boldsymbol{a}\boldsymbol{a}^\top\boldsymbol{a}\boldsymbol{a}^\top\boldsymbol{W}_2^{0\top}\boldsymbol{W}_1^{0\top}\boldsymbol{X}^\top\boldsymbol{y}\boldsymbol{y}^\top\boldsymbol{X}\boldsymbol{X}^\top\boldsymbol{y}\boldsymbol{y}^\top\boldsymbol{X}\boldsymbol{W}_1^0\boldsymbol{W}_2^0\boldsymbol{a}\boldsymbol{a}^\top\right]\right) \\
&= \frac{\eta_1^2\eta_2^2}{n^4h^3} tr\left(\mathbb{E}\left[\boldsymbol{a}\boldsymbol{a}^\top\boldsymbol{a}\boldsymbol{a}^\top\boldsymbol{a}\boldsymbol{a}^\top\boldsymbol{W}_2^{0\top}\boldsymbol{W}_1^{0\top}\boldsymbol{X}^\top\boldsymbol{y}\boldsymbol{y}^\top\boldsymbol{X}\boldsymbol{X}^\top\boldsymbol{y}\boldsymbol{y}^\top\boldsymbol{X}\boldsymbol{W}_1^0\boldsymbol{W}_2^0\right]\right) \\
&= \frac{\eta_1^2\eta_2^2}{n^4h^3} tr\left(\mathbb{E}\left[\boldsymbol{a}\boldsymbol{a}^\top\boldsymbol{a}\boldsymbol{a}^\top\boldsymbol{a}\boldsymbol{a}^\top\right]\mathbb{E}\left[\boldsymbol{W}_2^{0\top}\boldsymbol{W}_1^{0\top}\boldsymbol{X}^\top\boldsymbol{y}\boldsymbol{y}^\top\boldsymbol{X}\boldsymbol{X}^\top\boldsymbol{y}\boldsymbol{y}^\top\boldsymbol{X}\boldsymbol{W}_1^0\boldsymbol{W}_2^0\right]\right)
\end{aligned} \tag{170}$$

For $\mathbb{E}\left[\boldsymbol{a}\boldsymbol{a}^\top\boldsymbol{a}\boldsymbol{a}^\top\boldsymbol{a}\boldsymbol{a}^\top\right]$, we have

$$\begin{aligned}
\mathbb{E}\left[\boldsymbol{a}\boldsymbol{a}^\top\boldsymbol{a}\boldsymbol{a}^\top\boldsymbol{a}\boldsymbol{a}^\top\right]_{11} &= \mathbb{E}\left[\left(\sum_{i=1}^h a_i^2\right)^2 a_1^2\right] = \frac{1}{h} + O(\frac{1}{h^2}), \\
\mathbb{E}\left[\boldsymbol{a}\boldsymbol{a}^\top\boldsymbol{a}\boldsymbol{a}^\top\boldsymbol{a}\boldsymbol{a}^\top\right]_{11} &= \mathbb{E}\left[\left(\sum_{i=1}^h a_i^2\right)^2 a_1 a_2\right] = 0.
\end{aligned} \tag{171}$$

Which means $\mathbb{E}\left[\boldsymbol{a}\boldsymbol{a}^\top\boldsymbol{a}\boldsymbol{a}^\top\boldsymbol{a}\boldsymbol{a}^\top\right] = \left(\frac{1}{h} + O(\frac{1}{h^2})\right)\boldsymbol{I}_h$, taking it into $T_{16}$

$$\begin{aligned}
T_{16} &= \frac{\eta_1^2\eta_2^2}{n^4h^3}\left(\frac{1}{h} + O(\frac{1}{h^2})\right) tr\mathbb{E}\left[\boldsymbol{W}_2^{0\top}\boldsymbol{W}_1^{0\top}\boldsymbol{X}^\top\boldsymbol{y}\boldsymbol{y}^\top\boldsymbol{X}\boldsymbol{X}^\top\boldsymbol{y}\boldsymbol{y}^\top\boldsymbol{X}\boldsymbol{W}_1^0\boldsymbol{W}_2^0\right] \\
&= \frac{\eta_1^2\eta_2^2}{n^4h^3}\left(\frac{1}{h} + O(\frac{1}{h^2})\right) tr\mathbb{E}\left[\boldsymbol{W}_2^0\boldsymbol{W}_2^{0\top}\boldsymbol{W}_1^{0\top}\boldsymbol{X}^\top\boldsymbol{y}\boldsymbol{y}^\top\boldsymbol{X}\boldsymbol{X}^\top\boldsymbol{y}\boldsymbol{y}^\top\boldsymbol{X}\boldsymbol{W}_1^0\right] \\
&= \frac{\eta_1^2\eta_2^2}{n^4h^3}\left(\frac{1}{h} + O(\frac{1}{h^2})\right) tr\mathbb{E}\left[\boldsymbol{W}_1^{0\top}\boldsymbol{X}^\top\boldsymbol{y}\boldsymbol{y}^\top\boldsymbol{X}\boldsymbol{X}^\top\boldsymbol{y}\boldsymbol{y}^\top\boldsymbol{X}\boldsymbol{W}_1^0\right] \\
&= \frac{\eta_1^2\eta_2^2}{n^4h^3}\left(\frac{1}{h} + O(\frac{1}{h^2})\right) tr\mathbb{E}\left[\boldsymbol{W}_1^0\boldsymbol{W}_1^{0\top}\boldsymbol{X}^\top\boldsymbol{y}\boldsymbol{y}^\top\boldsymbol{X}\boldsymbol{X}^\top\boldsymbol{y}\boldsymbol{y}^\top\boldsymbol{X}\right] \\
&= \frac{\eta_1^2\eta_2^2}{n^4h^2d}\left(\frac{1}{h} + O(\frac{1}{h^2})\right) tr\mathbb{E}\left[\boldsymbol{X}^\top\boldsymbol{y}\boldsymbol{y}^\top\boldsymbol{X}\boldsymbol{X}^\top\boldsymbol{y}\boldsymbol{y}^\top\boldsymbol{X}\right]
\end{aligned} \tag{172}$$

For $tr\mathbb{E}\left[\boldsymbol{X}^\top \boldsymbol{y}\boldsymbol{y}^\top \boldsymbol{X}\boldsymbol{X}^\top \boldsymbol{y}\boldsymbol{y}^\top \boldsymbol{X}\right]$, we have

$$
\begin{aligned}
tr\mathbb{E}\left[\boldsymbol{X}^\top \boldsymbol{y}\boldsymbol{y}^\top \boldsymbol{X}\boldsymbol{X}^\top \boldsymbol{y}\boldsymbol{y}^\top \boldsymbol{X}\right] &= tr\mathbb{E}\left[\boldsymbol{X}^\top \boldsymbol{X}\boldsymbol{\beta}^*\boldsymbol{\beta}^{*^\top}\boldsymbol{X}^\top \boldsymbol{X}\boldsymbol{X}^\top \boldsymbol{\xi}\boldsymbol{\xi}^\top \boldsymbol{X}\right] \\
&+ tr\mathbb{E}\left[\boldsymbol{X}^\top \boldsymbol{\xi}\boldsymbol{\xi}^\top \boldsymbol{X}\boldsymbol{X}^\top \boldsymbol{X}\boldsymbol{\beta}^*\boldsymbol{\beta}^{*^\top}\boldsymbol{X}^\top \boldsymbol{X}\right] \\
&+ tr\mathbb{E}\left[\boldsymbol{X}^\top \boldsymbol{X}\boldsymbol{\beta}^*\boldsymbol{\xi}^\top \boldsymbol{X}\boldsymbol{X}^\top \boldsymbol{X}\boldsymbol{\beta}^*\boldsymbol{\xi}^\top \boldsymbol{X}\right] \\
&+ tr\mathbb{E}\left[\boldsymbol{X}^\top \boldsymbol{\xi}\boldsymbol{\beta}^{*^\top}\boldsymbol{X}^\top \boldsymbol{X}\boldsymbol{X}^\top \boldsymbol{\xi}\boldsymbol{\beta}^{*^\top}\boldsymbol{X}^\top \boldsymbol{X}\right] \\
&+ tr\mathbb{E}\left[\boldsymbol{X}^\top \boldsymbol{\xi}\boldsymbol{\xi}^\top \boldsymbol{X}\boldsymbol{X}^\top \boldsymbol{\xi}\boldsymbol{\xi}^\top \boldsymbol{X}\right] \\
&+ tr\mathbb{E}\left[\boldsymbol{X}^\top \boldsymbol{X}\boldsymbol{\beta}^*\boldsymbol{\xi}^\top \boldsymbol{X}\boldsymbol{X}^\top \boldsymbol{\xi}\boldsymbol{\beta}^{*^\top}\boldsymbol{X}^\top \boldsymbol{X}\right] \\
&+ tr\mathbb{E}\left[\boldsymbol{X}^\top \boldsymbol{\xi}\boldsymbol{\beta}^{*^\top}\boldsymbol{X}^\top \boldsymbol{X}\boldsymbol{X}^\top \boldsymbol{X}\boldsymbol{\beta}^*\boldsymbol{\xi}^\top \boldsymbol{X}\right] \\
&+ tr\mathbb{E}\left[\boldsymbol{X}^\top \boldsymbol{X}\boldsymbol{\beta}^*\boldsymbol{\beta}^{*^\top}\boldsymbol{X}^\top \boldsymbol{X}\boldsymbol{X}^\top \boldsymbol{X}\boldsymbol{\beta}^*\boldsymbol{\beta}^{*^\top}\boldsymbol{X}^\top \boldsymbol{X}\right]
\end{aligned}
$$

It is easy to see

$$
\begin{aligned}
tr\mathbb{E}\left[\boldsymbol{X}^\top \boldsymbol{X}\boldsymbol{\beta}^*\boldsymbol{\beta}^{*^\top}\boldsymbol{X}^\top \boldsymbol{X}\boldsymbol{X}^\top \boldsymbol{\xi}\boldsymbol{\xi}^\top \boldsymbol{X}\right] &= tr\mathbb{E}\left[\boldsymbol{X}^\top \boldsymbol{\xi}\boldsymbol{\xi}^\top \boldsymbol{X}\boldsymbol{X}^\top \boldsymbol{X}\boldsymbol{\beta}^*\boldsymbol{\beta}^{*^\top}\boldsymbol{X}^\top \boldsymbol{X}\right] \\
&= tr\mathbb{E}\left[\boldsymbol{X}^\top \boldsymbol{X}\boldsymbol{\beta}^*\boldsymbol{\xi}^\top \boldsymbol{X}\boldsymbol{X}^\top \boldsymbol{X}\boldsymbol{\beta}^*\boldsymbol{\xi}^\top \boldsymbol{X}\right] \\
&= tr\mathbb{E}\left[\boldsymbol{X}^\top \boldsymbol{\xi}\boldsymbol{\beta}^{*^\top}\boldsymbol{X}^\top \boldsymbol{X}\boldsymbol{X}^\top \boldsymbol{\xi}\boldsymbol{\beta}^{*^\top}\boldsymbol{X}^\top \boldsymbol{X}\right] \\
&= \frac{\rho_e^2}{d}tr\mathbb{E}\left[\boldsymbol{X}\boldsymbol{X}^\top \boldsymbol{X}\boldsymbol{X}\boldsymbol{X}^\top \boldsymbol{X}\right] \\
&= \frac{\rho_e^2}{d}\mathbb{E}\left\|\boldsymbol{X}\boldsymbol{X}^\top \boldsymbol{X}\right\|_F^2 \\
&\le \frac{\rho_e^2}{d}\mathbb{E}\left\|\boldsymbol{X}\right\|^2 \left\|\boldsymbol{X}\right\|^2 \left\|\boldsymbol{X}\right\|_F^2 \\
&= O(n^3)
\end{aligned}
$$

It is also easy to see $tr\mathbb{E}\left[\boldsymbol{X}^\top \boldsymbol{\xi}\boldsymbol{\xi}^\top \boldsymbol{X}\boldsymbol{X}^\top \boldsymbol{\xi}\boldsymbol{\xi}^\top \boldsymbol{X}\right] = \mathbb{E}\left\|\boldsymbol{X}\boldsymbol{X}^\top \boldsymbol{\xi}\boldsymbol{\xi}^\top\right\|_F^2 \le O(n^3)$.

Here we focus on computing

$$
tr\mathbb{E}\left[\boldsymbol{X}^\top \boldsymbol{X}\boldsymbol{\beta}^*\boldsymbol{\xi}^\top \boldsymbol{X}\boldsymbol{X}^\top \boldsymbol{\xi}\boldsymbol{\beta}^{*^\top}\boldsymbol{X}^\top \boldsymbol{X}\right], tr\mathbb{E}\left[\boldsymbol{X}^\top \boldsymbol{X}\boldsymbol{\beta}^*\boldsymbol{\beta}^{*^\top}\boldsymbol{X}^\top \boldsymbol{X}\boldsymbol{X}^\top \boldsymbol{X}\boldsymbol{\beta}^*\boldsymbol{\beta}^{*^\top}\boldsymbol{X}^\top \boldsymbol{X}\right]
$$

since we have

$$
tr\mathbb{E}\left[\boldsymbol{X}^\top \boldsymbol{X}\boldsymbol{\beta}^*\boldsymbol{\xi}^\top \boldsymbol{X}\boldsymbol{X}^\top \boldsymbol{\xi}\boldsymbol{\beta}^{*^\top}\boldsymbol{X}^\top \boldsymbol{X}\right] = tr\mathbb{E}\left[\boldsymbol{X}^\top \boldsymbol{\xi}\boldsymbol{\beta}^{*^\top}\boldsymbol{X}^\top \boldsymbol{X}\boldsymbol{X}^\top \boldsymbol{X}\boldsymbol{\beta}^*\boldsymbol{\xi}^\top \boldsymbol{X}\right]
$$

For $tr\mathbb{E}\left[\boldsymbol{X}^\top\boldsymbol{X}\boldsymbol{\beta}^*\boldsymbol{\xi}^\top\boldsymbol{X}\boldsymbol{X}^\top\boldsymbol{\xi}\boldsymbol{\beta}^{*^\top}\boldsymbol{X}^\top\boldsymbol{X}\right]$, we have

$$
\begin{aligned}
&tr\mathbb{E}\left[\boldsymbol{X}^\top\boldsymbol{X}\boldsymbol{\beta}^*\boldsymbol{\xi}^\top\boldsymbol{X}\boldsymbol{X}^\top\boldsymbol{\xi}\boldsymbol{\beta}^{*^\top}\boldsymbol{X}^\top\boldsymbol{X}\right]\\
=&tr\mathbb{E}\left[\boldsymbol{\xi}^\top\boldsymbol{X}\boldsymbol{X}^\top\boldsymbol{\xi}\boldsymbol{X}^\top\boldsymbol{X}\boldsymbol{\beta}^*\boldsymbol{\beta}^{*^\top}\boldsymbol{X}^\top\boldsymbol{X}\right]\\
=&tr\mathbb{E}\left[\boldsymbol{\beta}^*\boldsymbol{\beta}^{*^\top}\boldsymbol{X}^\top\boldsymbol{X}\boldsymbol{\xi}^\top\boldsymbol{X}\boldsymbol{X}^\top\boldsymbol{\xi}\boldsymbol{X}^\top\boldsymbol{X}\right]\\
=&\frac{\rho_e^2}{d}\mathbb{E}\left[tr\left(\boldsymbol{X}^\top\boldsymbol{X}\right)tr\left(\boldsymbol{X}^\top\boldsymbol{X}\boldsymbol{X}^\top\boldsymbol{X}\right)\right]\\
=&\frac{\rho_e^2}{d}\mathbb{E}\left[\left\|\boldsymbol{X}^\top\boldsymbol{X}\right\|_F^2\left\|\boldsymbol{X}\right\|_F^2\right]\\
=&\frac{\rho_e^2}{d}\mathbb{E}\left[\left(\sum_{i,j}X_{ij}^2\right)\left(\sum_{p,q}\left(X^\top X\right)_{pq}^2\right)\right]\\
=&\frac{\rho_e^2}{d}\mathbb{E}\left[\left(\sum_{i,j}X_{ij}^2\right)\left(\sum_{p=1}^d\left(\sum_{m=1}^n X_{mp}^2\right)^2+\sum_{\substack{p,q\\p\neq q}}\left(\sum_{k=1}^n X_{kp}X_{kq}\right)^2\right)\right]\\
=&n\rho_e^2\mathbb{E}\left[X_{11}^2\left(\sum_{p=1}^d\left(\sum_{m=1}^n X_{mp}^2\right)^2+\sum_{\substack{p,q\\p\neq q}}\left(\sum_{k=1}^n X_{kp}X_{kq}\right)^2\right)\right]
\end{aligned}
\tag{173}
$$

$$
\begin{aligned}
&n\rho_e^2\mathbb{E}\left[X_{11}^2\left(\sum_{p=1}^d\left(\sum_{m=1}^n X_{mp}^2\right)^2\right)\right]\\
=&nd\times\left[(d-1)(3n+n^2-n)+15+2\times(n-1)\times 3+n^2-2\times(n-1)-1\right]\\
=&n^3d\rho_e^2+O(n^2d)
\end{aligned}
\tag{174}
$$

$$
\begin{aligned}
n\rho_e^2\mathbb{E}\left[X_{11}^2\left(\sum_{\substack{p,q\\p\neq q}}\left(\sum_{k=1}^n X_{kp}X_{kq}\right)^2\right)\right]&=n\rho_e^2\times\left[(d^2-d-2)n+2\times 3n\right]\\
&=n^2d^2\rho_e^2+O(n^2d)
\end{aligned}
\tag{175}
$$

By ( 174) and ( 175) we have

$$
tr\mathbb{E}\left[\boldsymbol{X}^\top\boldsymbol{X}\boldsymbol{\beta}^*\boldsymbol{\xi}^\top\boldsymbol{X}\boldsymbol{X}^\top\boldsymbol{\xi}\boldsymbol{\beta}^{*^\top}\boldsymbol{X}^\top\boldsymbol{X}\right]=n^3d\rho_e^2+n^2d^2\rho_e^2+O(n^2d)
\tag{176}
$$

For $tr\mathbb{E}\left[\boldsymbol{X}^\top\boldsymbol{X}\boldsymbol{\beta}^*\boldsymbol{\beta}^{*^\top}\boldsymbol{X}^\top\boldsymbol{X}\boldsymbol{X}^\top\boldsymbol{X}\boldsymbol{\beta}^*\boldsymbol{\beta}^{*^\top}\boldsymbol{X}^\top\boldsymbol{X}\right]$, we have

$$
\begin{aligned}
&tr\mathbb{E}\left[\boldsymbol{X}^\top\boldsymbol{X}\boldsymbol{\beta}^*\boldsymbol{\beta}^{*^\top}\boldsymbol{X}^\top\boldsymbol{X}\boldsymbol{X}^\top\boldsymbol{X}\boldsymbol{\beta}^*\boldsymbol{\beta}^{*^\top}\boldsymbol{X}^\top\boldsymbol{X}\right]\\
=&\mathbb{E}\left\|\boldsymbol{X}^\top\boldsymbol{X}\boldsymbol{\beta}^*\boldsymbol{\beta}^{*^\top}\boldsymbol{X}^\top\boldsymbol{X}\right\|_F^2\\
=&\sum_{i,j}\left(\sum_{k=1}^n\sum_{m=1}^d\sum_{s=1}^d\sum_{q=1}^n X_{ki}X_{km}\beta_m^*\beta_s^{*^\top}X_{qs}X_{qj}\right)^2.
\end{aligned}
\tag{177}
$$

**Case 1** $i\neq j, m=s$. It is easy to see in this case, the main term holds when $3\leq m\leq d$ and $k\neq q$, since other conditions will only have up to $O(n^2d)$. Combining with the condition $3\leq m\leq d$, we have

$$
tr\mathbb{E}\left[\boldsymbol{X}^\top\boldsymbol{X}\boldsymbol{\beta}^*\boldsymbol{\beta}^{*^\top}\boldsymbol{X}^\top\boldsymbol{X}\boldsymbol{X}^\top\boldsymbol{X}\boldsymbol{\beta}^*\boldsymbol{\beta}^{*^\top}\boldsymbol{X}^\top\boldsymbol{X}\right]_1=O(n^2d).
$$

**Case 2** $i \neq j, m \neq s$. It is easy to see in this case, the main term holds when $s \geq 2, m \geq 2$, and other conditions will only have up to $O(n^2 d)$. Combining with the condition $s \geq 2, m \geq 2$, we have

$$tr\mathbb{E}\left[\boldsymbol{X}^\top \boldsymbol{X}\boldsymbol{\beta}^*\boldsymbol{\beta}^{*\top}\boldsymbol{X}^\top\boldsymbol{X}\boldsymbol{X}^\top\boldsymbol{X}\boldsymbol{\beta}^*\boldsymbol{\beta}^{*\top}\boldsymbol{X}^\top\boldsymbol{X}\right]_2 = 4n^2 d^2 + O(n^2 d).$$

**Case 3** $i = j, m = s$. In this case, the main term still holds when $3 \leq m \leq d$ and $k \neq q$. We have

$$tr\mathbb{E}\left[\boldsymbol{X}^\top \boldsymbol{X}\boldsymbol{\beta}^*\boldsymbol{\beta}^{*\top}\boldsymbol{X}^\top\boldsymbol{X}\boldsymbol{X}^\top\boldsymbol{X}\boldsymbol{\beta}^*\boldsymbol{\beta}^{*\top}\boldsymbol{X}^\top\boldsymbol{X}\right]_3 = O(n^2 d).$$

**Case 4** $i = j, m \neq s$. In this case, the main term holds when $s \geq 2, m \geq 2$, we have

$$tr\mathbb{E}\left[\boldsymbol{X}^\top \boldsymbol{X}\boldsymbol{\beta}^*\boldsymbol{\beta}^{*\top}\boldsymbol{X}^\top\boldsymbol{X}\boldsymbol{X}^\top\boldsymbol{X}\boldsymbol{\beta}^*\boldsymbol{\beta}^{*\top}\boldsymbol{X}^\top\boldsymbol{X}\right]_4 = O(n^2 d).$$

By Case 1 to Case 4, We finally get

$$tr\mathbb{E}\left[\boldsymbol{X}^\top \boldsymbol{X}\boldsymbol{\beta}^*\boldsymbol{\beta}^{*\top}\boldsymbol{X}^\top\boldsymbol{X}\boldsymbol{X}^\top\boldsymbol{X}\boldsymbol{\beta}^*\boldsymbol{\beta}^{*\top}\boldsymbol{X}^\top\boldsymbol{X}\right] = 4n^2 d^2 + O(n^2 d) \tag{178}$$

Here, we take ( 176) and( 178), we finally get

$$
\begin{aligned}
T_{16} &= \frac{\eta_1^2 \eta_2^2}{n^4 h^2 d}\left(\frac{1}{h} + O(\frac{1}{h^2})\right)\left[(2n^3 d + 2n^2 d^2)\rho_e^2 + 4n^2 d^2 + O(n^2 d)\right] \\
&= \frac{\eta_1^2 \eta_2^2 \rho_e^2}{nh^e} + \frac{\eta_1^2 \eta_2^2 d \rho_e^2}{n^2 h^3} + \frac{4\eta_1^2 \eta_2^2 d}{n^2 h^3} + O\left(\frac{\eta_1^2 \eta_2^2}{h^5}\right)
\end{aligned} \tag{179}
$$

$\square$

**Analysis of $T_2$, $T_3$, $T_5$, $T_8$, $T_9$, $T_{12}$, $T_{14}$, $T_{15}$, $T_{17}$, $T_{20}$, $T_{21}$ and $T_{24}$.** All terms involve the product of an odd number of identical random matrices with zero mean, and due to their independence from other random matrices, these terms are all 0. $\square$

**Analysis of $T_{18}$, $T_{19}$, $T_{22}$ and $T_{23}$.** We take $T_{18}$ as an example.

$$
\begin{aligned}
T_{18} &= tr\left(\mathbb{E}_{\boldsymbol{W}_1^0, \boldsymbol{W}_2^0, \boldsymbol{a}, \boldsymbol{\xi}, \boldsymbol{X}}\left[\frac{\eta_1}{nh}\boldsymbol{\beta}^*\boldsymbol{a}^\top \boldsymbol{W}_2^{0\top}\boldsymbol{W}_2^0\boldsymbol{a}\boldsymbol{y}^\top\boldsymbol{X}\right]\right) \\
&= \frac{\eta_1}{nh}\mathbb{E}tr\left[\boldsymbol{W}_2^{0\top}\boldsymbol{W}_2^0\boldsymbol{a}\boldsymbol{y}^\top\boldsymbol{X}\boldsymbol{\beta}^*\boldsymbol{a}^\top\right] \\
&= \frac{\eta_1}{nh}\mathbb{E}tr\left[\boldsymbol{a}^\top\boldsymbol{a}\boldsymbol{y}^\top\boldsymbol{X}\boldsymbol{\beta}^*\right] \\
&= \frac{\eta_1}{nh}\mathbb{E}tr\left[\boldsymbol{y}^\top\boldsymbol{X}\boldsymbol{\beta}^*\right] \\
&= \frac{\eta_1}{nh}\mathbb{E}tr\left[\boldsymbol{y}^\top\boldsymbol{X}\boldsymbol{\beta}^*\right] \\
&= \frac{\eta_1}{nh}\mathbb{E}tr\left[\boldsymbol{\beta}^{*\top}\boldsymbol{X}^\top\boldsymbol{X}\boldsymbol{\beta}^*\right] \\
&= \frac{\eta_1}{nh}\mathbb{E}tr\left[\boldsymbol{X}^\top\boldsymbol{X}\boldsymbol{\beta}^*\boldsymbol{\beta}^{*\top}\right] \\
&= \frac{\eta_1}{h}
\end{aligned} \tag{180}
$$

Similar to $T_{18}$, we can get

$$T_{19} = \frac{\eta_2}{d} \tag{181}$$

$$T_{22} = \frac{\eta_1}{h} \tag{182}$$

$$T_{23} = \frac{\eta_2}{d} \tag{183}$$

□

Finally, we get the exact loss

$$
\begin{aligned}
L_{\text{three-layer}} = & \frac{\eta_1^2}{h^2} + \frac{\eta_1^2 d(1+\rho_e^2)}{h^2 n} - 2\frac{\eta_1}{h} \\
& + \frac{\eta_2^2}{d^2} + \frac{\eta_2^2}{hd} + \frac{\eta_2^2(1+\rho_e^2)}{nh} + \frac{\eta_2^2(1+\rho_e^2)}{nd} - 2\frac{\eta_2}{d} \\
& + \frac{2\eta_1\eta_2}{hd} + \frac{2\eta_1\eta_2(1+\rho_e^2)}{nh} + \frac{\eta_1^2\eta_2^2\rho_e^2}{nh^3} + \frac{\eta_1^2\eta_2^2 d\rho_e^2}{n^2 h^3} + \frac{4\eta_1^2\eta_2^2 d}{n^2 h^3} \\
& + 1 + O(\frac{\eta_1^2}{h^3}) + O(\frac{\eta_2^2}{h^3}) + O(\frac{\eta_1\eta_2}{h^3}) + O\left(\frac{\eta_1^2\eta_2^2}{h^5}\right)
\end{aligned}
\tag{184}
$$

Under Assumption E.1, we have

$$
\begin{aligned}
L_{\text{three-layer}} = & \frac{\eta_1^2}{h^2} + \frac{\eta_1^2(1+\rho_e^2)}{hn} - 2\frac{\eta_1}{h} \\
& + \frac{2\eta_2^2}{h^2} + \frac{2\eta_2^2(1+\rho_e^2)}{nh} - 2\frac{\eta_2}{h} \\
& + \frac{2\eta_1\eta_2}{h^2} + \frac{2\eta_1\eta_2(1+\rho_e^2)}{nh} + \frac{\eta_1^2\eta_2^2\rho_e^2}{nh^3} + \frac{\eta_1^2\eta_2^2\rho_e^2}{n^2 h^2} + \frac{4\eta_1^2\eta_2^2}{n^2 h^2} \\
& + 1 + O(\frac{\eta_1^2}{h^3}) + O(\frac{\eta_2^2}{h^3}) + O(\frac{\eta_1\eta_2}{h^3}) + O\left(\frac{\eta_1^2\eta_2^2}{h^5}\right)
\end{aligned}
\tag{185}
$$

□

# F. Additional Experiments

## F.1. Spectral Analysis

To better understand Proposition 5.1, in this subsection, we perform spectral analysis of the key matrices like $\{A_l^0\}_{l=1}^2, \{B_l^0\}_{l=1}^2, \{G_l^0\}_{l=1}^2, \{\widetilde{A_l^1}\}_{l=1}^2, \{\widetilde{B_l^1}\}_{l=1}^2, \{\widetilde{G_l^1}\}_{l=1}^2$ arising after one-step and two-step updates in a two-layer linear neural network under orthogonal initialization. In Figure 3, we consider $\eta_1 = \eta_2 = h^{\frac{3}{2}}$ with $h = 1000$, we visualize the empirical spectral densities (ESDs) of the weight matrices, gradient matrices, and the decomposed gradient components represented as $A$ and $B$. Take $\{A_l^0\}_{l=1}^2, \{B_l^0\}_{l=1}^2, \{G_l^0\}_{l=1}^2$ as examples, we find that the eigenvalue scales of $\{A_l^0\}_{l=1}^2$ and $\{G_l^0\}_{l=1}^2$ are comparable, and are larger than those of $\{B_l^0\}_{l=1}^2$ by an $O(h)$ factor. This matches our norm analysis in Section C.1.2 and C.2.1, showing that $\|A_l^0\|$ exceeds $\|B_l^0\|$) by $O(\sqrt{h})$, since the ESD is computed from the eigenvalues of $W^\top W$. We visualize the norm gap in Figure 3q and 3r, which also confirm that the eigenvalue scales of $\{A_l^0\}_{l=1}^2$ and $\{G_l^0\}_{l=1}^2$ are comparable, and are larger than those of $\{B_l^0\}_{l=1}^2$ in magnitude, we also find the eigenvalue scales of $\{\eta_l G_l^0\}_{l=1}^2$ and $\{\eta_l A_l^0\}_{l=1}^2$ are comparable to $\{\widetilde{W_l^1}\}_{l=1}^2$, which matches our Proposition 5.1. Consequently, the ESDs provide an intuitive explanation for why $\{A_l^0\}_{l=1}^2$ and $\{G_l^0\}_{l=1}^2$ are close in norm, supporting our use of the approximate gradient when deriving the one-step and two-step exact losses. A similar phenomenon holds for $\{\widetilde{A_l^1}\}_{l=1}^2$ and $\{\overline{G_l^1}\}_{l=1}^2$, relative to $\{\widetilde{B_l^1}\}_{l=1}^2$.

## F.2. Theoretical Simulation

Here we present more experimental results.

**Orthogonal initialization.** In Figure 5 and 7, Under orthogonal initialization, we set $h = 1000$ and conducted experiments for steps $\in \{1, 2, 4, 8\}$ under the constraint $\eta_1 + \eta_2 \leq h^{\frac{3}{2}}$. Consistent with our earlier findings: after two updates the model exhibits local optimality at balanced layer-wise learning rates. We also try different $h$ and find that when $h$ satisfy the condition on $h$ in Corollary like Corollary 5.4, the balanced learning-rate allocation is locally optimal, otherwise not. See Figure 11, 12 and 13.

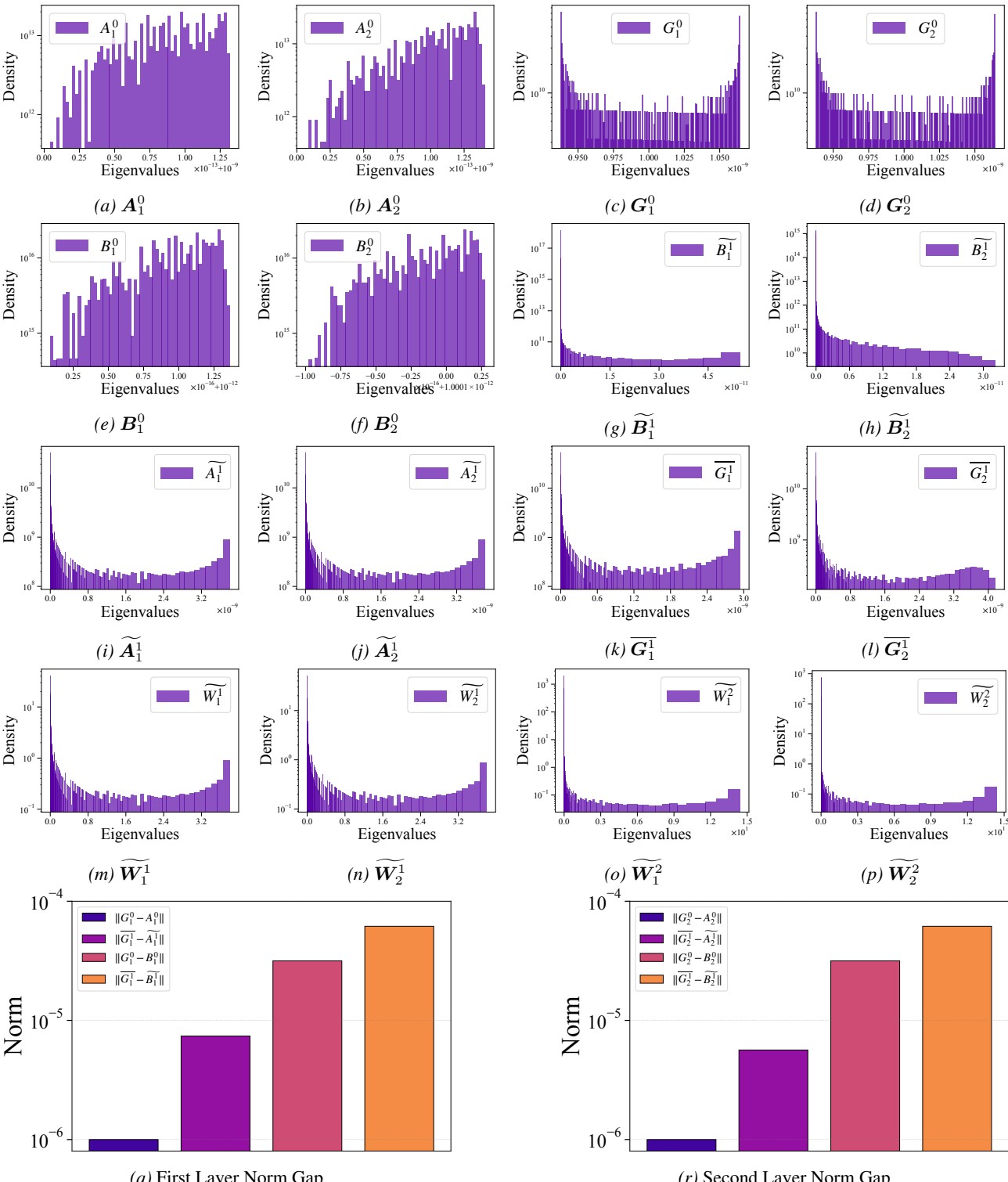

*Figure 3.* We visualize the ESDs of gradient matrices and weight matrices $\{A_l^0\}_{l=1}^2$, $\{B_l^0\}_{l=1}^2$, $\{G_l^0\}_{l=1}^2$, $\{\widetilde{A_l^1}\}_{l=1}^2$, $\{\widetilde{B_l^1}\}_{l=1}^2$, $\{\overline{G_l^1}\}_{l=1}^2$, $\{\widetilde{W_l^1}\}_{l=1}^2$ and $\{\widetilde{W_l^2}\}_{l=1}^2$ and norm gap with $\eta_1 = \eta_2 = h^{\frac{3}{2}}$ and $h = 1000$.

**Orthogonal initialization.** We also ran the same set of experiments under Gaussian initialization In Figure 6 and 8, with $h = 1000$ and $\eta_1 + \eta_2 \leq h^{\frac{2}{3}}$, again for steps $\in \{1, 2, 4, 8\}$. The results mirror those under orthogonal initialization:

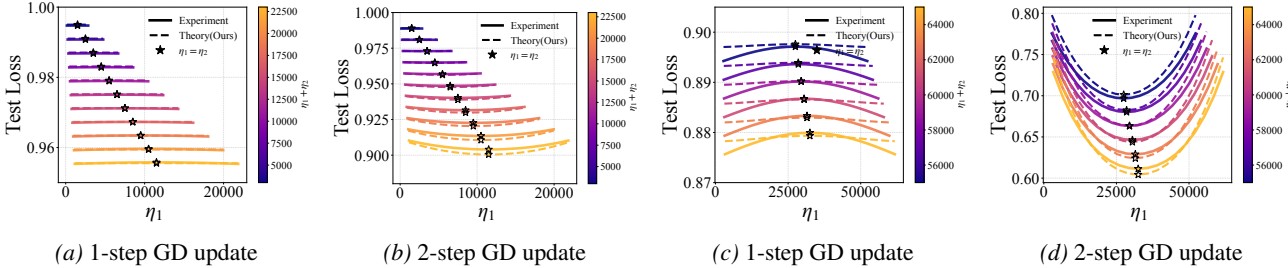

*Figure 4.* Test loss of a 2-layer NN under orthogonal initialization with width $h = 1000$. Here we set $\eta_1 + \eta_2 \leq O(h^{\frac{3}{2}})$. Our theory accurately predicts the test loss after 1-step and 2-step gradient descent updates with varying learning rates. In particular, we highlight the role of balancing learning rates across layers (i.e $\eta_1 = \eta_2$) on the test loss.

balanced learning rates become locally optimal after two updates, whereas after a single update an asymmetric learning-rate allocation performs better, which is consistent with the special cases of Theorem E.4 and Theorem E.6 for two-layer and three-layer neural networks.

**More discussions in Section 7.**  For the question about our paper shows symmetric learning rates are suboptimal for one step but optimal for two steps, which may be not exactly the same as using asymmetric learning rates early and symmetric ones later. Here we agree that our theory focuses on the result that symmetric learning rates are suboptimal for a single update step but become optimal after two steps. This suggests that 1. asymmetric learning rates may be preferable at the very beginning of training, 2. symmetric learning rates become optimal as cross-layer interactions develop over subsequent steps, even if the initial learning-rate allocation is not optimal. We believe this also points to a practical strategy that use asymmetric learning rates early in training and more symmetric ones later.

To better connect these two regimes, in Figure 16(a)(b) we consider a three layer linear network in which the first step uses an asymmetric learning rate allocation by training only the first layer. For the second step, we then search over the test loss as a function of $\eta_1$ under the constraint $\eta_1 + \eta_2 = C$. We find that the same transition still appears: from asymmetry at the first step to balance at the second step.

In Figure 16(c)(d)(e), we consider a 3-layer CNN whose first two layers are convolutional layers and whose final layer is a fixed linear readout layer. We consider a synthetic binary image classification problem on $16 \times 16$ grayscale images. Each sample belongs to one of two classes: Class $-1$: an image containing a horizontal bar. Class $+1$: an image containing a vertical bar. We consider two trainable $3 \times 3$ convolutional layers with no bias: the first maps from 1 input channel to 8 hidden channels, and the second maps from 8 channels to 8 channels, with a ReLU activation after each convolution. The resulting feature map is then globally average pooled over the spatial dimensions, producing an 8-dimensional representation, which is fed into a fixed random linear readout to produce a single scalar output.

In Figure 16(c)(d), for the first step, we consider only updating the first(second) layer, using learning rate $C$. For the second step, we study layer-wise learning-rate allocation under the constraint $\eta_1 + \eta_2 = C$. Although the optimum no longer occurs exactly at $\eta_1 = \eta_2$ because of the changed architecture and the presence of nonlinearities, the optimal performance is still attained when $\eta_1 \approx \eta_2$. This indicates that, after an asymmetric first step, learning rates that are approximately symmetric still remain preferable. In Figure 16(e), We study a 3-layer CNN in which the second update step uses a symmetric learning-rate allocation, $\eta_1 = \eta_2 = \frac{C}{2}$, while the first step is optimized under the constraint $\eta_1 + \eta_2 = C$. We find training the second layer (corresponding to $\eta_1 = 0$) at the first step is better than the symmetric allocation. Thus, the CNN experiments lead to similar conclusions as in the 3-layer linear neural network setting, further reinforcing our claim.

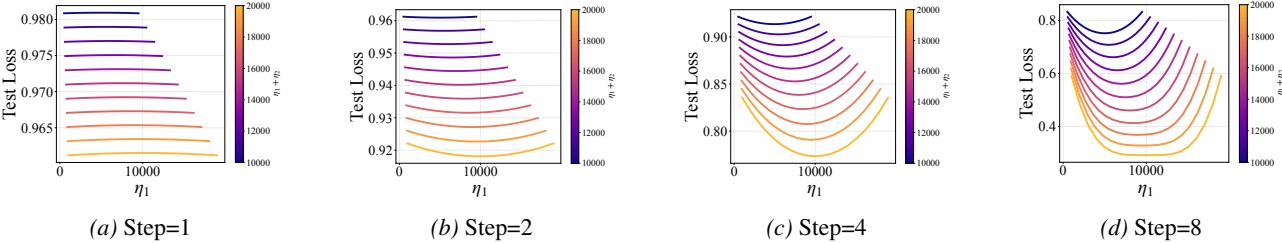

*(a)* Step=1  *(b)* Step=2  *(c)* Step=4  *(d)* Step=8

*Figure 5.* **More-steps-empirical-loss for 2-layer NN under Orthogonal initialization.** Here we set $\eta_1 + \eta_2 \leq O(h^{\frac{3}{2}})$ and $h = 1000$.

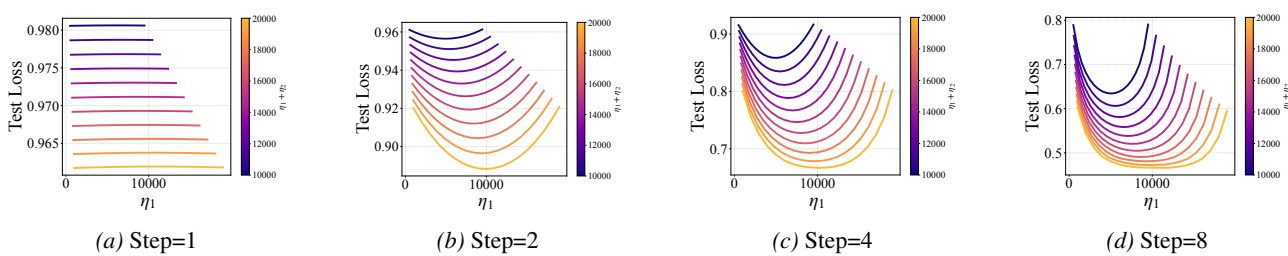

*(a)* Step=1  *(b)* Step=2  *(c)* Step=4  *(d)* Step=8

*Figure 6.* **More-steps-empirical-loss for 2-layer NN under Gaussian initialization.** Here we set $\eta_1 + \eta_2 \leq O(h^{\frac{3}{2}})$ and $h = 1000$.

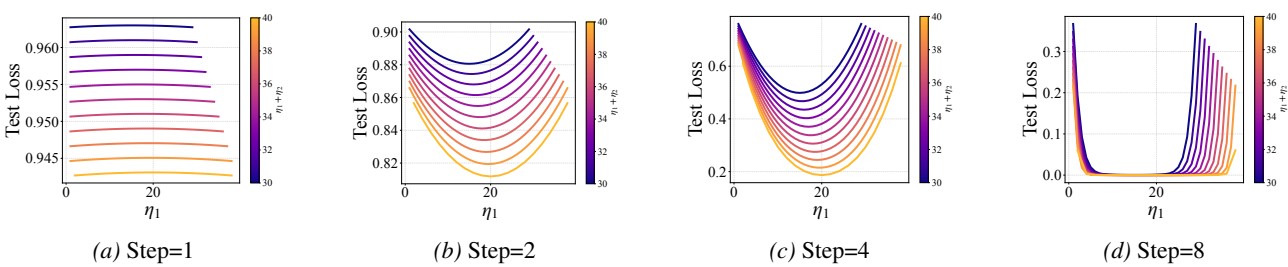

*(a)* Step=1  *(b)* Step=2  *(c)* Step=4  *(d)* Step=8

*Figure 7.* **More-steps-empirical-loss for 3-layer NN under Orthogonal initialization.** Here we set $\eta_1 + \eta_2 \leq O(h^{\frac{2}{3}})$ and $h = 1000$.

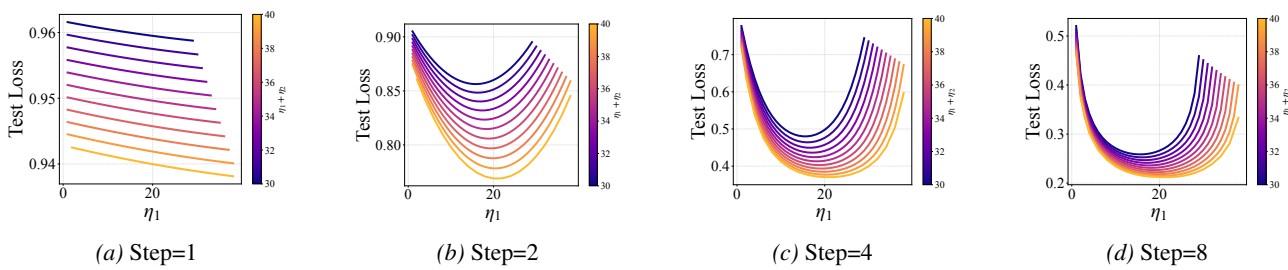

*(a)* Step=1  *(b)* Step=2  *(c)* Step=4  *(d)* Step=8

*Figure 8.* **More-steps-empirical-loss for 3-layer NN under Gaussian initialization.** Here we set $\eta_1 + \eta_2 \leq O(h^{\frac{2}{3}})$ and $h = 1000$.

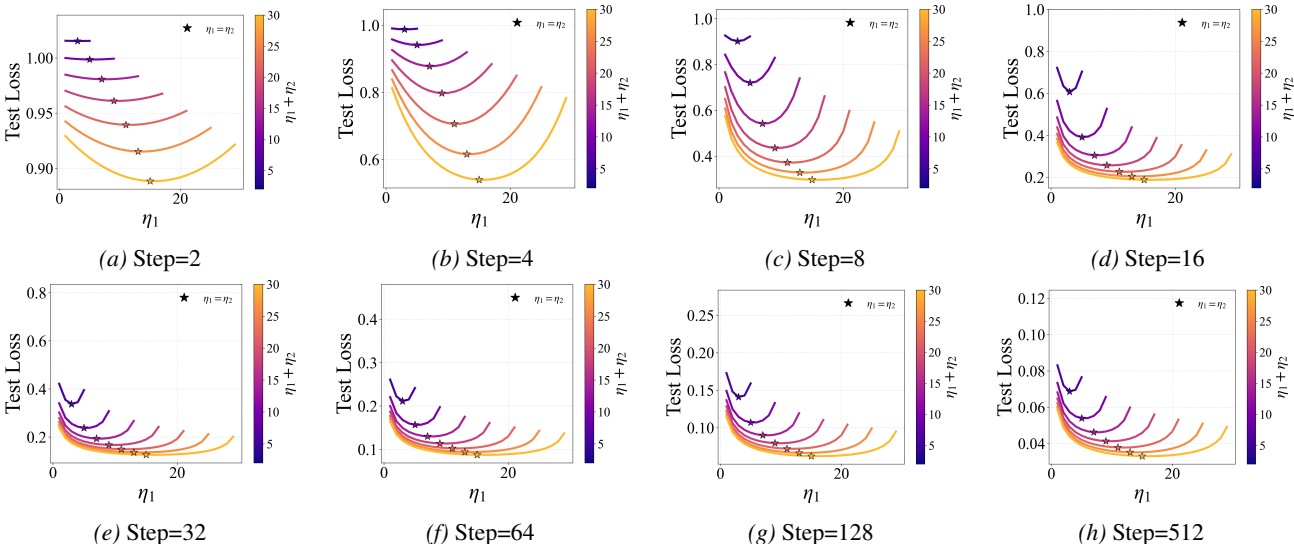

*Figure 9.* **More-steps-empirical-loss for 3-layer NN under Orthogonal initialization up to 512 steps.** Here we set $\eta_1 + \eta_2 \leq O(h^{\frac{2}{3}})$ and $h = 1000$.

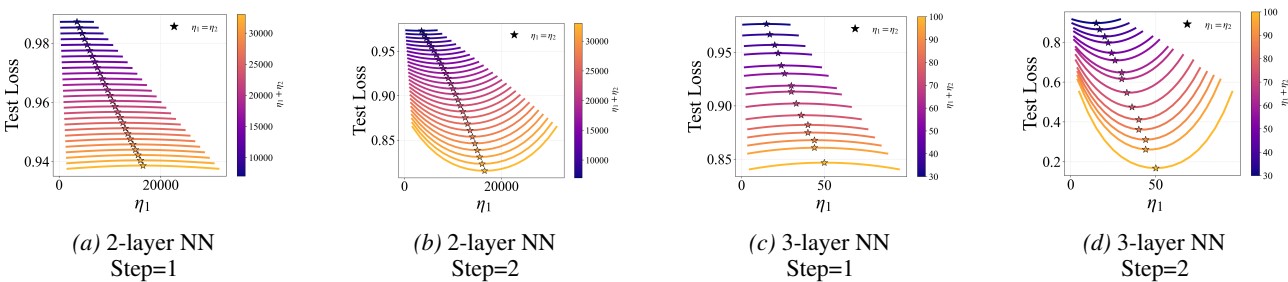

*Figure 10.* (a)(b)**2-layer NN with label noise** $\xi \in \mathcal{N}(0, \rho)$ **under orthogonal initialization.** Here we set $\eta_1 + \eta_2 \leq O(h^{\frac{3}{2}})$ with $h = 1000$ and $\rho = 0.001$. (c)(d)**3-layer NN with label noise** $\xi \in \mathcal{N}(0, \rho)$ **under orthogonal initialization.** Here we set $\eta_1 + \eta_2 \leq O(h^{\frac{2}{3}})$ with $h = 1000$ and $\rho = 0.001$.

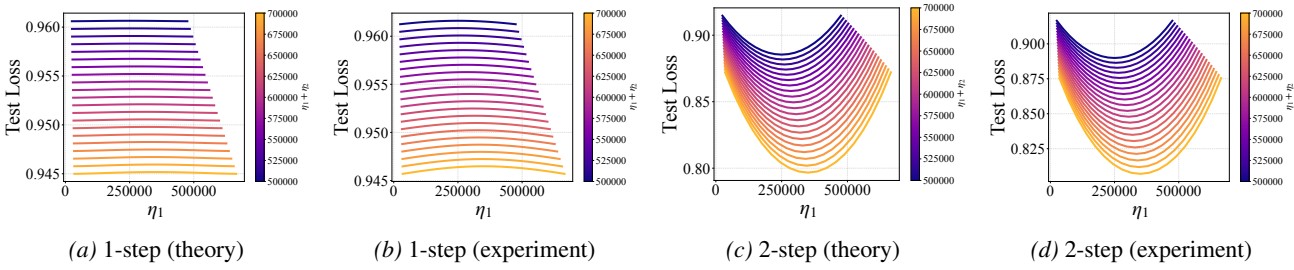

*Figure 11.* **2-layer NN under orthogonal initialization.** Here we set $\eta_1 + \eta_2 \leq O(h^{\frac{3}{2}})$ and $h = 5000$.

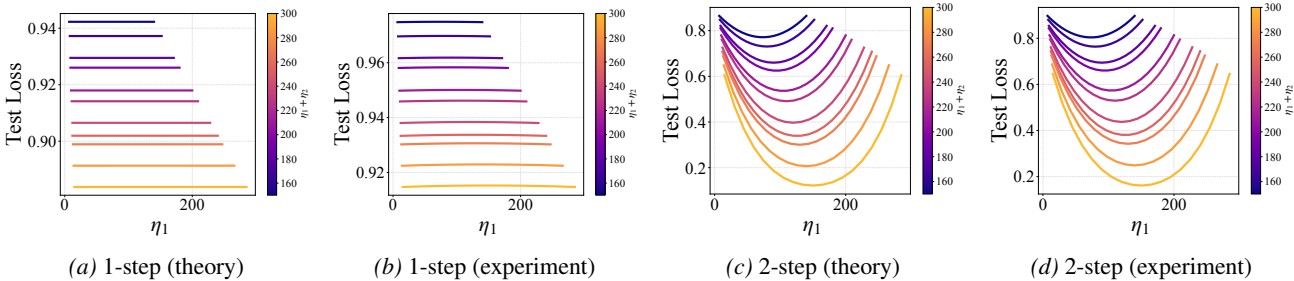

*(a)* 1-step (theory)  *(b)* 1-step (experiment)  *(c)* 2-step (theory)  *(d)* 2-step (experiment)

*Figure 12.* **3-layer NN under orthogonal initialization.** Here we set $\eta_1 + \eta_2 \leq O(h^{\frac{2}{3}})$ and $h = 5000$.

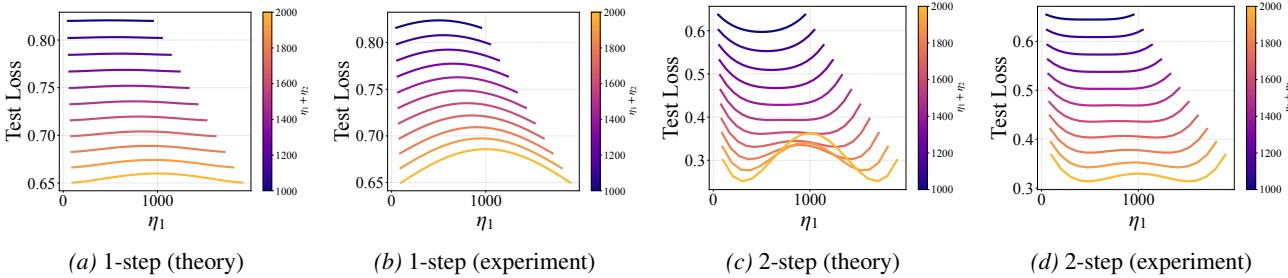

*(a)* 1-step (theory)  *(b)* 1-step (experiment)  *(c)* 2-step (theory)  *(d)* 2-step (experiment)

*Figure 13.* **2-layer NN under orthogonal initialization.** Here we set $\eta_1 + \eta_2 \leq O(h^{\frac{3}{2}})$ and $h = 100$. We can see since $h$ does not satisfy the condition on $h$ in Corollary 5.4, the balanced learning-rate allocation is not locally optimal.

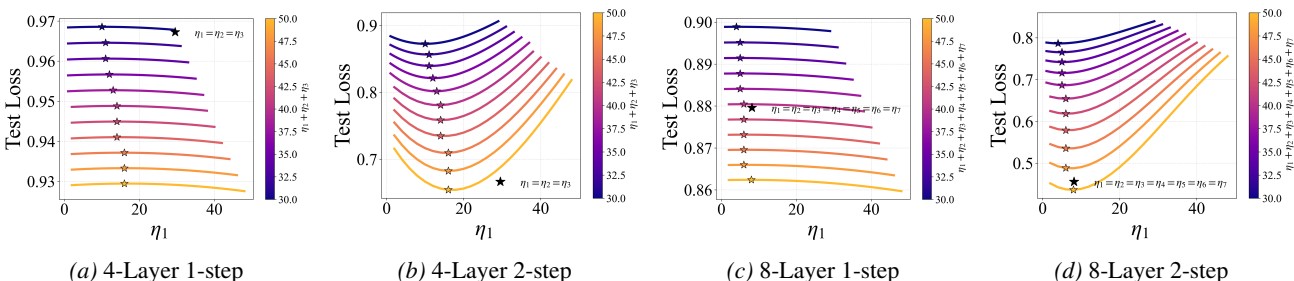

*(a)* 4-Layer 1-step  *(b)* 4-Layer 2-step  *(c)* 8-Layer 1-step  *(d)* 8-Layer 2-step

*Figure 14.* **4-layer and 8-layer NN under orthogonal initialization for 1 and 2-step updates.** For 4-NN, we set $\eta_1 + \eta_2 + \eta_3 = C \leq O(h^{\frac{2}{3}})$ with $h = 1000$ and we set $\eta_2 = \eta_3 = \frac{C-\eta_1}{2}$. For 8-NN, we set $\eta_1 + \eta_2 + \eta_3 + \eta_4 + \eta_5 + \eta_6 + \eta_7 = C \leq O(h^{\frac{2}{3}})$ with $h = 1000$ and we set $\eta_2 = \eta_3 = \eta_4 = \eta_5 = \eta_6 = \eta_7 = \frac{C-\eta_1}{6}$.

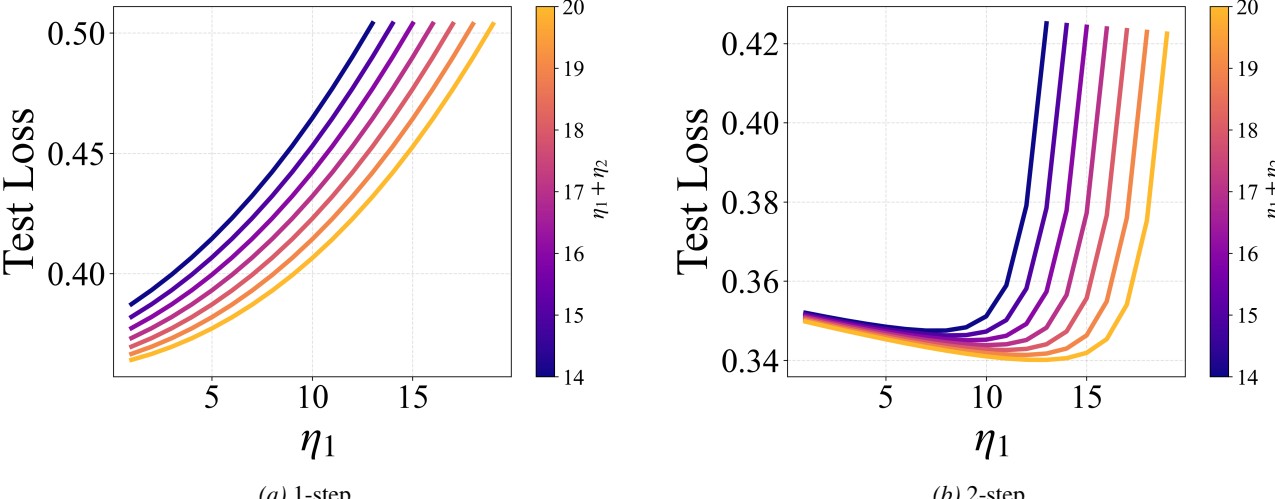

*(a)* 1-step

*(b)* 2-step

*Figure 15.* **3-NN nonlinear under orthogonal initialization for 1 and 8-step updates.** Here we consider student model is $f(\boldsymbol{x}_i) = \frac{1}{\sqrt{h}}\sigma(\sigma(\boldsymbol{x}_i^\top \boldsymbol{W}_1)\boldsymbol{W}_2)\boldsymbol{a}$, and the teacher model is $\boldsymbol{y}_i = \sigma(\boldsymbol{\beta}^{*\top}\boldsymbol{x}_i)$, with $\sigma$ being the ReLU activation.

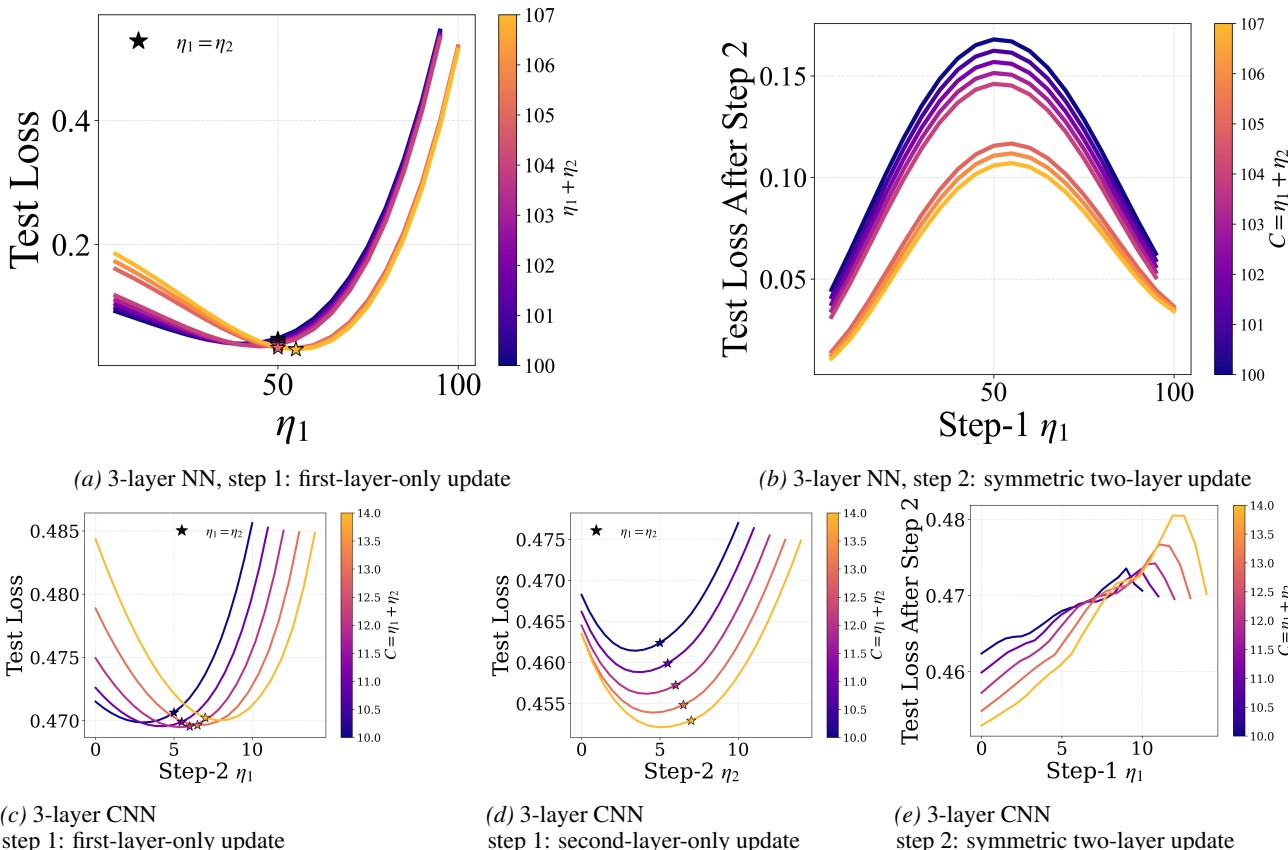

*(a)* 3-layer NN, step 1: first-layer-only update

*(b)* 3-layer NN, step 2: symmetric two-layer update

*(c)* 3-layer CNN
step 1: first-layer-only update

*(d)* 3-layer CNN
step 1: second-layer-only update

*(e)* 3-layer CNN
step 2: symmetric two-layer update

*Figure 16.* **(a)** 3-layer NN under orthogonal initialization for special 2-step update. Here we set $\eta_1 + \eta_2 = C \leq O(h^{\frac{2}{3}})$ with $h = 1000$. We set the first step update step to use an asymmetric learning-rate allocation: the first layer is updated with learning rate C, while the second layer is not trained. For the second update step, we then optimize under the constraint $\eta_1 + \eta_2 = C$. **(b)** 3-layer NN under orthogonal initialization for special 2-step update. Here we set $\eta_1 + \eta_2 = C$. We set the second update step to use an symmetric learning-rate allocation: $\eta_1 = \eta_2 = \frac{C}{2}$. For the first update step, we optimize under the constraint $\eta_1 + \eta_2 = C$. **(c)** 3-layer CNN for special 2-step update. Here we set $\eta_1 + \eta_2 = C$. We set the first update step to use an asymmetric learning-rate allocation: the first layer is updated with learning rate C, while the second layer is not trained. For the second update step, we then optimize under the constraint $\eta_1 + \eta_2 = C$. **(d)** 3-layer CNN for special 2-step update. Here we set $\eta_1 + \eta_2 = C$. We set the first update step to use an asymmetric learning-rate allocation: the second layer is updated with learning rate C, while the first layer is not trained. For the second update step, we then optimize under the constraint $\eta_1 + \eta_2 = C$. **(e)** 3-layer CNN for special 2-step update. Here we set $\eta_1 + \eta_2 = C$. We set the second update step to use an symmetric learning-rate allocation: $\eta_1 = \eta_2 = \frac{C}{2}$. For the first update step, we optimize under the constraint $\eta_1 + \eta_2 = C$.

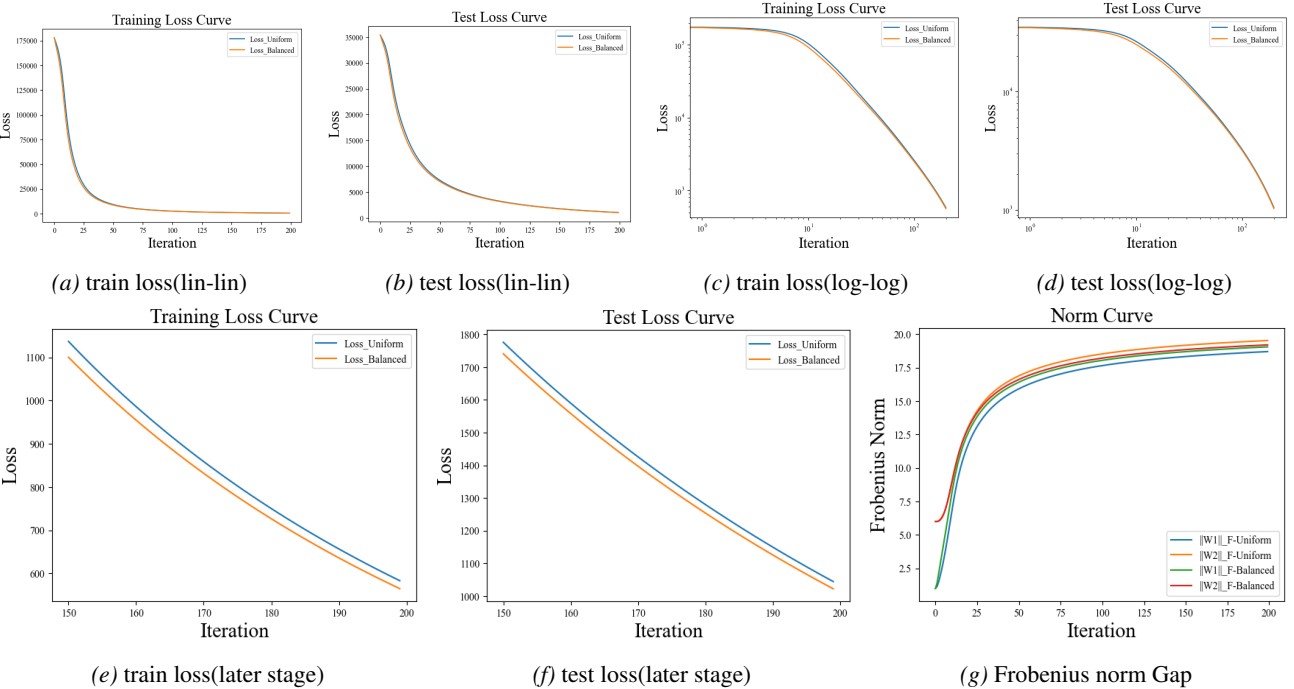

*(a)* train loss(lin-lin)  *(b)* test loss(lin-lin)  *(c)* train loss(log-log)  *(d)* test loss(log-log)

*(e)* train loss(later stage)  *(f)* test loss(later stage)  *(g)* Frobenius norm Gap

*Figure 17.* **Insights for designing layer-wise lr scheduler.** Here we condier $W_1 \in \mathbb{R}^{60 \times 100}$, $W_2 \in \mathbb{R}^{100 \times 60}$ and $M \in \mathbb{R}^{60 \times 60}$, with $\|W_1\|_F = 1$ and $\|W_2\|_F = 6$ at initialization. We use 100 training samples and 20 test samples, base $lr = 0.0001$, adopt the MSE loss, and train the model using gradient descent for 200 iterations.

