# OpenReview forum: "Balancing Learning Rates Across Layers: Exact Two-Step Dynamics and Optimal Scaling in Linear Neural Networks"
_ICML.cc/2026/Conference — ICML 2026 regular_

### Official Review · Reviewer_mafu · 2026-03-10

**Soundness:** 3
**Presentation:** 3
**Significance:** 3
**Originality:** 3
**Overall Recommendation:** 5
**Confidence:** 4

**Summary:**

This paper derives an expression for the exact test loss for a linear neural network with two layers and also with two trainable layers and a third fixed layer. This expression is a function of the learning rates of the two layers $\eta_1, \eta_2$. The expression assumes data from a linear teacher model with an orthogonal transform (for the two-layer case) and test inputs taken from an isotropic Gaussian. The authors show that the balanced learning rates $\eta_1=\eta_2$ are suboptimal for the single step update but optimal for the two-step update. The authors also include an interpretation of the exact solutions.

**Compliance With Llm Reviewing Policy:**

Affirmed.

**Final Justification:**

I have raised my score, the authors addressed all of my concerns, running extra experiments (which were probably somewhat out of scope of the original paper) to show that this phenomenon continues to hold.

Of my concerns:
* Phrasing: The authors have made clear they understand the difference between what they proved and what they claimed and will highlight this in the final version
* Experiments: The authors ran extra experiments, increasing depth, adding ReLU and moving to CNNs. While the CNN results do not perfectly match theory (symmetry is not quite optimal), they are close and I believe may with the correct scaling
* Changing LR after the first step: The authors ran good proxies for this, searching over layer 1/layer 2 independently while fixing the other at the point predicted by their theory, even extending this to CNNs on a toy image task

Even if this ends up not to be a practical learning algorithm due to small differences, I still believe it adds to our understanding of the field and is of interest to the community.

**Key Questions For Authors:**

Questions
1. From my understanding, W_1^t W_2^t in equation 9 will approach hM (assuming the network learns the data), and we can ignore B_l^t only for small t before it has learnt enough (assuming a low enough learning rate)? If so it would help to state this explicitly
2. Why is the 1-step update only correct up to terms of order O(\frac{\eta_1 \eta_2}{h^5}) but 2-step update exact? As far as I can tell from D.1, the only term skipped is specifically \frac{2 \eta_1 \eta_2}{h^5}, but \frac{\eta_1^2 \eta_2^2}{h^7} is included. Is there another approximation in the Equations for T_{1-24} that isn't stated explicitly? Does the analysis change at all if this is included?

**Limitations:**

No - see weaknesses

**Strengths And Weaknesses:**

Strengths
1. Exact analysis are very useful for giving insights into phenomena, in particular the breakdown of the 1-step and 2-step updates was very interesting
2. Understanding learning rate impact on generalization is important in practice and this takes a solid step in that direction

Weaknesses
1. The framing of this paper implies that for efficient training we want asymmetric learning rates early in training and symmetric ones later. In reality it shows a slightly different thing, namely that for a single step symmetric learning rates are suboptimal and optimal for two steps. This is distinct from taking a single step with asymmetric learning rates *followed* by a step with symmetric learning rates which is implied as the takeaway. There could be dependence here that makes these two different (perhaps not in the linear/convex setting considered though). The authors should revise the framing to make clear what was actually shown.
2. The difference between optimal and suboptimal for the first step is incredibly small and it is unclear how much this distinction is important, it would significantly strengthen the paper if the authors could provide a case where taking the first step with symmetric learning rates results in significant degradation of performance.
3. The limitation to two trained layers with a linear network is significant. While I understand the deeper/non-linear networks are out of scope, the paper could be improved by examining this empirically with a ReLU network and a deep linear network on a toy dataset.

Minor points:
* Phrasing around one step/two step is ambiguous, e.g. L199 'after a single gradient step, symmetric learning rates \eta_1=\eta_2 do not minimize the test loss' which I believe is meaning to say 'FOR the first gradient step', it currently sounds like it is talking about after having completed a single gradient step. Knowing the context now I understand what it is trying to say but the most obvious interpretation of that sentence to me implies the wrong thing. This type of phrasing is consistently ambiguous throughout the paper and took me longer than I'd like to admit to follow.
* The plots in Figure 1 are confusing to understand what symmetry means visually, to show symmetry I would find it easier to interpret having $\eta_1$ and $\eta_2$ on x/y and loss as color, with the y=x line superimposed. I think the current graphs are possibly easier to compare theory to practise on though so I'm not against them.
* Theorem 5.5, I believe should be $L_\text{three-layer}$ not $L_\text{two-layer}$.

---

> ### Author Rebuttal · Authors · 2026-03-30
>
> Dear reviewer, thank you for the constructive feedback. We'd like to address your concerns as follows. Please find the supplementary figures in the [rebuttal link here](https://anonymous.4open.science/r/ICML2026_13581-206C/).
> ## **W1**
> Thank you for pointing out this slight difference. We agree that, for tractability, our theory focuses on the result that symmetric learning rates are suboptimal for a single update step but become optimal after two steps. This suggests that 1. asymmetric learning rates may be preferable at the very beginning of training, 2. symmetric learning rates become optimal as cross-layer interactions develop over subsequent steps, even if the initial learning-rate allocation is not optimal. We believe this also points to a practical strategy that use asymmetric learning rates early in training and more symmetric ones later.
>
> To better connect these two regimes, in **Figure 16** of our rebuttal link  we consider a three layer linear network in which the first step uses an asymmetric learning rate allocation by training only the first layer. For the second step, we then search over the test loss as a function of $\eta_1$ under the constraint $\eta_1+\eta_2=C$. We find that the same transition still appears: from asymmetry at the first step to balance at the second step.
> ## **W2**
> In **Figure 17** of our rebuttal link, for the three layer linear neural network, we consider a larger learning rate range for one step of training. We observe that symmetric learning rates lead to significantly worse performance compared to **Figure 2** in our paper.
> ## **W3**
> * **(Nonlinear network)** We considered a three layer nonlinear neural network, where the student model is $f(x_i)= \frac{1}{\sqrt{h}}\sigma\big(\sigma(x_i^\top W_1)W_2\big)a,$ and the teacher model is $y_i=\sigma(\beta^{*\top}x_i),$ with $\sigma$ being the ReLU activation. We use the same orthogonal initialization and training pipeline as in the paper. In **Figure 14** of rebuttal link, we visualize the test loss as a function of $\eta_1$ after 1 and 8-step updates under $\eta_1+\eta_2< O(\sqrt{h})$. Although the curves are relatively less symmetric than in linear case, we still observe a similar asymmetry-to-balance transition.
> * Intuitively, even in the non-linear settings, the gradient updates can still be decomposed into signal aligned terms and residual terms, with the signal aligned part dominating in norm, although the decomposition becomes more technical. Hence, if the learning rates are chosen to address layerwise norm imbalance, we expect the same asymmetry-to-balance behavior to persist.
> * **(Deep linear network)** In **Figure 13** of our rebuttal link, we also consider four-layer and eight-layer linear neural networks, which generalize the two layer and three layer settings. We observe that, for one and two update steps, the same transition from asymmetry to balance still appears. This further increases our confidence that the theoretical findings capture a meaningful phenomenon rather than a narrow theoretical case.
> ## **Q1**
> We agree that when assuming a reasonable learning rate, we can ignore $B_l^t$ only at small t before it has learned enough. The one step and two step updates studied in our paper fall within this regime. We will make this point clearer in the revision.
> ## **Q2**
> We would like to clarify that, for example, in Theorem 5.3, the $O(\eta_1\eta_2/h^5)$ term in the one step expression of $L_{\mathrm{two\text{-}layer}}$ is not due to an additional approximation. In fact, in Lines 1325–1383, we can explicitly derive the coefficient of the $\eta_1\eta_2/h^5$ term. We will make this coefficient explicit in the revision to avoid any misunderstanding caused by the $O(\cdot)$ notation. Therefore, there are no further approximations in the equations for $T_{1}–T_{24}$, and it does not affect our analysis.
>
> ## **Minor Points.**
> * We will revise phrases such as “a single gradient step” that may be misleading. We sincerely apologize for the confusion.
> * In **Figure 18 and 19** of our rebuttal link, we add the stars which imply $\eta_1=\eta_2$ for better visualization, thanks for your suggestions.
> * Thanks for pointing out this typo. It should be $L_{\mathrm{three\text{-}layer}}$ in Theorem 5.5 , we will revise it.

---

> > ### Author Rebuttal · Reviewer_mafu · 2026-04-01
> >
> > Thank you to the authors for addressing many of these points so clearly, I have some remaining issues regarding W1.
> >
> > Thank you for Figure 16! I believe this gets halfway there (symmetric LRs are still better after an asymmetric first step), the remainder is performance on the final model as the first step symmetry is changed. E.g. fix the second step as symmetric (with the same $C$ as the first step: $\eta^{(2)}_1 = \eta^{(2)}_2 = \frac{C}{2}$) and do the same pan over the first step LR (i.e. vary $\eta^{(1)}_1$ subject to $\eta^{(1)}_1 + \eta^{(1)}_2 = C$). I think this would nicely show your claim is reasonable.
> >
> > I know I'm pushing quite hard on this point, it's mostly because the conclusion is so surprising if this is actually true in general which is why I'm hoping for clearer proof - it's saying we should always train a single layer for the first step then train normally. A strong supporting evidence here would be train some simple CNN on an image task and train just the first layer for the first step then train everything and compare the test loss to training everything for 2 steps. If training just the first layer for a step performs better then that's fascinating! (even more so if training for longer doesn't regain that lost performance).
> >
> > The other issue is just phrasing, I feel like there is still a mismatch between what is proven and what is stated, even in the response, it felt like this was stated in a way that if I hadn't read the math carefully I would interpret as the comparing symmetric steps *after* having taken an asymmetric step. Similarly, the phrasing of 'early' and 'late' do not imply 'first step' and 'everything else' to me.

---

> > > ### Author Response · Authors · 2026-04-02
> > >
> > > We thank the reviewer for the constructive and helpful experimental suggestion. We address the concerns below. We have updated our supplementary figures in the [rebuttal link here](https://anonymous.4open.science/r/ICML2026_13581-206C/).
> > >
> > >
> > > * **In Figure 16(b)**, we study a 3-layer NN in which the second update step uses a symmetric learning-rate allocation, $\eta_1 =\eta_2 = \frac{C}{2}$, while the first step is optimized under the constraint $\eta_1 + \eta_2 =  C$. We observe that the two-step loss is minimized at the asymmetric endpoints $\eta_1 = C$ or $\eta_1 = 0$, whereas a symmetric allocation in step 1 yields a suboptimal final loss. Together with **Figure 16(a)** (which is the figure before we update), this further validates our asymmetry-to-balance claim.
> > > *  To further strengthen our claim, following the reviewer’s suggestion, we consider a 3-layer CNN whose first two layers are convolutional layers and whose final layer is a fixed linear readout layer.
> > >
> > > **Dataset.** We consider a synthetic binary image classification problem on $16 \times 16$ grayscale images. Each sample belongs to one of two classes: Class $-1$: an image containing a horizontal bar. Class $+1$: an image containing a vertical bar.
> > >
> > > **Model.** We consider  two trainable $3\times3$ convolutional layers with no bias: the first maps from 1 input channel to 8 hidden channels, and the second maps from 8 channels to 8 channels, with a ReLU activation after each convolution. The resulting feature map is then globally average pooled over the spatial dimensions, producing an 8-dimensional representation, which is fed into a fixed random linear readout to produce a single scalar output.
> > >
> > > In **Figure 16 (c ) and (d)**, for the first step,  we consider only updating the first(second) layer, using learning rate C. For the second step, we study layer-wise learning-rate allocation under the constraint $\eta_1+\eta_2=C$. Although the  optimum no longer occurs exactly at $\eta_1=\eta_2$ because of the changed architecture and the presence of nonlinearities, the optimal performance is still attained when **$\eta_1 \approx \eta_2$**. This indicates that, after an asymmetric first step, learning rates that are approximately symmetric still remain preferable. In **Figure 16 (e)** We study a 3-layer CNN in which the second update step uses a symmetric learning-rate allocation, $\eta_1 = \eta_2 = \frac{C}{2}$, while the first step is optimized under the constraint $\eta_1 + \eta_2 =  C$. We find training the second layer (corresponding to $\eta_1=0$) at the first step is better than the symmetric allocation. Thus, the CNN experiments lead to  similar conclusions as in the 3-layer linear neural network setting, further reinforcing our claim.
> > > * We apologize for the confusion caused by our presentation. We acknowledge that our theory establishes the asymmetry-to-balance transition by showing that symmetric learning rates are suboptimal for a single step but optimal for two steps. We also acknowledge that this is not exactly the same as taking one asymmetric step followed by one symmetric step. We will make this distinction much clearer in the revision and state our claims more precisely. In addition, our experiments provide partial evidence connecting these two views, and we will expand this discussion to clarify the relationship. We will also revise the early/late terminology to use more accurate phrasing. We thank the reviewer again for these valuable suggestions.

---

### Official Review · Reviewer_sLK6 · 2026-03-11

**Soundness:** 3
**Presentation:** 3
**Significance:** 3
**Originality:** 4
**Overall Recommendation:** 5
**Confidence:** 3

**Summary:**

This paper studies the role of layer-wise learning rates in early training dynamics of linear neural networks. The authors analyze two-layer and three-layer linear networks trained on a single-index model and derive exact closed-form expressions for gradients and test loss after one and two steps of gradient descent.

Using these formulas, the paper develops a theoretical framework to analyze how learning rates should scale across layers in the early phase of training. The authors show that:

in the very first step, unequal layer-wise learning rates can minimize the test loss,

while equal learning rates become optimal in subsequent steps.

The analysis relies on decomposing gradients into dominant signal-aligned components and residual terms, and bounding the approximation error introduced by the proposed gradient approximation. Experiments validate the theoretical predictions and demonstrate the importance of balancing learning rates during early training.

**Compliance With Llm Reviewing Policy:**

Affirmed.

**Key Questions For Authors:**

1. Generalization beyond linear networks.
The theoretical analysis focuses on linear neural networks trained on a single-index model. To what extent do the conclusions about optimal layer-wise learning rate scaling hold for nonlinear networks (e.g., ReLU MLPs or CNNs)? In particular, do the authors observe the same phenomenon (asymmetric learning rates being beneficial in the very first step but symmetric ones later) in nonlinear settings?
Impact on evaluation: If empirical evidence shows that the phenomenon persists in nonlinear networks, this would significantly strengthen the practical relevance of the work.

2. Sensitivity to initialization assumptions.
The analysis relies on specific assumptions about initialization (e.g., Gaussian weights and independence properties). How sensitive are the derived optimal learning-rate scalings to these assumptions? For instance, do similar conclusions hold under other common initialization schemes (e.g., orthogonal initialization or scaled initializations used in deep networks)?
Impact on evaluation: Demonstrating robustness to initialization would increase confidence that the results are not artifacts of the theoretical setup.

3. Extension beyond the first two optimization steps.
The core theoretical contribution analyzes the first one or two steps of gradient descent. Do the authors have theoretical or empirical evidence that similar learning-rate balancing principles remain relevant beyond the very early phase of training?
Impact on evaluation: If the insights extend beyond the first few steps, the contribution would appear more broadly useful for understanding optimization dynamics.

4. Practical implications for optimizer design.
The paper derives theoretically optimal layer-wise learning rate scalings in the analyzed setting. How might these results translate into practical training strategies or optimizer modifications for modern deep learning models? For example, could the theory inform adaptive layer-wise learning-rate schedules?
Impact on evaluation: Clarifying the practical implications would help assess the significance of the contribution for real-world deep learning practice.

5. Empirical validation scope.
The empirical validation appears limited relative to the theoretical analysis. Could the authors provide additional experiments that test the predicted learning-rate scaling behavior across different architectures, depths, and datasets?
Impact on evaluation: Stronger empirical validation would increase confidence that the theoretical findings capture a meaningful phenomenon rather than a narrow theoretical case.

**Limitations:**

The paper discusses some limitations of the theoretical setting, particularly the focus on linear neural networks and early training dynamics. However, the discussion could be strengthened by more explicitly addressing how these assumptions limit the applicability of the results to modern deep learning models with nonlinear activations and deeper architectures. In particular, clarifying whether the proposed insights might extend to nonlinear networks or remain primarily theoretical would improve the limitations section.

Regarding societal impact, the work is primarily theoretical and does not introduce immediate societal risks.

**Strengths And Weaknesses:**

Strengths

1. Precise analysis of early GD dynamics

The paper provides an exact analysis of the first two gradient descent steps, which is uncommon. Most theoretical works focus on:

gradient flow,

asymptotic regimes,

NTK approximations.

This work instead analyzes finite-step discrete dynamics, which is an interesting perspective.

2. Closed-form loss expressions

The derivation of explicit formulas for the test loss after one and two updates is technically strong and could be useful for future theoretical work.

3. Insight into layer-wise learning rates

The result that asymmetric learning rates may be beneficial only at the very beginning is an interesting conceptual insight.

4. Theoretical rigor

The appendix contains detailed proofs and concentration arguments. The paper appears mathematically careful.

Weaknesses

1. Limited model setting

The analysis is restricted to:

linear neural networks

single-index target functions

early training (1–2 steps)

It is unclear how these results extend to nonlinear networks, which limits practical relevance.

2. Empirical validation seems limited

From the paper structure, the experimental section appears relatively small compared to the theoretical development. It would strengthen the paper to include:

experiments on deeper networks

nonlinear activations

modern architectures

to demonstrate whether the phenomenon persists beyond linear settings.

3. Practical implications are somewhat unclear

Although the theory suggests optimal scaling for layer-wise learning rates, the paper does not clearly show:

how these results translate into practical optimizer design

whether the predicted scaling improves training in realistic settings.

4. Clarity of presentation

Some parts of the paper are difficult to follow:

the gradient decomposition and approximation could be explained more intuitively

the high-level intuition behind the optimal learning-rate scaling is not always clear.

---

> ### Author Rebuttal · Authors · 2026-03-30
>
> We thank the reviewer for the constructive feedback. Please find the supplementary figures in the [rebuttal link here](https://anonymous.4open.science/r/ICML2026_13581-206C/).
> ##  **W1+W2+Q1+Q5**
> * Due to space limitation, for discussion of the assumptions of our setup,  as well as intuition and preliminary evidence for whether a similar asymmetry to balance transition persists in nonlinear networks, please see **our responses to Reviewer m6Zx’s W1 and 4utP’s W1.**
> * In **Figure 13** of our rebuttal link, we also consider 4-layer and 8-layer linear NNs, which generalize the 2-layer and 3-layer settings. We observe that, for 1 and 2-step updates, the same transition from asymmetry to balance still appears. This further increases our confidence that the theoretical findings capture a meaningful phenomenon rather than a narrow theoretical case.
> ## **Q3**
> **(Beyond first two updates)** In our paper, **Figures 4–7** empirically extend the training dynamics of 2-layer and 3-layer linear NNs under both orthogonal and Gaussian initialization to 4 and 8 update steps. In addition, to further examine longer horizon behavior, **Figure 15** in our rebuttal link extends the three layer linear network setting to 512 update steps. These results suggest that the asymmetry to balance transition can hold more generally beyond the 1 step and 2 step regimes.
> ## **W3+Q4**
> **(Practical implications)** Since our paper reveals the asymmetry to balance transition in layer-wise learning rate allocation, the insights offer theoretical support for layer-wise learning rate schedulers that aim to promote layer balance at later stages of training. Here we provide a simple example based on a two layer linear network, to illustrate how the insights from our theory can be used to design a practical layer-wise lr scheduler or optimizer. We consider a teacher model $y_i=M^\top x_i$ and a student model $f(x_i)=x_iW_1W_2$, where $x_i$ is the input and $W_1, W_2$ are the two trainable matrices. Since Frobenius norm  is a classic generalization metric, we can consider leveraing the Frobenius norms of $W_1$ and $W_2$  as a metric to design a learning rate scheduler.
>
> First, we expect the layer with the larger Frobenius norm to be assigned a smaller learning rate, due to the property of the metric. More importantly, based on the theoretical insights in our paper, we expect the lrs of the two layers to become balanced in the later stages of training, which motivates us to promote balance between the layer norms. As a result, at each step $t$,  we set the lrs for ${W_1}^t$ and ${W_2}^t$ as:
> $\eta_{W_1}^{(t)}=\frac{2|{W_2}^t|_F}{|{W_1}^t|_F+|{W_2}^t|_F}\mathrm{lr} $ where $\mathrm{lr}$ is a uniform base learning rate.
>
> Similarly, $\eta_{W_2}^{(t)}=\frac{2|{W_1}^t|_F}{{W_1}^t|_F+|{W_2}^t|_F}\mathrm{lr}$. As training enters the later stage, this balance-driven learning rate scheduler $\big||{W_1}|_F-|W_2|_F\big|\to 0$, the learning rates also become balanced and vice versa. It is worthing noting that for this matrix-factorization type linear network, the curvature at convergence, measured by the largest Hessian eigenvalue, is related to $\big||W_1|_F-|W_2|_F\big|$; in particular, smaller norm gap corresponds to a flatter solution[1]. Therefore, the transition of the learning rates from asymmetry to balance also corresponds to the process by which the model gradually converges to a flatter minima.
>
> In **Figure 20** of our rebuttal link, we initialize with $|W_2|_F=6$ and $|W_1|_F=1$, and compare this design with a uniform learning rate used throughout training. We find that this layer-wise schedule captures the asymmetry-to-balance transition observed in our paper, and achieves lower training loss and test loss than the fully uniform baseline. More specifically, we observe that $\big||W_1|_F-|W_2|_F\big|$ approaches zero in the middle and late stages of training, which corresponds to increasingly balanced learning rates.
> ## **W4**
> * To better understand the approximation, we use the 2-layer network as an example. In (9) of our paper, we can see that $A_1^t$ is related to the teacher target matrix $M$, whereas $B_1^t$ depends only on the weight matrices. Intuitively, $A_1^t$ captures the information provided by the data and thus represents the dominant, signal aligned component, while $B_1^t$ can be viewed as a smaller residual term. We then rigorously bound these residual terms in operator norm, establishing conditions under which the approximate gradients accurately capture the test loss dynamics.
> * Due to space limitation, for the high-level intuition behind the optimal learning-rate scaling, please **see our responses to Reviewer m6Zx’s W3.**
>
> ## **Q2**
> Appendix E and F of our paper extend the main results to Gaussian initialization, deriving a theoretical loss expression for one-step GD and complementing it with simulation results (**Figure 5 and 7** in our paper) for the multi-step setting, where we find similar asymmetry-to-balance transition.

---

> > ### Author Rebuttal · Reviewer_sLK6 · 2026-04-02
> >
> > I thank the authors for the detailed and constructive rebuttal. The additional explanations and empirical results help clarify several important points.
> >
> > On generalization and empirical validation (Q1, Q5).
> > The additional experiments on deeper linear networks (4- and 8-layer models) and longer training horizons (up to 512 steps) strengthen the empirical evidence that the asymmetry-to-balance transition is not limited to the very first optimization steps or to shallow architectures. This addresses part of my concern regarding the limited temporal scope of the analysis.
> >
> > However, the question of whether this phenomenon extends to nonlinear networks remains only partially addressed. While the rebuttal points to intuition and preliminary evidence, the absence of clear experimental validation in nonlinear settings still limits the practical significance of the claims.
> >
> > On extension beyond two steps (Q3).
> > The additional results showing that the transition persists beyond the first two steps are convincing and significantly strengthen the contribution. This alleviates my concern that the analysis might be restricted to a very narrow regime.
> >
> > On practical implications (Q4).
> > The proposed layer-wise learning rate scheduler based on Frobenius norms is an interesting and concrete illustration of how the theoretical insights could translate into practice. The connection to flatter minima via norm balancing is also insightful. The empirical improvement over a uniform learning rate baseline is encouraging.
> >
> > That said, the proposed method is still demonstrated only in linear settings, and it remains unclear how well this approach would perform in modern deep learning architectures.
> >
> > On theoretical assumptions and approximation (W4, Q2).
> > The clarification of the gradient decomposition and the role of dominant vs. residual terms is helpful. The extensions to Gaussian initialization further support the robustness of the theoretical findings within the considered framework.

---

> > > ### Author Response · Authors · 2026-04-03
> > >
> > > Thank you for your thoughtful and encouraging feedback. We are pleased that our rebuttal helped clarify several important points and helped address many of your concerns.
> > >
> > > We would also like to respectfully clarify that we did include experimental validation in a nonlinear setting to examine whether the asymmetry-to-balance transition observed in linear neural networks extends beyond the linear case. Specifically, in **Figure 14** of the [rebuttal link](https://anonymous.4open.science/r/ICML2026_13581-206C/), we consider a three-layer nonlinear neural network, where the student model is $f(x_i)= \frac{1}{\sqrt{h}}\sigma\big(\sigma(x_i^\top W_1)W_2\big)a$, and the teacher model is $y_i=\sigma(\beta^{*\top}x_i)$,with $\sigma$ denoting the ReLU activation. Using the same orthogonal initialization and training pipeline as in the paper, we visualize the test loss as a function of $\eta_1$ after 1-step and 8-step updates, and still observe a similar asymmetry-to-balance transition. Also, based on Reviewer mafu's suggestion, we were also able to validate our theory for CNN setting in **Figure 16(c )(d)(e)** of the [rebuttal link](https://anonymous.4open.science/r/ICML2026_13581-206C/), the CNN experiments lead to similar asymmetry-to-balance transition as shown in the 3-layer linear neural network setting, further strengthening our claim.
> > >
> > > We again sincerely thank the reviewer for the valuable suggestions.

---

### Official Review · Reviewer_4utP · 2026-03-13

**Soundness:** 2
**Presentation:** 3
**Significance:** 2
**Originality:** 2
**Overall Recommendation:** 4
**Confidence:** 3

**Summary:**

This paper studies how to choose learning rates across layers in early training stages for two-layer and three-layer networks. Their main findings can be summarized as: equal learning rates are suboptimal in the first step but become optimal in the subsequent steps. Their numerical experiments also support the theoretical findings.

**Compliance With Llm Reviewing Policy:**

Affirmed.

**Final Justification:**

This paper is mathematical rigor and provide experimental supports. During the rebuttal, the authors have addressed my concern regarding feasibility of step sizes, the significance of studying convergence of GD for one/two steps, etc. Therefore, I will increase the score to 4.

**Key Questions For Authors:**

**Question 1** In In Figure 1 and 2, the learning rates are extremely large compared to what people use in practice. Can the authors comment on how these empirical insights can be transferred to practice?

**Question 2** The theoretical results cover only one and two gradient steps. While Figure 4 shows empirically that the balance optimality extends to 4 and 8 steps, there is no theorem supporting this. For practitioners, knowing what happens at step 100 or 1000 matters far more than at step 2. I think a more interesting regime is that for fine-tuning, how choosing learning rates across layers since in the fine-tuning, we often only fine-tune for very few epochs.

**Question 3** The gradient decomposition into signal-aligned (A) and self-interaction (B) terms relies heavily on the linear structure. Do you have any intuition or preliminary evidence for whether similar asymmetry-to-balance transition occurs in networks with nonlinear activations such as ReLU?

**Question 4** The admissible learning rate regime shrinks from $h\sqrt{h}$ to $h$ when moving from two to three layers. Do you expect this trend to continue, i.e., $\sqrt{h}$ for four layers?

**Limitations:**

Please see the weakness and questions for details.

**Strengths And Weaknesses:**

**Strength**

- Clean test error evolution in early stage: Theorem 5.3 and Theorem 5.5 show that for two-layer and three-layer linear networks, how the test loss evolve after one-step and two-step GD update. These equations provide intuitions of why balanced learning rates are suboptimal in early training.

- Interesting corollary 5.4: Corollary 5.4 shows that in certain regime, balanced learning rates are not local minimum for one-step loss but is a local minimum of two-step loss. These findings are interesting.

**Weakness**

- Restrictive assumptions: the theoretical results of the work are derived under restrictive assumptions, such as orthogonal initialization, fixed last-layer weight for three-layer linear models, focusing on only two-step horizons.

- Conditions in Corollary 5.4: the authors require $h>256$ but the 256 seems arbitrary. The paper does not provide easily interpretable sufficient conditions on $h$ in terms of $\alpha$ alone, making it hard to know when the theorem actually applies in practice. Moreover, can the authors comment on why assuming $\eta_1+\eta_2=2h^{\alpha}$ is the regime worth studying?

- Restriction to linear models: the theory in this paper only applies to linear models. But I don't think this is a big issue. In the deep learning theory community, many interesting results start with linear models. However, what concerns me is that there are experiments showing that the theoretical insights can guide the practitioners in choosing learning rates for more realistic networks.

---

> ### Author Rebuttal · Authors · 2026-03-30
>
> We thank the reviewer for constructive feedback and address the concerns as follows. The supplementary figures are available in [rebuttal link here](https://anonymous.4open.science/r/ICML2026_13581-206C/).
> ## **W1+W3+Q3**
> **(Assumptions/scope of our setup)** We acknowledge the assumptions needed for our setup. However, unlike prior work that mainly focuses on asymptotic convergence, gradient flow, or kernel-based analyses, our work derives exact closed-form expressions for the gradients and test loss after one and two GD steps, enabling a precise characterization of early training dynamics. In particular, we directly link finite-step layer-wise lr allocation to test loss. This is already nontrivial even for two and three-layer linear NNs: at each step, we decompose the gradient into dominant signal-aligned components and smaller residual terms, and rigorously bound the residual terms in operator norm to justify when the approximate gradients accurately capture the test loss dynamics.  We will extend the framework to more realistic training  settings in the future.
>
> Due to space limitation, we  kindly refer the reviewer to **our responses to Reviewer m6Zx’s W1 and W2** for results extend to nonlinear networks and more training steps.
> ## **W2**
> * In our paper, **Lines 1510–1515** give the derivative of $L_{\mathrm{two\text{-}layer}}$ with respect to $\eta_1$. By factoring out $(h^\alpha-\eta_1)$, the remaining expression contains only one negative term $-8h^{\alpha-5}$. To ensure that $\eta_1=\eta_2=h^\alpha$ is a local minimum of $L_{\mathrm{two\text{-}layer}}$, we need the other terms to be of smaller order than $O(h^{\alpha-5})$.
> This yields the admissible range of $\alpha$, shown in **Lines 1519–1530**. Once this range is obtained, we can further derive an explicit condition on $h$ in terms of $\alpha$ that guarantees $\eta_1=\eta_2=h^\alpha$ is a local minimum of $L_{\mathrm{two\text{-}layer}}$. For example, when $\alpha=\tfrac{3}{2}$, substituting into the derivative shows that it suffices to require $h>256$.
> * We focus on $\eta_1+\eta_2=2h^\alpha$ regime as layer-wise lr allocation is typically studied under a fixed lr budget across layers. Imposing such a budget enables a fair comparison between different layer-wise allocation strategies and makes it meaningful to ask whether a balanced allocation across layers is beneficial.
> ## **Q1**
> * We acknowledge that for one or two-step updates, (18) and (19) in our paper imply that the large lr regime can indeed be fairly large, similar to the regime studied in [1]. In practice, however, training runs for many more steps. As shown in **Figure 15(h)** of our rebuttal, when a three-layer linear NN is trained for 512 steps, compared with **Figure 15(a)**, it achieves a smaller loss with smaller lr's than in the two-step setting. This suggests that the large lr regime shrinks as the training horizon increases. Therefore, in practical many-step settings, the relevant lrs are expected to lie in a more realistic and acceptable range.
> * We also would like to emphasize that the lr scale is tied to the choice of initialization. If we consider an initialization whose scale is smaller than the standard orthogonal initialization, we can obtain a smaller lr regime, consistent with the observations in Appendix D of [2].
> ## **Q2**
> We agree that fine-tuning is a practically interesting regime and we plan to explore this regime in future work. At the same time, **Figure 15** of our rebuttal extends the three layer linear NN to 512 steps, providing empirical evidence that the asymmetry to balance transition can persist beyond the 1 and 2 step regimes.
> ## **Q4**
> We kindly note that the two and three-layer architectures are structurally different: the three layer model is scalar-output, whereas the two layer model is closer to a matrix-factorization type model. Accordingly, to keep the test loss of both models in a reasonable range,  (1) and (2) in our paper use different scalings for initializations when defining the models. This in turn makes the admissible learning-rate regime for the two layer model one order larger than that for the three layer model.
>
> Since the 4-layer model is structurally similar to the three layer model, we do not expect the admissible learning-rate regime to shrink by another factor of $\sqrt{h}$. In **Figures 13(a) and 13(b)** of our rebuttal link, we study the four layer model and compare it with Figure 2 in our paper. We find that the four layer model does not exhibit an additional $\sqrt{h}$ shrinkage. However it can indeed achieve smaller loss with same lr compare to 3-layer NN, which suggests that, relative to the three layer model, the large lr regime does become somewhat narrower.
>
> [1] Ba, Jimmy, et al. "High-dimensional asymptotics of feature learning: How one gradient step improves the representation." NeurIPS 2022
>
> [2] Kothapalli, Vignesh, et al. "From Spikes to Heavy Tails: Unveiling the Spectral Evolution of Neural Networks." TMLR 2025

---

> > ### Author Rebuttal · Reviewer_4utP · 2026-04-02
> >
> > I appreciate the mathematical rigor of the derivations and find the phase transition result genuinely interesting. However, I have a broader concern about the significance of studying only one or two steps of gradient descent. Specifically:
> >
> > **What is the scientific justification** for focusing on such a small number of steps? Most practical training runs involve thousands to millions of steps, and it is unclear how insights about the first two steps translate to long-run behavior. The paper acknowledges this limitation but does not provide a compelling argument for why early-step dynamics are sufficient to understand generalization.
> >
> > **Is there a research program** where one/two-step analyses serve as a foundation for full convergence results? Could the authors point to concrete examples in the literature where this approach, starting with exact finite-step dynamics and gradually extending to general convergence, has been successfully executed?
> >
> > I raise these questions not to dismiss the contribution, but because I believe the paper would be significantly strengthened by situating the one/two-step approach within a broader scientific program with a clear path toward more general results.

---

> > > ### Author Response · Authors · 2026-04-02
> > >
> > > We appreciate the reviewers suggestion. Our motivation to explore one/two-step GD analysis is motivated by the literature on feature learning in shallow NNs [1,2,3,4]. The main idea of these works is as follows:
> > >
> > > **Setup** Consider a single index model $y_i=\sigma'(\beta'^T x_i)$ with a target direction ($\beta'$) and link function $\sigma' : \mathbb{R} \to \mathbb{R}$. The goal is to train a two layer NN: $f(x_i) = \frac{1}{\sqrt{N}}a^\top\sigma(W^\top x_i)$ with point-wise activation function $\sigma$ such that $\frac{1}{n}\sum_{i=1}^n||y_i - f(x_i)||^2$ is minimized for all data points. In essence, the student network $f(.)$ is trained to learn the (teacher) single index model.
> > >
> > > **Key message:** The key takeaway of [1, 2] is that, one step of GD update with a large learning rate is sufficient for the NN to learn the unknown target direction of the single index teacher model. [3] extends this line of analysis with a setup considering multi-index target functions and multiple finite GD updates. This is an example of the related work which gradually extend the analysis from one-step to multiple steps. Recent work [4] also extended the one step GD analysis to Full batch Adam as well.
> > >
> > > **Comparison with our paper:** In this growing body of work, a common limitation is that only two layer non-linear NNs with a fixed learning rate are considered for theory (that too with a layerwise training strategy). In particular, [1] fixes the second layer weight vector ($a$) for the first GD update and instead only updates the first layer $W$ matrix. The $a$ weight vector is then obtained via a ridge regression algorithm. Such a setup is not common in practise, but joint training of both layers comes with its own technical challenges for theory. However, our work shows that this issue can be avoided by removing the non-linearity in the NNs. This allows us to consider the joint training of all layers with different learning rates and showcase the cross-layer interactions during training. Also, based on our follow up experiments with non-linear NNs and CNNs during the rebuttal (see our responses to Reviewer mafu), we noticed that the insights from our theory on learning rate balancing indeed translate to these practical networks as well. One of the challenging yet impactful extensions of our work (in this research program) is to theoretically analyze the cross-layer feature interactions over multiple steps (t > 2).
> > >
> > > Overall, our setup and the decision to consider a small number of steps and employ large learning rates are inspired by these previous works (which are gaining more interest nowadays). We will extend the related work section to discuss these aspects and broaden the scope of our work.
> > >
> > >
> > > [1] Ba, Jimmy, et al. "High-dimensional asymptotics of feature learning: How one gradient step improves the representation." NeurIPS 2022
> > >
> > > [2] Dandi, Yatin, et al. "How two-layer neural networks learn, one (giant) step at a time." JMLR 2024
> > >
> > > [3] Dandi, Yatin, et al. "The benefits of reusing batches for gradient descent in two-layer networks: breaking the curse of information and leap exponents." ICML 2024.
> > >
> > > [4] Kothapalli, Vignesh, et al. "From Spikes to Heavy Tails: Unveiling the Spectral Evolution of Neural Networks." TMLR 2025

---

### Official Review · Reviewer_m6Zx · 2026-03-13

**Soundness:** 3
**Presentation:** 3
**Significance:** 3
**Originality:** 3
**Overall Recommendation:** 4
**Confidence:** 3

**Summary:**

This paper studies the early-stage dynamics of layer-wise learning-rate allocation in linear neural networks under a teacher-student setting. The analysis focuses on two-layer and three-layer linear networks trained with mean squared error, primarily under random orthogonal initialization. The main technical strategy is to decompose the gradient into a dominant signal-aligned term and a residual term, and then use this approximation to derive explicit expressions for the test loss after one and two steps of gradient descent.

Based on these formulas, the paper analyzes the local optimality of layer-wise learning-rate allocations $(\eta_1,\eta_2)$ under a fixed total learning-rate budget. The main conclusion is that, after the first update step, the balanced allocation $\eta_1=\eta_2$ is generally not locally optimal, whereas after two update steps, under suitable width and learning-rate scaling conditions, the balanced allocation becomes locally optimal. In this sense, the paper identifies a transition from "asymmetry is better'' to "balancing is better'' in early training. The paper develops this analysis for both two-layer and three-layer linear models and supports the theoretical predictions with numerical experiments.

**Compliance With Llm Reviewing Policy:**

Affirmed.

**Final Justification:**

The author provided meaningful clarifications during the rebuttal, so I tend to maintain my original positive score.

**Key Questions For Authors:**

1. Can the authors further clarify the more general mechanism behind the restoration of symmetry from one step to two steps, beyond a term-by-term comparison in the explicit expansions?

2. Can the authors add experiments on the simplest nonlinear networks to test whether the phenomenon has at least some qualitative robustness beyond linear models?

3. Under SGD noise, label noise, or non-orthogonal initialization, does the same qualitative phenomenon still hold?

4. Can the authors position the paper more clearly relative to the existing literature on layerwise adaptive optimization and balancing, and specify more explicitly what the new theoretical contribution is?

5.  Is there a possibility of a more general phase characterization over multiple training steps?

**Limitations:**

Yes

**Strengths And Weaknesses:**

Strengths:
1. The paper studies a clear and meaningful question: how layer-wise learning-rate allocation affects early training dynamics, and in particular whether the balanced choice $\eta_1=\eta_2$ is actually optimal in the first few gradient descent steps. This is a well-motivated problem and the main conclusion is easy to state and interesting.

2. The finite-step perspective is a notable strength. Rather than focusing on asymptotic convergence, gradient flow, or steady-state behavior, the paper analyzes the explicit test loss after one and two gradient descent steps. This gives the work a distinctive angle and makes the results conceptually sharp.

3. The technical development is coherent. The paper decomposes the gradient into a dominant signal-aligned term and a residual term, identifies a regime in which the dominant term controls the dynamics, derives explicit one-step and two-step test-loss expressions, and then uses these formulas to study the local optimality of different learning-rate allocations. The extension from two-layer to three-layer linear networks also strengthens the technical story.

4. The main phenomenon identified by the paper is nontrivial and valuable: after one gradient step, an asymmetric allocation can be preferable, while after two steps the balanced allocation becomes locally optimal under suitable scaling conditions. This gives a clean theoretical example showing that learning-rate balancing is not a static principle but a dynamic one.

5. The experimental results, while limited, are aligned with the theoretical claims. In particular, the plots reproduce the predicted transition from imbalance-favoring behavior after one step to balance-favoring behavior after two steps, and the paper also includes evidence beyond the exact orthogonal-initialization setting.

Weaknesses:
1. The main limitation is the narrow scope of the theoretical setting. The analysis is developed for linear teacher-student models with strong assumptions such as noise-free regression, full-batch gradient descent, and structured initialization. These assumptions make the problem tractable, but they also limit the extent to which the conclusions can be transferred to modern nonlinear deep networks.

2. The strongest results are concentrated on the first one or two update steps. This is enough to support the central claim of the paper, but it leaves open whether the observed transition from asymmetry to balance reflects a more general multi-step principle or only a very early-stage effect.

3. Although the formulas clearly establish the phenomenon, the higher-level mechanism is somewhat less developed. The paper shows what happens and gives reasonable intuition, but the conceptual explanation for why symmetry re-emerges after additional updates could be made more distilled and transferable.

4. The empirical section mainly validates the theory in settings that are still very close to the theoretical assumptions. This is appropriate for a theory paper, but even modest experiments on simple nonlinear networks or under mild stochasticity would have made the broader significance more convincing.

5. The paper's significance is therefore somewhat constrained by its scope. I find the theoretical result interesting and technically sound, but the current version provides limited evidence that the insights will extend in a strong way to practical layer-wise learning-rate design in realistic modern architectures.

---

> ### Author Rebuttal · Authors · 2026-03-30
>
> We appreciate the reviewer's constructive feedback. Please find the supplementary figures in the [rebuttal link here](https://anonymous.4open.science/r/ICML2026_13581-206C/).
>
> ## **W1+Q2**
> * Due to space limitation, please refer to our response to **Reviewer 4utP’s W1** for a discussion on why the  assumptions/scope of our setup are needed and why the setting is nontrivial.
> * **(Nonlinear network)** Here we consider a three layer nonlinear neural network, where the student model is $f(x_i)= \frac{1}{\sqrt{h}}\sigma\big(\sigma(x_i^\top W_1)W_2\big)a,$ and the teacher model is $y_i=\sigma(\beta^{*\top}x_i),$ with $\sigma$ being the ReLU activation. We use the same orthogonal initialization and training pipeline as in the paper. In **Figure 14** of rebuttal link, we visualize the test loss as a function of $\eta_1$ after 1 and 8-step updates under $\eta_1+\eta_2< O(\sqrt{h})$. Although the curves are relatively less symmetric than in linear case, we still observe a similar asymmetry-to-balance transition.
> * Intuitively, even in the non-linear settings, the gradient updates can still be decomposed into signal aligned terms and residual terms, with the signal aligned part dominating in norm, although the decomposition becomes more technical. Hence, if the learning rates are chosen to address layerwise norm imbalance, we expect the same asymmetry-to-balance behavior to persist.
> ## **W2+Q5**
> **(Beyond first two updates)** In our paper, **Figures 4–7** empirically extend the training dynamics of two layer and three layer linear neural networks under both orthogonal and Gaussian initialization to 4 and 8 update steps. In addition, to further examine longer horizon behavior, **Figure 15** in our rebuttal link extends the three layer linear network setting to 512 update steps. These results suggest that the asymmetry-to-balance transition can hold more generally beyond the 1 step and 2 step regimes.
>
> ## **W3+Q1**
> **(More general mechanism and high-level intuition)** From a high-level perspective, the asymmetry-to-balance transition arises because, as training proceeds, the learning dynamics become increasingly coupled across layers. In this regime, balanced lrs promote more effective layer interaction and cross-layer feature learning, which leads to better performance. More specifically, take the 2-layer NN with orthgonal initialization as an example, at the first update step，the dominant signal-aligned components $A_{1}^0=\frac{1}{h}M{W_{2}^0}^{\top}$ and  $A_{2}^0=\frac{1}{h}{W_{1}^0}^{\top}M$, since  $W_{1}^0$ and ${W_{2}^0}$ are random matrix in the initialization, there is little interaction or information exchange between the two layers, so this manifests in the loss as only positive lower-order $O(\eta_1\eta_2)$ terms and the best strategy is just letting one layer learn the signal. As a result, using the same lr for both layers is suboptimal. However, after one update, we have  $\widetilde{A_{1}^1}=\frac{1}{h}M\widetilde{{W_{2}^1}^{\top}}$ and  $\widetilde{A_{2}^1}=\frac{1}{h}\widetilde{{W_{1}^1}^{\top}}M$, in this regime cross-layer feature learning begins to emerge, and the two layers start to coordinate in shaping the learned representation which is reflected in the fact that the test loss exhibits a higher order dependence on the product $\eta_1\eta_2$, together with the symmetry of the updates across the two layers, the balanced lrs become more beneficial.
>
> ## **W4+Q3**
> * **(Label noise)** In **Figure 11 and 12** of our rebuttal link, we consider adding label noise $\xi \sim \mathcal{N}(0, \rho)$ to the teacher model in both two layer and three layer linear neural networks. We find that the asymmetry-to-balance transition still persists.
> * **(Gaussian initialization)** We kindly point out that Appendix E and F of our paper extend the main results to Gaussian initialization, deriving a theoretical loss expression for one-step GD and complementing it with simulation results for the multi-step setting(**Figure 5 and 7** in our paper), where we find similar asymmetry-to-balance transition.
>
> ##  **W5+Q4**
>
> * The new theoretical contribution of our paper is that, by deriving exact closed form expressions for the test loss after one and two update steps, we provide a precise characterization of early training dynamics. This in turn reveals the asymmetry-to-balance transition in layer-wise lr allocation and offers theoretical support for layer-wise lr schedulers that aim to promote layer balance at later stages of training [1].
> * Due to space limitation, for the insights that extend to practical layer-wise learning rate design, please see  **our responses to Reviewer sLK6's W3.**
>
> [1] Zhou, Yefan, et al. "Temperature balancing, layer-wise weight analysis, and neural network training." NeurIPS 2023

---

> > ### Author Rebuttal · Reviewer_m6Zx · 2026-04-04
> >
> > Thanks for your detailed rebuttal. The authors addressed my main questions with useful clarifications. I do not have further questions.

---

### Decision · Program_Chairs · 2026-04-30

**Decision:**

Accept (regular)

**Comment:**

This paper provides an exact analysis of the first two gradient descent steps of the gradient descent training of two-layer and three-layer linear neural networks. Concerns from the reviewers mainly include: theoretical setting too narrow (Reviews m6Zx, 4utP, sLK6, and mafu, but obviously, this may be necessary for the exact analysis, and I see that the rebuttal is largely accepted); analysis limited to the first two steps only (Reviewers m6Zx, 4utP, and sLK6), while the rebuttal is accepted since similar analysis is very active in the literature, but not the restriction of just this paper; insufficient experiments (Reviewers m6Zx, 4utP, sLK6, and mafu), while the rebuttal is accepted since the authors added new experiments; lack of explanation of higher-level mechanism (Reviewers m6Zx and sLK6), while the rebuttal is accepted since the authors do provide more explanation. Overall, reviewers accept that the paper provides solid new analysis and insights.